# Improved Scaling Laws via Weak-to-Strong Generalization in Random Feature Ridge Regression

Diyuan Wu [1]   Lehan Chen [2]   Theodor Misiakiewicz [3] [*]   Marco Mondelli [1] [*]

## Abstract

It is increasingly common in machine learning to use learned models to label data and then employ such data to train more capable models. The phenomenon of *weak-to-strong generalization* exemplifies the advantage of this two-stage procedure: a strong student is trained on imperfect labels obtained from a weak teacher, and yet the strong student outperforms the weak teacher. In this paper, we show that the potential improvement is substantial, in the sense that it affects the scaling law followed by the test error. Specifically, we consider students and teachers trained via random feature ridge regression (RFRR). Our main technical contribution is to derive a deterministic equivalent for the excess test error of the student trained on labels obtained via the teacher. Via this deterministic equivalent, we then identify regimes in which the scaling law of the student improves upon that of the teacher, unveiling that the improvement can be achieved both in bias-dominated and variance-dominated settings. Strikingly, the student may attain the minimax optimal rate regardless of the scaling law of the teacher—in fact, when the test error of the teacher does not even decay with the sample size.

## 1. Introduction

Across modern machine learning pipelines, models often generate synthetic labels (or synthetic data altogether), which are then used to train other models via a cheaper—but imperfect—supervision. This two-stage paradigm appears in knowledge distillation (Hinton et al., 2015; Panigrahi

[*]Equal contribution  [1]Institute of Science and Technology Austria (ISTA) [2]Department of Applied and Computational Mathematics, Yale University [3]Department of Statistics and Data Science, Yale University. Correspondence to: Diyuan Wu <diyuan.wu@ist.ac.at>.

*Proceedings of the 43$^{rd}$ International Conference on Machine Learning*, Seoul, South Korea. PMLR 306, 2026. Copyright 2026 by the author(s).

et al., 2025), self-training (Xie et al., 2020) and, more recently, in proposals where weaker agents supervise stronger ones (Burns et al., 2023). Weak-to-strong generalization (W2SG) formalizes a particularly striking version of such a proposal: a strong student (a pretrained GPT-4 model in the experiments by Burns et al. (2023)) is fine-tuned on labels produced by a weak teacher (a pretrained GPT-2 model) and, despite the labels being imperfect, the student outperforms the teacher. This motivates the central question of *how much a strong student can improve upon its weak teacher* and, in particular, whether this improvement can happen at the level of the exponent in the *scaling law* (Hestness et al., 2017; Kaplan et al., 2020; Hoffmann et al., 2022).

On the empirical side, distillation scaling laws have characterized how student performance depends on compute allocation between teacher and student (Busbridge et al., 2025), and related questions have been explored for scalable oversight (Engels et al., 2025). On the theoretical side, a rich line of work has established scaling laws for linear regression (Maloney et al., 2022; Bahri et al., 2024; Lin et al., 2024; 2025; Atanasov et al., 2024) and simple neural networks (Paquette et al., 2024; Bordelon et al., 2024; Ferbach et al., 2025; Ren et al., 2025; Defilippis et al., 2024). Several recent papers have also given theoretical insights into W2SG (Charikar et al., 2024; Lang et al., 2024; Somerstep et al., 2025; Yao et al., 2025; Moniri & Hassani, 2025; Xue et al., 2025), identifying mechanisms that allow the strong student to outperform the weak teacher, such as early stopping (Medvedev et al., 2025), feature discrepancy (Dong et al., 2025), or benign overfitting (Wu & Sahai, 2024). However, the only existing result relating W2SG to scaling laws is *negative*: Ildiz et al. (2025) show that, for ridgeless linear regression, training on surrogate teacher labels can improve performance, but it is unable to improve the scaling law.

Our paper demonstrates that the interplay of regularization and over-parameterization drastically changes the picture: we establish that *improved scaling laws* are already possible in a tractable non-linear model, i.e., *random feature ridge regression (RFRR)*. More precisely, we consider a setting where both teacher and student are trained via RFRR: first, the teacher is fit on $n_t$ labeled samples using $p_t$ random features and ridge parameter $\lambda_t$; then, the student draws $n_s$

fresh unlabeled inputs, and it is trained purely on teacher-generated labels using $p_s$ random features and ridge parameter $\lambda_s$. Our contributions are summarized as follows:

- We show a dimension-free *deterministic equivalent*, together with non-asymptotic approximation guarantees, for the excess test error incurred by the student trained with labels obtained from the teacher (Theorem 2).

- Then, by leveraging the deterministic equivalent, we obtain *scaling laws* for the student error under *source and capacity conditions*, i.e., power-law decay of target coefficients and covariance spectrum (Theorem 4).

- Finally, we compare the exponent in the scaling law of the student with the exponent in the scaling law of the teacher (the latter was derived in earlier work by Defilippis et al. (2024)). This allows us to identify regimes in which the error of the student decays faster than that of the teacher (Corollaries 2 and 3).

Overall, our analysis sheds light into the enabling mechanisms for weak-to-strong generalization, underscoring the importance of regularization and number of features in the model: *(i)* if the teacher is optimally tuned, then the student cannot improve the scaling law; *(ii)* if the teacher is variance-dominated (and not optimal), then the student can always improve the scaling law by properly selecting regularization and model size; *(iii)* if the teacher is bias-dominated (and not optimal), then there are still settings in which the student improves the scaling law; and *(iv)* the student may attain the minimax-optimal decay rate, no matter the scaling law of the teacher—even in scenarios where the error of the teacher does not decay to 0 as the sample size grows.

## 2. Related work

**Weak-to-strong generalization (W2SG).** Several recent theoretical works have analyzed mechanisms leading to W2SG. Charikar et al. (2024) focus on the misfit error incurred by the strong model on labels generated by the weaker model, and Yao et al. (2025) extend these findings beyond the square loss to output distribution divergence. Lang et al. (2024) suggest that W2SG occurs when the strong model cannot reproduce the teacher's mistakes without sacrificing performance. A link to benign overfitting is provided in Wu & Sahai (2024), and Xue et al. (2025) show that W2SG occurs when the parts of the weak model's mistakes representable by the strong model are small. In high-dimensional ridgeless linear regression, Ildiz et al. (2025) show that training on teacher labels can be better than training on clean labels, but it does not improve the exponent in the scaling law. In contrast, we show that by allowing over-parameterization and ridge regularization, the scaling law can improve. Ridge regularization is considered in Moniri & Hassani (2025), who show that the student can

compensate for the teacher's under-regularization. Closer to our setting, Medvedev et al. (2025) focus on the random feature model, but train the teacher on the population and consider students with infinitely many features, so that W2SG can only emerge as a consequence of early-stopping. W2SG arises in the variance-dominated regime in Dong et al. (2025), due to the fact that fine-tuning occurs in intrinsically low-dimensional spaces. In contrast, our work unveils that W2SG is possible both in variance-dominated and bias-dominated regimes. Overall, we highlight that the key distinguishing factor of this paper w.r.t. earlier work is the improvement in scaling laws and, at the technical level, the fact that we provide a dimension-free deterministic equivalent for the student error.

**Scaling laws.** Bahri et al. (2024) propose a theory that explains the emergence of neural scaling laws in terms of variance-limited and resolution-limited scaling regimes. Compute/data/parameter tradeoffs are characterized by Lin et al. (2024); Paquette et al. (2024). Lin et al. (2025); Ferbach et al. (2025) focus on the role of data re-use and on the choice of the optimizer, respectively; and shallow neural networks are considered in (Defilippis et al., 2024; Atanasov et al., 2024; Ren et al., 2025). Most closely related to our work are scaling law results for kernel and random feature methods under source/capacity conditions (Caponnetto & De Vito, 2007; Rudi & Rosasco, 2017; Cui et al., 2023; Defilippis et al., 2024). Our results contribute to this landscape by establishing the excess test error of a student trained on imperfect labels generated from a teacher, and by comparing student and teacher scaling laws.

**Random features and deterministic equivalents.** Introduced by Rahimi & Recht (2007); Balcan et al. (2006), random feature models have been used to understand various phenomena in deep learning, including double descent (Mei & Montanari, 2022), feature learning (Ba et al., 2022; Moniri et al., 2024), robustness under adversarial attacks (Dohmatob & Bietti, 2022; Bombari et al., 2023; Hassani & Javanmard, 2024), spurious features and correlations (Bombari & Mondelli, 2024; 2025), and distribution shift (Tripuraneni et al., 2021; Lee et al., 2023). Dimension-free deterministic equivalents for ridge regression and random feature regression have recently enabled refined, non-asymptotic predictions of test error and scaling laws (Cheng & Montanari, 2024; Defilippis et al., 2024), and analogous deterministic-equivalent techniques have been developed for kernel ridge regression (Misiakiewicz & Saeed, 2024). The deterministic equivalent we prove in this paper extends these tools to a two-stage learning pipeline, which introduces additional dependencies and cross-terms absent in the one-stage setting considered in earlier work.

## 3. Setting

Let $(\mathcal{X}, \mu_x)$ be an input probability space and $(\mathcal{W}, \mu_w)$ a weight probability space. We denote by $L^2(\mathcal{X}, \mu_x)$ and $L^2(\mathcal{W}, \mu_w)$ the associated Hilbert spaces of square-integrable functions.

**Random Feature (RF) model.** Let $\varphi : \mathcal{X} \times \mathcal{W} \to \mathbb{R}$ be a feature map with $\varphi \in L^2(\mathcal{X} \times \mathcal{W}, \mu_x \otimes \mu_w)$. The associated *random feature model* is

$$\hat{f}(\boldsymbol{x}; \boldsymbol{a}) = \frac{1}{\sqrt{p}} \sum_{j=1}^{p} a_j \varphi(\boldsymbol{x}; \boldsymbol{w}_j), \tag{1}$$

where $\boldsymbol{a} \in \mathbb{R}^p$ are trainable coefficients and $\boldsymbol{w}_j \sim_{\text{i.i.d.}} \mu_w$ are fixed random weights. Viewing $\varphi$ as a compact linear operator from $L^2(\mathcal{X}, \mu_x)$ to $L^2(\mathcal{W}, \mu_w)$, it admits a diagonalization (Mei et al., 2022; Defilippis et al., 2024):

$$\varphi(\boldsymbol{x}; \boldsymbol{w}) = \sum_{k=1}^{d} \xi_k \psi_k(\boldsymbol{w}) \phi_k(\boldsymbol{x}),$$

where $|\xi_1| \geq |\xi_2| \geq \cdots$ are the eigenvalues, and $\{\psi_k\}_{k \geq 1}$ and $\{\phi_k\}_{k \geq 1}$ are orthonormal bases of $L^2(\mathcal{W}, \mu_w)$ and $L^2(\mathcal{X}, \mu_x)$, respectively. Define the (potentially infinite-dimensional $d = \infty$) diagonal operator

$$\boldsymbol{\Sigma} = \text{diag}(\xi_1^2, \xi_2^2, \ldots), \qquad \text{with} \qquad \text{Tr}(\boldsymbol{\Sigma}) < \infty,$$

which follows from $\varphi \in L^2(\mathcal{X} \times \mathcal{W}, \mu_x \otimes \mu_w)$. It is convenient to rewrite the feature map in inner-product form

$$\varphi(\boldsymbol{x}; \boldsymbol{w}) = \langle \boldsymbol{f}, \boldsymbol{g} \rangle, \tag{2}$$

where $\boldsymbol{f} := \boldsymbol{f}(\boldsymbol{w})$ and $\boldsymbol{g} := \boldsymbol{g}(\boldsymbol{x})$ with

$$\boldsymbol{f}(\boldsymbol{w}) = (\xi_k \psi_k(\boldsymbol{w}))_{k \geq 1}, \quad \boldsymbol{g}(\boldsymbol{x}) = (\phi_k(\boldsymbol{x}))_{k \geq 1}.$$

Under $\boldsymbol{w} \sim \mu_w$ and $\boldsymbol{x} \sim \mu_x$, the random vectors $\boldsymbol{f}$ and $\tilde{\boldsymbol{g}} := \boldsymbol{\Sigma}^{1/2} \boldsymbol{g}$ are elements of $\ell_2$ (the space of squared-summable sequences), with covariance $\boldsymbol{\Sigma}$. The RF model (1) becomes

$$\hat{f}(\boldsymbol{x}; \boldsymbol{a}) = \frac{1}{\sqrt{p}} \boldsymbol{a}^\mathsf{T} \boldsymbol{F} \boldsymbol{g},$$

where $\boldsymbol{F} := [\boldsymbol{f}_1, \ldots, \boldsymbol{f}_p]^\mathsf{T} \in \mathbb{R}^{p \times d}$ with $\boldsymbol{f}_j := \boldsymbol{f}(\boldsymbol{w}_j)$.

For our formal results, we follow Cheng & Montanari (2024); Misiakiewicz & Saeed (2024); Defilippis et al. (2024) and impose the following concentration property on the eigenfunctions of the feature map:

**Assumption 1** (Concentration of the eigenfunctions). *There exists a constant $\mathsf{C}_x > 0$ such that for any p.s.d. operator $\boldsymbol{A} \in \mathbb{R}^{d \times d}$ with $\text{Tr}(\boldsymbol{A}\boldsymbol{\Sigma}) < \infty$, we have*

$$\mathbb{P}\big(|\boldsymbol{z}^\mathsf{T} \boldsymbol{A} \boldsymbol{z} - \text{Tr}(\boldsymbol{\Sigma}\boldsymbol{A})| \geq t \|\boldsymbol{\Sigma}^{1/2} \boldsymbol{A} \boldsymbol{\Sigma}^{1/2}\|_F\big) \leq \mathsf{C}_x e^{-\frac{t}{\mathsf{C}_x}},$$

*where $\boldsymbol{z}$ is either $\boldsymbol{f}(\boldsymbol{w})$ or $\tilde{\boldsymbol{g}}(\boldsymbol{x})$.*

This condition subsumes several standard assumptions in the literature, including independent sub-Gaussian coordinates, log-Sobolev concentration, and convex Lipschitz concentration (Cheng & Montanari, 2024). Moreover, Defilippis et al. (2024) provide numerical evidence that the deterministic equivalent predictions remain accurate well beyond these conditions, including for non-linear feature maps and real-data settings, and Misiakiewicz & Saeed (2024) develop relaxations of Assumption 1 for kernel regression. We leave similar extensions to RF models, and to the W2SG setting, to future work.

**Random Feature Ridge Regression (RFRR).** Let $(y_i, \boldsymbol{x}_i)_{i \leq n}$ be i.i.d. samples with $\boldsymbol{x}_i \sim \mu_x$ and

$$y_i = f_*(\boldsymbol{x}_i) + \varepsilon_i, \quad f_* \in L^2(\mathcal{X}, \mu_x), \quad \varepsilon_i \sim \mathcal{N}(0, \tau_\varepsilon^2). \tag{3}$$

The target function decomposes in the eigenbasis as (in $L^2$)

$$f_*(\boldsymbol{x}) = \sum_{k=1}^{\infty} \beta_{*,k} \phi_k(\boldsymbol{x}) = \langle \boldsymbol{\beta}_*, \boldsymbol{g} \rangle,$$

where $\|\boldsymbol{\beta}_*\|_2 = \|f_*\|_{L^2} < \infty$ by assumption. We fit this training data using ridge regression over the class of RF models (1):

$$\hat{\boldsymbol{a}} = \arg\min_{\boldsymbol{a} \in \mathbb{R}^p} \left\{ \sum_{i=1}^{n} \big(y_i - \hat{f}(\boldsymbol{x}_i; \boldsymbol{a})\big)^2 + \lambda \|\boldsymbol{a}\|_2^2 \right\}$$
$$= (\boldsymbol{Z}^\mathsf{T} \boldsymbol{Z} + \lambda \boldsymbol{I}_p)^{-1} \boldsymbol{Z}^\mathsf{T} \boldsymbol{y}, \tag{4}$$

where $\lambda \geq 0$ is the regularization parameter and we denote the label vector $\boldsymbol{y} = (y_i)_{i \in [n]}$ and the feature matrix $\boldsymbol{Z} = (p^{-1/2} \varphi(\boldsymbol{x}_i; \boldsymbol{w}_j))_{i \in [n], j \in [p]} \in \mathbb{R}^{n \times p}$. The excess test error (or population risk) of RFRR is

$$\mathcal{R}_{\text{test}}(\hat{f}, f_*) := \mathbb{E}_{\boldsymbol{x} \sim \mu_x}\big[\big(f_*(\boldsymbol{x}) - \hat{f}(\boldsymbol{x}; \hat{\boldsymbol{a}})\big)^2\big]$$
$$= \|\boldsymbol{\beta}_* - p^{-1/2} \boldsymbol{F}^\mathsf{T} \hat{\boldsymbol{a}}\|_2^2. \tag{5}$$

**Weak-to-Strong training.** In this paper, we study a two-stage learning scheme consisting of a *teacher* model trained on the target data followed by a *student* model trained on data labeled by the teacher model.

**Teacher.** We observe $n_\mathsf{t}$ i.i.d. samples $(y_i^{(\mathsf{t})}, \boldsymbol{x}_i^{(\mathsf{t})})_{i \leq n}$ with $\boldsymbol{x}_i^{(\mathsf{t})} \sim \mu_x$ and

$$y_i^{(\mathsf{t})} = \langle \boldsymbol{\beta}_*, \boldsymbol{g}_i^{(\mathsf{t})} \rangle + \varepsilon_i, \quad \varepsilon_i \sim \mathcal{N}(0, \tau_\mathsf{t}^2),$$

where we denote $\boldsymbol{g}_i^{(\mathsf{t})} = \boldsymbol{g}(\boldsymbol{x}_i^{(\mathsf{t})})$. We fit this data with a *teacher* model with $p_\mathsf{t}$ i.i.d. random features $\boldsymbol{F}_\mathsf{t} = [\boldsymbol{f}_1^{(\mathsf{t})}, \ldots, \boldsymbol{f}_{p_\mathsf{t}}^{(\mathsf{t})}]^\mathsf{T} \in \mathbb{R}^{p_\mathsf{t} \times d}$ and regularization parameter $\lambda_\mathsf{t} \geq 0$. The teacher model is then given by

$$\hat{f}_\mathsf{t}(\boldsymbol{x}) = \frac{1}{\sqrt{p_\mathsf{t}}} \langle \hat{\boldsymbol{a}}_\mathsf{t}, \boldsymbol{F}_\mathsf{t} \boldsymbol{g} \rangle,$$

where $\hat{a}_t$ is the RFRR solution (4):

$$\hat{a}_t = (Z_t^\top Z_t + \lambda_t I_{p_t})^{-1} Z_t^\top y_t,$$

with $y_t = (y_i^{(t)})_{i \leq n}$ and $Z_t = G_t F_t^\top / \sqrt{p_t}$, where $G_t := [g_1^{(t)}, \ldots, g_{n_t}^{(t)}]^\top \in \mathbb{R}^{n_t \times d}$. The teacher excess test error is

$$\mathcal{R}_{\text{test}}(\hat{f}_t, f_*) = \|\beta_* - p_t^{-1/2} F_t^\top \hat{a}_t\|_2^2. \qquad (6)$$

**Student.** We draw $n_s$ fresh inputs $(x_i^{(s)})_{i \leq n_s} \sim_{\text{i.i.d.}} \mu_x$ and label them using the teacher model:

$$y_i^{(s)} = \hat{f}_t(x_i^{(s)}) = \frac{1}{\sqrt{p_t}} \langle g_i^{(s)}, F_t^\top \hat{a}_t \rangle.$$

Define the teacher coefficient vector $\beta_t := p_t^{-1/2} F_t^\top \hat{a}_t$. We fit this data using a new *student* model with $p_s$ i.i.d. random features $F_s = [f_1^{(s)}, \ldots, f_{p_s}^{(s)}]^\top$ and regularization parameter $\lambda_s \geq 0$:

$$\hat{f}_s(x) = \frac{1}{\sqrt{p_s}} \langle \hat{a}_s, F_s g \rangle,$$

where $\hat{a}_s$ is the RFRR solution (4):

$$\hat{a}_s = (Z_s^\top Z_s + \lambda_s I_{p_s})^{-1} Z_s^\top y_s,$$

with $Z_s = G_s F_s^\top / \sqrt{p_s}$. The student excess test error is evaluated against the *original target*:

$$\mathcal{R}_{\text{test}}(\hat{f}_s, f_*) = \|\beta_* - p_s^{-1/2} F_s^\top \hat{a}_s\|_2^2. \qquad (7)$$

Our main technical contribution is to derive a *deterministic equivalent* for the student test error and to prove a non-asymptotic approximation guarantee. We now introduce a few quantities required to state these results.

**Fixed points.** Consider integers $(n, p) \in \mathbb{N}^2$ and a regularization parameter $\lambda > 0$. We define $(\mu_1, \mu_2) \in \mathbb{R}_{>0}^2$ as the unique positive solution of the following self-consistency equations:

$$1 + \frac{n}{p} - \sqrt{\left(1 - \frac{n}{p}\right)^2 + 4\frac{\lambda}{p\mu_2}} = \frac{2}{p}\text{Tr}(\Sigma(\Sigma + \mu_2)^{-1}),$$

$$1 - \frac{n}{p} + \sqrt{\left(1 - \frac{n}{p}\right)^2 + 4\frac{\lambda}{p\mu_2}} = 2\frac{\mu_1}{\mu_2}. \qquad (8)$$

We will denote by

$$(\mu_{t,1}, \mu_{t,2}) \quad \text{and} \quad (\mu_{s,1}, \mu_{s,2}) \qquad (9)$$

the teacher and student fixed points associated to parameters $(n_t, p_t, \lambda_t)$ and $(n_s, p_s, \lambda_s)$, respectively. The quantities $(\mu_{t,1}, \mu_{t,2})$ and $(\mu_{s,1}, \mu_{s,2})$ can be viewed as effective regularization parameters that appear in the deterministic equivalent for the random feature model. Informally, we have that $(Z^\top Z + \lambda)^{-1} \approx \frac{\mu_1}{\lambda}(\frac{1}{p}F^\top F + \mu_{t,1})^{-1} \approx \frac{\mu_2}{\lambda}(\Sigma + \mu_{t,2})^{-1}$. Note that this intuition holds for linear functionals and there will be correction terms for functionals of higher order, see (36)-(37) in Theorem 5 and (40)-(41) in Theorem 6.

**Functionals.** For a p.s.d. operator $A \in \mathbb{R}^{d \times d}$ and fixed points $(\mu_1, \mu_2) \in \mathbb{R}_{>0}^2$, define the functionals

$$\Upsilon_{n,p}(A; \mu_1, \mu_2) = \frac{1}{n}\text{Tr}(A\Sigma(\Sigma + \mu_2)^{-1})$$

$$- \frac{\mu_1}{n}\frac{\text{Tr}(A\Sigma(\Sigma + \mu_2)^{-2})}{1 - \frac{1}{p}\text{Tr}(\Sigma^2(\Sigma + \mu_2)^{-2})}, \qquad (10)$$

$$\chi_{n,p}(A; \mu_2) = \frac{\text{Tr}(A\Sigma(\Sigma + \mu_2)^{-2})}{p - \text{Tr}(\Sigma^2(\Sigma + \mu_2)^{-2})}.$$

Note that we always have $\Upsilon_{n,p}(A; \mu_1, \mu_2) \geq 0$. For $A = I$, we simply write $\Upsilon_{n,p}(\mu_1, \mu_2) := \Upsilon_{n,p}(I; \mu_1, \mu_2)$ and $\chi_{n,p}(\mu_2) := \chi_{n,p}(I; \mu_2)$.

# 4. Deterministic equivalents of the test error

This section provides a sharp characterization of the excess test errors for both teacher and student models in the weak-to-strong framework introduced in Section 3. A deterministic equivalent for the teacher test error (6) was already derived in Defilippis et al. (2024); we briefly recall this result for completeness.

Our main technical contribution is a dimension-free *deterministic equivalent* for the student test error (7). This characterization is an explicit analytical expression that only depends on the problem parameters and the population eigenvalues of the feature map. In particular, we establish non-asymptotic approximation guarantees between the (random) student test error and its deterministic equivalent.

## 4.1. Test error of the teacher

We begin by recalling the dimension-free characterization of the teacher test error (6) from Defilippis et al. (2024).

**Definition 1** (Teacher deterministic equivalent). *Let $(\mu_{t,1}, \mu_{t,2}) \in \mathbb{R}_{>0}^2$ be the unique positive solution of the self-consistency equations (8) with parameters $(n_t, p_t, \lambda_t)$. The deterministic equivalent of the teacher test error (6) is*

$$R_t := \langle \beta_*, \Lambda_t \beta_* \rangle + \tau_t^2 \frac{\Upsilon_t}{1 - \Upsilon_t}, \qquad (11)$$

$$\Lambda_t := \frac{\mu_{t,2}^2}{1 - \Upsilon_t}\left[(\Sigma + \mu_{t,2})^{-2} + \chi_t \Sigma(\Sigma + \mu_{t,2})^{-2}\right],$$

*where $\Upsilon_t := \Upsilon_{n_t,p_t}(\mu_{t,1}, \mu_{t,2})$ and $\chi_t := \chi_{n_t,p_t}(\mu_{t,2})$ are defined in (10). The term $B_t := \langle \beta_*, \Lambda_t \beta_* \rangle$ corresponds to the* bias *of $R_t$ and the term $V_t := \tau_t^2 \frac{\Upsilon_t}{1 - \Upsilon_t}$ to its* variance.

Defilippis et al. (2024) establish a non-asymptotic approximation guarantee between the test error (6) and its deterministic equivalent (11), under the technical condition below.

**Assumption 2.** *There exists a constant $C_* > 0$ such that*

$$\frac{\text{Tr}(\Sigma(\Sigma + \mu_{t,2})^{-1})}{\text{Tr}(\Sigma^2(\Sigma + \mu_{t,2})^{-2})} \vee \frac{\langle \beta_*, (\Sigma + \mu_{t,2})^{-1}\beta_* \rangle}{\mu_{t,2}\langle \beta_*, (\Sigma + \mu_{t,2})^{-2}\beta_* \rangle} \leq C_*.$$

In the following, we use the shorthand notations $(a \vee b) := \max(a, b)$ and $(a \wedge b) := \min(a, b)$. Assumption 2 is mild and satisfied, for instance, under the source–capacity conditions introduced in Section 5. The approximation bounds depend on the feature spectrum $\boldsymbol{\Sigma}$ and regularization $\lambda > 0$ through the following quantities:

$$
\begin{aligned}
r_{\boldsymbol{\Sigma}}(k) &:= \frac{\sum_{j \geq k} \xi_j^2}{\xi_k^2}, \\
M_{\boldsymbol{\Sigma}}(k) &:= 1 + \frac{r_{\boldsymbol{\Sigma}}(k_*) \vee k}{k} \log\left(r_{\boldsymbol{\Sigma}}(k_*) \vee k\right), \\
\rho_\lambda(p) &:= 1 + \frac{p \cdot \xi_{p_*}^2}{\lambda} M_{\boldsymbol{\Sigma}}(p), \\
\tilde{\rho}_\lambda(n, p) &:= 1 + \left\{ \frac{n \xi_{n_*}^2}{\lambda} + \frac{n}{p} \rho_\lambda(p) \right\} M_{\boldsymbol{\Sigma}}(n) \mathbb{1}_{n_* \leq p},
\end{aligned}
\tag{12}
$$

where $k_* := \lfloor \eta_* k \rfloor$, $n_* := \lfloor \eta_* n \rfloor$, and $p_* := \lfloor \eta_* p \rfloor$ for some $\eta_* \in (0, 1/2)$ specified in the theorem. Typically, e.g., for regularly varying spectrum, we will have $\rho_\lambda(p) \lesssim (\log p)^C / \lambda$ and $\tilde{\rho}_\lambda(n, p) \lesssim (\log p \wedge n)^C / \lambda$.

For convenience, we introduce the shorthand notations:

$$
\begin{aligned}
\rho_{\mathsf{t}} &:= \rho_{p_{\mathsf{t}} \mu_{\mathsf{t},1}}(p_{\mathsf{t}}), & \tilde{\rho}_{\mathsf{t}} &:= \tilde{\rho}_{p_{\mathsf{t}} \lambda_{\mathsf{t}} / n_{\mathsf{t}}}(n_{\mathsf{t}}, p_{\mathsf{t}}), \\
\rho_{\mathsf{s}} &:= \rho_{p_{\mathsf{s}} \mu_{\mathsf{s},1}}(p_{\mathsf{s}}), & \tilde{\rho}_{\mathsf{s}} &:= \tilde{\rho}_{p_{\mathsf{s}} \lambda_{\mathsf{s}} / n_{\mathsf{s}}}(n_{\mathsf{s}}, p_{\mathsf{s}}).
\end{aligned}
\tag{13}
$$

We are now ready to state the deterministic equivalent approximation theorem for the teacher test error (6).

**Theorem 1.** *(Defilippis et al., 2024, Theorem 3.3) There exist absolute constants $C_0, C_1 > 0$ such that the following holds. Under Assumptions 1 and 2, for any $D, K > 0$, there exist constants $\eta_* \in (0, 1/2)$ and $C_* > 0$ depending only on $D, K$ and the constants in the assumptions, such that for all $n_{\mathsf{t}}, p_{\mathsf{t}} \geq C_*$, $\lambda_{\mathsf{t}} > 0$, and target $\|\boldsymbol{\beta}_*\|_2 < \infty$, if*

$$
\lambda_{\mathsf{t}} \geq \max(n_{\mathsf{t}}^{-K}, n_{\mathsf{t}} p_{\mathsf{t}}^{-1-K}), \quad \tilde{\rho}_{\mathsf{t}}^{C_0} \leq n_{\mathsf{t}}, \quad \rho_{\mathsf{t}}^{C_0} \leq p_{\mathsf{t}},
$$

*then with probability at least $1 - \min(n_{\mathsf{t}}, p_{\mathsf{t}})^{-D}$,*

$$
\left| \mathcal{R}_{\text{test}}(\hat{f}_{\mathsf{t}}, f_*) - \mathsf{R}_{\mathsf{t}} \right| \leq C_* \mathcal{E}_{n_{\mathsf{t}}, p_{\mathsf{t}}} \mathsf{R}_{\mathsf{t}},
$$

*where the approximation rate is given by*

$$
\mathcal{E}_{n_{\mathsf{t}}, p_{\mathsf{t}}} := \frac{(\tilde{\rho}_{\mathsf{t}} \log n_{\mathsf{t}})^{C_1}}{\sqrt{n_{\mathsf{t}}}} + \frac{(\tilde{\rho}_{\mathsf{t}} \rho_{\mathsf{t}} \log p_{\mathsf{t}})^{C_1}}{\sqrt{p_{\mathsf{t}}}}.
$$

This theorem is non-asymptotic, pointwise in the target function (no minimax or randomization), and dimension-free, covering in particular the infinite-dimensional regime $d = \infty$. Furthermore, the approximation is *multiplicative*, implying that the deterministic equivalent remains accurate even when the test error itself vanishes polynomially in $(n_{\mathsf{t}}, p_{\mathsf{t}})$, provided $\mathcal{E}_{n_{\mathsf{t}}, p_{\mathsf{t}}}$ is sufficiently small. In particular, $\mathcal{E}_{n_{\mathsf{t}}, p_{\mathsf{t}}} = o(1)$ under the source–capacity conditions considered in Section 5 for $\gamma_{\lambda_{\mathsf{t}}}$ not too large, including the optimal regularization. See Defilippis et al. (2024, Remark 4.1) for a discussion.

### 4.2. Test error of the student

We now turn to the student test error (7) and introduce an analogous dimension-free deterministic equivalent.

**Definition 2** (Student deterministic equivalent). *Let $(\mu_{\mathsf{t},1}, \mu_{\mathsf{t},2}) \in \mathbb{R}_{>0}^2$ and $(\mu_{\mathsf{s},1}, \mu_{\mathsf{s},2}) \in \mathbb{R}_{>0}^2$ be the unique positive solutions of the self-consistency equations (8) with parameters $(n_{\mathsf{t}}, p_{\mathsf{t}}, \lambda_{\mathsf{t}})$ and $(n_{\mathsf{s}}, p_{\mathsf{s}}, \lambda_{\mathsf{s}})$ respectively. The deterministic equivalent of the student test error is*

$$
\mathsf{R}_{\mathsf{s}} := \langle \boldsymbol{\beta}_*, \boldsymbol{\Lambda} \boldsymbol{\beta}_* \rangle + \tau_{\mathsf{t}}^2 \frac{\Upsilon_{\mathsf{t}}(\boldsymbol{\Lambda}_0)}{1 - \Upsilon_{\mathsf{t}}},
\tag{14}
$$

*where the p.s.d. operators $\boldsymbol{\Lambda}$ and $\boldsymbol{\Lambda}_0$ are defined by*

$$
\begin{aligned}
\boldsymbol{\Lambda} &:= \boldsymbol{I} - 2\boldsymbol{\Sigma}^2(\boldsymbol{\Sigma} + \mu_{\mathsf{s},2})^{-1}(\boldsymbol{\Sigma} + \mu_{\mathsf{t},2})^{-1} \\
&\quad + \boldsymbol{\Lambda}_0 \boldsymbol{\Sigma}^2(\boldsymbol{\Sigma} + \mu_{\mathsf{t},2})^{-2} + \frac{\Upsilon_{\mathsf{t}}(\boldsymbol{\Lambda}_0)}{1 - \Upsilon_{\mathsf{t}}} \mu_{\mathsf{t},2}^2(\boldsymbol{\Sigma} + \mu_{\mathsf{t},2})^{-2} \\
&\quad + \left[ \frac{\Upsilon_{\mathsf{t}}(\boldsymbol{\Lambda}_0)}{1 - \Upsilon_{\mathsf{t}}} \chi_{\mathsf{t}} + \chi_{\mathsf{t}}(\boldsymbol{\Lambda}_0) \right] \mu_{\mathsf{t},2}^2 \boldsymbol{\Sigma}(\boldsymbol{\Sigma} + \mu_{\mathsf{t},2})^{-2}, \\
\boldsymbol{\Lambda}_0 &:= \boldsymbol{\Sigma}^2(\boldsymbol{\Sigma} + \mu_{\mathsf{s},2})^{-2} + \frac{\Upsilon_{\mathsf{s}}}{1 - \Upsilon_{\mathsf{s}}} \mu_{\mathsf{s},2}^2(\boldsymbol{\Sigma} + \mu_{\mathsf{s},2})^{-2} \\
&\quad + \frac{\chi_{\mathsf{s}}}{1 - \Upsilon_{\mathsf{s}}} \mu_{\mathsf{s},2}^2 \boldsymbol{\Sigma}(\boldsymbol{\Sigma} + \mu_{\mathsf{s},2})^{-2},
\end{aligned}
$$

*and we use the shorthand notations*

$$
\begin{aligned}
\Upsilon_s &:= \Upsilon_{n_{\mathsf{s}}, p_{\mathsf{s}}}(\mu_{\mathsf{s},1}, \mu_{\mathsf{s},2}), & \chi_{\mathsf{s}} &:= \chi_{n_{\mathsf{s}}, p_{\mathsf{s}}}(\mu_{\mathsf{s},2}), \\
\Upsilon_{\mathsf{t}}(\boldsymbol{\Lambda}_0) &:= \Upsilon_{n_{\mathsf{t}}, p_{\mathsf{t}}}(\boldsymbol{\Lambda}_0; \mu_{\mathsf{t},1}, \mu_{\mathsf{t},2}), & \Upsilon_{\mathsf{t}} &:= \Upsilon_{\mathsf{t}}(\boldsymbol{I}), \\
\chi_{\mathsf{t}}(\boldsymbol{\Lambda}_0) &:= \chi_{n_{\mathsf{t}}, p_{\mathsf{t}}}(\boldsymbol{\Lambda}_0; \mu_{\mathsf{t},2}), & \chi_{\mathsf{t}} &:= \chi_{\mathsf{t}}(\boldsymbol{I}).
\end{aligned}
$$

We note that the student receives labels from the teacher without additional noise. Thus, there is no variance term in $\mathsf{R}_{\mathsf{s}}$, and the two terms $\mathsf{B}_{bias,\mathsf{s}} := \langle \boldsymbol{\beta}_*, \boldsymbol{\Lambda} \boldsymbol{\beta}_* \rangle$, $\mathsf{B}_{var,\mathsf{s}} := \tau_{\mathsf{t}}^2 \frac{\Upsilon_{\mathsf{t}}(\boldsymbol{\Lambda}_0)}{1 - \Upsilon_{\mathsf{t}}}$ are respectively induced by $\mathsf{B}_{\mathsf{t}}$ and $\mathsf{V}_{\mathsf{t}}$. We further introduce the approximation functional

$$
\begin{aligned}
\widetilde{\mathcal{R}}_{\mathsf{s}} &:= \|\boldsymbol{\beta}_{\mathsf{t}}\|_2 \|\boldsymbol{\beta}_* - \boldsymbol{\beta}_{\mathsf{t}}\|_2 + \|\boldsymbol{\beta}_*\|_2^2 (\|\boldsymbol{\Lambda}_0\|_{\text{op}} \vee 1) \\
&\quad + \frac{1}{n_{\mathsf{t}}} \frac{\mu_{\mathsf{t},2}}{1 - \Upsilon_{\mathsf{t}}} \text{Tr}(\boldsymbol{\Lambda}_0(\boldsymbol{\Sigma} + \mu_{\mathsf{t},2})^{-1}),
\end{aligned}
\tag{15}
$$

where $\boldsymbol{\beta}_{\mathsf{t}} = p_{\mathsf{t}}^{-1/2} \boldsymbol{F}_{\mathsf{t}}^{\mathsf{T}} \hat{\boldsymbol{a}}_{\mathsf{t}}$ is the teacher coefficient vector. Although $\widetilde{\mathcal{R}}_{\mathsf{s}}$ is random through $\boldsymbol{\beta}_{\mathsf{t}}$, it is of order $O(1)$ with high probability under mild conditions.

We now establish an approximation guarantee between the student test error (7) and the deterministic equivalent (14).

**Theorem 2.** *There exist absolute constants $C_0, C_1 > 0$ such that the following holds. Under Assumption 1, for any $D, K > 0$, there exist constants $\eta_* \in (0, 1/2)$ and $C_* > 0$ depending only on $D, K$ and the constant in the assumption, such that for all $n_{\mathsf{t}}, p_{\mathsf{t}}, n_{\mathsf{s}}, p_{\mathsf{s}} \geq C_*$, $\lambda_{\mathsf{t}}, \lambda_{\mathsf{s}} > 0$, and target $\|\boldsymbol{\beta}_*\|_2 < \infty$, if*

$$
\begin{aligned}
\lambda_{\mathsf{t}} &\geq \max(n_{\mathsf{t}}^{-K}, n_{\mathsf{t}} p_{\mathsf{t}}^{-1-K}), \quad \tilde{\rho}_{\mathsf{t}}^{C_0} \leq n_{\mathsf{t}}, \quad \rho_{\mathsf{t}}^{C_0} \leq p_{\mathsf{t}}, \\
\lambda_{\mathsf{s}} &\geq \max(n_{\mathsf{s}}^{-K}, n_{\mathsf{s}} p_{\mathsf{s}}^{-1-K}), \quad \tilde{\rho}_{\mathsf{s}}^{C_0} \leq n_{\mathsf{s}}, \quad \rho_{\mathsf{s}}^{C_0} \leq p_{\mathsf{s}},
\end{aligned}
$$

*then with probability at least* $1 - \min(n_t, p_t, n_s, p_s)^{-D}$,

$$|\mathcal{R}_{\text{test}}(\hat{f}_s, f_*) - \mathsf{R}_s| \leq C_* \mathcal{E}_{n_s, p_s, n_t, p_t} \widetilde{\mathcal{R}}_s,$$

*where the approximation rate is given by*

$$\mathcal{E}_{n_s, p_s, n_t, p_t} := \mathcal{E}_{n_t, p_t} + \frac{(\tilde{\rho}_s \log n_s)^{C_1}}{\sqrt{n_s}} + \frac{(\tilde{\rho}_s \rho_s \log p_s)^{C_1}}{\sqrt{p_s}}.$$

The proof (deferred to Appendix A) proceeds in two stages: *(i)* a deterministic equivalent for the student conditional on the teacher coefficients $\boldsymbol{\beta}_t$, and *(ii)* a deterministic equivalent for $\boldsymbol{\beta}_t$ itself. These steps rely on approximation results for functionals of random features from (Misiakiewicz & Saeed, 2024; Defilippis et al., 2024), together with new bounds for asymmetric functionals (stated in Appendix B and proved in Appendix C).

**Discussion.** The key difference with Theorem 1 is that the error bound of Theorem 2 is *additive* rather than multiplicative, with potentially $\widetilde{\mathcal{R}}_s \gg \mathsf{R}_s$. This is due to the more intricate test error induced by the two-stage learning pipeline, with asymmetric terms. We conjecture that a multiplicative bound of the form

$$\left|\mathcal{R}_{\text{test}}(\hat{f}_s, f_*) - \mathsf{R}_s\right| \leq C_* \mathcal{E}_{n_s, p_s, n_t, p_t} \mathsf{R}_s$$

holds for a broad class of feature covariances $\boldsymbol{\Sigma}$ and targets $\boldsymbol{\beta}_*$. We leave this as a technically challenging problem for future work.

Nevertheless, Theorem 2 already yields a sharp characterization of the student risk whenever $\mathcal{E}_{n_s, p_s, n_t, p_t} \widetilde{\mathcal{R}}_s \ll \mathsf{R}_s$. Under mild conditions, $\widetilde{\mathcal{R}}_s = O(1)$ with high probability, and $\mathcal{E}_{n_s, p_s, n_t, p_t} = \tilde{O}\big(\min(n_t, n_s, p_t, p_s)^{-1/2}\big)$ for a broad range of feature covariances $\boldsymbol{\Sigma}$ and regularization parameters $\lambda_s, \lambda_t$. In particular, by taking[1] $r$ close to 0 and $\alpha$ close to 1, the result covers various W2SG regimes under the source–capacity conditions of Section 5, including settings where the student improves upon the teacher's scaling law and settings where the teacher risk does not even decay with the sample size while the student is minimax optimal. More broadly, we expect the deterministic equivalents to remain accurate well beyond the regimes covered by Theorem 2 and beyond the assumptions made therein. To showcase this, Figure 2 (deferred to Appendix H) shows that the deterministic equivalent in Definition 2 closely tracks the behavior of the student test error for a non-linear feature map and data coming from either a single-index target function (top plots) or from a standard dataset (MNIST, bottom plots). In

---

[1] In this regime, the minimax rate is close to 0 and one can choose the parameters in (17) such that $(p_t \mu_{t,1}, p_s \mu_{s,1}, p_t \lambda_t / n_t, p_s \lambda_s / n_s)$ appearing in the denominator of $\mathcal{E}_{n_s, p_s, n_t, p_t}$ decay at rates close to 0, slower than $\min(n_t, n_s, p_t, p_s)^{-1/2}$.

both cases, the deterministic equivalents are evaluated by estimating $(\boldsymbol{\beta}_*, \boldsymbol{\Sigma})$ from a large dataset (see Appendix C.3 in Defilippis et al. (2024)).

## 5. Scaling laws for the test error

Following prior work on scaling laws in linear and random-feature models (Maloney et al., 2022; Bahri et al., 2024; Defilippis et al., 2024; Paquette et al., 2024; Lin et al., 2024), we adopt the classical *source* and *capacity* conditions (Caponnetto & De Vito, 2007; Rudi & Rosasco, 2017). Specifically, we assume a power-law spectrum and target coefficients:

$$\boldsymbol{\Sigma}_{k,k} = k^{-\alpha}, \qquad \boldsymbol{\beta}_{*,k} = k^{-\frac{(1+2\alpha r)}{2}}, \qquad (16)$$

where $\alpha > 1$ and $r > 0$. For instance, $r > \frac{1}{2}$ corresponds to $f_*$ belonging to the RKHS of the RF kernel.

**Scaling of model widths and regularizations.** We consider the two-stage teacher–student procedure with $n_t$ ground-truth samples in the teacher stage. We parameterize the remaining quantities as power laws in $n_t$:

$$\begin{aligned} p_t &= n_t^{\gamma_{p_t}}, \quad \lambda_t = n_t^{-\gamma_{\lambda_t}}; \\ p_s &= n_t^{\gamma_{p_s}}, \quad n_s = n_t^{\gamma_{n_s}}, \quad \lambda_s = n_t^{-\gamma_{\lambda_s}}. \end{aligned} \qquad (17)$$

Using the deterministic equivalents (11) and (14), we derive sharp decay rates for the teacher and student test errors as $n_t \to \infty$ under source–capacity conditions (16) and hyperparameter scaling (17).

To state these rates, it will be convenient to introduce

$$z_s = \left(\frac{\gamma_{n_s} + \gamma_{\lambda_s}}{\alpha} \wedge \gamma_{n_s} \wedge \gamma_{p_s}\right), \quad z_t = \left(\frac{1 + \gamma_{\lambda_t}}{\alpha} \wedge 1 \wedge \gamma_{p_t}\right).$$

**Teacher scaling laws.** The scaling laws for the teacher model were derived in Defilippis et al. (2024); we restate them here for completeness.

**Theorem 3.** *(Defilippis et al., 2024, Theorem 4.1) Under the scaling assumptions* (16) *and* (17)*, the teacher deterministic equivalent* $\mathsf{R}_t = \mathsf{B}_t + \mathsf{V}_t$ *satisfies*

$$\mathsf{B}_t = \Theta(n_t^{-\gamma_{t,\mathsf{B}}}), \qquad \mathsf{V}_t = \Theta(\tau_t^2 n_t^{-\gamma_{t,\mathsf{V}}}),$$

*where*

$$\gamma_{t,\mathsf{B}} := [2\alpha(r \wedge 1)z_t] \wedge [\gamma_{p_t} + (2\alpha(r \wedge 1/2) - 1)z_t],$$
$$\gamma_{t,\mathsf{V}} := 1 - z_t.$$

*Furthermore, if* $\tau_t^2 = \Theta(1)$*, the optimal teacher exponent is*

$$\gamma_{t,*} = \max_{p_t, \lambda_t} \{\gamma_{t,\mathsf{B}} \wedge \gamma_{t,\mathsf{V}}\} = \frac{2\alpha(r \wedge 1)}{1 + 2\alpha(r \wedge 1)},$$

*and it is achieved when* $z_t = \frac{1}{1+2\alpha(r \wedge 1)}$ *and* $\gamma_{p_t} \geq 1 - \frac{2\alpha(r \wedge \frac{1}{2})}{1+2\alpha(r \wedge 1)}$.

**Student scaling laws.** We now derive the scaling laws for the student model using the deterministic equivalent of Theorem 2.

**Theorem 4** (Student scaling laws). *Under the scaling assumptions* (16) *and* (17), *the student deterministic equivalent* $R_s = B_{bias,s} + B_{var,s}$ *satisfies*

$$B_{bias,s} = \Theta(n_t^{-\gamma_{s,bias}}), \quad B_{var,s} = \Theta(\tau_t^2 n_t^{-\gamma_{s,var}}),$$

*where*

$$\gamma_{s,bias} := \begin{cases} [2\alpha(r \wedge 1)z_t] \wedge [\gamma_{p_t} + (2\alpha(r \wedge \frac{1}{2}) - 1)z_t] \\ \wedge[(2\alpha(r \wedge \frac{1}{2}) - \alpha)z_t + (\alpha - 1)z_s + \gamma_{p_s}], \text{ for } z_t \leq z_s \\ [2\alpha(r \wedge 1)z_s] \wedge [2\alpha(r \wedge \frac{1}{2})z_s - z_s + \gamma_{p_s}] \\ \wedge[2\alpha(r \wedge \frac{1}{2})z_t + \alpha(z_t - z_s) - z_s + \gamma_{p_t}] \\ \wedge[2\alpha(r \wedge \frac{1}{2})z_t - z_t + 1 - z_s + \gamma_{p_t}] \\ \wedge[2\alpha(r \wedge \frac{1}{2})z_t + \gamma_{p_t} + \gamma_{n_s} - z_s - z_t], \text{ for } z_t > z_s \end{cases}$$

$$\gamma_{s,var} := \begin{cases} 1 - z_t, \text{ for } z_t \leq z_s \\ [1 - z_s] \wedge [1 + \gamma_{n_s} - z_s - z_t], \quad \text{for } z_t > z_s \end{cases}$$

*Furthermore, if* $\tau_t^2 = \Theta(1)$, *the optimal student exponent is*

$$\gamma_{s,*} := \max_{p_t, \lambda_t, n_s, p_s, \lambda_s} \{\gamma_{s,bias} \wedge \gamma_{s,var}\} = \frac{2\alpha(r \wedge 1)}{1 + 2\alpha(r \wedge 1)},$$

*and it is achieved when* $(z_s \wedge z_t) = \frac{1}{1+2\alpha(r \wedge 1)}$ *and one of the following two conditions is satisfied:*

1. $z_t \leq z_s$ *and*

$$\gamma_{p_s} \geq (1 - \alpha)z_s + \frac{2\alpha(r \wedge 1) - 2\alpha(r \wedge \frac{1}{2}) + \alpha}{1 + 2\alpha(r \wedge 1)},$$

$$\gamma_{p_t} \geq 1 - \frac{2\alpha(r \wedge \frac{1}{2})}{1 + 2\alpha(r \wedge 1)}.$$

$$(18)$$

2. $z_t > z_s$ *and*

$$\gamma_{n_s} \geq z_t, \gamma_{p_s} \geq 1 - \frac{2\alpha(r \wedge \frac{1}{2})}{1 + 2\alpha(r \wedge 1)},$$

$$\gamma_{p_t} \geq \max\left\{ \frac{2\alpha(r \wedge 1) + \alpha + 1}{1 + 2\alpha(r \wedge 1)} - \left(\alpha + 2\alpha\left(r \wedge \frac{1}{2}\right)\right)z_t, \right.$$

$$\left. 1 + \left(1 - 2\alpha\left(r \wedge \frac{1}{2}\right)\right)z_t - \gamma_{n_s}, \left(1 - 2\alpha\left(r \wedge \frac{1}{2}\right)\right)z_t \right\}.$$

$$(19)$$

The proof of Theorem 4 is deferred to Appendix G.2.

**Stable regularization.** For any fixed $(\gamma_{p_t}, \gamma_{p_s}, \gamma_{n_s})$, we call $(\gamma_{\lambda_t}, \gamma_{\lambda_s})$ *stable* if $\gamma_{\lambda_t} \geq -1$ and $\gamma_{\lambda_s} \geq -\gamma_{n_s}$. This condition implies that $z_t, z_s \geq 0$, ensuring that student and teacher risks $R_t$ and $R_s$ remain $O(1)$ (i.e., do not diverge). Without loss of generality, we restrict our attention to stable regularizations in the following.

**Optimal and minimax rates.** For any fixed $(\alpha, r)$, the best achievable exponent for the student and the teacher are equal $\gamma_{s,*} = \gamma_{t,*}$. In particular, a student trained *only* on teacher-generated labels cannot outperform the *optimally tuned* teacher trained on ground-truth labels in terms of asymptotic scaling. It is also useful to compare to the minimax rate under source–capacity conditions, which is $n_t^{-\frac{2\alpha r}{1+2\alpha r}}$ (Caponnetto & De Vito, 2007). Thus, an optimally tuned student trained *only* on teacher-generated labels achieves the minimax rate when $r \leq 1$, same as the optimally tuned teacher trained on ground-truth labels.

### 5.1. Weak-to-Strong Generalization

We now leverage the teacher and student scaling laws in Theorems 3 and 4 to address the central question of this paper: *When can a student achieve a better scaling law than its teacher in the RFRR setting?*

**Necessary condition for weak-to-strong generalization.** Our first observation is a necessary condition for the improvement in scaling law under the source–capacity model, which immediately follows by inspecting the decay rates in Theorems 3 and 4.

**Corollary 1.** *Take any* $\alpha > 1$, $r > 0$, *and any relative scaling coefficients* $(\gamma_{p_t}, \gamma_{p_s}, \gamma_{n_s}, \gamma_{\lambda_t}, \gamma_{\lambda_s})$. *On the one hand, if* $z_t \leq z_s$, *then*

$$B_{bias,s} = \Omega(B_t), \qquad B_{var,s} = \Theta(V_t).$$

*On the other hand, if* $z_t > z_s$, *then* $B_{var,s} = o(V_t)$.

Corollary 1 implies that the improvement in scaling law is possible only if $z_t > z_s$. In fact, if $z_t \leq z_s$, then the bias/variance of the teacher induces error terms at least of the same order in the student (with $B_{bias,s}$ being potentially larger than $B_t$). Below, we discuss two mechanisms leading to weak-to-strong generalization: *variance reduction* and *bias reduction*. At a high level, $z_t > z_s$ suggests that the teacher is "richer" than the student.

**Variance reduction.** When the teacher is variance-dominated, with $V_t \gg B_t$, a student can *always* improve the scaling law of the teacher by reducing the variance term.

**Corollary 2.** *Assume* $\tau_t^2 = \Theta(1)$. *Take* $\alpha > 1$, $r > 0$, *and teacher scalings* $(\gamma_{p_t}, \gamma_{\lambda_t})$ *such that* $V_t \gg B_t$, *i.e.,*

$$z_t > \frac{1}{1 + 2\alpha(r \wedge 1)} \vee \frac{1 - \gamma_{p_t}}{2\alpha(r \wedge \frac{1}{2})}, \quad \gamma_{p_t} > \frac{1}{1 + 2\alpha(r \wedge \frac{1}{2})}.$$

$$(20)$$

*Then, the student achieves a better scaling law than its teacher (namely,* $\gamma_{s,bias} \wedge \gamma_{s,var} > \gamma_{t,B} \wedge \gamma_{t,V}$) *if and only if*

$$2\alpha(r \wedge 1/2)z_s - z_s + \gamma_{p_s} > 1 - z_t, \quad \gamma_{n_s} \wedge z_t > z_s > \frac{1 - z_t}{2\alpha(r \wedge 1)}.$$

Corollary 2 (proved in Appendix G.3) characterizes the W2SG region when the teacher is variance-dominated, in terms of $(z_t, z_s, \gamma_{p_t}, \gamma_{p_s}, \gamma_{n_s})$. In particular, by taking a student such that

$$z_s = \frac{1}{1 + 2\alpha(r \wedge 1)}, \quad \gamma_{p_s} \geq 1 - \frac{2\alpha(r \wedge \frac{1}{2})}{1 + 2\alpha(r \wedge 1)}, \quad \gamma_{n_s} \geq z_t,$$

we have that by Theorem 4, the student achieves the optimal rate $\gamma_{s,*}$, for any teacher model satisfying (19) and (20). A notable consequence is that when $0 < r < 1$, *the student can achieve the minimax exponent under source–capacity conditions even if the teacher does not* (e.g., when due to mis-regularization, $z_t$ is too large and the teacher is variance-limited). In fact, the student can achieve the minimax rate even in regimes where the teacher variance exponent satisfies $\gamma_{t,V} = 0$ (i.e. the teacher risk does not decay with $n_t$). This illustrates that W2SG improvements can be substantial in RFRR.

**Bias reduction.** Even if the teacher is not variance-dominated, i.e., $V_t \leq B_t$, it is still possible to improve the scaling law, and the corollary below (proved in Appendix G.4) characterizes the W2SG region in this setting.

**Corollary 3.** *Take $\alpha > 1$, $r > 0$, teacher scalings $(\gamma_{p_t}, \gamma_{\lambda_t})$ and $\tau_t$ such that $V_t \leq B_t$. Then, the student achieves a better scaling law than its teacher (namely, $\gamma_{s,bias} \wedge \gamma_{s,var} > \gamma_{t,B} \wedge \gamma_{t,V}$) if and only if*

$$z_s > \max\left\{ \frac{(\alpha-1)z_t + \gamma_{p_t}}{2\alpha R}, \frac{(\alpha-1)z_t + \gamma_{p_t} - \gamma_{p_s}}{\alpha - 1} \right\},$$

$$z_s < \min\left\{ z_t, 1 - (\alpha-1)z_t - \gamma_{p_t}, 1 + \gamma_{n_s} - \alpha z_t - \gamma_{p_t}, \gamma_{n_s} \right\},$$

$$\frac{\gamma_{p_t}}{1 + 2\alpha R - \alpha} < z_t < \frac{1}{\alpha - 1}\left( \frac{2\alpha R}{1 + 2\alpha R} - \gamma_{p_t} \right),$$

$$\gamma_{p_t} < 1 - \frac{\alpha}{1 + 2\alpha R},$$

$$\gamma_{p_s} > \max\left\{ \gamma_{p_t}, \alpha\big((\alpha-1)z_t + \gamma_{p_t}\big) - (\alpha - 1) \right\},$$

$$\gamma_{n_s} > \max\left\{ \frac{(\alpha-1)z_t + \gamma_{p_t}}{2\alpha R}, \alpha z_t + \gamma_{p_t} - 1 + \frac{(\alpha-1)z_t + \gamma_{p_t}}{2\alpha R} \right\},$$

$$\gamma_{p_s} + (\alpha - 1)\gamma_{n_s} >$$
$$\max\left\{ (\alpha-1)z_t + \gamma_{p_t}, (\alpha^2 - 1)z_t + \alpha\gamma_{p_t} - (\alpha - 1) \right\}, \tag{21}$$

*where $R = (r \wedge 1)$.*

We note that W2SG can only happen if the student width is larger than the teacher width $\gamma_{p_s} > \gamma_{p_t}$ (see (21)) and $r > 1/2$ (see the discussions below (80)). Notably, in this regime, the scaling law can be improved even when $\tau_t^2 = 0$, i.e., the teacher is trained with clean labels.

**Discussion.** Corollaries 2-3 above identify a set of conditions on the number of training samples, model size and

regularization allowing the student to improve upon the scaling law of the teacher. To interpret such conditions, we focus on two special cases: *(i)* the *kernel limit*, in which both teacher and student have a very large number of features, i.e., $\gamma_{p_t} = \gamma_{p_s} = \infty$, and *(ii)* the *approximation limit*, in which the student has a very large number of samples labeled by the teacher, i.e., $\gamma_{n_s} = \infty$.

In the *kernel limit*, W2SG improvements come only from variance reduction, since the upper bound on $\gamma_{p_t}$ required by Corollary 3 does not hold. Furthermore, the student improves the scaling law only of teachers that are not already optimal, due to a sub-optimal choice of the regularization. Formally, this corresponds to $z_t$ larger than its optimal value $\frac{1}{1+2\alpha(r \wedge 1)}$ (Theorem 3 shows that the optimal teacher exponent is reached when $z_t = \frac{1}{1+2\alpha(r \wedge 1)}$, and (20) requires $z_t > \frac{1}{1+2\alpha(r \wedge 1)}$). Now, for any sub-optimal teacher, W2SG holds upon choosing properly sample size and regularization of the student:

(a) If the student has enough samples ($\gamma_{n_s} > z_t$, implied when the student has more samples than the teacher), the scaling law improvement occurs for a range of regularizations ($\gamma_{\lambda_s} \in (\frac{1 - z_t}{2(r \wedge 1)} - \gamma_{n_s}, \alpha z_t - \gamma_{n_s})$).

(b) For small regularizations ($\gamma_{\lambda_s} \geq (\alpha - 1)\gamma_{n_s}$), the scaling law improvement occurs for a range of sample sizes ($\gamma_{n_s} \in (\frac{1 - z_t}{2\alpha(r \wedge 1)}, z_t)$); the improvement also occurs for large regularizations ($\gamma_{\lambda_s} < (\alpha - 1)\gamma_{n_s}$), albeit in a different range of sample sizes ($\gamma_{n_s} \in (\frac{1 - z_t}{2(r \wedge 1)} - \gamma_{\lambda_s}, \alpha z_t - \gamma_{\lambda_s})$).

In the *approximation limit*, W2SG improvements also come only from variance reduction. In fact, $\gamma_{n_s} = \infty$ implies $z_s = \gamma_{p_s}$ and, hence, the second and fifth line in (21) imply that $\gamma_{p_t} < z_t$, which is impossible. Given that the number of student samples is large, its regularization becomes irrelevant (as $z_s = \gamma_{p_s}$, $\gamma_{\lambda_s}$ does not appear in the conditions of Corollary 2). Thus, the only relevant student quantity is its number of features, which has to be chosen carefully in order to improve the scaling law. As mentioned earlier, when W2SG occurs due to a variance reduction, the student width has to be smaller than the teacher width.

**Numerical results.** We illustrate our theory via the simulations of Figure 1. We pick a Gaussian linear model, i.e., $g \sim \mathcal{N}(0, I)$, $f \sim \mathcal{N}(0, \Sigma)$, with $d = 20000$. For different choices of $(\alpha, r, \gamma_{p_t}, \gamma_{\lambda_t})$, we compare the test error of the teacher (in blue), the test error of various students (in other colors), and the minimax rate (the grey dashed line with slope $-\frac{2\alpha r}{1 + 2\alpha r}$). For each teacher and student, we plot points for the empirical test error (Equations (6) and (7)), solid lines for deterministic equivalent predictions (Theorems 1 and 2), and dotted lines for theoretical decay rates

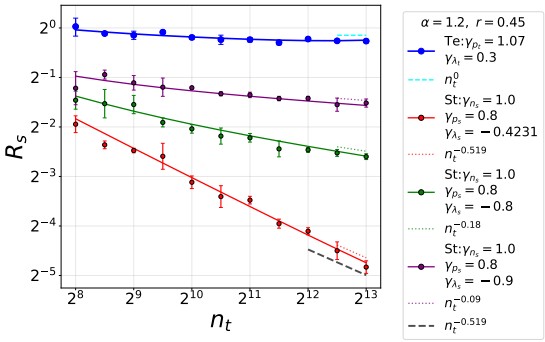

*(a)* Variance dominated: W2SG.

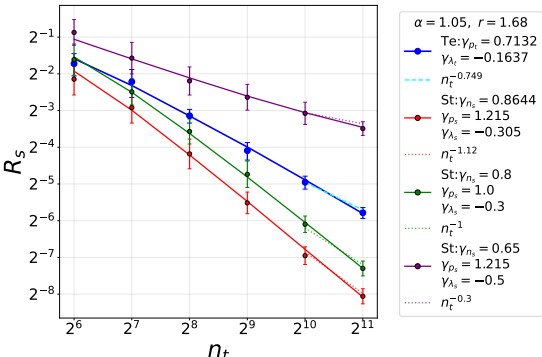

*(b)* Bias dominated: W2SG.

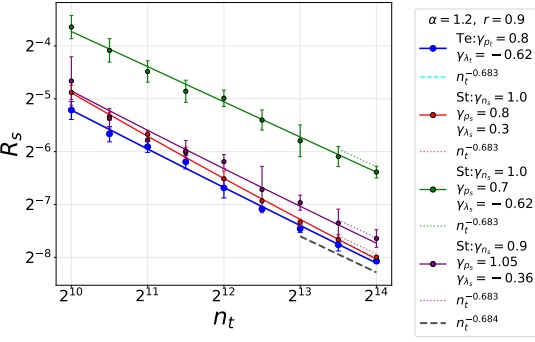

*(c)* Optimally tuned teacher: no W2SG.

*Figure 1.* Excess test errors of various students and teachers, as a function of the number of ground-truth samples $n_t$. We consider random feature ridge regression with weak-to-strong training, under the source and capacity conditions in (16) and with the hyperparameter scaling in (17). Each plot corresponds to a different choice of $(\alpha, r, \gamma_{p_t}, \gamma_{\lambda_t})$. Points are empirical experiments, solid lines are deterministic equivalent predictions (Theorems 1 and 2), dotted lines are theoretical decay rates (Theorems 3 and 4), and the grey dashed line is the minimax rate. The teacher test error is in blue, while the other colors correspond to student test errors for different choices of $(\gamma_{n_s}, \gamma_{p_s}, \gamma_{\lambda_s})$.

(Theorems 3 and 4). We run 25 independent experiments, reporting the average and the confidence interval at 1 standard deviation. We note that, in all the cases considered, the theoretical predictions are in excellent agreement with the empirical test errors. Each of the three subplots corresponds to a different scenario:

(a) *Variance-dominated regime:* The teacher test error does not decay with $n_t$ (zero decay rate), while a properly tuned student (in red) achieves the minimax rate.

(b) *Bias-dominated regime ($\tau_t^2 = 0$):* Two of the three students reduce the bias, thus improving the scaling law.

(c) *Optimally regularized teacher:* The teacher achieves the minimax rate, and no student can improve the scaling law.

## 6. Conclusion

In this paper, we show that, in random feature ridge regression, a student trained on labels generated via a teacher can attain a scaling law which is more favorable than that of the teacher itself. This demonstrates the benefit of proper regularization and over-parameterization in the context of weak-to-strong generalization, since the same claim does not hold for ridgeless linear regression (Ildiz et al., 2025). We obtain this result through the derivation of a novel deterministic equivalent for the excess test error of the student after a two-stage learning procedure. In contrast with deterministic equivalents obtained in earlier work (Defilippis et al., 2024; Misiakiewicz & Saeed, 2024), our derivation is made challenging by the necessity to handle two sources of randomness (from both the student and the teacher), and we expect the analysis carried out in this paper to be applicable to various problems in high-dimensional statistics involving multiple data sources, including distribution shift (Patil et al., 2024; Mallinar et al., 2024) and transfer learning (Yang et al., 2025; Song et al., 2024). We conclude by noting that our framework is better suited to handle ridge regression, and it does not directly capture the effect of early stopping in W2SG, despite the high-level connection between early stopping and regularization (Raskutti et al., 2014). We consider this an interesting future direction.

## Impact Statement

This paper presents work whose goal is to advance the field of Machine Learning. There are many potential societal consequences of our work, none of which we feel must be specifically highlighted here.

## Acknowledgements

DW and MM were funded in part by the Austrian Science Fund (FWF) 10.55776/COE12. For the purpose of open access, the authors have applied a CC BY public copyright license to any Author Accepted Manuscript version arising from this submission. MM was partially funded by the European Union (ERC, INF2, project number 101161364). Views and opinions expressed are however those of the author(s) only and do not necessarily reflect those of the European Union or the European Research Council Executive Agency. Neither the European Union nor the granting authority can be held responsible for them. The authors thank Yalda Shabanzadeh for pointing out a mistake in an earlier version of Appendix F.1.

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

## Notation

Recall that we denote $\boldsymbol{F}_\mathsf{t} \in \mathbb{R}^{p_\mathsf{t} \times d}$ the teacher random features with i.i.d. rows with covariance $\boldsymbol{\Sigma}$, $\boldsymbol{G}_\mathsf{t} \in \mathbb{R}^{n_\mathsf{t} \times d}$ the teacher input data with i.i.d. rows with covariance $\boldsymbol{I}_d$, $\boldsymbol{\beta}_*$ the target function coefficients, such that $y_i = \langle \boldsymbol{\beta}_*, \boldsymbol{g}_{\mathsf{t},i} \rangle + \varepsilon_i$, and $\hat{\boldsymbol{a}}_\mathsf{t}$ the solution of RFRR. We denote $\boldsymbol{y}_\mathsf{t} \in \mathbb{R}^{n_\mathsf{t}}$ the vector of labels from the true target, and $\boldsymbol{\varepsilon} \in \mathbb{R}^{n_\mathsf{t}}$ the vector of noise, such that

$$\boldsymbol{y}_\mathsf{t} = \boldsymbol{G}_\mathsf{t} \boldsymbol{\beta}_* + \boldsymbol{\varepsilon}.$$

We further define

$$\boldsymbol{\beta}_\mathsf{t} = \frac{1}{\sqrt{p_\mathsf{t}}} \boldsymbol{F}_\mathsf{t}^\top \hat{\boldsymbol{a}}_\mathsf{t}, \qquad \boldsymbol{Z}_\mathsf{t} = \frac{1}{\sqrt{p_\mathsf{t}}} \boldsymbol{G}_\mathsf{t} \boldsymbol{F}_\mathsf{t}^\top.$$

For the student model, we define the corresponding $\boldsymbol{F}_\mathsf{s}, \boldsymbol{G}_\mathsf{s}, \boldsymbol{Z}_\mathsf{s}, \hat{\boldsymbol{a}}_\mathsf{s}$ in the same way.

Recall that the student has solution

$$\hat{\boldsymbol{a}}_\mathsf{s} = (\boldsymbol{Z}_\mathsf{s}^\top \boldsymbol{Z}_\mathsf{s} + \lambda_\mathsf{s})^{-1} \boldsymbol{Z}_\mathsf{s}^\top \boldsymbol{y}_\mathsf{s},$$

where $\boldsymbol{y}_\mathsf{s}$ is the data coming from the teacher given by

$$\boldsymbol{y}_\mathsf{s} = \boldsymbol{G}_\mathsf{s} \boldsymbol{\beta}_\mathsf{t}.$$

We denote

$$\boldsymbol{\Xi}_\mathsf{s} := (\boldsymbol{Z}_\mathsf{s}^\top \boldsymbol{Z}_\mathsf{s} + \lambda_\mathsf{s})^{-1}, \qquad \widehat{\boldsymbol{\Sigma}}_\mathsf{s} := \frac{\boldsymbol{F}_\mathsf{s} \boldsymbol{F}_\mathsf{s}^\top}{p_\mathsf{s}}.$$

The test error of the student model can be characterized as

$$\mathcal{R}(f_*; \boldsymbol{G}_\mathsf{s}, \boldsymbol{F}_\mathsf{s}, \lambda, \boldsymbol{\beta}_\mathsf{t}) = \mathbb{E}_{\boldsymbol{g}} \left[ \left( \boldsymbol{g}^\top \boldsymbol{\beta}_* - \frac{1}{\sqrt{p_\mathsf{s}}} \boldsymbol{g}^\top \boldsymbol{F}_\mathsf{s}^\top \hat{\boldsymbol{a}}_\mathsf{s} \right)^2 \right] =: \mathcal{B}_\mathsf{s}(\boldsymbol{\beta}_*, \boldsymbol{\beta}_\mathsf{t}; \boldsymbol{G}_\mathsf{s}, \boldsymbol{F}_\mathsf{s}, \lambda_\mathsf{s}),$$

where we have

$$\mathcal{B}_\mathsf{s}(\boldsymbol{\beta}_*, \boldsymbol{\beta}_\mathsf{t}; \boldsymbol{G}_\mathsf{s}, \boldsymbol{F}_\mathsf{s}, \lambda_\mathsf{s}) := \| \boldsymbol{\beta}_* - p_\mathsf{s}^{-1/2} \boldsymbol{F}_\mathsf{s}^\top \boldsymbol{\Xi}_\mathsf{s} \boldsymbol{Z}_\mathsf{s}^\top \boldsymbol{G}_\mathsf{s} \boldsymbol{\beta}_\mathsf{t} \|_2^2,$$

We also define the following error scaling term:

$$\mathcal{E}_{n_\mathsf{s}, p_\mathsf{s}} := \frac{(\tilde{\rho}_\mathsf{s} \log n_\mathsf{s})^{C_1}}{\sqrt{n_\mathsf{s}}} + \frac{(\tilde{\rho}_\mathsf{s} \rho_\mathsf{s} \log p_\mathsf{s})^{C_1}}{\sqrt{p_\mathsf{s}}}, \qquad \mathcal{E}_{n_\mathsf{t}, p_\mathsf{t}} := \frac{(\tilde{\rho}_\mathsf{t} \log n_\mathsf{t})^{C_1}}{\sqrt{n_\mathsf{t}}} + \frac{(\tilde{\rho}_\mathsf{t} \rho_\mathsf{t} \log p_\mathsf{t})^{C_1}}{\sqrt{p_\mathsf{t}}};$$
$$\mathcal{E}_{n_\mathsf{s}, p_\mathsf{s}} := \mathcal{E}_{n_\mathsf{s}, p_\mathsf{s}} + \mathcal{E}_{n_\mathsf{t}, p_\mathsf{t}}$$

## A. Proof of Theorem 2: Deterministic equivalent for the student

In order to prove the deterministic equivalent of the student's test error, we first compute the deterministic equivalent of the test error conditioned on the teacher's weights $\boldsymbol{\beta}_\mathsf{t}$, and then compute the deterministic equivalent w.r.t $\boldsymbol{\beta}_\mathsf{t}$.

In particular, we prove that, with probability at least $1 - n^{-D}$,

$$|\mathcal{B}_\mathsf{s}(\boldsymbol{\beta}_*, \boldsymbol{\beta}_\mathsf{t}; \boldsymbol{G}_\mathsf{s}, \boldsymbol{F}_\mathsf{s}, \lambda_\mathsf{s}) - \mathcal{B}_\mathsf{s}(\boldsymbol{\beta}_*, \boldsymbol{\beta}_\mathsf{t})| \leq C_* \mathcal{E}_{n_\mathsf{s}, p_\mathsf{s}} \tilde{\mathcal{B}}_\mathsf{s}(\boldsymbol{\beta}_*, \boldsymbol{\beta}_\mathsf{t})$$
$$|\mathcal{B}_\mathsf{s}(\boldsymbol{\beta}_*, \boldsymbol{\beta}_\mathsf{t}) - \mathsf{B}_\mathsf{s}(\boldsymbol{\beta}_*)| \leq C_* \mathcal{E}_{n_\mathsf{t}, p_\mathsf{t}} \tilde{\mathsf{B}}_\mathsf{s}(\boldsymbol{\beta}_*)$$

and then by triangle inequality, we have

$$|\mathcal{B}_\mathsf{s}(\boldsymbol{\beta}_*, \boldsymbol{\beta}_\mathsf{t}; \boldsymbol{G}_\mathsf{s}, \boldsymbol{F}_\mathsf{s}, \lambda_\mathsf{s}) - \mathsf{B}_\mathsf{s}(\boldsymbol{\beta}_*)| \leq C_* \mathcal{E}_{n_\mathsf{s}, p_\mathsf{s}, n_\mathsf{t}, p_\mathsf{t}} \left( \left\{ \mathcal{B}_\mathsf{s}(\boldsymbol{\beta}_*, \boldsymbol{\beta}_\mathsf{t}) + \tilde{\mathsf{B}}_\mathsf{s}(\boldsymbol{\beta}_*) \right\} \right).$$

The proof will follow a similar approach as in Misiakiewicz & Saeed (2024); Defilippis et al. (2024) and we will omit repetitive details. All inequality between random variables are understood to hold with probability at least $1 - n^{-D}$ for a fixed arbitrary $D > 0$. We will allow the universal constants $C_0, C_1$ (appearing in $\mathcal{E}_{n_\mathsf{s}, p_\mathsf{s}, n_\mathsf{t}, p_\mathsf{t}}$), and the constant $C_*$ to change from line to line (in particular, $C_*$ depends on $D$).

### A.1. Deterministic approximation conditioned on the teacher

Conditioning on the teacher's weights $\boldsymbol{\beta}_{\mathsf{t}}$, we have

$$\mathcal{R}(\boldsymbol{G}_{\mathsf{s}}, \boldsymbol{F}_{\mathsf{s}}, \lambda_{\mathsf{s}}; \boldsymbol{\beta}_*, \boldsymbol{\beta}_{\mathsf{t}}) = \|\boldsymbol{\beta}_* - p_{\mathsf{s}}^{-1/2} \boldsymbol{F}_{\mathsf{s}}^{\mathsf{T}} \boldsymbol{\Xi}_{\mathsf{s}} \boldsymbol{Z}_{\mathsf{s}}^{\mathsf{T}} \boldsymbol{G}_{\mathsf{s}} \boldsymbol{\beta}_{\mathsf{t}}\|_2^2 = \|\boldsymbol{\beta}_* - \boldsymbol{\beta}_{\mathsf{t}}\|_2^2 + \|\boldsymbol{\beta}_{\mathsf{t}} - p_{\mathsf{s}}^{-1/2} \boldsymbol{F}_{\mathsf{s}}^{\mathsf{T}} \boldsymbol{\Xi}_{\mathsf{s}} \boldsymbol{Z}_{\mathsf{s}}^{\mathsf{T}} \boldsymbol{G}_{\mathsf{s}} \boldsymbol{\beta}_{\mathsf{t}}\|_2^2$$
$$+ 2 \langle \boldsymbol{\beta}_* - \boldsymbol{\beta}_{\mathsf{t}}, \boldsymbol{\beta}_{\mathsf{t}} - p_{\mathsf{s}}^{-1/2} \boldsymbol{F}_{\mathsf{s}}^{\mathsf{T}} \boldsymbol{\Xi}_{\mathsf{s}} \boldsymbol{Z}_{\mathsf{s}}^{\mathsf{T}} \boldsymbol{G}_{\mathsf{s}} \boldsymbol{\beta}_{\mathsf{t}} \rangle.$$

The first term is deterministic after conditioning on the teacher. The deterministic equivalent of the second term is known from Misiakiewicz & Saeed (2024); Defilippis et al. (2024), and we have

$$\left| \|\boldsymbol{\beta}_{\mathsf{t}} - p_{\mathsf{s}}^{-1/2} \boldsymbol{F}_{\mathsf{s}}^{\mathsf{T}} \boldsymbol{\Xi}_{\mathsf{s}} \boldsymbol{Z}_{\mathsf{s}}^{\mathsf{T}} \boldsymbol{G}_{\mathsf{s}} \boldsymbol{\beta}_{\mathsf{t}}\|_2^2 - \boldsymbol{\beta}_{\mathsf{t}}^{\mathsf{T}} \boldsymbol{\Lambda}_{\mathsf{t}} \boldsymbol{\beta}_{\mathsf{t}} \right| \le C_* \mathcal{E}_{n_{\mathsf{s}}, p_{\mathsf{s}}} \frac{\mu_{\mathsf{s},2} \boldsymbol{\beta}_{\mathsf{t}}^{\mathsf{T}} (\boldsymbol{\Sigma} + \mu_{\mathsf{s},2})^{-1} \boldsymbol{\beta}_{\mathsf{t}}}{1 - \Upsilon_{n,p}(\mu_{\mathsf{s},1}, \mu_{\mathsf{s},2})},$$

where

$$\boldsymbol{\Lambda}_{\mathsf{t}} = \frac{\mu_{\mathsf{s},2}^2}{1 - \Upsilon_{n,p}(\mu_{\mathsf{s},1}, \mu_{\mathsf{s},2})} \left[ (\boldsymbol{\Sigma} + \mu_{\mathsf{s},2})^{-2} + \chi_{n,p}(\mu_{\mathsf{s},2}) \boldsymbol{\Sigma} (\boldsymbol{\Sigma} + \mu_{\mathsf{s},2})^{-2} \right].$$

We note that the above bound is non-asymptotic compared to (Defilippis et al., 2024, Theorem B.12), since we do not require that Assumption 2 holds for $\boldsymbol{\beta}_{\mathsf{t}}$. The difference is that in (Defilippis et al., 2024, Proof of Lemma B.13, step 3), we upper bound

$$\mu_{\mathsf{s},2} \boldsymbol{\beta}_{\mathsf{t}}^{\mathsf{T}} (\boldsymbol{\Sigma} + \mu_{\mathsf{s},2})^{-1} \boldsymbol{\beta}_{\mathsf{t}} + (p_{\mathsf{s}} \mu_{\mathsf{s},1}) \frac{\boldsymbol{\beta}_{\mathsf{t}}^{\mathsf{T}} \boldsymbol{\Sigma} (\boldsymbol{\Sigma} + \mu_{\mathsf{s},2})^{-2} \boldsymbol{\beta}_{\mathsf{t}}}{p_{\mathsf{s}} - \text{Tr}(\boldsymbol{\Sigma}^2 (\boldsymbol{\Sigma} + \mu_{\mathsf{s},2})^{-2})},$$

by $\mu_{\mathsf{s},2} \boldsymbol{\beta}_{\mathsf{t}}^{\mathsf{T}} (\boldsymbol{\Sigma} + \mu_{\mathsf{s},2})^{-1} \boldsymbol{\beta}_{\mathsf{t}}$ instead of $\boldsymbol{\beta}_{\mathsf{t}}^{\mathsf{T}} \boldsymbol{\Lambda}_{\mathsf{t}} \boldsymbol{\beta}_{\mathsf{t}}$ due to the missing Assumption 2 on $\boldsymbol{\beta}_{\mathsf{t}}$. The rest follows the same argument as in (Defilippis et al., 2024, Proof of Theorem B.12).

It remains to bound the cross term, and we first decompose

$$\boldsymbol{\beta}_{\mathsf{t}} - p_{\mathsf{s}}^{-1/2} \boldsymbol{F}_{\mathsf{s}}^{\mathsf{T}} \boldsymbol{\Xi}_{\mathsf{s}} \boldsymbol{Z}_{\mathsf{s}}^{\mathsf{T}} \boldsymbol{G}_{\mathsf{s}} \boldsymbol{\beta}_{\mathsf{t}} = \boldsymbol{\beta}_{\mathsf{t}} - \frac{1}{\sqrt{p_{\mathsf{s}}}} \boldsymbol{F}_{\mathsf{s}}^{\mathsf{T}} (\boldsymbol{Z}_{\mathsf{s}}^{\mathsf{T}} \boldsymbol{Z}_{\mathsf{s}} + \lambda_{\mathsf{s}})^{-1} \boldsymbol{Z}_{\mathsf{s}}^{\mathsf{T}} \boldsymbol{Z}_{\mathsf{s}} \boldsymbol{\beta}_{\mathsf{t}, \boldsymbol{F}_{\mathsf{s}}} - \frac{1}{\sqrt{p_{\mathsf{s}}}} \boldsymbol{F}_{\mathsf{s}}^{\mathsf{T}} (\boldsymbol{Z}_{\mathsf{s}}^{\mathsf{T}} \boldsymbol{Z}_{\mathsf{s}} + \lambda_{\mathsf{s}})^{-1} \boldsymbol{Z}_{\mathsf{s}}^{\mathsf{T}} \boldsymbol{R}_{\mathsf{s}} \boldsymbol{\beta}_{\mathsf{t}, \boldsymbol{F}_{\mathsf{s}}, \perp},$$

and compute the deterministic equivalent w.r.t. $\boldsymbol{F}_{\mathsf{s}}$. Here we introduced

$$\boldsymbol{\beta}_{\mathsf{t}, \boldsymbol{F}_{\mathsf{s}}} = \sqrt{p_{\mathsf{s}}} \boldsymbol{F}_{\mathsf{s}} (\boldsymbol{F}_{\mathsf{s}}^{\mathsf{T}} \boldsymbol{F}_{\mathsf{s}})^{\dagger} \boldsymbol{\beta}_{\mathsf{t}}, \qquad \boldsymbol{\beta}_{\mathsf{t}, \boldsymbol{F}_{\mathsf{s}}, \perp} = \mathsf{P}_{\perp, \boldsymbol{F}_{\mathsf{s}}} \boldsymbol{\beta}_{\mathsf{t}}, \qquad \mathsf{P}_{\perp, \boldsymbol{F}_{\mathsf{s}}} = \boldsymbol{I} - (\boldsymbol{F}_{\mathsf{s}}^{\mathsf{T}} \boldsymbol{F}_{\mathsf{s}})(\boldsymbol{F}_{\mathsf{s}}^{\mathsf{T}} \boldsymbol{F}_{\mathsf{s}})^{\dagger}.$$

We will also use $\mathsf{P}_{\boldsymbol{F}_{\mathsf{s}}} = (\boldsymbol{F}_{\mathsf{s}}^{\mathsf{T}} \boldsymbol{F}_{\mathsf{s}})(\boldsymbol{F}_{\mathsf{s}}^{\mathsf{T}} \boldsymbol{F}_{\mathsf{s}})^{\dagger}$ to denote the projection onto the span of $\boldsymbol{F}_{\mathsf{s}}$.

For the deterministic equivalent of $\langle \boldsymbol{\beta}_* - \boldsymbol{\beta}_{\mathsf{t}}, \frac{1}{\sqrt{p_{\mathsf{s}}}} \boldsymbol{F}_{\mathsf{s}}^{\mathsf{T}} (\boldsymbol{Z}_{\mathsf{s}}^{\mathsf{T}} \boldsymbol{Z}_{\mathsf{s}} + \lambda_{\mathsf{s}})^{-1} \boldsymbol{Z}_{\mathsf{s}}^{\mathsf{T}} \boldsymbol{Z}_{\mathsf{s}} \boldsymbol{\beta}_{\mathsf{t}, \boldsymbol{F}_{\mathsf{s}}} \rangle$, we note that

$$\left\langle \boldsymbol{\beta}_* - \boldsymbol{\beta}_{\mathsf{t}}, \frac{1}{\sqrt{p_{\mathsf{s}}}} \boldsymbol{F}_{\mathsf{s}}^{\mathsf{T}} (\boldsymbol{Z}_{\mathsf{s}}^{\mathsf{T}} \boldsymbol{Z}_{\mathsf{s}} + \lambda_{\mathsf{s}})^{-1} \boldsymbol{Z}_{\mathsf{s}}^{\mathsf{T}} \boldsymbol{Z}_{\mathsf{s}} \boldsymbol{\beta}_{\mathsf{t}, \boldsymbol{F}_{\mathsf{s}}} \right\rangle = \Phi_2(\boldsymbol{Z}_{\mathsf{s}}; \boldsymbol{v} \boldsymbol{u}^{\top}, \lambda_{\mathsf{s}}),$$

where

$$\boldsymbol{u} = \hat{\boldsymbol{\Sigma}}_{\boldsymbol{F}_{\mathsf{s}}}^{1/2} \boldsymbol{\beta}_{t, \boldsymbol{F}_{\mathsf{s}}}, \quad \boldsymbol{v} = \frac{1}{\sqrt{p_{\mathsf{s}}}} \hat{\boldsymbol{\Sigma}}_{\boldsymbol{F}_{\mathsf{s}}}^{-1/2} \boldsymbol{F}_{\mathsf{s}}^{\top} (\boldsymbol{\beta}_* - \boldsymbol{\beta}_{\mathsf{t}}).$$

By Theorem 6, we have

$$\left| \Phi_2(\boldsymbol{Z}_{\mathsf{s}}; \boldsymbol{v} \boldsymbol{u}^{\top}, \lambda_{\mathsf{s}}) - \Psi_1(\mu_{\mathsf{s},1}; \boldsymbol{v} \boldsymbol{u}^{\top}) \right| \le C_* \mathcal{E}_{n_{\mathsf{s}}, p_{\mathsf{s}}} \tilde{\Psi}_1(\mu_{\mathsf{s},1}; \boldsymbol{v} \boldsymbol{u}^{\top}),$$

where

$$\Psi_1(\mu_{\mathsf{s},1}; \boldsymbol{v} \boldsymbol{u}^{\top}) = \langle \boldsymbol{\beta}_* - \boldsymbol{\beta}_{\mathsf{t}}, \frac{1}{\sqrt{p_{\mathsf{s}}}} \boldsymbol{F}_{\mathsf{s}}^{\top} (\hat{\boldsymbol{\Sigma}}_{\mathsf{s}} + \mu_{\mathsf{s},1})^{-1} \hat{\boldsymbol{\Sigma}}_{\mathsf{s}} \boldsymbol{\beta}_{t, \boldsymbol{F}_{\mathsf{s}}} \rangle,$$

$$\tilde{\Psi}_1(\mu_{\mathsf{s},1}; \boldsymbol{v} \boldsymbol{u}^{\top}) = \left\{ \|\boldsymbol{u}\|_2 \sqrt{\boldsymbol{v}^{\top} (\hat{\boldsymbol{\Sigma}}_{\mathsf{s}} + \mu_{\mathsf{s},1})^{-2} \hat{\boldsymbol{\Sigma}}_{\mathsf{s}}^2 \boldsymbol{v}} \right\} \vee \sqrt{\boldsymbol{v}^{\top} (\hat{\boldsymbol{\Sigma}}_{\mathsf{s}} + \mu_{\mathsf{s},1})^{-1} \hat{\boldsymbol{\Sigma}}_{\mathsf{s}} \boldsymbol{v} \boldsymbol{u}^{\top} (\boldsymbol{\Sigma}_{\mathsf{s}} + \mu_{\mathsf{s},1})^{-1} \boldsymbol{\Sigma}_{\mathsf{s}} \boldsymbol{u}}.$$

For the determinisic equivalent of $\langle \boldsymbol{\beta}_* - \boldsymbol{\beta}_t, \frac{1}{\sqrt{p_s}} \boldsymbol{F}_s^\top (\boldsymbol{Z}_s^\top \boldsymbol{Z}_s + \lambda_s)^{-1} \boldsymbol{Z}_s^\top \boldsymbol{R}_s \boldsymbol{\beta}_{t,\boldsymbol{F}_s,\perp} \rangle$, by Lemma 1 we have:

$$|\langle \boldsymbol{\beta}_* - \boldsymbol{\beta}_t, \frac{1}{\sqrt{p_s}} \boldsymbol{F}_s^\top (\boldsymbol{Z}_s^\top \boldsymbol{Z}_s + \lambda_s)^{-1} \boldsymbol{Z}_s^\top \boldsymbol{R}_s \boldsymbol{\beta}_{t,\boldsymbol{F}_s,\perp} \rangle| \leq C_* \mathcal{E}_{n_s} \|\boldsymbol{\beta}_{t,\boldsymbol{F}_s,\perp}\|_2 \|\boldsymbol{\beta}_* - \boldsymbol{\beta}_t\|_2,$$

with

$$n_s^2 \Psi_2(\mu_{s,1}; \boldsymbol{v}\boldsymbol{v}^\top; \boldsymbol{I}) = \frac{\boldsymbol{v}^\top (\hat{\boldsymbol{\Sigma}}_s + \mu_{s,1})^{-2} \hat{\boldsymbol{\Sigma}}_s^2 \boldsymbol{v}}{1 - \frac{1}{n_s} \mathrm{Tr}(\boldsymbol{F}_s^\top \boldsymbol{F}_s (\boldsymbol{F}_s^\top \boldsymbol{F}_s + p_s \mu_{s,1})^{-1})}.$$

Note that

$$\langle \boldsymbol{\beta}_* - \boldsymbol{\beta}_t, \frac{1}{\sqrt{p_s}} \boldsymbol{F}_s^\top (\hat{\boldsymbol{\Sigma}}_s + \mu_{s,1})^{-1} \hat{\boldsymbol{\Sigma}}_s \boldsymbol{\beta}_{t,\boldsymbol{F}_s} \rangle$$
$$= \langle \boldsymbol{\beta}_* - \boldsymbol{\beta}_t, \boldsymbol{F}_s^\top \boldsymbol{F}_s (\boldsymbol{F}_s^\top \boldsymbol{F}_s + p_s \mu_{s,1})^{-1} \boldsymbol{F}_s^\top \boldsymbol{F}_s (\boldsymbol{F}_s^\top \boldsymbol{F}_s)^\dagger \boldsymbol{\beta}_t \rangle$$
$$= \langle \boldsymbol{\beta}_* - \boldsymbol{\beta}_t, \boldsymbol{F}_s^\top \boldsymbol{F}_s (\boldsymbol{F}_s^\top \boldsymbol{F}_s + p_s \mu_{s,1})^{-1} (\boldsymbol{I} - \mathrm{P}_{\perp,\boldsymbol{F}_s}) \boldsymbol{\beta}_t \rangle$$
$$= \langle \boldsymbol{\beta}_* - \boldsymbol{\beta}_t, \boldsymbol{F}_s^\top \boldsymbol{F}_s (\boldsymbol{F}_s^\top \boldsymbol{F}_s + p_s \mu_{s,1})^{-1} \boldsymbol{\beta}_t \rangle,$$

which implies that

$$\langle \boldsymbol{\beta}_* - \boldsymbol{\beta}_t, \boldsymbol{\beta}_t \rangle - \langle \boldsymbol{\beta}_* - \boldsymbol{\beta}_t, \frac{1}{\sqrt{p_s}} \boldsymbol{F}_s^\top (\hat{\boldsymbol{\Sigma}}_s + \mu_{s,1})^{-1} \hat{\boldsymbol{\Sigma}}_s \boldsymbol{\beta}_{t,\boldsymbol{F}_s} \rangle$$
$$= (p_s \mu_s) \langle \boldsymbol{\beta}_* - \boldsymbol{\beta}_t, (\boldsymbol{F}_s^\top \boldsymbol{F}_s + p_s \mu_{s,1})^{-1} \boldsymbol{\beta}_t \rangle,$$

and by Theorem 6, we have

$$|(p_s \mu_s) \langle \boldsymbol{\beta}_* - \boldsymbol{\beta}_t, (\boldsymbol{F}_s^\top \boldsymbol{F}_s + p_s \mu_{s,1})^{-1} \boldsymbol{\beta}_t \rangle - \mu_{s,2} \langle \boldsymbol{\beta}_* - \boldsymbol{\beta}_t, (\boldsymbol{\Sigma} + \mu_{s,2})^{-1} \boldsymbol{\beta}_t \rangle|$$
$$\leq C_* \mathcal{E}_{n_s, p_s} \mu_{s,2} \sqrt{\langle \boldsymbol{\beta}_* - \boldsymbol{\beta}_t, (\boldsymbol{\Sigma} + \mu_{s,2})^{-1} (\boldsymbol{\beta}_* - \boldsymbol{\beta}_t) \rangle \langle \boldsymbol{\beta}_t, (\boldsymbol{\Sigma} + \mu_{s,2})^{-1} \boldsymbol{\beta}_t \rangle}.$$

By combining the terms, we obtain:

$$|\|\boldsymbol{\beta}_* - p_s^{-1/2} \boldsymbol{F}_s^\top \boldsymbol{\Xi}_s \boldsymbol{Z}_s^\top \boldsymbol{G}_s \boldsymbol{\beta}_t\|_2^2 - \mathcal{B}_s(\boldsymbol{\beta}_*, \boldsymbol{\beta}_t)| \leq \tilde{\mathcal{B}}_s(\boldsymbol{\beta}_*, \boldsymbol{\beta}_t),$$

where

$$\mathcal{B}_s(\boldsymbol{\beta}_t) = \|\boldsymbol{\beta}_* - \boldsymbol{\beta}_t\|_2^2 + 2\mu_{s,2} \langle \boldsymbol{\beta}_* - \boldsymbol{\beta}_t, (\boldsymbol{\Sigma} + \mu_{s,2})^{-1} \boldsymbol{\beta}_t \rangle + \boldsymbol{\beta}_t^\top \boldsymbol{\Lambda}_t \boldsymbol{\beta}_t$$
$$= \|\boldsymbol{\beta}_* - \boldsymbol{\Sigma}(\boldsymbol{\Sigma} + \mu_{s,2})^{-1} \boldsymbol{\beta}_t\|_2^2 + \boldsymbol{\beta}_t^\top \overline{\boldsymbol{\Lambda}}_0 \boldsymbol{\beta}_t,$$

and

$$\overline{\boldsymbol{\Lambda}}_0 = \frac{\Upsilon_{n_s, p_s}(\mu_{s,1}, \mu_{s,2})}{1 - \Upsilon_{n,p}(\mu_{s,1}, \mu_{s,2})} \mu_{s,2}^2 (\boldsymbol{\Sigma} + \mu_{s,2})^{-2} + \frac{\chi_{n,p}(\mu_{s,2})}{1 - \Upsilon_{n,p}(\mu_{s,1}, \mu_{s,2})} \mu_{s,2}^2 \boldsymbol{\Sigma} (\boldsymbol{\Sigma} + \mu_{s,2})^{-2}.$$

Finally, we compute the deterministic equivalent of the upper bounds. We note that each term in the approximation functional is at most $\|\boldsymbol{\beta}_t\|_2 \|\boldsymbol{\beta}_* - \boldsymbol{\beta}_t\|_2$, thus we have

$$\tilde{\mathcal{B}}_s(\boldsymbol{\beta}_*, \boldsymbol{\beta}_t) \leq \|\boldsymbol{\beta}_t\|_2 \|\boldsymbol{\beta}_* - \boldsymbol{\beta}_t\|_2.$$

## A.2. Deterministic equivalent on the teacher's model

Next, we compute the deterministic equivalent of $\mathcal{B}_s(\boldsymbol{\beta}_*, \boldsymbol{\beta}_t)$. We first decompose $\mathcal{B}_s(\boldsymbol{\beta}_*, \boldsymbol{\beta}_t)$ as

$$\mathcal{B}_s(\boldsymbol{\beta}_*, \boldsymbol{\beta}_t) = \|\boldsymbol{\beta}_* - \boldsymbol{\Sigma}(\boldsymbol{\Sigma} + \mu_{s,2})^{-1} \boldsymbol{\beta}_t\|_2^2 + \boldsymbol{\beta}_t^\top \boldsymbol{\Lambda}_0 \boldsymbol{\beta}_t$$
$$= \|\boldsymbol{\beta}_*\|_2^2 - 2\langle \boldsymbol{\beta}_*, \boldsymbol{\Sigma}(\boldsymbol{\Sigma} + \mu_{s,2})^{-1} \boldsymbol{\beta}_t \rangle + \langle \boldsymbol{\beta}_t, \boldsymbol{\Lambda}_0 \boldsymbol{\beta}_t \rangle,$$

where

$$\boldsymbol{\Lambda}_0 = \boldsymbol{\Sigma}^2(\boldsymbol{\Sigma} + \mu_{\mathsf{s},2})^{-2} + \frac{\Upsilon_{n_\mathsf{s},p_\mathsf{s}}(\mu_{\mathsf{s},1},\mu_{\mathsf{s},2})}{1 - \Upsilon_{n,p}(\mu_{\mathsf{s},1},\mu_{\mathsf{s},2})}\mu_{\mathsf{s},2}^2(\boldsymbol{\Sigma} + \mu_{\mathsf{s},2})^{-2} + \frac{\chi_{n,p}(\mu_{\mathsf{s},2})}{1 - \Upsilon_{n,p}(\mu_{\mathsf{s},1},\mu_{\mathsf{s},2})}\mu_{\mathsf{s},2}^2\boldsymbol{\Sigma}(\boldsymbol{\Sigma} + \mu_{\mathsf{s},2})^{-2}.$$

We split $\boldsymbol{\beta}_\mathsf{t}$ as

$$\begin{aligned}\boldsymbol{\beta}_\mathsf{t} = &\frac{1}{\sqrt{p_\mathsf{t}}}\boldsymbol{F}_\mathsf{t}^\top(\boldsymbol{Z}_\mathsf{t}^\top\boldsymbol{Z}_\mathsf{t} + \lambda_\mathsf{t})^{-1}\boldsymbol{Z}_\mathsf{t}^\top\boldsymbol{Z}_\mathsf{t}\boldsymbol{\beta}_{\boldsymbol{F}_\mathsf{t}} + \frac{1}{\sqrt{p_\mathsf{t}}}\boldsymbol{F}_\mathsf{t}^\top(\boldsymbol{Z}_\mathsf{t}^\top\boldsymbol{Z}_\mathsf{t} + \lambda_\mathsf{t})^{-1}\boldsymbol{Z}_\mathsf{t}^\top\boldsymbol{R}_\mathsf{t}\boldsymbol{\beta}_{\perp,\boldsymbol{F}_\mathsf{t}} \\ &+ \frac{1}{\sqrt{p_\mathsf{t}}}\boldsymbol{F}_\mathsf{t}^\top(\boldsymbol{Z}_\mathsf{t}^\top\boldsymbol{Z}_\mathsf{t} + \lambda_\mathsf{t})^{-1}\boldsymbol{Z}_\mathsf{t}^\top\boldsymbol{\varepsilon}.\end{aligned}$$

We decompose the test error into the terms that depend on the teacher variance (i.e., terms that depend on $\varepsilon$), and terms that depend on the teacher bias (i.e., terms that do not depends on $\varepsilon$). We define

$$\begin{aligned}\mathcal{B}_{\mathsf{s},bias}(\boldsymbol{Z}_\mathsf{t}) = &\boldsymbol{\beta}_{\boldsymbol{F}_\mathsf{t}}^\top(\boldsymbol{Z}_\mathsf{t}^\top\boldsymbol{Z}_\mathsf{t})(\boldsymbol{Z}_\mathsf{t}^\top\boldsymbol{Z}_\mathsf{t} + \lambda_\mathsf{t})^{-1}\frac{\boldsymbol{F}_\mathsf{t}\boldsymbol{\Lambda}_0\boldsymbol{F}_\mathsf{t}^\top}{p_\mathsf{t}}(\boldsymbol{Z}_\mathsf{t}^\top\boldsymbol{Z}_\mathsf{t} + \lambda_\mathsf{t})^{-1}(\boldsymbol{Z}_\mathsf{t}^\top\boldsymbol{Z}_\mathsf{t})\boldsymbol{\beta}_{\boldsymbol{F}_\mathsf{t}} \\ &+ \boldsymbol{\beta}_{\perp,\mathsf{t}}^\top\boldsymbol{R}_\mathsf{t}^\top\boldsymbol{Z}_\mathsf{t}(\boldsymbol{Z}_\mathsf{t}^\top\boldsymbol{Z}_\mathsf{t} + \lambda_\mathsf{t})^{-1}\frac{\boldsymbol{F}_\mathsf{t}\boldsymbol{\Lambda}_0\boldsymbol{F}_\mathsf{t}^\top}{p_\mathsf{t}}(\boldsymbol{Z}_\mathsf{t}^\top\boldsymbol{Z}_\mathsf{t} + \lambda_\mathsf{t})^{-1}\boldsymbol{Z}_\mathsf{t}^\top\boldsymbol{R}_\mathsf{t}\boldsymbol{\beta}_{\perp,\mathsf{t}} \\ &+ \boldsymbol{\beta}_{\perp,\mathsf{t}}^\top\boldsymbol{R}_\mathsf{t}^\top\boldsymbol{Z}_\mathsf{t}(\boldsymbol{Z}_\mathsf{t}^\top\boldsymbol{Z}_\mathsf{t} + \lambda_\mathsf{t})^{-1}\frac{\boldsymbol{F}_\mathsf{t}\boldsymbol{\Lambda}_0\boldsymbol{F}_\mathsf{t}^\top}{p_\mathsf{t}}(\boldsymbol{Z}_\mathsf{t}^\top\boldsymbol{Z}_\mathsf{t} + \lambda_\mathsf{t})^{-1}(\boldsymbol{Z}_\mathsf{t}^\top\boldsymbol{Z}_\mathsf{t})\boldsymbol{\beta}_{\boldsymbol{F}_\mathsf{t}} \\ &+ \frac{1}{\sqrt{p_\mathsf{t}}}\langle\boldsymbol{\beta}_*, \boldsymbol{\Lambda}_1\boldsymbol{F}_\mathsf{t}^\top(\boldsymbol{Z}_\mathsf{t}^\top\boldsymbol{Z}_\mathsf{t} + \lambda_\mathsf{t})^{-1}\boldsymbol{Z}_\mathsf{t}^\top\boldsymbol{Z}_\mathsf{t}\boldsymbol{\beta}_{\boldsymbol{F}_\mathsf{t}}\rangle \\ &+ \frac{1}{\sqrt{p_\mathsf{t}}}\langle\boldsymbol{\beta}_*, \boldsymbol{\Lambda}_1\boldsymbol{F}_\mathsf{t}^\top(\boldsymbol{Z}_\mathsf{t}^\top\boldsymbol{Z}_\mathsf{t} + \lambda_\mathsf{t})^{-1}\boldsymbol{Z}_\mathsf{t}^\top\boldsymbol{R}_\mathsf{t}\boldsymbol{\beta}_{\perp,\boldsymbol{F}_\mathsf{t}}\rangle, \\ \mathcal{B}_{\mathsf{s},var}(\boldsymbol{Z}_\mathsf{t}) = &\tau_\mathsf{t}^2\frac{1}{p_\mathsf{t}}\mathrm{Tr}(\boldsymbol{F}_\mathsf{t}\boldsymbol{\Lambda}_0\boldsymbol{F}_\mathsf{t}^\top(\boldsymbol{Z}_\mathsf{t}^\top\boldsymbol{Z}_\mathsf{t} + \lambda_\mathsf{t})^{-1}\boldsymbol{Z}_\mathsf{t}^\top\boldsymbol{Z}_\mathsf{t}(\boldsymbol{Z}_\mathsf{t}^\top\boldsymbol{Z}_\mathsf{t} + \lambda_\mathsf{t})^{-1}).\end{aligned}$$

### A.2.1. TERMS DEPENDING ON THE TEACHER BIAS

In this part, we show the deterministic equivalent of $\mathcal{B}_{\mathsf{s},bias}(\boldsymbol{Z}_\mathsf{t})$.

**Quadratic term.** Let us consider the following quadratic term:

$$\begin{aligned}&\boldsymbol{\beta}_{\boldsymbol{F}_\mathsf{t}}^\top(\boldsymbol{Z}_\mathsf{t}^\top\boldsymbol{Z}_\mathsf{t})(\boldsymbol{Z}_\mathsf{t}^\top\boldsymbol{Z}_\mathsf{t} + \lambda_\mathsf{t})^{-1}\frac{\boldsymbol{F}_\mathsf{t}\boldsymbol{\Lambda}_0\boldsymbol{F}_\mathsf{t}^\top}{p_\mathsf{t}}(\boldsymbol{Z}_\mathsf{t}^\top\boldsymbol{Z}_\mathsf{t} + \lambda_\mathsf{t})^{-1}(\boldsymbol{Z}_\mathsf{t}^\top\boldsymbol{Z}_\mathsf{t})\boldsymbol{\beta}_{\boldsymbol{F}_\mathsf{t}} \\ &+ \boldsymbol{\beta}_{\perp,\mathsf{t}}^\top\boldsymbol{R}_\mathsf{t}^\top\boldsymbol{Z}_\mathsf{t}(\boldsymbol{Z}_\mathsf{t}^\top\boldsymbol{Z}_\mathsf{t} + \lambda_\mathsf{t})^{-1}\frac{\boldsymbol{F}_\mathsf{t}\boldsymbol{\Lambda}_0\boldsymbol{F}_\mathsf{t}^\top}{p_\mathsf{t}}(\boldsymbol{Z}_\mathsf{t}^\top\boldsymbol{Z}_\mathsf{t} + \lambda_\mathsf{t})^{-1}\boldsymbol{Z}_\mathsf{t}^\top\boldsymbol{R}_\mathsf{t}\boldsymbol{\beta}_{\perp,\mathsf{t}} \\ &+ \boldsymbol{\beta}_{\perp,\mathsf{t}}^\top\boldsymbol{R}_\mathsf{t}^\top\boldsymbol{Z}_\mathsf{t}(\boldsymbol{Z}_\mathsf{t}^\top\boldsymbol{Z}_\mathsf{t} + \lambda_\mathsf{t})^{-1}\frac{\boldsymbol{F}_\mathsf{t}\boldsymbol{\Lambda}_0\boldsymbol{F}_\mathsf{t}^\top}{p_\mathsf{t}}(\boldsymbol{Z}_\mathsf{t}^\top\boldsymbol{Z}_\mathsf{t} + \lambda_\mathsf{t})^{-1}(\boldsymbol{Z}_\mathsf{t}^\top\boldsymbol{Z}_\mathsf{t})\boldsymbol{\beta}_{\boldsymbol{F}_\mathsf{t}}.\end{aligned}$$

For the first term, we note that it is equal to $\Phi_5(\boldsymbol{Z}_\mathsf{t}; \boldsymbol{u}\boldsymbol{u}^\top, \boldsymbol{B})$ with

$$\boldsymbol{u} = \hat{\boldsymbol{\Sigma}}_{\boldsymbol{F}_\mathsf{t}}^{1/2}\boldsymbol{\beta}_{\boldsymbol{F}_\mathsf{t}}, \quad \boldsymbol{B} = \hat{\boldsymbol{\Sigma}}_{\boldsymbol{F}_\mathsf{t}}^{-1/2}\frac{\boldsymbol{F}_\mathsf{t}\boldsymbol{\Lambda}_0\boldsymbol{F}_\mathsf{t}^\top}{p_\mathsf{t}}\hat{\boldsymbol{\Sigma}}_{\boldsymbol{F}_\mathsf{t}}^{-1/2}.$$

By Theorem 6, we have:

$$|\Phi_5(\boldsymbol{Z}_\mathsf{t}; \boldsymbol{u}\boldsymbol{u}^\top, \boldsymbol{B}) - \Psi_3(\mu_{\mathsf{t},1}; \boldsymbol{u}\boldsymbol{u}^\top, \boldsymbol{B})| \leq C_*\mathcal{E}_{n_\mathsf{t},p_\mathsf{t}}\tilde{\Psi}_3(\mu_{\mathsf{t},1}; \boldsymbol{u}\boldsymbol{u}^\top, \boldsymbol{B}),$$

where

$$
\begin{aligned}
\Psi_3(\mu_{\mathsf{t},1}; \boldsymbol{u}\boldsymbol{u}^\top, \boldsymbol{B}) =& \boldsymbol{\beta}_{\boldsymbol{F}_\mathsf{t}}^\top \hat{\boldsymbol{\Sigma}}_{\boldsymbol{F}_\mathsf{t}} (\hat{\boldsymbol{\Sigma}}_{\boldsymbol{F}_\mathsf{t}} + \mu_{\mathsf{t},1})^{-1} \frac{\boldsymbol{F}_\mathsf{t}\boldsymbol{\Lambda}_0\boldsymbol{F}_\mathsf{t}^\top}{p_\mathsf{t}} (\hat{\boldsymbol{\Sigma}}_{\boldsymbol{F}_\mathsf{t}} + \mu_{\mathsf{t},1})^{-1} \hat{\boldsymbol{\Sigma}}_{\boldsymbol{F}_\mathsf{t}} \boldsymbol{\beta}_{\boldsymbol{F}_\mathsf{t}} \\
&+ \frac{(p_\mathsf{t}\mu_{\mathsf{t},1})^2 \mathrm{Tr}(\boldsymbol{\Lambda}_0(\boldsymbol{F}_\mathsf{t}^\top\boldsymbol{F}_\mathsf{t})^2(\boldsymbol{F}_\mathsf{t}^\top\boldsymbol{F}_\mathsf{t} + p_\mathsf{t}\mu_{\mathsf{t},1})^{-2})}{n_\mathsf{t} - \mathrm{Tr}(\boldsymbol{F}_\mathsf{t}^\top\boldsymbol{F}_\mathsf{t}(\boldsymbol{F}_\mathsf{t}^\top\boldsymbol{F}_\mathsf{t} + p_\mathsf{t}\mu_{\mathsf{t},1})^{-1})} \boldsymbol{\beta}_{\boldsymbol{F}_\mathsf{t}}^\top \hat{\boldsymbol{\Sigma}}_{\boldsymbol{F}_\mathsf{t}} (\boldsymbol{F}_\mathsf{t}^\top\boldsymbol{F}_\mathsf{t} + p_\mathsf{t}\mu_{\mathsf{t},1})^{-2} \boldsymbol{\beta}_{\boldsymbol{F}_\mathsf{t}}, \\
\tilde{\Psi}_3(\mu_{\mathsf{t},1}; \boldsymbol{u}\boldsymbol{u}^\top, \boldsymbol{B}) =& \|\hat{\boldsymbol{\Sigma}}_{\boldsymbol{F}_\mathsf{t}}^{1/2} \boldsymbol{\beta}_{\boldsymbol{F}_\mathsf{t}}\|_2^2 \left\| \hat{\boldsymbol{\Sigma}}_{\boldsymbol{F}_\mathsf{t}}^{-1/2} \frac{\boldsymbol{F}_\mathsf{t}\boldsymbol{\Lambda}_0\boldsymbol{F}_\mathsf{t}^\top}{p_\mathsf{t}} \hat{\boldsymbol{\Sigma}}_{\boldsymbol{F}_\mathsf{t}}^{-1/2} \right\|_{op} \\
\leq& \|(\boldsymbol{I} - \mathtt{P}_{\perp, \boldsymbol{F}_\mathsf{t}}) \boldsymbol{\beta}_*\|_2^2 \|\boldsymbol{\Lambda}_0\|_{op} \leq \|\boldsymbol{\beta}_*\|_2^2 \|\boldsymbol{\Lambda}_0\|_{op}.
\end{aligned}
$$

For the second term, by Lemma 1 we have

$$
\begin{aligned}
|\boldsymbol{\beta}_{\perp,\mathsf{t}}^\top \boldsymbol{R}_\mathsf{t}^\top \boldsymbol{Z}_\mathsf{t} (\boldsymbol{Z}_\mathsf{t}^\top\boldsymbol{Z}_\mathsf{t} + \lambda_\mathsf{t})^{-1} \frac{\boldsymbol{F}_\mathsf{t}\boldsymbol{\Lambda}_0\boldsymbol{F}_\mathsf{t}^\top}{p_\mathsf{t}} (\boldsymbol{Z}_\mathsf{t}^\top\boldsymbol{Z}_\mathsf{t} + \lambda_\mathsf{t})^{-1}\boldsymbol{Z}_\mathsf{t}^\top\boldsymbol{R}_\mathsf{t}\boldsymbol{\beta}_{\perp,\mathsf{t}} - \|\boldsymbol{\beta}_{\perp,\mathsf{t}}\|_2^2 \Psi_2(\mu_{\mathsf{t},1}; \boldsymbol{B}, \boldsymbol{I})| \leq& C_* \mathcal{E}_{n_\mathsf{t},p_\mathsf{t}} \|\boldsymbol{\beta}_{\perp,\mathsf{t}}\|_2^2 \|\boldsymbol{B}\|_{op} \\
\leq& C_* \mathcal{E}_{n_\mathsf{t},p_\mathsf{t}} \|\boldsymbol{\beta}_*\|_2^2 \|\boldsymbol{B}\|_{op},
\end{aligned}
$$

where

$$
\Psi_2(\mu_{\mathsf{t},1}; \boldsymbol{B}, \boldsymbol{I}) = \frac{\mathrm{Tr}(\boldsymbol{\Lambda}_0 \hat{\boldsymbol{\Sigma}}_{\boldsymbol{F}_\mathsf{t}} (\hat{\boldsymbol{\Sigma}}_{\boldsymbol{F}_\mathsf{t}} + \mu_{\mathsf{t},1})^{-2})}{n_\mathsf{t} - \mathrm{Tr}(\hat{\boldsymbol{\Sigma}}_{\boldsymbol{F}_\mathsf{t}}^2 (\hat{\boldsymbol{\Sigma}}_{\boldsymbol{F}_\mathsf{t}} + \mu_{\mathsf{t},1})^{-2})} = \frac{\mathrm{Tr}(\boldsymbol{\Lambda}_0(\boldsymbol{F}_\mathsf{t}^\top\boldsymbol{F}_\mathsf{t})^2(\boldsymbol{F}_\mathsf{t}^\top\boldsymbol{F}_\mathsf{t} + \mu_{\mathsf{t},1})^{-2})}{n_\mathsf{t} - \mathrm{Tr}((\boldsymbol{F}_\mathsf{t}^\top\boldsymbol{F}_\mathsf{t})^2(\boldsymbol{F}_\mathsf{t}^\top\boldsymbol{F}_\mathsf{t} + \mu_{\mathsf{t},1})^{-2})}.
$$

Note that

$$
\begin{aligned}
(p_\mathsf{t}\mu_{\mathsf{t},1})^2 \boldsymbol{\beta}_{\boldsymbol{F}_\mathsf{t}}^\top \hat{\boldsymbol{\Sigma}}_{\boldsymbol{F}_\mathsf{t}} (\boldsymbol{F}_\mathsf{t}^\top\boldsymbol{F}_\mathsf{t} + p_\mathsf{t}\mu_{\mathsf{t},1})^{-2} \boldsymbol{\beta}_{\boldsymbol{F}_\mathsf{t}} =& (p_\mathsf{t}\mu_{\mathsf{t},1})^2 \boldsymbol{\beta}_*^\top (\boldsymbol{I} - \mathtt{P}_{\perp,\boldsymbol{F}_\mathsf{t}})(\boldsymbol{F}_\mathsf{t}^\top\boldsymbol{F}_\mathsf{t} + p_\mathsf{t}\mu_{\mathsf{t},1})^{-2}(\boldsymbol{I} - \mathtt{P}_{\perp,\boldsymbol{F}_\mathsf{t}})\boldsymbol{\beta}_* \\
=& (p_\mathsf{t}\mu_{\mathsf{t},1})^2 \boldsymbol{\beta}_*^\top (\boldsymbol{F}_\mathsf{t}^\top\boldsymbol{F}_\mathsf{t} + p_\mathsf{t}\mu_{\mathsf{t},1})^{-2}\boldsymbol{\beta}_* - \|\boldsymbol{\beta}_{\perp,\boldsymbol{F}_\mathsf{t}}\|_2^2.
\end{aligned}
$$

Thus, we have

$$
\begin{aligned}
\Psi_3(\mu_{\mathsf{t},1}; & \boldsymbol{u}\boldsymbol{u}^\top, \boldsymbol{B}) + \Psi_2(\mu_{\mathsf{t},1}; \boldsymbol{B}, \boldsymbol{I}) \\
=& \boldsymbol{\beta}_{\boldsymbol{F}_\mathsf{t}}^\top \hat{\boldsymbol{\Sigma}}_{\boldsymbol{F}_\mathsf{t}} (\hat{\boldsymbol{\Sigma}}_{\boldsymbol{F}_\mathsf{t}} + \mu_{\mathsf{t},1})^{-1} \frac{\boldsymbol{F}_\mathsf{t}\boldsymbol{\Lambda}_0\boldsymbol{F}_\mathsf{t}^\top}{p_\mathsf{t}} (\hat{\boldsymbol{\Sigma}}_{\boldsymbol{F}_\mathsf{t}} + \mu_{\mathsf{t},1})^{-1} \hat{\boldsymbol{\Sigma}}_{\boldsymbol{F}_\mathsf{t}} \boldsymbol{\beta}_{\boldsymbol{F}_\mathsf{t}} \\
&+ \frac{(p_\mathsf{t}\mu_{\mathsf{t},1})^2 \mathrm{Tr}(\boldsymbol{\Lambda}_0(\boldsymbol{F}_\mathsf{t}^\top\boldsymbol{F}_\mathsf{t})^2(\boldsymbol{F}_\mathsf{t}^\top\boldsymbol{F}_\mathsf{t} + p_\mathsf{t}\mu_{\mathsf{t},1})^{-2})}{n_\mathsf{t} - \mathrm{Tr}((\boldsymbol{F}_\mathsf{t}^\top\boldsymbol{F}_\mathsf{t})^2(\boldsymbol{F}_\mathsf{t}^\top\boldsymbol{F}_\mathsf{t} + p_\mathsf{t}\mu_{\mathsf{t},1})^{-2})} \boldsymbol{\beta}_*^\top (\boldsymbol{F}_\mathsf{t}^\top\boldsymbol{F}_\mathsf{t} + p_\mathsf{t}\mu_{\mathsf{t},1})^{-2} \boldsymbol{\beta}_*.
\end{aligned}
$$

The third term concentrates to zero, and by Lemma 1, we have

$$
\begin{aligned}
|\boldsymbol{\beta}_{\perp,\mathsf{t}}^\top \boldsymbol{R}_\mathsf{t}^\top \boldsymbol{Z}_\mathsf{t} (\boldsymbol{Z}_\mathsf{t}^\top\boldsymbol{Z}_\mathsf{t} + \lambda_\mathsf{t})^{-1} \frac{\boldsymbol{F}_\mathsf{t}\boldsymbol{\Lambda}_0\boldsymbol{F}_\mathsf{t}^\top}{p_\mathsf{t}} (\boldsymbol{Z}_\mathsf{t}^\top\boldsymbol{Z}_\mathsf{t} + \lambda_\mathsf{t})^{-1}(\boldsymbol{Z}_\mathsf{t}^\top\boldsymbol{Z}_\mathsf{t})\boldsymbol{\beta}_{\boldsymbol{F}_\mathsf{t}}| \leq& C_* \mathcal{E}_{n_\mathsf{t},p_\mathsf{t}} \|\boldsymbol{\beta}_{\perp,\mathsf{t}}\|_2 \|\boldsymbol{B}\|_{op} \|\boldsymbol{u}\|_2 \\
\leq& C_* \mathcal{E}_{n_\mathsf{t},p_\mathsf{t}} \|\boldsymbol{\beta}_*\|_2^2 \|\boldsymbol{\Lambda}_0\|_{op}.
\end{aligned}
$$

Next, we do the deterministic equivalent w.r.t. $\boldsymbol{F}_\mathsf{t}$, and we have

$$
\begin{aligned}
\boldsymbol{\beta}_{\boldsymbol{F}_\mathsf{t}}^\top \hat{\boldsymbol{\Sigma}}_{\boldsymbol{F}_\mathsf{t}} &(\hat{\boldsymbol{\Sigma}}_{\boldsymbol{F}_\mathsf{t}} + \mu_{\mathsf{t},1})^{-1} \frac{\boldsymbol{F}_\mathsf{t}\boldsymbol{\Lambda}_0\boldsymbol{F}_\mathsf{t}^\top}{p_\mathsf{t}} (\hat{\boldsymbol{\Sigma}}_{\boldsymbol{F}_\mathsf{t}} + \mu_{\mathsf{t},1})^{-1} \hat{\boldsymbol{\Sigma}}_{\boldsymbol{F}_\mathsf{t}} \boldsymbol{\beta}_{\boldsymbol{F}_\mathsf{t}} \\
=& \boldsymbol{\beta}_*^\top (\boldsymbol{I} - \mathtt{P}_{\perp,\boldsymbol{F}_\mathsf{t}})(\boldsymbol{F}_\mathsf{t}^\top\boldsymbol{F}_\mathsf{t})(\boldsymbol{F}_\mathsf{t}^\top\boldsymbol{F}_\mathsf{t} + p_\mathsf{t}\mu_{\mathsf{t},1})^{-1}\boldsymbol{\Lambda}_0(\boldsymbol{F}_\mathsf{t}^\top\boldsymbol{F}_\mathsf{t})(\boldsymbol{F}_\mathsf{t}^\top\boldsymbol{F}_\mathsf{t} + p_\mathsf{t}\mu_{\mathsf{t},1})^{-1}(\boldsymbol{I} - \mathtt{P}_{\perp,\boldsymbol{F}_\mathsf{t}})\boldsymbol{\beta}_* \\
=& \boldsymbol{\beta}_*^\top (\boldsymbol{F}_\mathsf{t}^\top\boldsymbol{F}_\mathsf{t})(\boldsymbol{F}_\mathsf{t}^\top\boldsymbol{F}_\mathsf{t} + p_\mathsf{t}\mu_{\mathsf{t},1})^{-1}\boldsymbol{\Lambda}_0(\boldsymbol{F}_\mathsf{t}^\top\boldsymbol{F}_\mathsf{t})(\boldsymbol{F}_\mathsf{t}^\top\boldsymbol{F}_\mathsf{t} + p_\mathsf{t}\mu_{\mathsf{t},1})^{-1}\boldsymbol{\beta}_* \\
=& \|\boldsymbol{\beta}_*\|_2^2 - 2(p_\mathsf{t}\mu_\mathsf{t})\boldsymbol{\beta}_*^\top (\boldsymbol{F}_\mathsf{t}^\top\boldsymbol{F}_\mathsf{t} + p_\mathsf{t}\mu_{\mathsf{t},1})^{-1}\boldsymbol{\Lambda}_0\boldsymbol{\beta}_* \\
&+ (p_\mathsf{t}\mu_{\mathsf{t},1})^2 \boldsymbol{\beta}_*^\top (\boldsymbol{F}_\mathsf{t}^\top\boldsymbol{F}_\mathsf{t} + p_\mathsf{t}\mu_{\mathsf{t},1})^{-1}\boldsymbol{\Lambda}_0(\boldsymbol{F}_\mathsf{t}^\top\boldsymbol{F}_\mathsf{t} + p_\mathsf{t}\mu_{\mathsf{t},1})^{-1}\boldsymbol{\beta}_*.
\end{aligned}
$$

The second term satisfies

$$
\begin{aligned}
|(p_\mathsf{t}\mu_\mathsf{t})\boldsymbol{\beta}_*^\top &(\boldsymbol{F}_\mathsf{t}^\top\boldsymbol{F}_\mathsf{t} + p_\mathsf{t}\mu_{\mathsf{t},1})^{-1}\boldsymbol{\Lambda}_0\boldsymbol{\beta}_* - \mu_{\mathsf{t},2}\boldsymbol{\beta}_*^\top (\boldsymbol{\Sigma} + \mu_{\mathsf{t},2})^{-1}\boldsymbol{\Lambda}_0\boldsymbol{\beta}_*| \\
&\leq C_* \mathcal{E}_{n_\mathsf{t},p_\mathsf{t}}\mu_{\mathsf{t},2}\sqrt{\boldsymbol{\beta}_*^\top(\boldsymbol{\Sigma} + \mu_{\mathsf{t},2})^{-1}\boldsymbol{\beta}_*\boldsymbol{\beta}_*^\top\boldsymbol{\Lambda}_0(\boldsymbol{\Sigma} + \mu_{\mathsf{t},2})^{-1}\boldsymbol{\Lambda}_0\boldsymbol{\beta}_*}.
\end{aligned}
$$

The third term can be written as

$$\boldsymbol{\beta}_*^\top (\boldsymbol{F}_\mathsf{t}^\top \boldsymbol{F}_\mathsf{t} + p_\mathsf{t}\mu_{\mathsf{t},1})^{-1}\boldsymbol{\Lambda}_0(\boldsymbol{F}_\mathsf{t}^\top \boldsymbol{F}_\mathsf{t} + p_\mathsf{t}\mu_{\mathsf{t},1})^{-1}\boldsymbol{\beta}_* = \Phi_3(\boldsymbol{F}_\mathsf{t}; \boldsymbol{A}, \boldsymbol{B}, p_\mathsf{t}\mu_{\mathsf{t},1}),$$

with $\boldsymbol{A} = \boldsymbol{\Sigma}^{-1/2}\boldsymbol{\beta}_*\boldsymbol{\beta}_*^\top\boldsymbol{\Sigma}^{-1/2}, \boldsymbol{B} = \boldsymbol{\Sigma}^{-1/2}\boldsymbol{\Lambda}_0\boldsymbol{\Sigma}^{-1/2}$.

By Theorem 6, we have

$$|(p_\mathsf{t}\mu_{\mathsf{t},1})^2\Phi_3(\boldsymbol{F}_\mathsf{t}; \boldsymbol{A}, \boldsymbol{B}) - \mu_{\mathsf{t},2}^2\Psi_2(\mu_{\mathsf{t},2}; \boldsymbol{A}, \boldsymbol{B})| \le C_*\mathcal{E}_{n_\mathsf{t},p_\mathsf{t}}\mu_{\mathsf{t},2}^2\widetilde{\Psi}_2(\mu_{\mathsf{t},2}; \boldsymbol{A}, \boldsymbol{B}),$$

where

$$\begin{aligned}
\mu_{\mathsf{t},2}^2\Psi_2(\mu_{\mathsf{t},2}; \boldsymbol{A}, \boldsymbol{B}) =& \mu_{\mathsf{t},2}^2\boldsymbol{\beta}_*^\top(\boldsymbol{\Sigma} + \mu_{\mathsf{t},2})^{-1}\boldsymbol{\Lambda}_0(\boldsymbol{\Sigma} + \mu_{\mathsf{t},2})^{-1}\boldsymbol{\beta}_* + \mu_{\mathsf{t},2}^2\frac{\mathrm{Tr}(\boldsymbol{\Lambda}_0\boldsymbol{\Sigma}(\boldsymbol{\Sigma} + \mu_{\mathsf{t},2})^{-2})}{p_\mathsf{t} - \mathrm{Tr}(\boldsymbol{\Sigma}^2(\boldsymbol{\Sigma} + \mu_{\mathsf{t},2})^{-2})}\boldsymbol{\beta}_*^\top\boldsymbol{\Sigma}(\boldsymbol{\Sigma} + \mu_{\mathsf{t},2})^{-2}\boldsymbol{\beta}_* \\
=& \mu_{\mathsf{t},2}^2\boldsymbol{\beta}_*^\top(\boldsymbol{\Sigma} + \mu_{\mathsf{t},2})^{-1}\boldsymbol{\Lambda}_0(\boldsymbol{\Sigma} + \mu_{\mathsf{t},2})^{-1}\boldsymbol{\beta}_* + \mu_{\mathsf{t},2}^2\chi_{n_\mathsf{t},p_\mathsf{t}}(\boldsymbol{\Lambda}_0; \mu_{\mathsf{t},2})\boldsymbol{\beta}_*^\top\boldsymbol{\Sigma}(\boldsymbol{\Sigma} + \mu_{\mathsf{t},2})^{-2}\boldsymbol{\beta}_*, \\
\mu_{\mathsf{t},2}^2\widetilde{\Psi}_2(\mu_{\mathsf{t},2}; \boldsymbol{A}, \boldsymbol{B}) =& \mu_{\mathsf{t},2}^2\boldsymbol{\beta}_*^\top(\boldsymbol{\Sigma} + \mu_{\mathsf{t},2})^{-1}\boldsymbol{\beta}_*\|\boldsymbol{\Lambda}_0(\boldsymbol{\Sigma} + \mu_{\mathsf{t},2})^{-1}\|_{op} + \mu_{\mathsf{t},2}^2\frac{\mathrm{Tr}(\boldsymbol{\Lambda}_0\boldsymbol{\Sigma}(\boldsymbol{\Sigma} + \mu_{\mathsf{t},2})^{-2})}{p_\mathsf{t} - \mathrm{Tr}(\boldsymbol{\Sigma}^2(\boldsymbol{\Sigma} + \mu_{\mathsf{t},2})^{-2})}\boldsymbol{\beta}_*^\top\boldsymbol{\Sigma}(\boldsymbol{\Sigma} + \mu_{\mathsf{t},2})^{-2}\boldsymbol{\beta}_* \\
=& \mu_{\mathsf{t},2}^2\boldsymbol{\beta}_*^\top(\boldsymbol{\Sigma} + \mu_{\mathsf{t},2})^{-1}\boldsymbol{\beta}_*\|\boldsymbol{\Lambda}_0(\boldsymbol{\Sigma} + \mu_{\mathsf{t},2})^{-1}\|_{op} + \mu_{\mathsf{t},2}^2\chi_{n_\mathsf{t},p_\mathsf{t}}(\boldsymbol{\Lambda}_0; \mu_{\mathsf{t},2})\boldsymbol{\beta}_*^\top\boldsymbol{\Sigma}(\boldsymbol{\Sigma} + \mu_{\mathsf{t},2})^{-2}\boldsymbol{\beta}_*.
\end{aligned}$$

Combining all terms, we have

$$\begin{aligned}
&\left|\boldsymbol{\beta}_{\boldsymbol{F}_\mathsf{t}}^\top\hat{\boldsymbol{\Sigma}}_{\boldsymbol{F}_\mathsf{t}}(\hat{\boldsymbol{\Sigma}}_{\boldsymbol{F}_\mathsf{t}} + \mu_{\mathsf{t},1})^{-1}\frac{\boldsymbol{F}_\mathsf{t}\boldsymbol{\Lambda}_0\boldsymbol{F}_\mathsf{t}^\top}{p_\mathsf{t}}(\hat{\boldsymbol{\Sigma}}_{\boldsymbol{F}_\mathsf{t}} + \mu_{\mathsf{t},1})^{-1}\hat{\boldsymbol{\Sigma}}_{\boldsymbol{F}_\mathsf{t}}\boldsymbol{\beta}_{\boldsymbol{F}_\mathsf{t}}\right. \\
&\left. - \boldsymbol{\beta}_*^\top\boldsymbol{\Sigma}(\boldsymbol{\Sigma} + \mu_{\mathsf{t},2})^{-1}\boldsymbol{\Lambda}_0(\boldsymbol{\Sigma} + \mu_{\mathsf{t},2})^{-1}\boldsymbol{\Sigma}\boldsymbol{\beta}_* + \mu_{\mathsf{t},2}^2\chi_{n_\mathsf{t},p_\mathsf{t}}(\boldsymbol{\Lambda}_0; \mu_{\mathsf{t},2})\boldsymbol{\beta}_*^\top\boldsymbol{\Sigma}(\boldsymbol{\Sigma} + \mu_{\mathsf{t},2})^{-2}\boldsymbol{\beta}_*\right| \\
&\le C_*\mathcal{E}_{n_\mathsf{t},p_\mathsf{t}}\tilde{T}_{q,1},
\end{aligned}$$

where

$$\begin{aligned}
\tilde{T}_{q,1} =& \mu_{\mathsf{t},2}^2\boldsymbol{\beta}_*^\top(\boldsymbol{\Sigma} + \mu_{\mathsf{t},2})^{-1}\boldsymbol{\beta}_*\|\boldsymbol{\Lambda}_0(\boldsymbol{\Sigma} + \mu_{\mathsf{t},2})^{-1}\|_{op} + \mu_{\mathsf{t},2}^2\chi_{n_\mathsf{t},p_\mathsf{t}}(\boldsymbol{\Lambda}_0; \mu_{\mathsf{t},2})\boldsymbol{\beta}_*^\top\boldsymbol{\Sigma}(\boldsymbol{\Sigma} + \mu_{\mathsf{t},2})^{-2}\boldsymbol{\beta}_* \\
&+ \mu_{\mathsf{t},2}\sqrt{\boldsymbol{\beta}_*^\top(\boldsymbol{\Sigma} + \mu_{\mathsf{t},2})^{-1}\boldsymbol{\beta}_*\boldsymbol{\beta}_*^\top\boldsymbol{\Lambda}_0(\boldsymbol{\Sigma} + \mu_{\mathsf{t},2})^{-1}\boldsymbol{\Lambda}_0\boldsymbol{\beta}_*}.
\end{aligned}$$

For the term $\frac{(p_\mathsf{t}\mu_{\mathsf{t},1})^2\mathrm{Tr}(\boldsymbol{\Lambda}_0(\boldsymbol{F}_\mathsf{t}^\top\boldsymbol{F}_\mathsf{t})^2(\boldsymbol{F}_\mathsf{t}^\top\boldsymbol{F}_\mathsf{t}+p_\mathsf{t}\mu_{\mathsf{t},1})^{-2})}{n_\mathsf{t} - \mathrm{Tr}(\boldsymbol{F}_\mathsf{t}^\top\boldsymbol{F}_\mathsf{t}(\boldsymbol{F}_\mathsf{t}^\top\boldsymbol{F}_\mathsf{t}+p_\mathsf{t}\mu_{\mathsf{t},1})^{-1})}\boldsymbol{\beta}_{\boldsymbol{F}_\mathsf{t}}^\top(\boldsymbol{F}_\mathsf{t}^\top\boldsymbol{F}_\mathsf{t}+p_\mathsf{t}\mu_{\mathsf{t},1})^{-2}\boldsymbol{\beta}_{\boldsymbol{F}_\mathsf{t}}$, we first note that from (Defilippis et al., 2024, Proof of Theorem B.12), we have:

$$\begin{aligned}
&\left|\frac{(p_\mathsf{t}\mu_{\mathsf{t},1})^2\boldsymbol{\beta}_*^\top(\boldsymbol{F}_\mathsf{t}^\top\boldsymbol{F}_\mathsf{t}+p_\mathsf{t}\mu_{\mathsf{t},1})^{-2}\boldsymbol{\beta}_*}{n_\mathsf{t} - \mathrm{Tr}(\boldsymbol{F}_\mathsf{t}^\top\boldsymbol{F}_\mathsf{t}(\boldsymbol{F}_\mathsf{t}^\top\boldsymbol{F}_\mathsf{t}+p_\mathsf{t}\mu_{\mathsf{t},1})^{-1})} - \frac{1}{n_\mathsf{t}}\frac{\tilde{\mathsf{B}}_{n_\mathsf{t},p_\mathsf{t}}}{1 - \Upsilon(\mu_{\mathsf{t},1}, \mu_{\mathsf{t},2})}\right| \\
&\qquad\qquad\qquad \le C_*\mathcal{E}_{n_\mathsf{t},p_\mathsf{t}}\frac{1}{n_\mathsf{t}}\frac{\tilde{\mathsf{B}}_{n_\mathsf{t},p_\mathsf{t}}}{1 - \Upsilon(\mu_{\mathsf{t},1}, \mu_{\mathsf{t},2})}
\end{aligned}$$

with

$$\tilde{\mathsf{B}}_{n_\mathsf{t},p_\mathsf{t}} = \mu_{\mathsf{t},2}^2(\boldsymbol{\beta}_*^\top(\boldsymbol{\Sigma} + \mu_{\mathsf{t},2})^{-2}\boldsymbol{\beta}_* + \chi(\mu_{\mathsf{t},2})\boldsymbol{\beta}_*^\top\boldsymbol{\Sigma}(\boldsymbol{\Sigma} + \mu_{\mathsf{t},2})^{-2}\boldsymbol{\beta}_*).$$

We then do the deterministic equivalent of $\mathrm{Tr}(\boldsymbol{\Lambda}_0(\boldsymbol{F}_\mathsf{t}^\top\boldsymbol{F}_\mathsf{t})^2(\boldsymbol{F}_\mathsf{t}^\top\boldsymbol{F}_\mathsf{t} + p_\mathsf{t}\mu_{\mathsf{t},1})^{-2})$. We have

$$\mathrm{Tr}(\boldsymbol{\Lambda}_0(\boldsymbol{F}_\mathsf{t}^\top\boldsymbol{F}_\mathsf{t})^2(\boldsymbol{F}_\mathsf{t}^\top\boldsymbol{F}_\mathsf{t} + p_\mathsf{t}\mu_{\mathsf{t},1})^{-2}) = \mathrm{Tr}(\boldsymbol{\Lambda}_0(\boldsymbol{F}_\mathsf{t}^\top\boldsymbol{F}_\mathsf{t})(\boldsymbol{F}_\mathsf{t}^\top\boldsymbol{F}_\mathsf{t} + p_\mathsf{t}\mu_{\mathsf{t},1})^{-1}) - (p_\mathsf{t}\mu_{\mathsf{t},1})\mathrm{Tr}(\boldsymbol{\Lambda}_0(\boldsymbol{F}_\mathsf{t}^\top\boldsymbol{F}_\mathsf{t})(\boldsymbol{F}_\mathsf{t}^\top\boldsymbol{F}_\mathsf{t} + p_\mathsf{t}\mu_{\mathsf{t},1})^{-2})$$

For the first term, we decompose

$$\mathrm{Tr}(\boldsymbol{\Lambda}_0(\boldsymbol{F}_\mathsf{t}^\top\boldsymbol{F}_\mathsf{t})(\boldsymbol{F}_\mathsf{t}^\top\boldsymbol{F}_\mathsf{t} + p_\mathsf{t}\mu_{\mathsf{t},1})^{-1}) = \mathrm{Tr}(\boldsymbol{\Lambda}_0) - (p_\mathsf{t}\mu_{\mathsf{t},1})\mathrm{Tr}(\boldsymbol{\Lambda}_0(\boldsymbol{F}_\mathsf{t}^\top\boldsymbol{F}_\mathsf{t} + p_\mathsf{t}\mu_{\mathsf{t},1})^{-1}),$$

and by Theorem 5 we have

$$|(p_\mathsf{t}\mu_{\mathsf{t},1})\mathrm{Tr}(\boldsymbol{\Lambda}_0(\boldsymbol{F}_\mathsf{t}^\top\boldsymbol{F}_\mathsf{t} + p_\mathsf{t}\mu_{\mathsf{t},1})^{-1}) - \mu_{\mathsf{t},2}\mathrm{Tr}(\boldsymbol{\Lambda}_0(\boldsymbol{\Sigma} + \mu_{\mathsf{t},2})^{-1})| \le C_*\mathcal{E}_{n_\mathsf{t},p_\mathsf{t}}\mu_{\mathsf{t},2}\mathrm{Tr}(\boldsymbol{\Lambda}_0(\boldsymbol{\Sigma} + \mu_{\mathsf{t},2})^{-1}).$$

For the second term, from Theorem 5 we have

$$\left|(p_{\mathsf{t}}\mu_{\mathsf{t},1})\mathrm{Tr}(\boldsymbol{\Lambda}_0(\boldsymbol{F}_{\mathsf{t}}^{\top}\boldsymbol{F}_{\mathsf{t}})(\boldsymbol{F}_{\mathsf{t}}^{\top}\boldsymbol{F}_{\mathsf{t}}+p_{\mathsf{t}}\mu_{\mathsf{t},1})^{-2}) - (p_{\mathsf{t}}\mu_{\mathsf{t},1})\frac{\mathrm{Tr}(\boldsymbol{\Lambda}_0\boldsymbol{\Sigma}(\boldsymbol{\Sigma}+\mu_{\mathsf{t},2})^{-2})}{p_{\mathsf{t}}-\mathrm{Tr}(\boldsymbol{\Sigma}^2(\boldsymbol{\Sigma}+\mu_{\mathsf{t},2})^{-2})}\right|$$
$$\leq C_*\mathcal{E}_{n_{\mathsf{t}},p_{\mathsf{t}}}(p_{\mathsf{t}}\mu_{\mathsf{t},1})\frac{\mathrm{Tr}(\boldsymbol{\Lambda}_0\boldsymbol{\Sigma}(\boldsymbol{\Sigma}+\mu_{\mathsf{t},2})^{-2})}{p_{\mathsf{t}}-\mathrm{Tr}(\boldsymbol{\Sigma}^2(\boldsymbol{\Sigma}+\mu_{\mathsf{t},2})^{-2})}.$$

Using the same arguments in (Defilippis et al., 2024, Proof of Lemma B.14), we have

$$(p_{\mathsf{t}}\mu_{\mathsf{t},1})\frac{\mathrm{Tr}(\boldsymbol{\Lambda}_0\boldsymbol{\Sigma}(\boldsymbol{\Sigma}+\mu_{\mathsf{t},2})^{-2})}{p_{\mathsf{t}}-\mathrm{Tr}(\boldsymbol{\Sigma}^2(\boldsymbol{\Sigma}+\mu_{\mathsf{t},2})^{-2})} \leq \mu_{\mathsf{t},2}\mathrm{Tr}(\boldsymbol{\Lambda}_0\boldsymbol{\Sigma}(\boldsymbol{\Sigma}+\mu_{\mathsf{t},2})^{-2})) \leq \mu_{\mathsf{t},2}\mathrm{Tr}(\boldsymbol{\Lambda}_0(\boldsymbol{\Sigma}+\mu_{\mathsf{t},2})^{-1}).$$

Thus, combining all terms, we obtain

$$\left|\mathrm{Tr}(\boldsymbol{\Lambda}_0(\boldsymbol{F}_{\mathsf{t}}^{\top}\boldsymbol{F}_{\mathsf{t}})^2(\boldsymbol{F}_{\mathsf{t}}^{\top}\boldsymbol{F}_{\mathsf{t}}+p_{\mathsf{t}}\mu_{\mathsf{t},1})^{-2}) - n_{\mathsf{t}}\Upsilon_{n_{\mathsf{t}},p_{\mathsf{t}}}(\boldsymbol{\Lambda}_0;\mu_{\mathsf{t},1},\mu_{\mathsf{t},1})\right|$$
$$\leq C_*\mathcal{E}_{n_{\mathsf{t}},p_{\mathsf{t}}}\mu_{\mathsf{t},2}\mathrm{Tr}(\boldsymbol{\Lambda}_0(\boldsymbol{\Sigma}+\mu_{\mathsf{t},2})^{-1}), \tag{22}$$

which implies that

$$\left|\frac{(p_{\mathsf{t}}\mu_{\mathsf{t},1})^2\mathrm{Tr}(\boldsymbol{\Lambda}_0(\boldsymbol{F}_{\mathsf{t}}^{\top}\boldsymbol{F}_{\mathsf{t}})^2(\boldsymbol{F}_{\mathsf{t}}^{\top}\boldsymbol{F}_{\mathsf{t}}+p_{\mathsf{t}}\mu_{\mathsf{t},1})^{-2})}{n_{\mathsf{t}}-\mathrm{Tr}(\boldsymbol{F}_{\mathsf{t}}^{\top}\boldsymbol{F}_{\mathsf{t}}(\boldsymbol{F}_{\mathsf{t}}^{\top}\boldsymbol{F}_{\mathsf{t}}+p_{\mathsf{t}}\mu_{\mathsf{t},1})^{-1})}\boldsymbol{\beta}_{\boldsymbol{F}_{\mathsf{t}}}^{\top}(\boldsymbol{F}_{\mathsf{t}}^{\top}\boldsymbol{F}_{\mathsf{t}}+p_{\mathsf{t}}\mu_{\mathsf{t},1})^{-2}\boldsymbol{\beta}_{\boldsymbol{F}_{\mathsf{t}}} - \frac{\tilde{\mathsf{B}}_{n_{\mathsf{t}},p_{\mathsf{t}}}}{1-\Upsilon(\mu_{\mathsf{t},1},\mu_{\mathsf{t},2})}\Upsilon_{n_{\mathsf{t}},p_{\mathsf{t}}}(\boldsymbol{\Lambda}_0;\mu_{\mathsf{t},1},\mu_{\mathsf{t},1})\right|$$
$$\leq C_*\mathcal{E}_{n_{\mathsf{t}},p_{\mathsf{t}}}\frac{\tilde{\mathsf{B}}_{n_{\mathsf{t}},p_{\mathsf{t}}}}{1-\Upsilon(\mu_{\mathsf{t},1},\mu_{\mathsf{t},2})}\frac{1}{n_{\mathsf{t}}}\mu_{\mathsf{t},2}\mathrm{Tr}(\boldsymbol{\Lambda}_0(\boldsymbol{\Sigma}+\mu_{\mathsf{t},2})^{-1}).$$

**Cross term.** Let us now consider the following cross term:

$$\frac{1}{\sqrt{p_{\mathsf{t}}}}\langle\boldsymbol{\beta}_*,\boldsymbol{\Lambda}_1\boldsymbol{F}_{\mathsf{t}}^{\top}(\boldsymbol{Z}_{\mathsf{t}}^{\top}\boldsymbol{Z}_{\mathsf{t}}+\lambda_{\mathsf{t}})^{-1}\boldsymbol{Z}_{\mathsf{t}}^{\top}\boldsymbol{Z}_{\mathsf{t}}\boldsymbol{\beta}_{\boldsymbol{F}_{\mathsf{t}}}\rangle$$
$$+\frac{1}{\sqrt{p_{\mathsf{t}}}}\langle\boldsymbol{\beta}_*,\boldsymbol{\Lambda}_1\boldsymbol{F}_{\mathsf{t}}^{\top}(\boldsymbol{Z}_{\mathsf{t}}^{\top}\boldsymbol{Z}_{\mathsf{t}}+\lambda_{\mathsf{t}})^{-1}\boldsymbol{Z}_{\mathsf{t}}^{\top}\boldsymbol{R}_{\mathsf{t}}\boldsymbol{\beta}_{\perp,\boldsymbol{F}_{\mathsf{t}}},$$

with $\boldsymbol{\Lambda}_1 = \boldsymbol{\Sigma}(\boldsymbol{\Sigma}+\mu_{\mathsf{s},2})^{-1}$.

For the first term, we write

$$\frac{1}{\sqrt{p_{\mathsf{t}}}}\langle\boldsymbol{\beta}_*,\boldsymbol{\Lambda}_1\boldsymbol{F}_{\mathsf{t}}^{\top}(\boldsymbol{Z}_{\mathsf{t}}^{\top}\boldsymbol{Z}_{\mathsf{t}}+\lambda_{\mathsf{t}})^{-1}\boldsymbol{Z}_{\mathsf{t}}^{\top}\boldsymbol{Z}_{\mathsf{t}}\boldsymbol{\beta}_{\boldsymbol{F}_{\mathsf{t}}}\rangle = \frac{1}{\sqrt{p_{\mathsf{t}}}}\Phi_2(\boldsymbol{Z}_{\mathsf{t}};\boldsymbol{v}\boldsymbol{u}^{\top}),$$

with

$$\boldsymbol{u} = \hat{\boldsymbol{\Sigma}}_{\boldsymbol{F}_{\mathsf{t}}}^{1/2}\boldsymbol{\beta}_{\boldsymbol{F}_{\mathsf{t}}}, \quad \boldsymbol{v} = \hat{\boldsymbol{\Sigma}}_{\boldsymbol{F}_{\mathsf{t}}}^{-1/2}\boldsymbol{F}_{\mathsf{t}}\boldsymbol{\Lambda}_1\boldsymbol{\beta}_*,$$

and by Theorem 6 we obtain that

$$|\frac{1}{\sqrt{p_{\mathsf{t}}}}\Phi_2(\boldsymbol{Z}_{\mathsf{t}};\boldsymbol{v}\boldsymbol{u}^{\top}) - \frac{1}{\sqrt{p_{\mathsf{t}}}}\Psi_1(\mu_{\mathsf{t},1};\boldsymbol{v}\boldsymbol{u}^{\top})| \leq C_*\mathcal{E}_{n_{\mathsf{t}},p_{\mathsf{t}}}\frac{1}{\sqrt{p_{\mathsf{t}}}}\tilde{\Psi}_1(\boldsymbol{Z}_{\mathsf{t}};\boldsymbol{v}\boldsymbol{u}^{\top}),$$

with

$$\frac{1}{\sqrt{p_{\mathsf{t}}}}\Psi_1(\mu_{\mathsf{t},1};\boldsymbol{v}\boldsymbol{u}^{\top}) = \langle\boldsymbol{\beta}_*,\boldsymbol{\Lambda}_1\boldsymbol{F}_{\mathsf{t}}^{\top}(\hat{\boldsymbol{\Sigma}}_{\mathsf{t}}+\mu_{\mathsf{s},1})^{-1}\hat{\boldsymbol{\Sigma}}_{\mathsf{t}}\boldsymbol{\beta}_{\boldsymbol{F}_{\mathsf{t}}}\rangle,$$
$$\frac{1}{\sqrt{p_{\mathsf{t}}}}\tilde{\Psi}_1(\mu_{\mathsf{t},1};\boldsymbol{v}\boldsymbol{u}^{\top}) = \|(\boldsymbol{I}-\mathsf{P}_{\perp,\boldsymbol{F}_{\mathsf{t}}})\|_2\sqrt{\langle\boldsymbol{\beta}_*,\boldsymbol{\Lambda}_1(\boldsymbol{F}_{\mathsf{t}}^{\top}\boldsymbol{F}_{\mathsf{t}})^2(\boldsymbol{F}_{\mathsf{t}}^{\top}\boldsymbol{F}_{\mathsf{t}}+p_{\mathsf{t}}\mu_{\mathsf{t},1})^{-2}\boldsymbol{\Lambda}_1\boldsymbol{\beta}_*\rangle}$$
$$\vee\sqrt{\langle\boldsymbol{\beta}_*,(\boldsymbol{F}_{\mathsf{t}}^{\top}\boldsymbol{F}_{\mathsf{t}})(\boldsymbol{F}_{\mathsf{t}}^{\top}\boldsymbol{F}_{\mathsf{t}}+p_{\mathsf{t}}\mu_{\mathsf{s},1})^{-1}\boldsymbol{\beta}_*\rangle\langle\boldsymbol{\beta}_*,\boldsymbol{\Lambda}_1(\boldsymbol{F}_{\mathsf{t}}^{\top}\boldsymbol{F}_{\mathsf{t}})(\boldsymbol{F}_{\mathsf{t}}^{\top}\boldsymbol{F}_{\mathsf{t}}+p_{\mathsf{t}}\mu_{\mathsf{s},1})^{-1}\boldsymbol{\Lambda}_1\boldsymbol{\beta}_*\rangle}.$$

Then, note that

$$\langle \boldsymbol{\beta}_*, \boldsymbol{\Lambda}_1 \boldsymbol{F}_{\mathsf{t}}^\top (\hat{\boldsymbol{\Sigma}}_{\mathsf{t}} + \mu_{\mathsf{s},1})^{-1} \hat{\boldsymbol{\Sigma}}_{\mathsf{t}} \boldsymbol{\beta}_{\boldsymbol{F}_{\mathsf{t}}} \rangle = \langle \boldsymbol{\beta}_*, \boldsymbol{\Lambda}_1 (\hat{\boldsymbol{\Sigma}}_{\mathsf{t}} + p_{\mathsf{t}} \mu_{\mathsf{t},1})^{-1} (\boldsymbol{F}_{\mathsf{t}}^\top \boldsymbol{F}_{\mathsf{t}}) (\boldsymbol{I} - \mathsf{P}_{\perp, \boldsymbol{F}_{\mathsf{t}}}) \boldsymbol{\beta}_* \rangle$$
$$= \langle \boldsymbol{\beta}_*, \boldsymbol{\Lambda}_1 (\boldsymbol{F}_{\mathsf{t}}^\top \boldsymbol{F}_{\mathsf{t}} + p_{\mathsf{t}} \mu_{\mathsf{t},1})^{-1} (\boldsymbol{F}_{\mathsf{t}}^\top \boldsymbol{F}_{\mathsf{t}}) \boldsymbol{\beta}_* \rangle.$$

Again by Theorem 5, we have

$$|\langle \boldsymbol{\beta}_*, \boldsymbol{\Lambda}_1 (\boldsymbol{F}_{\mathsf{t}}^\top \boldsymbol{F}_{\mathsf{t}} + p_{\mathsf{t}} \mu_{\mathsf{t},1})^{-1} (\boldsymbol{F}_{\mathsf{t}}^\top \boldsymbol{F}_{\mathsf{t}}) \boldsymbol{\beta}_* \rangle - \langle \boldsymbol{\beta}_*, \boldsymbol{\Lambda}_1 (\boldsymbol{\Sigma} + \mu_{\mathsf{t},2})^{-1} \boldsymbol{\Sigma} \boldsymbol{\beta}_* \rangle|$$
$$\leq C_* \mathcal{E}_{n_{\mathsf{t}}, p_{\mathsf{t}}} \mu_{\mathsf{t},2} \sqrt{\boldsymbol{\beta}_*^\top \boldsymbol{\Lambda}_1 (\boldsymbol{\Sigma} + \mu_{\mathsf{t},2})^{-1} \boldsymbol{\Lambda}_1 \boldsymbol{\beta}_* \boldsymbol{\beta}_*^\top (\boldsymbol{\Sigma} + \mu_{\mathsf{t},2})^{-1} \boldsymbol{\beta}_*}.$$

**Combining all the terms.** We define

$$\mathsf{B}_{\mathsf{s},bias} = \|\boldsymbol{\beta}_*\|_2^2 + \boldsymbol{\beta}_*^\top \boldsymbol{\Sigma} (\boldsymbol{\Sigma} + \mu_{\mathsf{t},2})^{-1} \boldsymbol{\Lambda}_0 (\boldsymbol{\Sigma} + \mu_{\mathsf{t},2})^{-1} \boldsymbol{\Sigma} \boldsymbol{\beta}_* + \mu_{\mathsf{t},2}^2 \chi_{n_{\mathsf{t}}, p_{\mathsf{t}}} (\boldsymbol{\Lambda}_0; \mu_{\mathsf{t},2}) \boldsymbol{\beta}_*^\top \boldsymbol{\Sigma} (\boldsymbol{\Sigma} + \mu_{\mathsf{t},2})^{-2} \boldsymbol{\beta}_*$$
$$+ \frac{\tilde{\mathsf{B}}_{n_{\mathsf{t}}, p_{\mathsf{t}}}}{1 - \Upsilon(\mu_{\mathsf{t},1}, \mu_{\mathsf{t},2})} \Upsilon_{n_{\mathsf{t}}, p_{\mathsf{t}}} (\boldsymbol{\Lambda}_0; \mu_{\mathsf{t},1}, \mu_{\mathsf{t},2})$$
$$- 2 \boldsymbol{\beta}_*^\top \boldsymbol{\Lambda}_1 (\boldsymbol{\Sigma} + \mu_{\mathsf{t},2})^{-1} \boldsymbol{\Sigma} \boldsymbol{\beta}_*$$
$$= \boldsymbol{\beta}_*^\top \boldsymbol{\Lambda} \boldsymbol{\beta}_*,$$

with

$$\boldsymbol{\Lambda} = \boldsymbol{I} - 2 \boldsymbol{\Sigma}^2 (\boldsymbol{\Sigma} + \mu_{\mathsf{s},2})^{-1} (\boldsymbol{\Sigma} + \mu_{\mathsf{t},2})^{-1} + \boldsymbol{\Lambda}_0 \boldsymbol{\Sigma}^2 (\boldsymbol{\Sigma} + \mu_{\mathsf{t},2})^{-2}$$
$$+ \frac{\Upsilon_{n_{\mathsf{t}}, p_{\mathsf{t}}} (\boldsymbol{\Lambda}_0; \mu_{\mathsf{t},1}, \mu_{\mathsf{t},2})}{1 - \Upsilon(\mu_{\mathsf{t},1} \mu_{\mathsf{t},2})} \mu_{\mathsf{t},2}^2 (\boldsymbol{\Sigma} + \mu_{\mathsf{t},2})^{-2}$$
$$+ \left[ \frac{\Upsilon_{n_{\mathsf{t}}, p_{\mathsf{t}}} (\boldsymbol{\Lambda}_0; \mu_{\mathsf{t},1}, \mu_{\mathsf{t},2})}{1 - \Upsilon(\mu_{\mathsf{t},1} \mu_{\mathsf{t},2})} + \chi_{n_{\mathsf{t}}, p_{\mathsf{t}}} (\boldsymbol{\Lambda}_0; \mu_{\mathsf{t},2}) \right] \mu_{\mathsf{t},2}^2 \boldsymbol{\Sigma} (\boldsymbol{\Sigma} + \mu_{\mathsf{t},2})^{-2},$$

and the approximation error is given by

$$\tilde{\mathsf{B}}_{\mathsf{s},bias} (\boldsymbol{Z}_{\mathsf{t}}) = \|\boldsymbol{\beta}_*\|_2^2 \|\boldsymbol{\Lambda}_0\|_{op}$$
$$+ \frac{\mathrm{Tr}(\boldsymbol{\Lambda}_0 (\boldsymbol{F}_{\mathsf{t}}^\top \boldsymbol{F}_{\mathsf{t}})^2 (\boldsymbol{F}_{\mathsf{t}}^\top \boldsymbol{F}_{\mathsf{t}} + \mu_{\mathsf{t},1})^{-2})}{n_{\mathsf{t}} - \mathrm{Tr}((\boldsymbol{F}_{\mathsf{t}}^\top \boldsymbol{F}_{\mathsf{t}})^2 (\boldsymbol{F}_{\mathsf{t}}^\top \boldsymbol{F}_{\mathsf{t}} + \mu_{\mathsf{t},1})^{-2})}$$
$$+ \mu_{\mathsf{t},2}^2 \boldsymbol{\beta}_*^\top (\boldsymbol{\Sigma} + \mu_{\mathsf{t},2})^{-1} \boldsymbol{\beta}_* \|\boldsymbol{\Lambda}_0 (\boldsymbol{\Sigma} + \mu_{\mathsf{t},2})^{-1}\|_{op} + \mu_{\mathsf{t},2}^2 \chi_{n_{\mathsf{t}}, p_{\mathsf{t}}} (\boldsymbol{\Lambda}_0; \mu_{\mathsf{t},2}) \boldsymbol{\beta}_*^\top \boldsymbol{\Sigma} (\boldsymbol{\Sigma} + \mu_{\mathsf{t},2})^{-2} \boldsymbol{\beta}_*$$
$$+ \mu_{\mathsf{t},2} \sqrt{\boldsymbol{\beta}_*^\top (\boldsymbol{\Sigma} + \mu_{\mathsf{t},2})^{-1} \boldsymbol{\beta}_* \boldsymbol{\beta}_*^\top \boldsymbol{\Lambda}_0 (\boldsymbol{\Sigma} + \mu_{\mathsf{t},2})^{-1} \boldsymbol{\Lambda}_0 \boldsymbol{\beta}_*}$$
$$+ \frac{\tilde{\mathsf{B}}_{n_{\mathsf{t}}, p_{\mathsf{t}}}}{1 - \Upsilon(\mu_{\mathsf{t},1}, \mu_{\mathsf{t},2})} \frac{1}{n_{\mathsf{t}}} \mu_{\mathsf{t},2} \mathrm{Tr}(\boldsymbol{\Lambda}_0 (\boldsymbol{\Sigma} + \mu_{\mathsf{t},2})^{-1})$$
$$+ \|(\boldsymbol{I} - \mathsf{P}_{\perp, \boldsymbol{F}_{\mathsf{t}}})\|_2 \sqrt{\langle \boldsymbol{\beta}_*, \boldsymbol{\Lambda}_1 (\boldsymbol{F}_{\mathsf{t}}^\top \boldsymbol{F}_{\mathsf{t}})^2 (\boldsymbol{F}_{\mathsf{t}}^\top \boldsymbol{F}_{\mathsf{t}} + p_{\mathsf{t}} \mu_{\mathsf{t},1})^{-2} \boldsymbol{\Lambda}_1 \boldsymbol{\beta}_* \rangle}$$
$$\vee \sqrt{\langle \boldsymbol{\beta}_*, (\boldsymbol{F}_{\mathsf{t}}^\top \boldsymbol{F}_{\mathsf{t}}) (\boldsymbol{F}_{\mathsf{t}}^\top \boldsymbol{F}_{\mathsf{t}} + p_{\mathsf{t}} \mu_{\mathsf{s},1})^{-1} \boldsymbol{\beta}_* \rangle \langle \boldsymbol{\beta}_*, \boldsymbol{\Lambda}_1 (\boldsymbol{F}_{\mathsf{t}}^\top \boldsymbol{F}_{\mathsf{t}}) (\boldsymbol{F}_{\mathsf{t}}^\top \boldsymbol{F}_{\mathsf{t}} + p_{\mathsf{t}} \mu_{\mathsf{s},1})^{-1} \boldsymbol{\Lambda}_1 \boldsymbol{\beta}_* \rangle}$$
$$+ \mu_{\mathsf{t},2} \sqrt{\boldsymbol{\beta}_*^\top \boldsymbol{\Lambda}_1 (\boldsymbol{\Sigma} + \mu_{\mathsf{t},2})^{-1} \boldsymbol{\Lambda}_1 \boldsymbol{\beta}_* \boldsymbol{\beta}_*^\top (\boldsymbol{\Sigma} + \mu_{\mathsf{t},2})^{-1} \boldsymbol{\beta}_*}.$$

Thus, we have

$$|\mathcal{B}_{\mathsf{s},bias} (\boldsymbol{Z}_{\mathsf{t}}) - \overline{\mathsf{B}}_{\mathsf{s},bias}| \leq C_* \mathcal{E}_{n_{\mathsf{t}}, p_{\mathsf{t}}} \tilde{\mathsf{B}}_{\mathsf{s},bias} (\boldsymbol{Z}_{\mathsf{t}}).$$

**Bounding the approximation error.** It remains to bound $\tilde{\mathsf{B}}_{\mathsf{s},bias} (\boldsymbol{Z}_{\mathsf{t}})$. We first note that from the previous computation we have

$$\frac{\mathrm{Tr}(\boldsymbol{\Lambda}_0 (\boldsymbol{F}_{\mathsf{t}}^\top \boldsymbol{F}_{\mathsf{t}})^2 (\boldsymbol{F}_{\mathsf{t}}^\top \boldsymbol{F}_{\mathsf{t}} + \mu_{\mathsf{t},1})^{-2})}{n_{\mathsf{t}} - \mathrm{Tr}((\boldsymbol{F}_{\mathsf{t}}^\top \boldsymbol{F}_{\mathsf{t}})^2 (\boldsymbol{F}_{\mathsf{t}}^\top \boldsymbol{F}_{\mathsf{t}} + \mu_{\mathsf{t},1})^{-2})} \leq C_* \frac{1}{n_{\mathsf{t}}} \frac{\mu_{\mathsf{t},2} \mathrm{Tr}(\boldsymbol{\Lambda}_0 (\boldsymbol{\Sigma} + \mu_{\mathsf{t},2})^{-1})}{1 - \Upsilon(\mu_{\mathsf{t},1}, \mu_{\mathsf{t},2})},$$

and this is the only term that is not necessarily upper bounded by $C_* \|\boldsymbol{\beta}_*\|_2^2 (\|\boldsymbol{\Lambda}_0\|_{op} \vee 1)$. Thus, we write

$$\tilde{\mathsf{B}}_{\mathsf{s},bias} (\boldsymbol{Z}_{\mathsf{t}}) \leq \tilde{\mathsf{B}}_{\mathsf{s},bias} := \|\boldsymbol{\beta}_*\|_2^2 (\|\boldsymbol{\Lambda}_0\|_{op} \vee 1) + \frac{1}{n_{\mathsf{t}}} \frac{\mu_{\mathsf{t},2} \mathrm{Tr}(\boldsymbol{\Lambda}_0 (\boldsymbol{\Sigma} + \mu_{\mathsf{t},2})^{-1})}{1 - \Upsilon(\mu_{\mathsf{t},1}, \mu_{\mathsf{t},2})}.$$

A.2.2. TERMS DEPENDING ON THE TEACHER VARIANCE

In this part, we show the deterministic equivalent of $\mathcal{B}_{\mathsf{t},var}(\boldsymbol{Z}_\mathsf{t})$.

From Theorem 5, we immediately have

$$\left|\mathcal{B}_{\mathsf{t},var}(\boldsymbol{Z}_\mathsf{t}) - \tau_\mathsf{t}^2 \frac{\mathrm{Tr}(\boldsymbol{\Lambda}_0(\boldsymbol{F}_\mathsf{t}^\top \boldsymbol{F}_\mathsf{t})^2(\boldsymbol{F}_\mathsf{t}^\top \boldsymbol{F}_\mathsf{t} + p_\mathsf{t}\mu_{\mathsf{t},1})^{-2})}{n_\mathsf{t} - \mathrm{Tr}((\boldsymbol{F}_\mathsf{t}^\top \boldsymbol{F}_\mathsf{t})^2(\boldsymbol{F}_\mathsf{t}^\top \boldsymbol{F}_\mathsf{t} + p_\mathsf{t}\mu_{\mathsf{t},1})^{-2})}\right| \leq C_* \mathcal{E}_{n_\mathsf{s},p_\mathsf{s}} \tau_\mathsf{t}^2 \frac{\mathrm{Tr}(\boldsymbol{\Lambda}_0(\boldsymbol{F}_\mathsf{t}^\top \boldsymbol{F}_\mathsf{t})^2(\boldsymbol{F}_\mathsf{t}^\top \boldsymbol{F}_\mathsf{t} + p_\mathsf{t}\mu_{\mathsf{t},1})^{-2})}{n_\mathsf{t} - \mathrm{Tr}((\boldsymbol{F}_\mathsf{t}^\top \boldsymbol{F}_\mathsf{t})^2(\boldsymbol{F}_\mathsf{t}^\top \boldsymbol{F}_\mathsf{t} + p_\mathsf{t}\mu_{\mathsf{t},1})^{-2})}.$$

Using (22), we obtain that

$$\left|\tau_\mathsf{t}^2 \frac{\mathrm{Tr}(\boldsymbol{\Lambda}_0(\boldsymbol{F}_\mathsf{t}^\top \boldsymbol{F}_\mathsf{t})^2(\boldsymbol{F}_\mathsf{t}^\top \boldsymbol{F}_\mathsf{t} + p_\mathsf{t}\mu_{\mathsf{t},1})^{-2})}{n_\mathsf{t} - \mathrm{Tr}((\boldsymbol{F}_\mathsf{t}^\top \boldsymbol{F}_\mathsf{t})^2(\boldsymbol{F}_\mathsf{t}^\top \boldsymbol{F}_\mathsf{t} + p_\mathsf{t}\mu_{\mathsf{t},1})^{-2})} - \tau_\mathsf{t}^2 \frac{\Upsilon(\boldsymbol{\Lambda}_0; \mu_{\mathsf{t},1}, \mu_{\mathsf{t},2})}{1 - \Upsilon(\mu_{\mathsf{t},1}, \mu_{\mathsf{t},2})}\right|$$
$$\leq C_* \mathcal{E}_{n_\mathsf{t},p_\mathsf{t}} \frac{1}{n_\mathsf{t}} \frac{\mu_{\mathsf{t},2} \mathrm{Tr}(\boldsymbol{\Lambda}_0(\boldsymbol{\Sigma} + \mu_{\mathsf{t},2})^{-1})}{1 - \Upsilon(\mu_{\mathsf{t},1}, \mu_{\mathsf{t},2})},$$

which concludes the proof of Theorem 2.

# B. Basic deterministic equivalents

We consider a feature vector $\boldsymbol{x} \in \mathbb{R}^d$, with $d \in \mathbb{N} \cup \{\infty\}$. Denote by $\boldsymbol{\Sigma} = \mathbb{E}[\boldsymbol{x}\boldsymbol{x}^\top]$ the covariance of $\boldsymbol{x}$ and by $\xi_1^2 \geq \xi_2^2 \geq \xi_3^2 \geq \cdots$ the eigenvalues of $\boldsymbol{\Sigma}$ in non-increasing order. When $d = \infty$, we assume that $\mathrm{Tr}(\boldsymbol{\Sigma}) < \infty$.

We assume that the feature $\boldsymbol{x}$ satisfies the following Hanson-Wright-type inequality:

**Assumption 3** (Feature concentration). *There exist* $\mathsf{c}_x, \mathsf{C}_x > 0$ *such that for any p.s.d. matrix* $\boldsymbol{A} \in \mathbb{R}^{d \times d}$ *with* $\mathrm{Tr}(\boldsymbol{\Sigma}\boldsymbol{A}) < \infty$, *we have*

$$\mathbb{P}\left(\left|\boldsymbol{x}^\top \boldsymbol{A}\boldsymbol{x} - \mathrm{Tr}(\boldsymbol{\Sigma}\boldsymbol{A})\right| \geq t \cdot \|\boldsymbol{\Sigma}^{1/2}\boldsymbol{A}\boldsymbol{\Sigma}^{1/2}\|_F\right) \leq \mathsf{C}_x \exp\{-\mathsf{c}_x t\}. \tag{23}$$

**Definition 3** (Intrinsic dimension). *For a covariance matrix* $\boldsymbol{\Sigma} \in \mathbb{R}^{d \times d}$ *with eigenvalues in nonincreasing order* $\xi_1^2 \geq \xi_2^2 \geq \xi_3^2 \geq \cdots$, *we define the* intrinsic dimension $r_{\boldsymbol{\Sigma}}(k)$ *at level* $k \in \mathbb{N}$ *of* $\boldsymbol{\Sigma}$ *to be the intrinsic dimension of the covariance matrix* $\boldsymbol{\Sigma}_{\geq k} = \mathrm{diag}(\xi_k^2, \xi_{k+1}^2, \ldots)$, *i.e., the covariance matrix projected orthogonally to the top* $k - 1$ *eigenspaces, which is given by*

$$r_{\boldsymbol{\Sigma}}(k) := \frac{\mathrm{Tr}(\boldsymbol{\Sigma}_{\geq k})}{\|\boldsymbol{\Sigma}_{\geq k}\|_{\mathrm{op}}} = \frac{\sum_{j=k}^d \xi_j^2}{\xi_k^2}.$$

Let $\lambda_* > 0$ be the unique positive solution of the following self-consistency equation

$$n - \frac{\lambda}{\lambda_*} = \mathrm{Tr}(\boldsymbol{\Sigma}(\boldsymbol{\Sigma} + \lambda_*)^{-1}).$$

We further introduce $\mu_* = \frac{\lambda}{\lambda_*}$.

The rate of approximation will depend on the following quantity:

$$\rho_\lambda(k) := 1 + \frac{k \cdot \xi_{\lfloor \eta_* \cdot k \rfloor}^2}{\lambda} \left\{1 + \frac{r_{\boldsymbol{\Sigma}}(\lfloor \eta_* \cdot k \rfloor) \vee k}{k} \log\left(r_{\boldsymbol{\Sigma}}(\lfloor \eta_* \cdot k \rfloor) \vee k\right)\right\},$$

where $\eta_* \in (0, 1/2)$ is a constant that will only depend on the constants in Assumption 3.

We will consider the following functionals of the feature matrix:

$$\Phi_1(\boldsymbol{X}; \boldsymbol{A}, \lambda) := \mathrm{Tr}\left(\boldsymbol{A}\boldsymbol{\Sigma}^{1/2}(\boldsymbol{X}^\top \boldsymbol{X} + \lambda)^{-1}\boldsymbol{\Sigma}^{1/2}\right), \tag{24}$$

$$\Phi_2(\boldsymbol{X}; \boldsymbol{A}, \lambda) := \mathrm{Tr}\left(\boldsymbol{A}\boldsymbol{\Sigma}^{-1/2}\boldsymbol{X}^\top \boldsymbol{X}(\boldsymbol{X}^\top \boldsymbol{X} + \lambda)^{-1}\boldsymbol{\Sigma}^{1/2}\right), \tag{25}$$

$$\Phi_3(\boldsymbol{X}; \boldsymbol{A}, \boldsymbol{B}, \lambda) := \mathrm{Tr}\left(\boldsymbol{A}\boldsymbol{\Sigma}^{1/2}(\boldsymbol{X}^\top \boldsymbol{X} + \lambda)^{-1}\boldsymbol{\Sigma}^{1/2}\boldsymbol{B}\boldsymbol{\Sigma}^{1/2}(\boldsymbol{X}^\top \boldsymbol{X} + \lambda)^{-1}\boldsymbol{\Sigma}^{1/2}\right), \tag{26}$$

$$\Phi_4(\boldsymbol{X}; \boldsymbol{A}, \lambda) := \mathrm{Tr}\left(\boldsymbol{A}\boldsymbol{\Sigma}^{1/2}(\boldsymbol{X}^\top \boldsymbol{X} + \lambda)^{-1}\frac{\boldsymbol{X}^\top \boldsymbol{X}}{n}(\boldsymbol{X}^\top \boldsymbol{X} + \lambda)^{-1}\boldsymbol{\Sigma}^{1/2}\right), \tag{27}$$

$$\Phi_5(\boldsymbol{X}; \boldsymbol{A}, \boldsymbol{B}, \lambda) := \mathrm{Tr}\left(\boldsymbol{A}\boldsymbol{\Sigma}^{-1/2}\boldsymbol{X}^\top \boldsymbol{X}(\boldsymbol{X}^\top \boldsymbol{X} + \lambda)^{-1}\boldsymbol{\Sigma}^{1/2}\boldsymbol{B}\boldsymbol{\Sigma}^{1/2}(\boldsymbol{X}^\top \boldsymbol{X} + \lambda)^{-1}\boldsymbol{X}^\top \boldsymbol{X}\boldsymbol{\Sigma}^{-1/2}\right). \tag{28}$$

The deterministic equivalents of these functionals can be written in terms of:

$$\Psi_1(\lambda_*; \boldsymbol{A}) := \mathrm{Tr}\left(\boldsymbol{A}\boldsymbol{\Sigma}(\boldsymbol{\Sigma} + \lambda_*)^{-1}\right), \tag{29}$$

$$\Psi_2(\lambda_*; \boldsymbol{A}, \boldsymbol{B}) := \frac{1}{n^2} \cdot \mathrm{Tr}\left(\boldsymbol{A}\boldsymbol{\Sigma}(\boldsymbol{\Sigma} + \lambda_*)^{-1}\boldsymbol{B}\boldsymbol{\Sigma}(\boldsymbol{\Sigma} + \lambda_*)^{-1}\right) \tag{30}$$

$$+ \frac{1}{n^2} \cdot \frac{\mathrm{Tr}\left(\boldsymbol{A}\boldsymbol{\Sigma}^2(\boldsymbol{\Sigma} + \lambda_*)^{-2}\right) \mathrm{Tr}\left(\boldsymbol{B}\boldsymbol{\Sigma}^2(\boldsymbol{\Sigma} + \lambda_*)^{-2}\right)}{n - \mathrm{Tr}(\boldsymbol{\Sigma}^2(\boldsymbol{\Sigma} + \lambda_*)^{-2})}, \tag{31}$$

$$\Psi_3(\lambda_*; \boldsymbol{A}) := \mathrm{Tr}(\boldsymbol{A}\boldsymbol{\Sigma}(\boldsymbol{\Sigma} + \lambda_*)^{-1}\boldsymbol{B}\boldsymbol{\Sigma}(\boldsymbol{\Sigma} + \lambda_*)^{-1}) \tag{32}$$

$$+ \frac{\lambda_*^2 \mathrm{Tr}(\boldsymbol{A}\boldsymbol{\Sigma}^2(\boldsymbol{\Sigma} + \lambda_*)^{-2})\mathrm{Tr}(\boldsymbol{B}(\boldsymbol{\Sigma} + \lambda_*)^{-2})}{n - \mathrm{Tr}(\boldsymbol{\Sigma}^2(\boldsymbol{\Sigma} + \lambda_*)^{-2})}. \tag{33}$$

Note that when $\boldsymbol{B} = \boldsymbol{I}$, the second functional simplifies to

$$\Psi_2(\lambda_*; \boldsymbol{A}, \boldsymbol{I}) = \frac{1}{n} \cdot \frac{\mathrm{Tr}\left(\boldsymbol{A}\boldsymbol{\Sigma}^2(\boldsymbol{\Sigma} + \lambda_*)^{-2}\right)}{n - \mathrm{Tr}(\boldsymbol{\Sigma}^2(\boldsymbol{\Sigma} + \lambda_*)^{-2})}.$$

We also define the following approximation functionals:

$$\widetilde{\Psi}_1(\lambda_*; \boldsymbol{v}\boldsymbol{u}^\top) := \sqrt{\|\boldsymbol{u}\|_2^2 \boldsymbol{v}^\top \boldsymbol{\Sigma}^2(\boldsymbol{\Sigma} + \lambda_*)^{-2}\boldsymbol{v}} \vee \sqrt{\boldsymbol{u}^\top \boldsymbol{\Sigma}(\boldsymbol{\Sigma} + \lambda_*)^{-}\boldsymbol{u}\boldsymbol{v}^\top \boldsymbol{\Sigma}(\boldsymbol{\Sigma} + \lambda_*)^{-1}\boldsymbol{v}},$$

$$\widetilde{\Psi}_2(\lambda_*; \boldsymbol{v}\boldsymbol{u}^\top, \boldsymbol{B}) := \frac{1}{n^2} \cdot \sqrt{\boldsymbol{u}^\top \boldsymbol{\Sigma}(\boldsymbol{\Sigma} + \lambda_*)^{-1}\boldsymbol{u}\boldsymbol{v}^\top \boldsymbol{\Sigma}(\boldsymbol{\Sigma} + \lambda_*)^{-1}\boldsymbol{v}}\|\boldsymbol{B}^{1/2}\boldsymbol{\Sigma}(\boldsymbol{\Sigma} + \lambda_*)^{-1}\boldsymbol{B}^{1/2}\|_{op}$$

$$+ \frac{1}{n^2} \cdot \frac{\mathrm{Tr}\left(\boldsymbol{A}\boldsymbol{\Sigma}^2(\boldsymbol{\Sigma} + \lambda_*)^{-2}\right) \mathrm{Tr}\left(\boldsymbol{B}\boldsymbol{\Sigma}^2(\boldsymbol{\Sigma} + \lambda_*)^{-2}\right)}{n - \mathrm{Tr}(\boldsymbol{\Sigma}^2(\boldsymbol{\Sigma} + \lambda_*)^{-2})},$$

$$\widetilde{\Psi}_3(\lambda_*; \boldsymbol{v}\boldsymbol{u}^\top, \boldsymbol{B}) := \|\boldsymbol{u}\|_2 \|\boldsymbol{v}\|_2 \|\boldsymbol{B}\|_{op}.$$

**Theorem 5.** *([Misiakiewicz & Saeed, 2024](), Theorem 6,8,9) Assume the features $(\boldsymbol{x}_i)_{i \in [n]}$ satisfy Assumption 3 with some constants $\mathsf{c}_x, \mathsf{C}_x > 0$. For any $D, K > 0$, there exist constants $\eta := \eta_x \in (0, 1/2)$ (only depending on $\mathsf{c}_x, \mathsf{C}_x$), $C_{D,K} > 0$ (only depending on $D, K$), and $C_{x,K,D} > 0$ (only depending on $\mathsf{c}_x, \mathsf{C}_x, D, K$) such that the following holds. For all $n \geq C_{D,K}$ and $\lambda > 0$, if it holds that*

$$\lambda \cdot \rho_\lambda(n) \geq \|\boldsymbol{\Sigma}\|_{\mathrm{op}} \cdot n^{-K}, \qquad \rho_\lambda(n)^{5/2} \log^{3/2}(n) \leq K\sqrt{n}, \tag{34}$$

*then for any p.s.d. matrices $\boldsymbol{A}$, we have with probability at least $1 - n^{-D}$ that*

$$\left|\Phi_1(\boldsymbol{X}; \boldsymbol{A}, \lambda) - \frac{\lambda_*}{\lambda}\Psi_1(\lambda_*; \boldsymbol{A})\right| \leq C_{x,D,K}\frac{\rho_\lambda(n)^{5/2}\log^{3/2}(n)}{\sqrt{n}} \cdot \frac{\lambda_*}{\lambda}\Psi_1(\lambda_*; \boldsymbol{A}), \tag{35}$$

$$\left|\Phi_3(\boldsymbol{X}; \boldsymbol{A}, \boldsymbol{I}, \lambda) - \left(\frac{n\lambda_*}{\lambda}\right)^2\Psi_2(\lambda_*; \boldsymbol{A}, \boldsymbol{I})\right| \leq C_{x,D,K}\frac{\rho_\lambda(n)^6\log^{5/2}(n)}{\sqrt{n}}\left(\frac{n\lambda_*}{\lambda}\right)^2\Psi_2(\lambda_*; \boldsymbol{A}, \boldsymbol{I}), \tag{36}$$

$$\left|\Phi_4(\boldsymbol{X}; \boldsymbol{A}, \lambda) - \Psi_2(\lambda_*; \boldsymbol{A}, \boldsymbol{I})\right| \leq C_{x,D,K}\frac{\rho_\lambda(n)^6\log^{3/2}(n)}{\sqrt{n}}\Psi_2(\lambda_*; \boldsymbol{A}, \boldsymbol{I}). \tag{37}$$

The deterministic equivalents above assume that $\boldsymbol{A}$ is a p.s.d. matrix. In next theorem, instead we take $\boldsymbol{A} = \boldsymbol{v}\boldsymbol{u}^\top$, which is a non-p.s.d. rank-1 matrix.

**Theorem 6.** *Under the same assumptions of Theorem 5, let $\boldsymbol{A} = \boldsymbol{v}\boldsymbol{u}^\top$ and $\boldsymbol{B}$ be a p.s.d. matrix. Then, with probability at*

*least $1 - n^{-D}$, we have*

$$\left|\Phi_1(\boldsymbol{X}; \boldsymbol{A}, \lambda) - \frac{\lambda_*}{\lambda}\Psi_1(\lambda_*; \boldsymbol{A})\right| \le C_* \frac{\rho_\lambda(n)^{5/2}\log^{3/2}(n)}{\sqrt{n}}\frac{\lambda_*}{\lambda}\sqrt{\Psi_1(\lambda_*; \boldsymbol{uu}^\top)\Psi_1(\lambda_*; \boldsymbol{vv}^\top)}, \quad (38)$$

$$\left|\Phi_2(\boldsymbol{X}; \boldsymbol{A}, \lambda) - \Psi_1(\lambda_*; \boldsymbol{A})\right| \le C_{x,D,K}\frac{\rho_\lambda(n)^{5/2}\log^{3/2}(n)}{\sqrt{n}}\tilde{\Psi}_1(\lambda_*; \boldsymbol{vu}^\top), \quad (39)$$

$$\left|\Phi_3(\boldsymbol{X}; \boldsymbol{A}, \boldsymbol{B}, \lambda) - \left(\frac{n\lambda_*}{\lambda}\right)^2\Psi_2(\lambda_*; \boldsymbol{A}, \boldsymbol{B})\right| \le C_*\frac{\rho_\lambda^6(n)\log^{5/2}(n)}{\sqrt{n}}\cdot\left(\frac{n\lambda_*}{\lambda}\right)^2\widetilde{\Psi}_2(\lambda_*; \boldsymbol{A}, \boldsymbol{B}), \quad (40)$$

$$\left|\Phi_5(\boldsymbol{X}; \boldsymbol{A}, \boldsymbol{B}, \lambda) - \Psi_3(\lambda_*; \boldsymbol{A}, \boldsymbol{B})\right| \le C_*\frac{\rho_\lambda^6(n)\log^{5/2}(n)}{\sqrt{n}}\tilde{\Psi}_3(\lambda_*; \boldsymbol{A}, \boldsymbol{B}). \quad (41)$$

**Lemma 1.** *Under the same assumptions of Theorem 6, consider a deterministic vector $\boldsymbol{v}$ and a random vector $\boldsymbol{u}$ with i.i.d. entries $u_i$ such that $\mathbb{E}[u_i] = 0, \mathbb{E}[u_i^2] = 1$, and $\mathbb{E}[\boldsymbol{x}_i u_i] = 0$. Then, with probability at least $1 - n^{-D}$, we have*

$$|\langle \boldsymbol{v}, (\boldsymbol{X}^\top\boldsymbol{X} + \lambda)^{-1}\boldsymbol{X}^\top\boldsymbol{u}\rangle| \le C_*\frac{\log(n)\rho_\lambda(n)}{\sqrt{n}}\|\boldsymbol{\Sigma}^{-\frac{1}{2}}\boldsymbol{v}\|_2 \quad (42)$$

$$|\langle \boldsymbol{v}, \boldsymbol{X}^\top\boldsymbol{X}(\boldsymbol{X}^\top\boldsymbol{X} + \lambda)^{-1}\boldsymbol{B}(\boldsymbol{X}^\top\boldsymbol{X} + \lambda)^{-1}\boldsymbol{X}^\top\boldsymbol{u}\rangle| \le \frac{C_*\log(n)\rho_\lambda^2(n)}{n}\|\boldsymbol{\Sigma}^{-\frac{1}{2}}\boldsymbol{B}\boldsymbol{\Sigma}^{-\frac{1}{2}}\|_{op}\|\boldsymbol{\Sigma}^{\frac{1}{2}}\boldsymbol{v}\|_2 \quad (43)$$

$$|\langle \boldsymbol{u}, \boldsymbol{X}(\boldsymbol{X}^\top\boldsymbol{X} + \lambda)^{-1}\boldsymbol{B}(\boldsymbol{X}^\top\boldsymbol{X} + \lambda)^{-1}\boldsymbol{X}^\top\boldsymbol{u}\rangle - n\Psi_2(\lambda_*; \boldsymbol{\Sigma}^{-1/2}\boldsymbol{B}\boldsymbol{\Sigma}^{-1/2})| \le C_*\frac{\log(n)\rho_\lambda^6(n)}{\sqrt{n}}\|\boldsymbol{\Sigma}^{-1/2}\boldsymbol{B}\boldsymbol{\Sigma}^{-1/2}\|_{op}. \quad (44)$$

The proof of Lemma 1 is deferred to Appendix D.

## C. Proof of Theorem 6: Proof of new deterministic equivalent

We introduce the same notations as in (Misiakiewicz & Saeed, 2024, Appendix A). We first define

$$\boldsymbol{M} = \boldsymbol{\Sigma}^{1/2}(\boldsymbol{X}^\top\boldsymbol{X} + \lambda)^{-1}\boldsymbol{\Sigma}^{1/2}, \quad \overline{\boldsymbol{M}} = \boldsymbol{\Sigma}^{1/2}(\mu_*\boldsymbol{\Sigma} + \lambda)^{-1}\boldsymbol{\Sigma}^{1/2}.$$

Then, we define some notation for the leave-one-out arguments. Recalling $\boldsymbol{X} = [\boldsymbol{x}_1, \ldots, \boldsymbol{x}_n]^\top \in \mathbb{R}^{n\times d}$, we define $\boldsymbol{X}_i = [\boldsymbol{x}_1, \ldots, \boldsymbol{x}_{i-1}, \boldsymbol{x}_{i+1}, \ldots, \boldsymbol{x}_n]^\top$ and

$$\boldsymbol{M}_{-i} = \boldsymbol{\Sigma}^{1/2}(\boldsymbol{X}_i^\top\boldsymbol{X}_i + \lambda)^{-1}\boldsymbol{\Sigma}^{1/2}, \quad \overline{\boldsymbol{M}} = \boldsymbol{\Sigma}^{1/2}\left(\frac{n}{1+\kappa}\boldsymbol{\Sigma} + \lambda\right)^{-1}\boldsymbol{\Sigma}^{1/2}, \quad \kappa = \mathbb{E}[\text{Tr}(\boldsymbol{M}_{-i})].$$

When the subscript $i$ is not important, we write $\boldsymbol{M}_{-i}$ as $\boldsymbol{M}_-$ for short.

### C.1. General procedure of proving deterministic equivalent

We follow the same approach as in (Misiakiewicz & Saeed, 2024, Theorem 6). We take the proof of $\Phi_1(\boldsymbol{X}; \boldsymbol{uv}^\top, \lambda)$ as an illustrative example for the proof techniques, and the argument for the other functionals is analogous.

We first decompose the difference between the functional and its deterministic equivalent into two parts:

$$|\Phi_1(\boldsymbol{X}; \boldsymbol{uv}^\top, \lambda) - \Psi_1(\lambda_*; \boldsymbol{uv}^\top)|$$
$$\le \underbrace{|\Phi_1(\boldsymbol{X}; \boldsymbol{uv}^\top, \lambda) - \mathbb{E}_{\boldsymbol{X}}[\Phi_1(\boldsymbol{X}; \boldsymbol{uv}^\top, \lambda)]|}_{\text{Martingale part}} + \underbrace{|\mathbb{E}_{\boldsymbol{X}}[\Phi_1(\boldsymbol{X}; \boldsymbol{uv}^\top, \lambda)] - \Psi_1(\lambda_*; \boldsymbol{uv}^\top)|}_{\text{Deterministic part}}. \quad (45)$$

Then, we bound the two parts separately. For the deterministic part, we directly prove

$$|\mathbb{E}_{\boldsymbol{X}}[\Phi_1(\boldsymbol{X}; \boldsymbol{uv}^\top, \lambda)] - \Psi_1(\lambda_*; \boldsymbol{uv}^\top)| \le C_*\mathcal{E}_n\tilde{\Psi}_1(\lambda_*; \boldsymbol{uv}^\top),$$

with $\mathcal{E}_n = \tilde{\mathcal{O}}_n\left(\frac{1}{\sqrt{n}}\right)$.

For the martingale part, we decompose the difference into a martingale difference sequence, and show high probability bounds for the martingale difference term, as described in (Misiakiewicz & Saeed, 2024, Proof of Proposition 3). We briefly summarize the steps for completeness. In particular, we define

$$S_n := \Phi_1(\boldsymbol{X}; \boldsymbol{uv}^\top, \lambda) - \mathbb{E}_{\boldsymbol{X}}[\Phi_1(\boldsymbol{X}; \boldsymbol{uv}^\top, \lambda)] = \sum_{i=1}^n (\mathbb{E}_i - \mathbb{E}_{i-1})\Phi_1(\boldsymbol{X}; \boldsymbol{uv}^\top, \lambda) = \sum_{i=1}^n \Delta_i,$$

where $\mathbb{E}_i$ denote the partial expectation w.r.t. $\{\boldsymbol{x}_{i+1}, \dots \boldsymbol{x}_n\}$. For any constant $R$, we define the truncated martingale sequence and the remainder as

$$\tilde{S}_n := \sum_{i=1}^n \Delta_i \mathbf{1}_{\Delta_i \in [-R, R]} - \mathbb{E}_{i-1}[\Delta_i \mathbf{1}_{\Delta_i \in [-R, R]}],$$

$$R_n := S_n - \tilde{S}_n = \sum_{i=1}^n \Delta_i \mathbf{1}_{\Delta_i \notin [-R, R]} - \mathbb{E}_{i-1}[\Delta_i \mathbf{1}_{\Delta_i \notin [-R, R]}].$$

By Azuma-Hoeffding inequality, we have

$$\Pr\left[|\tilde{S}_n| \geq t\right] \leq 2\exp\left(-\frac{t^2}{2nR^2}\right).$$

Thus, there exists a constant $C_*$ such that $|\tilde{S}_n| \leq C_* \sqrt{n \log(n)} R$ with probability at least $1 - n^{-D}$. Next, we show that, with probability at least $1 - n^D$,

$$|\Delta_i| = \mathcal{O}\left(\frac{1}{n}\right), \quad \mathbb{E}_{i-1}[\Delta_i^2]^{1/2} = \mathcal{O}\left(\frac{1}{n}\right),$$

which implies $R_n = 0$ with probability at least $1 - n^{-D}$, and finishes the proof.

## C.2. Deterministic equivalent of $\mathrm{Tr}(\boldsymbol{AM})$

### C.2.1. BOUND THE DETERMINISTIC PART

Consider $\boldsymbol{A} = \boldsymbol{uv}^\mathsf{T}$. By following the same steps as in (Misiakiewicz & Saeed, 2024, Proof of Proposition 2), we obtain that

$$\begin{aligned}|\mathbb{E}_{\boldsymbol{X}}[\mathrm{Tr}(\boldsymbol{AM})] - \mathrm{Tr}(\boldsymbol{A}\overline{\boldsymbol{M}})| &= |\mathrm{Tr}(\boldsymbol{A}(\mathbb{E}[\boldsymbol{M}] - \overline{\boldsymbol{M}}))| \\ &\leq |\mathrm{Tr}(\boldsymbol{A}(\mathbb{E}[\boldsymbol{M}] - \overline{\boldsymbol{M}}_-))| + |\mathrm{Tr}(\boldsymbol{A}(\overline{\boldsymbol{M}}_- - \overline{\boldsymbol{M}}))|.\end{aligned} \tag{46}$$

*Bounding the term* $|\mathrm{Tr}(\boldsymbol{A}(\mathbb{E}[\boldsymbol{M}] - \overline{\boldsymbol{M}}_-))|$. We decompose the term as

$$|\mathbb{E}[\mathrm{Tr}(\boldsymbol{A}(\boldsymbol{M} - \overline{\boldsymbol{M}}_-))]| \leq n|\mathbb{E}[\mathrm{Tr}(\boldsymbol{A}\Delta_2)]| + n|\mathbb{E}[\mathrm{Tr}(\boldsymbol{A}\Delta_3)]|,$$

where

$$\Delta_2 := \frac{\boldsymbol{M}_- \boldsymbol{z}\boldsymbol{z}^\top \overline{\boldsymbol{M}}_-}{(1+\kappa)(1+\boldsymbol{z}^\top \boldsymbol{M}_- \boldsymbol{z})}(\boldsymbol{z}^\top \boldsymbol{M}_- \boldsymbol{z} - \kappa),$$

$$\Delta_3 := -\frac{\boldsymbol{M}_- \boldsymbol{z}\boldsymbol{z}^\top \boldsymbol{M}_- \overline{\boldsymbol{M}}_-}{1+\boldsymbol{z}^\top \boldsymbol{M}_- \boldsymbol{z}}(\boldsymbol{z}^\top \boldsymbol{M}_- \boldsymbol{z} - \kappa).$$

To bound $n|\mathbb{E}[\mathrm{Tr}(\boldsymbol{A}\Delta_2)]|$, we bound the numerator as

$$\mathbb{E}[|\boldsymbol{z}^\top \overline{\boldsymbol{M}}_- \boldsymbol{A}\boldsymbol{M}_- \boldsymbol{z}(\boldsymbol{z}^\top \boldsymbol{M}_- \boldsymbol{z} - \kappa)|] \leq \mathbb{E}_{\boldsymbol{z}}[(\boldsymbol{u}^\top \overline{\boldsymbol{M}}_- \boldsymbol{z})^3]^{1/3} \mathbb{E}_{\boldsymbol{M}_-}\left[\mathbb{E}_{\boldsymbol{z}}\left[(\boldsymbol{v}^\top \boldsymbol{M}_- \boldsymbol{z})^3\right]^{\frac{2}{3}}\right]^{\frac{1}{2}} \mathbb{E}_{\boldsymbol{M}_-}\left[\mathbb{E}_{\boldsymbol{z}}\left[(\boldsymbol{z}^\top \boldsymbol{M}_- \boldsymbol{z} - \kappa)^3\right]^{\frac{2}{3}}\right]^{\frac{1}{2}}.$$

We can bound each term using (Misiakiewicz & Saeed, 2024, Lemma 2, Lemma 4) as in (Misiakiewicz & Saeed, 2024, Proof of Proposition 2):

$$\mathbb{E}_{\boldsymbol{z}}[(\boldsymbol{u}^\top \overline{\boldsymbol{M}}_- \boldsymbol{z})^3]^{1/3} \leq C_* \sqrt{\frac{(1+\kappa)\rho_\lambda(n)}{n}\mathrm{Tr}(\boldsymbol{u}\boldsymbol{u}^\top \overline{\boldsymbol{M}})},$$

$$\mathbb{E}_{\boldsymbol{M}_-}\left[\mathbb{E}_{\boldsymbol{z}}\left[(\boldsymbol{v}^\top \boldsymbol{M}_- \boldsymbol{z})^3\right]^{\frac{2}{3}}\right]^{\frac{1}{2}} \leq C_* \frac{\rho_\lambda(n)}{\sqrt{n}}\sqrt{\mathrm{Tr}(\boldsymbol{v}\boldsymbol{v}^\top \overline{\boldsymbol{M}})},$$

$$\mathbb{E}_{\boldsymbol{M}_-}\left[\mathbb{E}_{\boldsymbol{z}}\left[(\boldsymbol{z}^\top \boldsymbol{M}_- \boldsymbol{z} - \kappa)^3\right]^{\frac{2}{3}}\right]^{\frac{1}{2}} \leq C_* \frac{\rho_\lambda(n)}{\sqrt{n}}.$$

Combining all the terms, we have that

$$n|\mathbb{E}[\mathrm{Tr}(\boldsymbol{A}\Delta_2)]| \leq C_* \frac{\rho_\lambda(n)^{\frac{5}{2}}}{\sqrt{n}}\sqrt{\mathrm{Tr}(\boldsymbol{u}\boldsymbol{u}^\top \overline{\boldsymbol{M}})\mathrm{Tr}(\boldsymbol{v}\boldsymbol{v}^\top \overline{\boldsymbol{M}})}.$$

To bound $n|\mathbb{E}[\mathrm{Tr}(\boldsymbol{A}\Delta_3)]|$, we follow the procedure in (Misiakiewicz & Saeed, 2024, eq (79)):

$$\begin{aligned}n|\mathbb{E}[\mathrm{Tr}(\boldsymbol{A}\Delta_3)]| \leq & n\mathbb{E}[(\boldsymbol{v}^\top \boldsymbol{M}_- \boldsymbol{z})^2]^{1/2}\mathbb{E}[(\boldsymbol{u}^\top \overline{\boldsymbol{M}}_- \boldsymbol{M}_- \boldsymbol{z})^2]^{1/2}\\ \leq & C_* \frac{\rho_\lambda(n)^3}{n}\sqrt{\mathrm{Tr}(\boldsymbol{u}\boldsymbol{u}^\top \overline{\boldsymbol{M}})\mathrm{Tr}(\boldsymbol{v}\boldsymbol{v}^\top \overline{\boldsymbol{M}})}\end{aligned}\quad.$$

Combining the two above display, we have

$$|\mathbb{E}[\mathrm{Tr}(\boldsymbol{A}(\boldsymbol{M} - \overline{\boldsymbol{M}}_-))]| \leq C_* \frac{\rho_\lambda(n)^{5/2}}{\sqrt{n}}\sqrt{\mathrm{Tr}(\boldsymbol{u}\boldsymbol{u}^\top \overline{\boldsymbol{M}})\mathrm{Tr}(\boldsymbol{v}\boldsymbol{v}^\top \overline{\boldsymbol{M}})}. \tag{47}$$

*Bounding the term* $|\mathrm{Tr}(\boldsymbol{A}(\overline{\boldsymbol{M}}_- - \overline{\boldsymbol{M}}))|$. We note that

$$\begin{aligned}|\mathrm{Tr}(\boldsymbol{A}(\overline{\boldsymbol{M}}_- - \overline{\boldsymbol{M}}))| = & n|\mathrm{Tr}(\boldsymbol{A}\overline{\boldsymbol{M}}_- \boldsymbol{\Sigma}\overline{\boldsymbol{M}})| \cdot \frac{|\kappa - \mathrm{Tr}(\overline{\boldsymbol{M}})|}{(1+\kappa)(1+\mathrm{Tr}(\overline{\boldsymbol{M}}))}\\ \leq & n\sqrt{\mathrm{Tr}(\boldsymbol{u}\boldsymbol{u}^\top \overline{\boldsymbol{M}}_- \boldsymbol{\Sigma}\overline{\boldsymbol{M}})\mathrm{Tr}(\boldsymbol{v}\boldsymbol{v}^\top \overline{\boldsymbol{M}}_- \boldsymbol{\Sigma}\overline{\boldsymbol{M}})} \cdot \frac{|\kappa - \mathrm{Tr}(\overline{\boldsymbol{M}})|}{(1+\kappa)(1+\mathrm{Tr}(\overline{\boldsymbol{M}}))}\\ \leq & \sqrt{\mathrm{Tr}(\boldsymbol{u}\boldsymbol{u}^\top \overline{\boldsymbol{M}})\mathrm{Tr}(\boldsymbol{v}\boldsymbol{v}^\top \overline{\boldsymbol{M}})} \cdot \frac{|\kappa - \mathrm{Tr}(\overline{\boldsymbol{M}})|}{1+\mathrm{Tr}(\overline{\boldsymbol{M}})},\end{aligned}$$

where in the first inequality we use Cauchy-Schwartz inequality and that $\overline{\boldsymbol{M}}_-, \boldsymbol{\Sigma}, \overline{\boldsymbol{M}}$ commute, and in the second inequality we use that $\|\overline{\boldsymbol{M}}_- \boldsymbol{\Sigma}\|_{op} \leq \frac{1+\kappa}{n}$.

Then we follow the same steps as in (Misiakiewicz & Saeed, 2024, Proof of Proposition 2), and obtain that

$$|\mathrm{Tr}(\boldsymbol{A}(\overline{\boldsymbol{M}}_- - \overline{\boldsymbol{M}}))| \leq C_* \frac{\rho_\lambda(n)^{5/2}}{\sqrt{n}}\sqrt{\mathrm{Tr}(\boldsymbol{u}\boldsymbol{u}^\top \overline{\boldsymbol{M}})\mathrm{Tr}(\boldsymbol{v}\boldsymbol{v}^\top \overline{\boldsymbol{M}})}. \tag{48}$$

Combining (46), (47), (48) we have

$$|\mathbb{E}_{\boldsymbol{X}}[\mathrm{Tr}(\boldsymbol{u}\boldsymbol{v}^\top \boldsymbol{M})] - \mathrm{Tr}(\boldsymbol{u}\boldsymbol{v}^\top \overline{\boldsymbol{M}})| \leq C_* \frac{\rho_\lambda(n)^{5/2}}{\sqrt{n}}\sqrt{\mathrm{Tr}(\boldsymbol{u}\boldsymbol{u}^\top \overline{\boldsymbol{M}})\mathrm{Tr}(\boldsymbol{v}\boldsymbol{v}^\top \overline{\boldsymbol{M}})}. \tag{49}$$

### C.2.2. BOUNDING THE MARTINGALE PART

Following the proof in (Misiakiewicz & Saeed, 2024, Proof of Proposition 3), we decompose the martingale parts as follows:

$$\mathrm{Tr}(\boldsymbol{u}\boldsymbol{v}^\top \boldsymbol{M}) - \mathbb{E}_{\boldsymbol{X}}[\mathrm{Tr}(\boldsymbol{u}\boldsymbol{v}^\top \boldsymbol{M})] = \sum_{i=1}^n \Delta_i,$$

where

$$\Delta_i = (\mathbb{E}_i - \mathbb{E}_{i-1})\mathrm{Tr}(\boldsymbol{A}(\boldsymbol{M} - \boldsymbol{M}_{-i})).$$

*Bounding $|\Delta_i|$ with high probability.* We decompose $\mathrm{Tr}(\boldsymbol{A}(\boldsymbol{M} - \boldsymbol{M}_{-i}))$ in the same way as (Misiakiewicz & Saeed, 2024, eq (86)), and we get

$$\mathrm{Tr}(\boldsymbol{A}(\boldsymbol{M} - \boldsymbol{M}_{-i})) = \frac{\boldsymbol{z}_i^\top \boldsymbol{M}_{-i} \boldsymbol{A} \boldsymbol{M}_{-i} \boldsymbol{z}_i}{1 + \mathrm{Tr}(\boldsymbol{M}_{-i})} - \frac{\boldsymbol{z}_i^\top \boldsymbol{M}_{-i} \boldsymbol{A} \boldsymbol{M}_{-i} \boldsymbol{z}_i (\boldsymbol{z}_i \boldsymbol{M}_{-i} \boldsymbol{z}_i - \mathrm{Tr}(\boldsymbol{M}_{-i}))}{(1 + \mathrm{Tr}(\boldsymbol{M}_{-i}))(1 + \boldsymbol{z}_i^\top \boldsymbol{M}_{-i} \boldsymbol{z}_i)}.$$

For the first term, we have

$$
\begin{aligned}
(\mathbb{E}_i - \mathbb{E}_{i-1})\boldsymbol{z}_i^\top \boldsymbol{M}_{-i} \boldsymbol{A} \boldsymbol{M}_{-i} \boldsymbol{z}_i &= \mathbb{E}_i[\boldsymbol{z}_i^\top \boldsymbol{M}_{-i} \boldsymbol{A} \boldsymbol{M}_{-i} \boldsymbol{z}_i - \mathbb{E}_{\boldsymbol{z}_i}[\boldsymbol{z}_i^\top \boldsymbol{M}_{-i} \boldsymbol{A} \boldsymbol{M}_{-i} \boldsymbol{z}_i]] \\
&= \mathbb{E}_i[\boldsymbol{z}_i^\top \boldsymbol{M}_{-i} \boldsymbol{A} \boldsymbol{M}_{-i} \boldsymbol{z}_i - \mathrm{Tr}(\boldsymbol{M}_{-i} \boldsymbol{A} \boldsymbol{M}_{-i})].
\end{aligned}
$$

By Lemma 2, we have that, with probability at least $1 - n^{-D}$,

$$
\begin{aligned}
\mathbb{E}_i[|\boldsymbol{z}_i^\top \boldsymbol{M}_{-i} \boldsymbol{A} \boldsymbol{M}_{-i} \boldsymbol{z}_i - \mathrm{Tr}(\boldsymbol{M}_{-i} \boldsymbol{A} \boldsymbol{M}_{-i})|] &\leq C_* \log(n) \varphi_1(p) \mathbb{E}_i[\|\boldsymbol{M}_{-i}\left(\frac{\boldsymbol{A} + \boldsymbol{A}^\top}{2}\right)\boldsymbol{M}_{-i})\|_F] \\
&\leq C_* \log(n) \varphi_1(d) \mathbb{E}_i[(\|\boldsymbol{M}_{-i} \boldsymbol{u} \boldsymbol{v}^\top \boldsymbol{M}_{-i}\|_F)].
\end{aligned}
$$

Then, we write

$$
\begin{aligned}
\mathbb{E}_i[(\|\boldsymbol{M}_{-i} \boldsymbol{u} \boldsymbol{v}^\top \boldsymbol{M}_{-i})\|_F] &\leq \mathbb{E}_i[(\|\boldsymbol{M}_{-i} \boldsymbol{u} \boldsymbol{v}^\top \boldsymbol{M}_{-i})\|_F^2]^{\frac{1}{2}} \\
&\leq \mathbb{E}_i[\mathrm{Tr}(\boldsymbol{u} \boldsymbol{u}^\top \boldsymbol{M}_{-i}^2)\mathrm{Tr}(\boldsymbol{v} \boldsymbol{v}^\top \boldsymbol{M}_{-i}^2)]^{\frac{1}{2}} \\
&\leq \mathbb{E}_i[\mathrm{Tr}(\boldsymbol{u} \boldsymbol{u}^\top \boldsymbol{M}_{-i}^2)^2]^{\frac{1}{4}} \mathbb{E}_i[\mathrm{Tr}(\boldsymbol{v} \boldsymbol{v}^\top \boldsymbol{M}_{-i}^2)^2]^{\frac{1}{4}} \\
&\leq C_* \frac{\rho_\lambda(n)^2}{n} \sqrt{\mathrm{Tr}(\boldsymbol{u} \boldsymbol{u}^\top \overline{\boldsymbol{M}})\mathrm{Tr}(\boldsymbol{v} \boldsymbol{v}^\top \overline{\boldsymbol{M}})},
\end{aligned}
$$

where in the last step we use (Misiakiewicz & Saeed, 2024, Lemma 4.(b)), and the same bound holds for $\|\boldsymbol{M}_{-i} \boldsymbol{v} \boldsymbol{u}^\top \boldsymbol{M}_{-i}\|_F$. Thus we obtain for the first term that

$$(\mathbb{E}_i - \mathbb{E}_{i-1})\boldsymbol{z}_i^\top \boldsymbol{M}_{-i} \boldsymbol{A} \boldsymbol{M}_{-i} \boldsymbol{z}_i \leq C_* \frac{\log(n)\rho_\lambda(n)^2}{n} \sqrt{\mathrm{Tr}(\boldsymbol{u} \boldsymbol{u}^\top \overline{\boldsymbol{M}})\mathrm{Tr}(\boldsymbol{v} \boldsymbol{v}^\top \overline{\boldsymbol{M}})}.$$

For the second term, for $j \in \{i, i-1\}$ we similarly have

$$
\begin{aligned}
\mathbb{E}_j[|\boldsymbol{z}_i^\top \boldsymbol{M}_{-i} \boldsymbol{A} \boldsymbol{M}_{-i} \boldsymbol{z}_i (\boldsymbol{z}_i \boldsymbol{M}_{-i} \boldsymbol{Z}_i - \mathrm{Tr}(\boldsymbol{M}_{-i}))|] & \\
\leq \mathbb{E}_j[|\boldsymbol{z}_i^\top \boldsymbol{M}_{-i} \boldsymbol{A} \boldsymbol{M}_{-i} \boldsymbol{z}_i|^2]^{\frac{1}{2}} & \mathbb{E}_i[|\boldsymbol{z}_i \boldsymbol{M}_{-i} \boldsymbol{z}_i - \mathrm{Tr}(\boldsymbol{M}_{-i})|^2]^{\frac{1}{2}}.
\end{aligned}
$$

Again by Lemma 2, we have

$$
\begin{aligned}
\mathbb{E}_i[|\boldsymbol{z}_i^\top \boldsymbol{M}_{-i} \boldsymbol{A} \boldsymbol{M}_{-i} \boldsymbol{z}_i|^2]^{\frac{1}{2}} &\leq C_* \log(n) \mathbb{E}_i[\mathrm{Tr}(\boldsymbol{u} \boldsymbol{u}^\top \boldsymbol{M}_{-i}^2)\mathrm{Tr}(\boldsymbol{v} \boldsymbol{v}^\top \boldsymbol{M}_{-i}^2)]^{1/2} \\
&\leq C_* \frac{\log(n)\rho_\lambda(n)^2}{n} \sqrt{\mathrm{Tr}(\boldsymbol{u} \boldsymbol{u}^\top \overline{\boldsymbol{M}})\mathrm{Tr}(\boldsymbol{v} \boldsymbol{v}^\top \overline{\boldsymbol{M}})},
\end{aligned}
$$

where in the first inequality we use (Misiakiewicz & Saeed, 2024, Lemma 4.(b)).

Following the same steps as in (Misiakiewicz & Saeed, 2024, Proof of Proposition 3), we have

$$\mathbb{E}_i[|\boldsymbol{z}_i \boldsymbol{M}_{-i} \boldsymbol{z}_i - \mathrm{Tr}(\boldsymbol{M}_{-i})|^2]^{\frac{1}{2}} \leq C_* \frac{\rho_\lambda(n)\log(n)}{n},$$

and the same bounds hold for $\mathbb{E}_{i-1}$ without the factor $\log(n)$. By combining the above bounds and apply a union bound to $\Delta_i, i \in [n]$, we have that

$$|\Delta_i| \leq C_* \frac{\rho_\lambda^2(n)\log(n)}{n} \sqrt{\mathrm{Tr}(\boldsymbol{u} \boldsymbol{u}^\top \overline{\boldsymbol{M}})\mathrm{Tr}(\boldsymbol{v} \boldsymbol{v}^\top \overline{\boldsymbol{M}})}.$$

The rest of the proof is exactly the same as Steps 3, 4 in the Proof of Proposition 3 of (Misiakiewicz & Saeed, 2024), and the only difference is the following bound:

$$
\begin{aligned}
\mathbb{E}_{i-1}[\Delta_i^2]^{1/2} &\le 2\mathbb{E}_{i-1}[\mathrm{Tr}(\boldsymbol{A}\boldsymbol{M})^2]^{1/2} \\
&\le 2\mathbb{E}_{i-1}\left[\mathrm{Tr}(\boldsymbol{u}\boldsymbol{u}^\top\boldsymbol{M}_{-i})\mathrm{Tr}(\boldsymbol{v}\boldsymbol{v}^\top\boldsymbol{M}_{-i})\right]^{1/2} \\
&\le 2\mathbb{E}_{i-1}\left[\mathrm{Tr}(\boldsymbol{u}\boldsymbol{u}^\top\boldsymbol{M}_{-i})^2\right]^{\frac{1}{4}}\mathbb{E}_{i-1}\left[\mathrm{Tr}(\boldsymbol{v}\boldsymbol{v}^\top\boldsymbol{M}_{-i})^2\right]^{\frac{1}{4}} \\
&\le C_*\rho_\lambda(n)\sqrt{\mathrm{Tr}(\boldsymbol{u}\boldsymbol{u}^\top\overline{\boldsymbol{M}})\mathrm{Tr}(\boldsymbol{v}\boldsymbol{v}^\top\overline{\boldsymbol{M}})}.
\end{aligned}
$$

Combining all the bounds and following the same steps as in (Misiakiewicz & Saeed, 2024, Steps 3, 4 in the Proof of Proposition 3), we have

$$
|\mathrm{Tr}(\boldsymbol{u}\boldsymbol{v}^\top\boldsymbol{M}) - \mathbb{E}_{\boldsymbol{X}}[\mathrm{Tr}(\boldsymbol{u}\boldsymbol{v}^\top\boldsymbol{M})]| \le C_* \frac{\rho_\lambda(n)^2\log^{\frac{3}{2}}(n)}{\sqrt{n}}\sqrt{\mathrm{Tr}(\boldsymbol{u}\boldsymbol{u}^\top\overline{\boldsymbol{M}})\mathrm{Tr}(\boldsymbol{v}\boldsymbol{v}^\top\overline{\boldsymbol{M}})}. \tag{50}
$$

**Combining all the bounds.** Combining (45), (49), (50), we obtain

$$
\left|\Phi_1(\boldsymbol{X};\boldsymbol{A},\lambda) - \frac{\lambda_*}{\lambda}\Psi_1(\lambda_*;\boldsymbol{A})\right| \le C_* \frac{\rho_\lambda(n)^{5/2}\log^{\frac{3}{2}}(n)}{\sqrt{n}}\frac{\lambda_*}{\lambda}\sqrt{\Psi_1(\lambda_*;\boldsymbol{u}\boldsymbol{u}^\top)\Psi_1(\lambda_*;\boldsymbol{v}\boldsymbol{v}^\top)}. \tag{51}
$$

### C.3. Deterministic equivalent of $\mathrm{Tr}(\boldsymbol{A}\boldsymbol{Z}^\top\boldsymbol{Z}\boldsymbol{M})$

C.3.1. BOUNDING THE DETERMINISTIC PART

By using exchangeability and Sherman-Morrison formula, we have

$$
\begin{aligned}
\mathbb{E}[\boldsymbol{u}^\top\boldsymbol{Z}^\top\boldsymbol{Z}\boldsymbol{M}\boldsymbol{v}] &= n\mathbb{E}[\boldsymbol{u}^\top\boldsymbol{z}_1\boldsymbol{z}_1^\top] \\
&= n\mathbb{E}\left[\frac{\boldsymbol{u}^\top\boldsymbol{z}_1\boldsymbol{z}_1\boldsymbol{M}_-\boldsymbol{v}}{1+\kappa}\right] + n\mathbb{E}\left[\frac{(\kappa - S_1)\boldsymbol{u}^\top\boldsymbol{z}_1\boldsymbol{z}_1^\top\boldsymbol{M}_-\boldsymbol{v}}{(1+\kappa)(1+S_1)}\right] \\
&= n\mathbb{E}\left[\frac{\boldsymbol{u}^\top\boldsymbol{M}_-\boldsymbol{v}}{1+\kappa}\right] + n\mathbb{E}\left[\frac{(\kappa - S_1)\boldsymbol{u}^\top\boldsymbol{z}_1\boldsymbol{z}_1^\top\boldsymbol{M}_-\boldsymbol{v}}{(1+\kappa)(1+S_1)}\right].
\end{aligned}
$$

For the first term, we note that (denoting $\tilde{\mu}_- = n/(1+\kappa)$)

$$
n\mathbb{E}\left[\frac{\boldsymbol{u}^\top\boldsymbol{M}_-\boldsymbol{v}}{1+\kappa}\right] = \tilde{\mu}_-\mathbb{E}[\boldsymbol{u}^\top\boldsymbol{M}_-\boldsymbol{v}],
$$

and we have

$$
\begin{aligned}
&|\tilde{\mu}_-\mathbb{E}[\boldsymbol{u}^\top\boldsymbol{M}_-\boldsymbol{v}] - \mu_*\mathbb{E}[\boldsymbol{u}^\top\overline{\boldsymbol{M}}\boldsymbol{v}]| \\
&\le \frac{\tilde{\mu}_-}{\mu_*}\mu_*|\mathbb{E}[\boldsymbol{u}^\top\boldsymbol{M}_-\boldsymbol{v}] - \mathbb{E}[\boldsymbol{u}^\top\overline{\boldsymbol{M}}\boldsymbol{v}]| + \frac{|\tilde{\mu}_- - \mu_*|}{\mu_*}\mu_*\mathbb{E}[\boldsymbol{u}^\top\overline{\boldsymbol{M}}\boldsymbol{v}] \\
&\le C_* \frac{\rho_\lambda(n)^{5/2}\log^{3/2}(n)}{\sqrt{n}}\mu_*\sqrt{\boldsymbol{u}^\top\overline{\boldsymbol{M}}\boldsymbol{u}\boldsymbol{v}^\top\overline{\boldsymbol{M}}\boldsymbol{v}} + \frac{\rho_\lambda(n)^{5/2}}{\sqrt{n}}\mu_*\mathbb{E}[\boldsymbol{u}^\top\overline{\boldsymbol{M}}\boldsymbol{v}],
\end{aligned} \tag{52}
$$

where we use the fact that

$$
\frac{|\tilde{\mu}_- - \mu_*|}{\mu_*} \le \frac{\rho_\lambda(n)^{5/2}}{\sqrt{n}}
$$

proved in (Misiakiewicz & Saeed, 2024, Proof of Proposition 6).

For the second term, we have

$$
\begin{aligned}
n\mathbb{E}\left[\frac{(\kappa - S_1)\boldsymbol{u}^\top \boldsymbol{z}_1 \boldsymbol{z}_1^\top \boldsymbol{M}_-\boldsymbol{v}}{(1+\kappa)(1+S_1)}\right] &\leq \tilde{\mu}_-\mathbb{E}[(\kappa - S_1)^2]^{1/2}\mathbb{E}[(\boldsymbol{z}_1\boldsymbol{M}_-\boldsymbol{v}\boldsymbol{u}^\top \boldsymbol{z}_1)^2]^{1/2} \\
&\leq \mu_*\mathbb{E}[(\kappa - S_1)^2]^{1/2}\mathbb{E}[(\boldsymbol{z}_1\boldsymbol{M}_-\boldsymbol{v}\boldsymbol{u}^\top \boldsymbol{z}_1)^2]^{1/2} \\
&\quad + \frac{|\tilde{\mu}_- - \mu_*|}{\mu_*}\tilde{\mu}_-\mathbb{E}[(\kappa - S_1)^2]^{1/2}\mathbb{E}[(\boldsymbol{z}_1\boldsymbol{M}_-\boldsymbol{v}\boldsymbol{u}^\top \boldsymbol{z}_1)^2]^{1/2} \\
&\leq C_*\frac{\rho_\lambda(n)}{\sqrt{n}}\mu_*\mathbb{E}[\|\boldsymbol{M}_-\boldsymbol{v}\boldsymbol{u}^\top\|_F] \\
&\leq C_*\frac{\rho_\lambda(n)}{\sqrt{n}}\mu_*\|\boldsymbol{u}\|_2\mathbb{E}[\sqrt{\boldsymbol{v}^\top \boldsymbol{M}_-^2\boldsymbol{v}}] \\
&\leq C_*\frac{\rho_\lambda(n)}{\sqrt{n}}\mu_*\|\boldsymbol{u}\|_2\mathbb{E}[\boldsymbol{v}^\top \boldsymbol{M}_-^2\boldsymbol{v}]^{1/2} \\
&\leq C_*\frac{\rho_\lambda(n)^{3/2}}{\sqrt{n}}\mu_*\|\boldsymbol{u}\|_2\sqrt{\boldsymbol{v}^\top \overline{\boldsymbol{M}}^2\boldsymbol{v}}.
\end{aligned}
\tag{53}
$$

Combining (52), (53), we have

$$
|\mathbb{E}[\boldsymbol{A}\boldsymbol{Z}^\top \boldsymbol{Z}\boldsymbol{M}] - \mu_*\boldsymbol{u}^\top \overline{\boldsymbol{M}}\boldsymbol{v}|C_*\frac{\rho_\lambda(n)^{5/2}\log^{3/2}(n)}{\sqrt{n}}\tilde{\Psi}_1(\lambda_*; \boldsymbol{A}, \boldsymbol{I}).
\tag{54}
$$

### C.3.2. BOUNDING THE MARTINGALE PART

Following the same methods, we have

$$
\mathrm{Tr}(\boldsymbol{A}\boldsymbol{Z}^\top \boldsymbol{Z}\boldsymbol{M}) - \mathbb{E}[\mathrm{Tr}(\boldsymbol{A}\boldsymbol{Z}^\top \boldsymbol{Z}\boldsymbol{M})] = \Sigma_{i=1}^n \Delta_i,
$$

with $\Delta_i = (\mathbb{E}_i - \mathbb{E}_{i-1})\mathrm{Tr}(\boldsymbol{A}\boldsymbol{Z}^\top \boldsymbol{Z}\boldsymbol{M})$.

We first have

$$
\begin{aligned}
\mathbb{E}_{i-1}[\Delta_i^2]^{1/2} &\leq 2n\mathbb{E}_{i-1}\left[\frac{(\boldsymbol{z}_1\boldsymbol{M}_-\boldsymbol{v}\boldsymbol{u}^\top \boldsymbol{z}_1)^2}{(1+S_1)^2}\right]^{1/2} \\
&\leq 4\frac{n}{(1+\kappa)}\mathbb{E}_{i-1}\left[(\boldsymbol{z}_1\boldsymbol{M}_-\boldsymbol{v}\boldsymbol{u}^\top \boldsymbol{z}_1)^2\right]^{1/2} + \mathbb{E}_{i-1}\left[\frac{(\kappa - S_1)(2+\kappa+S_1)(\boldsymbol{z}_1\boldsymbol{M}_-\boldsymbol{v}\boldsymbol{u}^\top \boldsymbol{z}_1)^2}{(1+S_1)^2}\right]^{1/2} \\
&\leq C_*\tilde{\mu}_-\mathbb{E}[\|\boldsymbol{M}_-\boldsymbol{v}\boldsymbol{u}^\top\|_F^2]^{1/2} \quad \text{(note that the second term has lower order due to the extra } \kappa - S_1) \\
&\leq C_*\rho_\lambda^{3/2}(n)\mu_*\|\boldsymbol{u}\|_2\sqrt{\boldsymbol{v}^\top \overline{\boldsymbol{M}}^2\boldsymbol{v}}.
\end{aligned}
$$

Next, we have

$$
\begin{aligned}
\mathrm{Tr}(\boldsymbol{A}\boldsymbol{Z}^\top \boldsymbol{Z}\boldsymbol{M}) - \mathrm{Tr}(\boldsymbol{A}\boldsymbol{Z}_{-i}^\top \boldsymbol{Z}_{-i}\boldsymbol{M}_{-i}) &= \frac{1}{(1+S_i)}\boldsymbol{z}_i^\top \boldsymbol{M}_{-i}\boldsymbol{v}\boldsymbol{u}^\top(\boldsymbol{I} - \boldsymbol{Z}_{-i}^\top \boldsymbol{Z}_{-i}\boldsymbol{M}_{-i})\boldsymbol{z}_i \\
&= \lambda\frac{1}{(1+S_i)}\boldsymbol{z}_i^\top \boldsymbol{M}_{-i}\boldsymbol{v}\boldsymbol{u}^\top \boldsymbol{\Sigma}^{-1}\boldsymbol{M}_{-i}\boldsymbol{z}_i,
\end{aligned}
$$

which implies that

$$\mathbb{E}_i[\text{Tr}(\boldsymbol{A}\boldsymbol{Z}^\top\boldsymbol{Z}\boldsymbol{M}) - \text{Tr}(\boldsymbol{A}\boldsymbol{Z}_{-i}^\top\boldsymbol{Z}_{-i}\boldsymbol{M}_{-i})]$$

$$=\frac{1}{n}\mathbb{E}_i\left[\lambda\frac{n}{(1+S_i)}\boldsymbol{z}_i^\top\boldsymbol{M}_{-i}\boldsymbol{v}\boldsymbol{u}^\top\boldsymbol{\Sigma}^{-1}\boldsymbol{M}_{-i}\boldsymbol{z}_i\right]$$

$$\leq C_*\frac{1}{n}\left(\lambda\frac{n}{1+\kappa}\mathbb{E}[\boldsymbol{z}_i^\top\boldsymbol{M}_{-i}\boldsymbol{v}\boldsymbol{u}^\top\boldsymbol{\Sigma}^{-1}\boldsymbol{M}_{-i}\boldsymbol{z}_i]\right) \quad \text{(we ignore the term with the extra } \kappa - S_1 \text{ factor as it is of lower order)}$$

$$\leq C_*\frac{1}{n}\left(\lambda\tilde{\mu}_-\mathbb{E}[(\boldsymbol{z}_i^\top\boldsymbol{M}_{-i}\boldsymbol{v}\boldsymbol{v}\boldsymbol{M}_{-i}\boldsymbol{z}_i)]^{1/2}\|\boldsymbol{u}\|_2\mathbb{E}[\boldsymbol{z}_i\boldsymbol{M}_{-1}\boldsymbol{\Sigma}^{-2}\boldsymbol{M}_{-i}\boldsymbol{z}_i]^{1/2}\right)$$

$$\leq C_*\frac{\rho_\lambda(n)}{n}\mu_*\|\boldsymbol{u}\|_2\sqrt{\boldsymbol{v}^\top\overline{\boldsymbol{M}}^2\boldsymbol{v}},$$

where we use $\boldsymbol{M}_i \preceq \lambda^{-1}\boldsymbol{\Sigma}$.

Combining all the bounds and following the same steps described in Appendix C.1, we have

$$|\text{Tr}(\boldsymbol{A}^\top\boldsymbol{Z}^\top\boldsymbol{Z}\boldsymbol{M}) - \mathbb{E}_{\boldsymbol{X}}[\text{Tr}(\boldsymbol{A}^\top\boldsymbol{Z}^\top\boldsymbol{Z}\boldsymbol{M})]| \leq C_*\frac{\rho_\lambda(n)}{\sqrt{n}}\mu_*\|\boldsymbol{u}\|_2\sqrt{\boldsymbol{v}^\top\overline{\boldsymbol{M}}^2\boldsymbol{v}}. \tag{55}$$

Combining (54), (55), we have that, with probability at least $1 - n^{-D}$,

$$|\text{Tr}(\boldsymbol{A}^\top\boldsymbol{Z}^\top\boldsymbol{Z}\boldsymbol{M}) - \mu_*\boldsymbol{u}^\top\overline{\boldsymbol{M}}\boldsymbol{v}| \leq \frac{\rho_\lambda(n)^{5/2}\log^{3/2}(n)}{\sqrt{n}}\tilde{\Psi}_1(\lambda_*; \boldsymbol{A}, \boldsymbol{I}).$$

### C.4. Deterministic equivalent of $\text{Tr}(\boldsymbol{A}\boldsymbol{M}\boldsymbol{B}\boldsymbol{M})$

#### C.4.1. BOUNDING THE DETERMINISTIC PART

We write $\Phi_3(\boldsymbol{X}; \boldsymbol{A}, \boldsymbol{B}, \lambda) = \text{Tr}(\boldsymbol{A}\boldsymbol{M}\boldsymbol{B}\boldsymbol{M})$, and recall that $\boldsymbol{A} = \boldsymbol{v}\boldsymbol{u}^\top$.

We first decompose $\mathbb{E}_{\boldsymbol{X}}[\text{Tr}(\boldsymbol{v}\boldsymbol{u}^\top\boldsymbol{M}\boldsymbol{B}\boldsymbol{M}) - \text{Tr}(\boldsymbol{v}\boldsymbol{u}^\top\overline{\boldsymbol{M}}_-\boldsymbol{B}\overline{\boldsymbol{M}}_-)]$ as

$$\mathbb{E}_{\boldsymbol{X}}[\text{Tr}(\boldsymbol{v}\boldsymbol{u}^\top\boldsymbol{M}\boldsymbol{B}\boldsymbol{M}) - \text{Tr}(\boldsymbol{v}\boldsymbol{u}^\top\overline{\boldsymbol{M}}_-\boldsymbol{B}\overline{\boldsymbol{M}}_-)]$$

$$= \underbrace{\mathbb{E}_{\boldsymbol{X}}[\boldsymbol{v}\boldsymbol{u}^\top(\boldsymbol{M} - \overline{\boldsymbol{M}}_-)\boldsymbol{B}\overline{\boldsymbol{M}}_-] + \mathbb{E}_{\boldsymbol{X}}[\boldsymbol{v}\boldsymbol{u}^\top\overline{\boldsymbol{M}}_-\boldsymbol{B}(\boldsymbol{M} - \overline{\boldsymbol{M}}_-)]}_{T_1(\boldsymbol{A}, \boldsymbol{B})} + \underbrace{\mathbb{E}_{\boldsymbol{X}}[\boldsymbol{v}\boldsymbol{u}^\top(\boldsymbol{M} - \overline{\boldsymbol{M}}_-)\boldsymbol{B}(\boldsymbol{M} - \overline{\boldsymbol{M}}_-)]}_{T_2(\boldsymbol{A}, \boldsymbol{B})}. \tag{56}$$

We have the following claims, which we will prove later.

**Claim 1.** $|T_1(\boldsymbol{A}, \boldsymbol{B})| \leq C_*\frac{\rho_\lambda(n)^{7/2}}{\sqrt{n}}\|\overline{\boldsymbol{M}}^{1/2}\boldsymbol{B}\overline{\boldsymbol{M}}^{1/2}\|_{op}\sqrt{\boldsymbol{u}^\top\overline{\boldsymbol{M}}\boldsymbol{u}\boldsymbol{v}^\top\overline{\boldsymbol{M}}\boldsymbol{v}}$

**Claim 2.** $|T_2(\boldsymbol{u}, \boldsymbol{v}) - \frac{n\mathbb{E}[\text{Tr}(\boldsymbol{B}\boldsymbol{M}^2)]}{(1+\kappa)^2}\text{Tr}(\overline{\boldsymbol{Q}}_-)| \leq C_*\frac{\rho_\lambda^5(n)}{\sqrt{n}}\|\overline{\boldsymbol{M}}^{1/2}\boldsymbol{B}\overline{\boldsymbol{M}}^{1/2}\|_{op}\sqrt{\boldsymbol{u}^\top\overline{\boldsymbol{M}}\boldsymbol{u}\boldsymbol{v}^\top\overline{\boldsymbol{M}}\boldsymbol{v}}$

We first prove the main deterministic bounds using the above claims. Combining Claim 1, Claim 2 and (56), we have

$$\left|\mathbb{E}_{\boldsymbol{X}}[\text{Tr}(\boldsymbol{A}\boldsymbol{M}\boldsymbol{B}\boldsymbol{M})] - \text{Tr}(\boldsymbol{A}\overline{\boldsymbol{M}}\boldsymbol{B}\overline{\boldsymbol{M}}) - \frac{n}{(1+\text{Tr}(\overline{\boldsymbol{M}}))^2}\frac{\text{Tr}(\boldsymbol{B}\overline{\boldsymbol{M}}^2)\text{Tr}(\boldsymbol{A}\overline{\boldsymbol{M}}^2)}{1 - \frac{\mu_*^2}{n}\text{Tr}(\overline{\boldsymbol{M}}^2)}\right|$$

$$\leq \left|\mathbb{E}_{\boldsymbol{X}}[\text{Tr}(\boldsymbol{A}\boldsymbol{M}\boldsymbol{B}\boldsymbol{M})] - \text{Tr}(\boldsymbol{A}\overline{\boldsymbol{M}}_-\boldsymbol{B}\overline{\boldsymbol{M}}_-) - \frac{n}{(1+\kappa)^2}\mathbb{E}[\text{Tr}(\boldsymbol{B}\boldsymbol{M}^2)]\text{Tr}(\boldsymbol{A}\overline{\boldsymbol{M}}_-^2)\right| \tag{57}$$

$$+ |\text{Tr}(\boldsymbol{A}\overline{\boldsymbol{M}}\boldsymbol{B}\overline{\boldsymbol{M}}) - \text{Tr}(\boldsymbol{A}\overline{\boldsymbol{M}}_-\boldsymbol{B}\overline{\boldsymbol{M}}_-)| \tag{58}$$

$$+ \left|\frac{n\text{Tr}(\boldsymbol{A}\overline{\boldsymbol{M}}_-^2)}{(1+\kappa)^2} - \frac{n\text{Tr}(\boldsymbol{A}\overline{\boldsymbol{M}}^2)}{(1+\text{Tr}(\overline{\boldsymbol{M}}))^2}\right|\mathbb{E}[\text{Tr}(\boldsymbol{B}\boldsymbol{M}^2)] \tag{59}$$

$$+ \left|\frac{n\text{Tr}(\boldsymbol{A}\overline{\boldsymbol{M}}^2)}{(1+\text{Tr}(\overline{\boldsymbol{M}}))^2}\right|\left|\mathbb{E}[\text{Tr}(\boldsymbol{B}\boldsymbol{M}^2)] - \frac{\text{Tr}(\boldsymbol{B}\overline{\boldsymbol{M}}^2)}{1 - \frac{\mu_*^2}{n}\text{Tr}(\overline{\boldsymbol{M}}^2)}\right|. \tag{60}$$

For (57), we use Claim 1 and Claim 2, to obtain that

$$(57) \leq C_* \frac{\rho_\lambda^5(n)}{\sqrt{n}} \|\overline{M}^{1/2} B \overline{M}^{1/2}\|_{op} \sqrt{u^\top \overline{M} u v^\top \overline{M} v}.$$

For (58), we first note that

$$\overline{M} - \overline{M}_- = \overline{M}(\tilde{\mu}_- \Sigma + \lambda)^{-1}(\tilde{\mu}_- - \mu_*)\Sigma$$
$$= \overline{M}^{1/2}(\tilde{\mu}_- \Sigma + \lambda)^{-1}(\tilde{\mu}_- - \mu_*)\Sigma \overline{M}^{1/2}.$$

Thus, we have

$$(58) = |2u^\top (\overline{M} - \overline{M}_-) B \overline{M} v + u^\top (\overline{M} - \overline{M}_-) B (\overline{M} - \overline{M}_-) v|$$
$$\leq 2\|(\tilde{\mu}_- \Sigma + \lambda)^{-1}(\tilde{\mu}_- - \mu_*)\|_{op} \sqrt{u^\top \overline{M} u v \overline{M} B \overline{M} B \overline{M} v}$$
$$+ \|(\tilde{\mu}_- \Sigma + \lambda)^{-1}(\tilde{\mu}_- - \mu_*)\|_{op}^2 \|\overline{M}^{1/2} B \overline{M}^{1/2}\|_{op} \sqrt{b u^\top \overline{M} u v^\top \overline{M} v}$$
$$\leq C_* \frac{\rho_\lambda(n)^{5/2}}{\sqrt{n}} \|\overline{M}^{1/2} B \overline{M}^{1/2}\|_{op} \sqrt{u^\top \overline{M} u v^\top \overline{M} v}.$$

For (59), we first note that

$$\overline{M}^2 - \overline{M}_-^2 = \overline{M}(\tilde{\mu}_- \Sigma + \lambda)^{-2}(\tilde{\mu}_- - \mu_*)\Sigma((\tilde{\mu}_- + \mu_*)\Sigma + \lambda)\overline{M},$$

which implies that

$$\left| \frac{n\mathrm{Tr}(A^\top \overline{M}_-^2)}{(1+\kappa)^2} - \frac{n\mathrm{Tr}(A\overline{M}^2)}{(1+\mathrm{Tr}(\overline{M}))^2} \right| = \left| \frac{nu^\top \overline{M}_-^2 v}{(1+\kappa)^2} - \frac{nu^\top \overline{M}^2 v}{(1+\mathrm{Tr}(\overline{M}))^2} \right|$$
$$\leq \left| \frac{\tilde{\mu}_-^2 u^\top \overline{M}_-^2 v}{n} - \frac{\mu_*^2 u^\top \overline{M}^2 v}{n} \right|,$$
$$\leq C_* \frac{\rho_\lambda(n)^{5/2}}{\sqrt{n}} n \sqrt{u^\top \overline{M}^2 u v^\top \overline{M}^2 v}$$
$$\leq C_* \frac{\rho_\lambda(n)^{7/2}}{\sqrt{n}} \sqrt{u^\top \overline{M} u v^\top \overline{M} v}.$$

Thus, we have

$$(59) \leq C_* \frac{\rho_\lambda(n)^{9/2}}{\sqrt{n}} \|\overline{M}^{1/2} B \overline{M}^{1/2}\|_{op} \sqrt{u^\top \overline{M} u v^\top \overline{M} v},$$

by using that $\mathbb{E}[\mathrm{Tr}(BM^2)] \leq \mathbb{E}[\|M^{1/2} B M^{1/2}\|_{op} \mathrm{Tr}(BM)] \leq \mathbb{E}[\|M^{1/2} B M^{1/2}\|_{op}^2]^{1/2} \mathbb{E}[\mathrm{Tr}(BM)^2]^{1/2}$, and then using Lemma 4 and (Misiakiewicz & Saeed, 2024, Lemma 4.(b)).

For (60), we simply use (Misiakiewicz & Saeed, 2024, Proposition 4), and we have

$$(60) \leq C_* \frac{\rho_\lambda^6(n) \log^{5/2}(n)}{\sqrt{n}} \frac{\mu_*^2}{n} \frac{\mathrm{Tr}(A\overline{M}^2)\mathrm{Tr}(B\overline{M}^2)}{1 - \frac{\mu_*}{n}\mathrm{Tr}(\overline{M}^2)}.$$

By combining the upper bounds for (57)-(60), we have

$$\left| \mathbb{E}_X[\mathrm{Tr}(AMBM)] - \mathrm{Tr}(A\overline{M} B \overline{M}) - \frac{\mu_*^2}{n} \frac{\mathrm{Tr}(B\overline{M}^2)\mathrm{Tr}(A\overline{M}^2)}{1 - \frac{\mu_*^2}{n}\mathrm{Tr}(\overline{M}^2)} \right|$$
$$\leq C_* \frac{\rho_\lambda^6(n) \log^{5/2}(n)}{\sqrt{n}} \frac{\mu_*^2}{n} \tilde{\Psi}_2(\lambda_*, vu^\top, B). \tag{61}$$

It remains to prove Claim 1 and Claim 2.

*Proof of Claim 1.* It is sufficient to bound $\mathbb{E}_{\boldsymbol{X}}[\text{Tr}(\boldsymbol{v}\boldsymbol{u}^\top(\boldsymbol{M} - \overline{\boldsymbol{M}}_-)\boldsymbol{B}\overline{\boldsymbol{M}}_-)]$, and the other one holds by symmetry. We note that

$$\text{Tr}(\boldsymbol{v}\boldsymbol{u}^\top(\boldsymbol{M} - \overline{\boldsymbol{M}}_-)\boldsymbol{B}\overline{\boldsymbol{M}}_-) = \text{Tr}(\boldsymbol{w}^\top\boldsymbol{u}(\boldsymbol{M} - \overline{\boldsymbol{M}}_-)),$$

with $\boldsymbol{w} = \boldsymbol{B}\overline{\boldsymbol{M}}_-\boldsymbol{v}$. Following the same steps as in C.2.1, we have

$$
\begin{aligned}
|T_1(\boldsymbol{A}, \boldsymbol{B})| \leq & \frac{\rho_\lambda^{5/2}(n)}{\sqrt{n}}\sqrt{\boldsymbol{u}^\top\overline{\boldsymbol{M}}\boldsymbol{u}\boldsymbol{v}^\top\overline{\boldsymbol{M}}_-\boldsymbol{B}\overline{\boldsymbol{M}}\boldsymbol{B}\overline{\boldsymbol{M}}_-\boldsymbol{v}} \\
\leq & \frac{\rho_\lambda^{5/2}(n)}{\sqrt{n}}\sqrt{\boldsymbol{u}^\top\overline{\boldsymbol{M}}\boldsymbol{u}\boldsymbol{v}^\top\overline{\boldsymbol{M}}_-\boldsymbol{v}\|\overline{\boldsymbol{M}}_-^{1/2}\boldsymbol{B}\overline{\boldsymbol{M}}\boldsymbol{B}\overline{\boldsymbol{M}}_-^{1/2}\|_{op}} \\
\leq & \frac{\rho_\lambda^3(n)}{\sqrt{n}}\sqrt{\boldsymbol{u}^\top\overline{\boldsymbol{M}}\boldsymbol{u}\boldsymbol{v}^\top\overline{\boldsymbol{M}}\boldsymbol{v}\|\boldsymbol{B}\overline{\boldsymbol{M}}\|_{op}\|\boldsymbol{B}\overline{\boldsymbol{M}}_-\|_{op}} \\
\leq & \frac{\rho_\lambda^{9/2}(n)}{\sqrt{n}}\sqrt{\boldsymbol{u}^\top\overline{\boldsymbol{M}}\boldsymbol{u}\boldsymbol{v}^\top\overline{\boldsymbol{M}}\boldsymbol{v}}\|\overline{\boldsymbol{M}}^{1/2}\boldsymbol{B}\overline{\boldsymbol{M}}^{1/2}\|_{op},
\end{aligned}
$$

where in the last step we use (Misiakiewicz & Saeed, 2024, Lemma 4.(b)). □

*Proof of Claim 2.* For the second term, we use the same decomposition as in (Misiakiewicz & Saeed, 2024, Proof of Claim 2). In particular, we have:

$$
\begin{aligned}
T_2(\boldsymbol{A}, \boldsymbol{B}) &= \mathbb{E}_{\boldsymbol{X}}\left[\text{Tr}\left(\overline{\boldsymbol{M}}_-\boldsymbol{v}\boldsymbol{u}^\top\overline{\boldsymbol{M}}_-\left(\frac{n\boldsymbol{I}}{1+\kappa} - \boldsymbol{Z}^\top\boldsymbol{Z}\right)\boldsymbol{M}\boldsymbol{B}\boldsymbol{M}\left(\frac{n\boldsymbol{I}}{1+\kappa} - \boldsymbol{Z}^\top\boldsymbol{Z}\right)\right)\right] \\
&= (I) + (II),
\end{aligned}
$$

where

$$
\begin{aligned}
(I) &= n\mathbb{E}\sum_{i,j\in[3]}\text{Tr}(\overline{\boldsymbol{Q}}_-\Delta_{1i}^\top\boldsymbol{M}_{-1}\boldsymbol{B}\boldsymbol{M}_{-1}\Delta_{1j}), \\
(II) &= n(n-1)\mathbb{E}\sum_{i,j\in[3]}\text{Tr}(\overline{\boldsymbol{Q}}_-\Delta_{1i}^\top\boldsymbol{M}_{-1}\boldsymbol{B}\boldsymbol{M}_{-2}\Delta_{2j}),
\end{aligned}
$$

and using the same notation as in (Misiakiewicz & Saeed, 2024, Proof of Claim 2) we define

$$\overline{\boldsymbol{Q}}_- = \overline{\boldsymbol{M}}_-\boldsymbol{A}\overline{\boldsymbol{M}}_-, \qquad \boldsymbol{Q} = \overline{\boldsymbol{M}}\boldsymbol{A}\overline{\boldsymbol{M}},$$

$$\Delta_{i1} = \frac{\boldsymbol{I} - \boldsymbol{z}_i\boldsymbol{z}_i^\top}{1+\kappa}, \quad \Delta_{i2}\frac{\boldsymbol{z}_i\boldsymbol{z}_i^\top(S_i - \kappa)}{(1+S_i)(1+\kappa)}, \quad \Delta_{i3} = -\frac{-\boldsymbol{z}_i\boldsymbol{z}_i^\top\boldsymbol{M}_i}{1+S_i}.$$

**Bounding** $\mathbb{E}[\text{Tr}(\overline{\boldsymbol{Q}}_-\Delta_{11}^\top\boldsymbol{M}_{-1}\boldsymbol{B}\boldsymbol{M}_{-1}\Delta_{11})]$

We expand the term in the same way as in (Misiakiewicz & Saeed, 2024, Proof of Claim 2) and we have

$$(1+\kappa)^2\mathbb{E}[\text{Tr}(\overline{\boldsymbol{Q}}_-\Delta_{11}^\top\boldsymbol{M}_{-1}\boldsymbol{B}\boldsymbol{M}_{-1}\Delta_{11})] = -\mathbb{E}[\text{Tr}(\overline{\boldsymbol{Q}}_-\boldsymbol{M}_{-1}\boldsymbol{B}\boldsymbol{M}_{-1})] + \mathbb{E}[\boldsymbol{z}_1^\top\overline{\boldsymbol{Q}}_-\boldsymbol{z}_1\boldsymbol{z}_1^\top\boldsymbol{M}_{-1}\boldsymbol{B}\boldsymbol{M}_{-1}\boldsymbol{z}_1].$$

Furthermore, we have

$$
\begin{aligned}
|\mathbb{E}[\text{Tr}(\overline{\boldsymbol{Q}}_-\boldsymbol{M}_{-1}\boldsymbol{B}\boldsymbol{M}_{-1})]| \leq & \mathbb{E}[\sqrt{\boldsymbol{u}^\top\overline{\boldsymbol{M}}_-\boldsymbol{M}_-\boldsymbol{B}\boldsymbol{M}_-\overline{\boldsymbol{M}}_-\boldsymbol{u}\boldsymbol{v}^\top\overline{\boldsymbol{M}}_-\boldsymbol{M}_-\boldsymbol{B}\boldsymbol{M}_-\overline{\boldsymbol{M}}_-\boldsymbol{v}}] \\
\leq & \mathbb{E}[\|\boldsymbol{M}_-\boldsymbol{B}\boldsymbol{M}_-\|_{op}]\sqrt{\boldsymbol{u}^\top\overline{\boldsymbol{M}}_-^2\boldsymbol{u}\boldsymbol{v}^\top\overline{\boldsymbol{M}}_-^2\boldsymbol{v}} \\
\leq & C_*\frac{\rho_\lambda^4(n)}{n^2}\|\overline{\boldsymbol{M}}^{1/2}\boldsymbol{B}\overline{\boldsymbol{M}}^{1/2}\|_{op}\sqrt{\boldsymbol{u}^\top\overline{\boldsymbol{M}}\boldsymbol{u}\boldsymbol{v}^\top\overline{\boldsymbol{M}}\boldsymbol{v}},
\end{aligned}
$$

where in the last step we use Lemma 4 and (Misiakiewicz & Saeed, 2024, Lemma 4.(a)).

For the second term, we use

$$
\begin{aligned}
|\mathbb{E}[\boldsymbol{z}_1^\top \overline{\boldsymbol{Q}}_- \boldsymbol{z}_1 \boldsymbol{z}_1^\top \boldsymbol{M}_{-1} \boldsymbol{B} \boldsymbol{M}_{-1} \boldsymbol{z}_1] &- \mathrm{Tr}(\overline{\boldsymbol{Q}}_-)\mathbb{E}[\mathrm{Tr}(\boldsymbol{B}\boldsymbol{M}_{-1}^2)]| \\
=&\mathbb{E}[(\boldsymbol{z}_1^\top \overline{\boldsymbol{Q}}_- \boldsymbol{z}_1 - \mathrm{Tr}(\overline{\boldsymbol{Q}}_-))(\boldsymbol{z}_1^\top \boldsymbol{M}_{-1} \boldsymbol{B} \boldsymbol{M}_{-1} \boldsymbol{z}_1 - \mathbb{E}[\mathrm{Tr}(\boldsymbol{B}\boldsymbol{M}_{-1}^2)])] \\
\leq&\mathbb{E}[(\boldsymbol{z}_1^\top \overline{\boldsymbol{Q}}_- \boldsymbol{z}_1 - \mathrm{Tr}(\overline{\boldsymbol{Q}}_-))^2]^{\frac{1}{2}}\mathbb{E}[(\boldsymbol{z}_1^\top \boldsymbol{M}_{-1} \boldsymbol{B} \boldsymbol{M}_{-1} \boldsymbol{z}_1 - \mathbb{E}[\mathrm{Tr}(\boldsymbol{B}\boldsymbol{M}_{-1}^2)])^2]^{\frac{1}{2}} \\
\leq&C_* \log(n)\|\overline{\boldsymbol{Q}}_-\|_F \mathbb{E}[\|\boldsymbol{M}_{-1}\boldsymbol{B}\boldsymbol{M}_{-1})\|_F] \\
\leq&C_* \log(n)\|\overline{\boldsymbol{Q}}_-\|_F \mathbb{E}[\|\boldsymbol{M}_{-1}^{1/2}\boldsymbol{B}\boldsymbol{M}_{-1}^{1/2}\|_{op}^2]^{1/2}\mathbb{E}[\|\boldsymbol{M}_{-1}\|_F^2]^{1/2} \\
\leq&C_* \frac{C_* \log(n)\rho_\lambda^2(n)}{\sqrt{n}}\|\overline{\boldsymbol{M}}^{1/2}\boldsymbol{B}\overline{\boldsymbol{M}}^{1/2}\|_{op}\sqrt{\boldsymbol{u}^\top \overline{\boldsymbol{M}}^2 \boldsymbol{u}\boldsymbol{v}^\top \overline{\boldsymbol{M}}^2 \boldsymbol{v}} \\
\leq&\frac{C_* \log(n)\rho_\lambda^{5/2}(n)}{n^{3/2}}\|\overline{\boldsymbol{M}}^{1/2}\boldsymbol{B}\overline{\boldsymbol{M}}^{1/2}\|_{op}\sqrt{\boldsymbol{u}^\top \overline{\boldsymbol{M}}\boldsymbol{u}\boldsymbol{v}^\top \overline{\boldsymbol{M}}\boldsymbol{v}}.
\end{aligned}
$$

Furthermore, we have

$$
\begin{aligned}
\mathrm{Tr}(\overline{\boldsymbol{Q}}_-)|\mathbb{E}[\mathrm{Tr}(\boldsymbol{B}\boldsymbol{M}_-^2)] - \mathbb{E}[\mathrm{Tr}(\boldsymbol{B}\boldsymbol{M}^2)]| \leq&C_*\sqrt{\boldsymbol{u}^\top \overline{\boldsymbol{M}}_-^2 \boldsymbol{u}\boldsymbol{v}^\top \overline{\boldsymbol{M}}_-^2 \boldsymbol{v}}\mathbb{E}[\mathrm{Tr}(\boldsymbol{B}(\boldsymbol{M}^2 - \boldsymbol{M}_-^2))]| \\
\leq&C_*\frac{\rho_\lambda(n)^2}{n}\sqrt{\boldsymbol{u}^\top \overline{\boldsymbol{M}}\boldsymbol{u}\boldsymbol{v}^\top \overline{\boldsymbol{M}}\boldsymbol{v}}\mathbb{E}[\mathrm{Tr}(\boldsymbol{B}(\boldsymbol{M}^2 - \boldsymbol{M}_-^2))]|.
\end{aligned}
$$

We bound $|\mathbb{E}[\mathrm{Tr}(\boldsymbol{B}(\boldsymbol{M}^2 - \boldsymbol{M}_-^2))]|$ by first noting that

$$
\begin{aligned}
\mathbb{E}[\mathrm{Tr}(\boldsymbol{B}(\boldsymbol{M}^2 - \boldsymbol{M}_-^2))] =&\mathbb{E}[\boldsymbol{z}^\top \boldsymbol{M}_-^2 \boldsymbol{z}\boldsymbol{z}^\top \boldsymbol{M}_- \boldsymbol{B}\boldsymbol{M}_- \boldsymbol{z} - 2\boldsymbol{z}^\top \boldsymbol{M}_- \boldsymbol{B}\boldsymbol{M}_-^2 \boldsymbol{z}] \\
\leq&\mathbb{E}[(\boldsymbol{z}^\top \boldsymbol{M}_-^2 \boldsymbol{z})^2]^{1/2}\mathbb{E}[(\boldsymbol{z}^\top \boldsymbol{M}_- \boldsymbol{B}\boldsymbol{M}_- \boldsymbol{z})^2]^{1/2} + 2\mathbb{E}[|\boldsymbol{z}^\top \boldsymbol{M}_- \boldsymbol{B}\boldsymbol{M}_-^2 \boldsymbol{z}|] \\
\leq&C_*(\mathbb{E}[\mathrm{Tr}(\boldsymbol{M}_-^2)]\mathbb{E}[\mathrm{Tr}(\boldsymbol{M}_- \boldsymbol{B}\boldsymbol{M}_-)] + \mathbb{E}[\mathrm{Tr}(\boldsymbol{M}_- \boldsymbol{B}\boldsymbol{M}_-^2)]) \\
\leq&\frac{C_*\rho_\lambda^4(n)}{n}\|\overline{\boldsymbol{M}}^{1/2}\boldsymbol{B}\overline{\boldsymbol{M}}^{1/2}\|_{op},
\end{aligned}
$$

where we use

$$
\mathrm{Tr}(\boldsymbol{M}_- \boldsymbol{B}\boldsymbol{M}_-) \leq \|\boldsymbol{M}_-^{1/2}\boldsymbol{B}\boldsymbol{M}_-^{1/2}\|_{op}\|\boldsymbol{M}_-\|_F, \qquad \mathrm{Tr}(\boldsymbol{M}_- \boldsymbol{B}\boldsymbol{M}_-^2) \leq \|\boldsymbol{M}_-\|_{op}\|\boldsymbol{M}_-^{1/2}\boldsymbol{B}\boldsymbol{M}_-^{1/2}\|_{op}\|\boldsymbol{M}_-\|_F,
$$

and then apply Lemma 4 and (Misiakiewicz & Saeed, 2024, Lemma 4).

Thus, combining the two parts, we have:

$$
\begin{aligned}
|n\mathbb{E}[\mathrm{Tr}(\overline{\boldsymbol{Q}}_- \Delta_{11}^\top \boldsymbol{M}_{-1} \boldsymbol{B}\boldsymbol{M}_{-1}\Delta_{11})] &- \frac{n}{(1+\kappa)^2}\mathrm{Tr}(\overline{Q}_-\mathbb{E}[\mathrm{Tr}(\boldsymbol{B}\boldsymbol{M}^2)])| \\
&\leq C_* \frac{\log(n)\rho_\lambda^{5/2}(n)}{\sqrt{n}}\|\overline{\boldsymbol{M}}^{1/2}\boldsymbol{B}\overline{\boldsymbol{M}}^{1/2}\|_{op}\sqrt{\boldsymbol{u}^\top \overline{\boldsymbol{M}}\boldsymbol{u}\boldsymbol{v}^\top \overline{\boldsymbol{M}}\boldsymbol{v}}.
\end{aligned}
$$

**Bounding** $\mathbb{E}[\mathrm{Tr}(\overline{\boldsymbol{Q}}_- \Delta_{11}^\top \boldsymbol{M}_{-1} \boldsymbol{B}\boldsymbol{M}_{-1}\Delta_{12})]$

We expand the term as follows:

$$
(1+\kappa)^2\mathbb{E}[\mathrm{Tr}(\overline{\boldsymbol{Q}}_- \Delta_{11}^\top \boldsymbol{M}_{-1} \boldsymbol{B}\boldsymbol{M}_{-1}\Delta_{11})] = \mathbb{E}\left[\frac{\mathrm{Tr}(\overline{\boldsymbol{Q}}_- \boldsymbol{M}_1 \boldsymbol{B}\boldsymbol{M}_1 \boldsymbol{z}_1 \boldsymbol{z}_1^\top)(S_1 - \kappa)}{(1+S_1)}\right] - \mathbb{E}\left[\frac{\mathrm{Tr}(\overline{\boldsymbol{Q}}_- \boldsymbol{z}_1 \boldsymbol{z}_1^\top \boldsymbol{M}_1 \boldsymbol{B}\boldsymbol{M}_1 \boldsymbol{z}_1 \boldsymbol{z}_1^\top)(S_1 - \kappa)}{(1+S_1)}\right].
$$

For the first term, we have

$$
\mathbb{E}\left[\frac{\mathrm{Tr}(\overline{\boldsymbol{Q}}_-\boldsymbol{M}_1\boldsymbol{B}\boldsymbol{M}_1\boldsymbol{z}_1\boldsymbol{z}_1^\top)(S_1-\kappa)}{(1+S_1)}\right]
$$

$$
\leq \mathbb{E}[(S_1-\kappa)^3]^{\frac{1}{3}}\mathbb{E}[(\boldsymbol{z}_1^\top\overline{\boldsymbol{M}}_-\boldsymbol{v})^3]^{\frac{1}{3}}\mathbb{E}[(\boldsymbol{u}\overline{\boldsymbol{M}}_-\boldsymbol{M}_-\boldsymbol{B}\boldsymbol{M}_-\boldsymbol{z}_1)^3]^{1/3}
$$

$$
\leq C_*\frac{\rho_\lambda(n)}{\sqrt{n}}\mathbb{E}[\sqrt{\boldsymbol{v}^\top\overline{\boldsymbol{M}}_-^2\boldsymbol{v}\boldsymbol{u}\overline{\boldsymbol{M}}_-(\boldsymbol{M}_-\boldsymbol{B}\boldsymbol{M}_-)^2\overline{\boldsymbol{M}}_-\boldsymbol{u}}]
$$

$$
\leq C_*\frac{\rho_\lambda(n)}{\sqrt{n}}\mathbb{E}[\|\boldsymbol{M}_-\boldsymbol{B}\boldsymbol{M}_-\|_{op}]\sqrt{\boldsymbol{v}^\top\overline{\boldsymbol{M}}_-^2\boldsymbol{v}\boldsymbol{u}\overline{\boldsymbol{M}}_-^2\boldsymbol{u}}
$$

$$
\leq C_*\frac{\rho_\lambda(n)^4}{n^{\frac{5}{2}}}\|\overline{\boldsymbol{M}}^{1/2}\boldsymbol{B}\overline{\boldsymbol{M}}^{1/2}\|_{op}\sqrt{\boldsymbol{v}^\top\overline{\boldsymbol{M}}\boldsymbol{v}\boldsymbol{u}\overline{\boldsymbol{M}}\boldsymbol{u}}.
$$

For the second term, we have

$$
\mathbb{E}\left[\frac{\mathrm{Tr}(\overline{\boldsymbol{Q}}_-\boldsymbol{z}_1\boldsymbol{z}_1^\top\boldsymbol{M}_1\boldsymbol{B}\boldsymbol{M}_1\boldsymbol{z}_1\boldsymbol{z}_1^\top)(S_1-\kappa)}{(1+S_1)}\right]
$$

$$
\leq \mathbb{E}[(S_1-\kappa)^3]^{\frac{1}{3}}\mathbb{E}[(\boldsymbol{z}_1^\top\boldsymbol{M}_{-1}\boldsymbol{B}\boldsymbol{M}_{-1}\boldsymbol{z}_1)^3]^{\frac{1}{3}}\mathbb{E}[(\boldsymbol{z}_1^\top\overline{\boldsymbol{Q}}_-\boldsymbol{z}_1)^3]^{\frac{1}{3}}
$$

$$
\leq C_*\frac{\rho_\lambda(n)}{\sqrt{n}}\mathbb{E}[\mathrm{Tr}(\boldsymbol{B}\boldsymbol{M}_-^2)]\|\overline{\boldsymbol{Q}}_-\|_F
$$

$$
\leq C_*\frac{\rho_\lambda(n)^4}{n^{\frac{3}{2}}}\|\overline{\boldsymbol{M}}^{1/2}\boldsymbol{B}\overline{\boldsymbol{M}}^{1/2}\|_{op}\sqrt{\boldsymbol{v}^\top\overline{\boldsymbol{M}}\boldsymbol{v}\boldsymbol{u}\overline{\boldsymbol{M}}\boldsymbol{u}}.
$$

Thus, by combining all the terms, we have

$$
|n\mathbb{E}[\mathrm{Tr}(\overline{\boldsymbol{Q}}_-\Delta_{11}^\top\boldsymbol{M}_{-1}\boldsymbol{B}\boldsymbol{M}_{-1}\Delta_{12})]| \leq C_*\frac{\rho_\lambda(n)^4}{\sqrt{n}}\|\overline{\boldsymbol{M}}^{1/2}\boldsymbol{B}\overline{\boldsymbol{M}}^{1/2}\|_{op}\sqrt{\boldsymbol{v}^\top\overline{\boldsymbol{M}}^2\boldsymbol{v}\boldsymbol{u}\overline{\boldsymbol{M}}^2\boldsymbol{u}}.
$$

**Bounding** $\mathbb{E}[\mathrm{Tr}(\overline{\boldsymbol{Q}}_-\Delta_{12}^\top\boldsymbol{M}_{-1}\boldsymbol{B}\boldsymbol{M}_{-1}\Delta_{13})]$

We have

$$
|(1+\kappa)^2\mathbb{E}[\mathrm{Tr}(\overline{\boldsymbol{Q}}_-\Delta_{12}^\top\boldsymbol{M}_{-1}\boldsymbol{B}\boldsymbol{M}_{-1}\Delta_{13})]| \leq \mathbb{E}[\mathrm{Tr}(\overline{\boldsymbol{Q}}_-\boldsymbol{z}_1\boldsymbol{z}_1^\top\boldsymbol{M}_1\boldsymbol{B}\boldsymbol{M}_1\boldsymbol{z}_1\boldsymbol{z}_1^\top\boldsymbol{M}_1)(S_1-\kappa)],
$$

which is of lower order compared to $\mathbb{E}[\mathrm{Tr}(\overline{\boldsymbol{Q}}_-\Delta_{11}^\top\boldsymbol{M}_{-1}\boldsymbol{B}\boldsymbol{M}_{-1}\Delta_{12})]$, due to the extra $\boldsymbol{M}_1$ term.

**Bounding** $\mathbb{E}[\mathrm{Tr}(\overline{\boldsymbol{Q}}_-\Delta_{11}^\top\boldsymbol{M}_{-1}\boldsymbol{B}\boldsymbol{M}_{-2}\Delta_{21})]$

Following the same steps as in (Defilippis et al., 2024, Proof of Claim 2), we have

$$
\mathbb{E}[\mathrm{Tr}(\overline{\boldsymbol{Q}}_-\Delta_{11}^\top\boldsymbol{M}_{-1}\boldsymbol{B}\boldsymbol{M}_{-2}\Delta_{21})] = \mathbb{E}\left[\frac{\mathrm{Tr}(\overline{\boldsymbol{Q}}_-\Delta_{11}^\top\boldsymbol{M}_{12}\boldsymbol{z}_2\boldsymbol{z}_2^\top\boldsymbol{M}_{12}\boldsymbol{B}\boldsymbol{M}_{12}\boldsymbol{z}_1\boldsymbol{z}_1^\top\boldsymbol{M}_{12}\Delta_{12})}{(1+\tilde{S}_1)(1+\tilde{S}_2)}\right],
$$

and

$$
(1+\kappa)^2\mathrm{Tr}(\overline{\boldsymbol{Q}}_-\Delta_{11}^\top\boldsymbol{M}_{-1}\boldsymbol{B}\boldsymbol{M}_{-2}\Delta_{21})
$$
$$
= \boldsymbol{z}_2^\top\boldsymbol{M}_{12}\boldsymbol{B}\boldsymbol{M}_{12}\boldsymbol{z}_1\boldsymbol{z}_1^\top\boldsymbol{M}_{12}\boldsymbol{z}_2(\boldsymbol{z}_2^\top\boldsymbol{M}_{12}\boldsymbol{z}_1\boldsymbol{z}_1^\top\overline{\boldsymbol{Q}}_-\boldsymbol{z}_2 - \mathrm{Tr}(\boldsymbol{M}_{12}\boldsymbol{z}_1\boldsymbol{z}_1^\top\overline{\boldsymbol{Q}}_-))
$$
$$
+ \boldsymbol{z}_2^\top\boldsymbol{M}_{12}\boldsymbol{B}\boldsymbol{M}_{12}\boldsymbol{z}_1\boldsymbol{z}_2^\top\overline{\boldsymbol{Q}}_-\boldsymbol{M}_{12}\boldsymbol{z}_2\boldsymbol{z}_1^\top\boldsymbol{M}_{12}\boldsymbol{z}_2 - \boldsymbol{z}_2^\top\boldsymbol{M}_{12}^2\boldsymbol{z}_1\boldsymbol{z}_1^\top\boldsymbol{M}_{12}\overline{\boldsymbol{Q}}_-\boldsymbol{M}_{12}\boldsymbol{z}_2.
$$

For the first term, we have

$$
|\mathbb{E}[\boldsymbol{z}_2^\top\boldsymbol{M}_{12}\boldsymbol{B}\boldsymbol{M}_{12}\boldsymbol{z}_1\boldsymbol{z}_1^\top\boldsymbol{M}_{12}\boldsymbol{z}_2(\boldsymbol{z}_2^\top\boldsymbol{M}_{12}\boldsymbol{z}_1\boldsymbol{z}_1^\top\overline{\boldsymbol{Q}}_-\boldsymbol{z}_2 - \mathrm{Tr}(\boldsymbol{M}_{12}\boldsymbol{z}_1\boldsymbol{z}_1^\top\overline{\boldsymbol{Q}}_-))]|
$$
$$
\leq \mathbb{E}[(\boldsymbol{z}_2^\top\boldsymbol{M}_{12}\boldsymbol{B}\boldsymbol{M}_{12}\boldsymbol{z}_1)^3]^{1/3}\mathbb{E}[(\boldsymbol{z}_1^\top\boldsymbol{M}_{12}\boldsymbol{z}_2)^3]^{1/2}\mathbb{E}[(\boldsymbol{z}_2^\top\boldsymbol{M}_{12}\boldsymbol{z}_1\boldsymbol{z}_1^\top\overline{\boldsymbol{Q}}_-\boldsymbol{z}_2 - \mathrm{Tr}(\boldsymbol{M}_{12}\boldsymbol{z}_1\boldsymbol{z}_1^\top\overline{\boldsymbol{Q}}_-))^3]^{1/3}.
$$

We first have

$$
\begin{aligned}
\mathbb{E}[(\boldsymbol{z}_2^\top \boldsymbol{M}_{12}\boldsymbol{B}\boldsymbol{M}_{12}\boldsymbol{z}_1)^3]^{1/3} \leq & C_*(\mathbb{E}[\|\boldsymbol{M}_{12}\boldsymbol{B}\boldsymbol{M}_{12}\|_F^3]^{1/3} + \mathbb{E}[\|\boldsymbol{M}_{12}\boldsymbol{B}\boldsymbol{M}_{12}^2\boldsymbol{B}\boldsymbol{M}_{12}\|_F^{3/2}]^{1/3}) \\
\leq & C_*\|\overline{\boldsymbol{M}}^{1/2}\boldsymbol{B}\overline{\boldsymbol{M}}^{1/2}\|_{op}(\mathbb{E}[\|\boldsymbol{M}_{12}\|_F^3]^{1/3} + \mathbb{E}[\|\boldsymbol{M}_{12}\|_{op}^{3/2}\|\boldsymbol{M}_{12}\|_F^{3/2}]^{1/3}).
\end{aligned}
$$

The rest of the proof follows the same steps in the corresponding part in (Defilippis et al., 2024, Proof of Claim 2), and we just note that

$$
\|\overline{\boldsymbol{Q}}_-\|_F = \sqrt{\boldsymbol{u}^\top \overline{\boldsymbol{M}}_-^2 \boldsymbol{u}\boldsymbol{v}^\top \overline{\boldsymbol{M}}_-^2 \boldsymbol{v}},
$$

which implies that

$$
\begin{aligned}
& |\mathbb{E}[\boldsymbol{z}_2^\top \boldsymbol{M}_{12}\boldsymbol{B}\boldsymbol{M}_{12}\boldsymbol{z}_1\boldsymbol{z}_1^\top \boldsymbol{M}_{12}\boldsymbol{z}_2(\boldsymbol{z}_2^\top \boldsymbol{M}_{12}\boldsymbol{z}_1\boldsymbol{z}_1^\top \overline{\boldsymbol{Q}}_-\boldsymbol{z}_2 - \mathrm{Tr}(\boldsymbol{M}_{12}\boldsymbol{z}_1\boldsymbol{z}_1^\top \overline{\boldsymbol{Q}}_-))]| \\
& \leq C_* \frac{\rho_\lambda^6(n)}{n^{3/2}}\|\overline{\boldsymbol{M}}^{1/2}\boldsymbol{B}\overline{\boldsymbol{M}}^{1/2}\|_{op}\sqrt{\boldsymbol{u}^\top \overline{\boldsymbol{M}}^2 \boldsymbol{u}\boldsymbol{v}^\top \overline{\boldsymbol{M}}^2 \boldsymbol{v}} \\
& \leq C_* \frac{\rho_\lambda^8(n)}{n^{5/2}}\|\overline{\boldsymbol{M}}^{1/2}\boldsymbol{B}\overline{\boldsymbol{M}}^{1/2}\|_{op}\sqrt{\boldsymbol{u}^\top \overline{\boldsymbol{M}}\boldsymbol{u}\boldsymbol{v}^\top \overline{\boldsymbol{M}}\boldsymbol{v}}.
\end{aligned}
$$

The second and the third terms are of lower order compared to the first, term following the same steps as in (Defilippis et al., 2024, Proof of Claim 2).

**Bounding** $\mathbb{E}[\mathrm{Tr}(\overline{\boldsymbol{Q}}_-\Delta_{12}^\top \boldsymbol{M}_{-1}\boldsymbol{B}\boldsymbol{M}_{-2}\Delta_{22})]$

We follow the same steps as in the corresponding parts in (Defilippis et al., 2024, Proof of Claim 2), where we define

$$
\boldsymbol{D}_i = \frac{S_i - \kappa}{1 + S_i}\boldsymbol{z}_i\boldsymbol{z}_i^\top, \quad \tilde{\boldsymbol{D}}_i = \frac{\tilde{S}_i - \kappa}{1 + \tilde{S}_i}\boldsymbol{z}_i\boldsymbol{z}_i^\top.
$$

Then, the term can be expanded as

$$
\begin{aligned}
(1 + \kappa)^2 \mathbb{E}[\mathrm{Tr}(\overline{\boldsymbol{Q}}_-\Delta_{12}^\top \boldsymbol{M}_{12}\boldsymbol{B}\boldsymbol{M}_{12}\Delta_{22})] \\
\leq \mathrm{Tr}(\overline{\boldsymbol{Q}}\tilde{\boldsymbol{D}}_1\boldsymbol{M}_{12}\boldsymbol{B}\boldsymbol{M}_{12}\tilde{\boldsymbol{D}}_2) - 2\mathbb{E}[\delta_1(\boldsymbol{z}_1\boldsymbol{M}_{12}\boldsymbol{z}_2)^2\boldsymbol{z}_1\boldsymbol{M}_{12}\boldsymbol{B}\boldsymbol{M}_{12}\tilde{\boldsymbol{D}}_{22}\tilde{\boldsymbol{Q}}_-\boldsymbol{z}_1] \\
+ \mathbb{E}[\delta_1\delta_2(\boldsymbol{z}_1^\top \boldsymbol{M}_{12}\boldsymbol{z}_2)^4\boldsymbol{z}_1^\top \boldsymbol{M}_{12}\boldsymbol{z}_2\boldsymbol{z}_2^\top \overline{\boldsymbol{Q}}_-\boldsymbol{z}_1].
\end{aligned}
$$

Following the same steps as in (Defilippis et al., 2024, Proof of Claim 2), we have

$$
\begin{aligned}
|\mathbb{E}[\mathrm{Tr}(\overline{\boldsymbol{Q}}\tilde{\boldsymbol{D}}_1\boldsymbol{M}_{12}\boldsymbol{B}\boldsymbol{M}_{12}\tilde{\boldsymbol{D}}_2)]| \leq & \|\overline{\boldsymbol{Q}}\|_F\mathbb{E}[\|\boldsymbol{M}_{12}\boldsymbol{B}\boldsymbol{M}_{12}\|_{op}^2]^{1/2}\mathbb{E}[\|\tilde{\boldsymbol{D}}\|_{op}^4]^{1/2} \\
\leq & \mathbb{E}[\|\boldsymbol{M}^{1/2}\boldsymbol{B}\boldsymbol{M}^{1/2}\|_{op}^4]^{1/4}\mathbb{E}[\|\boldsymbol{M}_{12}\|_{op}^4]^{1/4}\mathbb{E}[\|\tilde{\boldsymbol{D}}\|_{op}^4]^{1/2}\sqrt{\boldsymbol{u}^\top \overline{\boldsymbol{M}}_-^2 \boldsymbol{u}\boldsymbol{v}^\top \overline{\boldsymbol{M}}_-^2 \boldsymbol{v}} \\
\leq & \rho_\lambda(n)^2\|\overline{\boldsymbol{M}}^{1/2}\boldsymbol{B}\overline{\boldsymbol{M}}^{1/2}\|_{op}\mathbb{E}[\|\boldsymbol{M}_{12}\|_{op}^4]^{1/4}\mathbb{E}[\|\tilde{\boldsymbol{D}}\|_{op}^4]^{1/2}\sqrt{\boldsymbol{u}^\top \overline{\boldsymbol{M}}\boldsymbol{u}\boldsymbol{v}^\top \overline{\boldsymbol{M}}\boldsymbol{v}},
\end{aligned}
$$

and the remaining steps are the same.

For the rest of the term, we just show that $\mathbb{E}[(\boldsymbol{z}_1^\top \boldsymbol{M}_{12}\boldsymbol{B}\boldsymbol{M}_{12}\boldsymbol{z}_2)^q]^{1/q}$ can be upper bound similarly to $\mathbb{E}[(\boldsymbol{z}_1^\top \boldsymbol{M}_{12}\boldsymbol{z}_2)^q]^{1/q}$ with an extra factor of $\|\overline{\boldsymbol{M}}^{1/2}\boldsymbol{B}\overline{\boldsymbol{M}}^{1/2}\|_{op}$.

To see this, we have:

$$
\begin{aligned}
\mathbb{E}[(\boldsymbol{z}_1^\top \boldsymbol{M}_{12}\boldsymbol{B}\boldsymbol{M}_{12}\boldsymbol{z}_2)^q]^{1/q} \leq & \mathbb{E}[\|\boldsymbol{M}_{12}\boldsymbol{B}\boldsymbol{M}_{12}\|_F^q]^{1/q} + \mathbb{E}[\|\boldsymbol{M}_{12}\boldsymbol{B}\boldsymbol{M}_{12}^2\boldsymbol{B}\boldsymbol{M}_{12}\|_F^{q/2}]^{1/q} \\
\leq & \mathbb{E}[\|\boldsymbol{M}_{12}^{1/2}\|_{op}^{3q}]^{1/3q}\mathbb{E}[\|\boldsymbol{M}_{12}^{1/2}\boldsymbol{B}\boldsymbol{M}_{12}^{1/2}\|_{op}^{3q}]^{1/3q}\mathbb{E}[\|\boldsymbol{M}_{12}^{1/2}\|_F^{3q}]^{1/3q} \\
= & \mathbb{E}[\|\boldsymbol{M}_{12}\|_{op}^{3q/2}]^{1/3q}\mathbb{E}[\|\boldsymbol{M}_{12}^{1/2}\boldsymbol{B}\boldsymbol{M}_{12}^{1/2}\|_{op}^{3q}]^{1/3q}\mathbb{E}[\mathrm{Tr}(\boldsymbol{M}_{12})^{3q/2}]^{1/3q} \\
\leq & C_* \frac{\rho_\lambda^2(n)}{\sqrt{n}}\|\overline{\boldsymbol{M}}^{1/2}\boldsymbol{B}\overline{\boldsymbol{M}}^{1/2}\|_{op},
\end{aligned}
$$

where we use Lemma 4 and (Misiakiewicz & Saeed, 2024, Lemma 4.(b)).

**Combining all terms.** The rest of the terms, as discussed in (Misiakiewicz & Saeed, 2024, Proof of Proposition 4), are of lower order compared to the terms upper bounded in the previous section. Combining the upper bounds finishes the proof of Claim 2.

### C.4.2. BOUNDING THE MARTINGALE PART

Define

$$\Delta_i = (\mathbb{E}_i - \mathbb{E}_{i-1})(\mathrm{Tr}(\boldsymbol{AMBM}) - \mathrm{Tr}(\boldsymbol{AM}_i\boldsymbol{BM}_i)).$$

First, we have

$$
\begin{aligned}
\mathbb{E}_i[|\Delta_i|^2]^{1/2} &\leq \mathbb{E}[\|\boldsymbol{M}^{1/2}\boldsymbol{BM}^{1/2}\|_{op}^2]^{1/2}\mathbb{E}[(\boldsymbol{u}^\top \boldsymbol{Muv}^\top \boldsymbol{Mv})]^{1/2} \\
&\leq C_*\rho_\lambda(n)^2\|\overline{\boldsymbol{M}}^{1/2}\boldsymbol{B}\overline{\boldsymbol{M}}^{1/2}\|_{op}\sqrt{\boldsymbol{u}^\top \overline{\boldsymbol{M}}\boldsymbol{u}\boldsymbol{v}^\top \overline{\boldsymbol{M}}\boldsymbol{v}}.
\end{aligned}
$$

Next, we show a high probability bound on $|\Delta_i|$. We note that

$$
\begin{aligned}
\boldsymbol{u}^\top \boldsymbol{MBMv} - \boldsymbol{u}^\top \boldsymbol{M}_i\boldsymbol{BM}_i\boldsymbol{v} = &- \frac{\boldsymbol{u}^\top \boldsymbol{M}_i\boldsymbol{z}_i\boldsymbol{z}_i^\top \boldsymbol{M}_i\boldsymbol{BM}_i\boldsymbol{v}}{1 + S_i} - \frac{\boldsymbol{u}^\top \boldsymbol{M}_i\boldsymbol{BM}_i\boldsymbol{z}_i\boldsymbol{z}_i^\top \boldsymbol{M}_i\boldsymbol{v}}{1 + S_i} \\
&+ \frac{\boldsymbol{u}^\top \boldsymbol{M}_i\boldsymbol{z}_i\boldsymbol{z}_i\boldsymbol{M}_i\boldsymbol{BM}_i\boldsymbol{z}_i\boldsymbol{z}_i^\top \boldsymbol{M}_i\boldsymbol{v}}{(1 + S_i)^2}.
\end{aligned}
$$

We have that, with probability at least $1 - n^{-D}$,

$$
\begin{aligned}
\left|\mathbb{E}_i\left[\frac{\boldsymbol{u}^\top \boldsymbol{M}_i\boldsymbol{z}_i\boldsymbol{z}_i^\top \boldsymbol{M}_i\boldsymbol{BM}_i\boldsymbol{v}}{1 + S_i}\right]\right| &\leq \mathbb{E}_i[(\boldsymbol{u}^\top \boldsymbol{M}_i\boldsymbol{z}_i)^2]^{1/2}\mathbb{E}_i[(\boldsymbol{z}_i^\top \boldsymbol{M}_i\boldsymbol{BM}_i\boldsymbol{v})^2]^{1/2} \\
&\leq C_* \log(n)\mathbb{E}_i[\boldsymbol{u}^\top \boldsymbol{M}_i^2\boldsymbol{u}]^{1/2}\mathbb{E}_i[\boldsymbol{v}^\top \boldsymbol{M}_i\boldsymbol{BM}_i^2\boldsymbol{BM}_i\boldsymbol{v}]^{1/2} \\
&\leq C_* \log(n)\sqrt{\mathbb{E}_i[\|\boldsymbol{M}_i^{1/2}\boldsymbol{BM}^{1/2}\|_{op}^2\boldsymbol{u}^\top \boldsymbol{M}_i^2\boldsymbol{uv}^\top \boldsymbol{M}_i^2\boldsymbol{v}]} \\
&\leq C_* \frac{\log(n)\rho_\lambda^2(n)}{n}\|\overline{\boldsymbol{M}}^{1/2}\boldsymbol{B}\overline{\boldsymbol{M}}^{1/2}\|_{op}\sqrt{\boldsymbol{u}^\top \overline{\boldsymbol{M}}\boldsymbol{uv}^\top \overline{\boldsymbol{M}}\boldsymbol{v}},
\end{aligned}
$$

where we use (Misiakiewicz & Saeed, 2024, Lemma 1), that $\|\boldsymbol{M}_i\|_{op} \leq C_* \frac{\rho_\lambda(n)}{n}$, (Misiakiewicz & Saeed, 2024, Lemma 4), and Lemma 4. The same bounds hold for $\mathbb{E}_i\left[\frac{\boldsymbol{u}^\top \boldsymbol{M}_i\boldsymbol{BM}_i\boldsymbol{z}_i\boldsymbol{z}_i^\top \boldsymbol{M}_i\boldsymbol{v}}{1+S_i}\right]$ by symmetry of $\boldsymbol{u}, \boldsymbol{v}$.

Next, we have

$$
\begin{aligned}
\left|\mathbb{E}_i\frac{\boldsymbol{u}^\top \boldsymbol{M}_i\boldsymbol{z}_i\boldsymbol{z}_i\boldsymbol{M}_i\boldsymbol{BM}_i\boldsymbol{z}_i\boldsymbol{z}_i^\top \boldsymbol{M}_i\boldsymbol{v}}{(1 + S_i)^2}\right| &\leq \mathbb{E}_i[(\boldsymbol{z}_i\boldsymbol{M}_i\boldsymbol{uu}^\top \boldsymbol{M}_i\boldsymbol{z}_i)^{3/2}]^{1/3}\mathbb{E}_i[(\boldsymbol{z}_i\boldsymbol{M}_i\boldsymbol{vv}^\top \boldsymbol{M}_i\boldsymbol{z}_i)^{3/2}]^{1/3}\mathbb{E}_i[(\boldsymbol{z}_i^\top \boldsymbol{M}_i\boldsymbol{BM}_i\boldsymbol{z}_i)^3]^{1/3} \\
&\leq C_* \log(n)\mathbb{E}_i[(\boldsymbol{u}^\top \boldsymbol{M}_i^2\boldsymbol{u}]^{1/2}\mathbb{E}_i[\boldsymbol{v}^\top \boldsymbol{M}_i^2\boldsymbol{v}]^{1/2}\mathbb{E}_i[\mathrm{Tr}(\boldsymbol{M}_i\boldsymbol{BM}_i)] \\
&\leq C_* \frac{\log(n)\rho_\lambda^4(n)}{n}\|\overline{\boldsymbol{M}}^{1/2}\boldsymbol{B}\overline{\boldsymbol{M}}^{1/2}\|_{op}\mathrm{Tr}(\boldsymbol{B}\overline{\boldsymbol{M}})\sqrt{\boldsymbol{u}^\top \overline{\boldsymbol{M}}\boldsymbol{uv}^\top \overline{\boldsymbol{M}}\boldsymbol{v}} \\
&\leq C_* \frac{\log(n)\rho_\lambda^5(n)}{n}\|\overline{\boldsymbol{M}}^{1/2}\boldsymbol{B}\overline{\boldsymbol{M}}^{1/2}\|_{op}\sqrt{\boldsymbol{u}^\top \overline{\boldsymbol{M}}\boldsymbol{uv}^\top \overline{\boldsymbol{M}}\boldsymbol{v}},
\end{aligned}
$$

where in the last step we use $\mathrm{Tr}(\overline{\boldsymbol{M}}) \leq C_*\rho_\lambda(n)$.

Thus, we have that, with probability at least $1 - n^{-D}$,

$$\Delta_i \leq C_* \frac{\log(n)\rho_\lambda^5(n)}{n}\|\overline{\boldsymbol{M}}^{1/2}\boldsymbol{B}\overline{\boldsymbol{M}}^{1/2}\|_{op}\sqrt{\boldsymbol{u}^\top \overline{\boldsymbol{M}}\boldsymbol{uv}^\top \overline{\boldsymbol{M}}\boldsymbol{v}}.$$

Then, by following the same steps as in Appendix C.1, we obtain that, with probability at least $1 - n^{-D}$,

$$\mathrm{Tr}(\boldsymbol{AMBM}) - \mathbb{E}[\mathrm{Tr}(\boldsymbol{AMBM})] \leq \frac{\log(n)\rho_\lambda^5(n)}{n}\|\overline{\boldsymbol{M}}^{1/2}\boldsymbol{B}\overline{\boldsymbol{M}}^{1/2}\|_{op}\sqrt{\boldsymbol{u}^\top \overline{\boldsymbol{M}}\boldsymbol{uv}^\top \overline{\boldsymbol{M}}\boldsymbol{v}}. \tag{62}$$

Finally, combining (62) and (61), we obtain that, with probability at least $1 - n^{-D}$,

$$|\text{Tr}(\boldsymbol{A}\boldsymbol{M}\boldsymbol{B}\boldsymbol{M}) - \Psi_2(\lambda_*; \boldsymbol{v}\boldsymbol{u}^\top, \boldsymbol{B})| \leq C_* \frac{\rho_\lambda^6(n)\log^{5/2}(n)}{\sqrt{n}}\tilde{\Psi}_2(\lambda_*; \boldsymbol{v}\boldsymbol{u}^\top, \boldsymbol{B}).$$

$\square$

## C.5. Deterministic equivalent of $\text{Tr}(\boldsymbol{A}(\boldsymbol{Z}^\top\boldsymbol{Z})\boldsymbol{M}\boldsymbol{B}\boldsymbol{M}(\boldsymbol{Z}^\top\boldsymbol{Z}))$

For simplicity, we define $\boldsymbol{M}_{12}$ to be $\boldsymbol{M}$ removing $\boldsymbol{x}_1, \boldsymbol{x}_2$. We also define

$$\kappa = \mathbb{E}[\text{Tr}(\boldsymbol{M}_-)],$$
$$S_i = \boldsymbol{z}_i^\top\boldsymbol{M}_{-i}\boldsymbol{z}_i, \qquad \tilde{S}_i = \boldsymbol{z}_i^\top\boldsymbol{M}_{12}\boldsymbol{z}_i, \qquad i \in \{1, 2\}.$$

### C.5.1. BOUNDING THE DETERMINISTIC PART

By exchangeability, we decompose the term as follows:

$$\mathbb{E}[\text{Tr}(\boldsymbol{A}(\boldsymbol{Z}^\top\boldsymbol{Z})\boldsymbol{M}\boldsymbol{B}\boldsymbol{M}(\boldsymbol{Z}^\top\boldsymbol{Z}))] = n\mathbb{E}[\boldsymbol{u}^\top\boldsymbol{z}_1\boldsymbol{z}_1^\top\boldsymbol{M}\boldsymbol{B}\boldsymbol{M}\boldsymbol{z}_1\boldsymbol{z}_1^\top\boldsymbol{v}]$$
$$+ n(n-1)\mathbb{E}[\text{Tr}(\boldsymbol{u}^\top\boldsymbol{z}_1\boldsymbol{z}_1^\top\boldsymbol{M}\boldsymbol{B}\boldsymbol{M}\boldsymbol{z}_2\boldsymbol{z}_2^\top\boldsymbol{v})].$$

**Deterministic equivalent of $n\mathbb{E}[\boldsymbol{u}^\top\boldsymbol{z}_1\boldsymbol{z}_1^\top\boldsymbol{M}\boldsymbol{B}\boldsymbol{M}\boldsymbol{z}_1\boldsymbol{z}_1^\top\boldsymbol{v}]$.** By using Sherman-Morrison formula, we have

$$\mathbb{E}[\boldsymbol{u}^\top\boldsymbol{z}_1\boldsymbol{z}_1^\top\boldsymbol{M}\boldsymbol{B}\boldsymbol{M}\boldsymbol{z}_1\boldsymbol{z}_1^\top\boldsymbol{v}] = \frac{\mathbb{E}[(\boldsymbol{z}_1^\top\boldsymbol{v}\boldsymbol{u}^\top\boldsymbol{z}_1)(\boldsymbol{z}_1\boldsymbol{M}_{-1}\boldsymbol{B}\boldsymbol{M}_{-1}\boldsymbol{z}_1)]}{(1+S_1)^2}$$
$$= \frac{\mathbb{E}[(\boldsymbol{z}_1^\top\boldsymbol{v}\boldsymbol{u}^\top\boldsymbol{z}_1)(\boldsymbol{z}_1\boldsymbol{M}_{-1}\boldsymbol{B}\boldsymbol{M}_{-1}\boldsymbol{z}_1)]}{(1+\kappa)^2}$$
$$+ \mathbb{E}\left[\frac{(\kappa - S_1)(2 + S_1 + \kappa)}{(1+\kappa)^2(1+S_1)^2}(\boldsymbol{z}_1^\top\boldsymbol{v}\boldsymbol{u}^\top\boldsymbol{z}_1)(\boldsymbol{z}_1\boldsymbol{M}_{-1}\boldsymbol{B}\boldsymbol{M}_{-1}\boldsymbol{z}_1)\right].$$

For the first term, we have

$$n\left|\frac{\mathbb{E}[(\boldsymbol{z}_1^\top\boldsymbol{v}\boldsymbol{u}^\top\boldsymbol{z}_1)(\boldsymbol{z}_1\boldsymbol{M}_{-1}\boldsymbol{B}\boldsymbol{M}_{-1}\boldsymbol{z}_1)]}{(1+\kappa)^2} - \frac{\boldsymbol{v}^\top\boldsymbol{u}\mathbb{E}[\text{Tr}(\boldsymbol{M}_{-1}\boldsymbol{B}\boldsymbol{M}_{-1})]}{(1+\kappa)^2}\right|$$
$$= n\left|\frac{\mathbb{E}[(\boldsymbol{z}_1^\top\boldsymbol{v}\boldsymbol{u}^\top\boldsymbol{z}_1 - \boldsymbol{u}^\top\boldsymbol{v})(\boldsymbol{z}_1\boldsymbol{M}_{-1}\boldsymbol{B}\boldsymbol{M}_{-1}\boldsymbol{z}_1 - \text{Tr}(\boldsymbol{M}_{-1}\boldsymbol{B}\boldsymbol{M}_{-1}))]}{(1+\kappa)^2}\right|$$
$$\leq \frac{1}{n}\tilde{\mu}_-^2\mathbb{E}[(\boldsymbol{z}_1^\top\boldsymbol{v}\boldsymbol{u}^\top\boldsymbol{z}_1 - \boldsymbol{u}^\top\boldsymbol{v})^2]^{1/2}\mathbb{E}[(\boldsymbol{z}_1\boldsymbol{M}_{-1}\boldsymbol{B}\boldsymbol{M}_{-1}\boldsymbol{z}_1 - \text{Tr}(\boldsymbol{M}_{-1}\boldsymbol{B}\boldsymbol{M}_{-1}))^2]^{1/2}$$
$$\leq C_* \frac{\tilde{\mu}_-^2}{n}\|\boldsymbol{v}\boldsymbol{u}^\top\|_F\mathbb{E}[\|\boldsymbol{M}_{-1}\boldsymbol{B}\boldsymbol{M}_{-1}\|_F]$$
$$\leq C_* \frac{\rho_\lambda(n)^2}{\sqrt{n}}\frac{1}{n^2}\mu_*^2\|\boldsymbol{u}\|_2\|\boldsymbol{v}\|_2\|\boldsymbol{B}\|_{op}^{1/2}\text{Tr}(\boldsymbol{B}\overline{\boldsymbol{M}})^{1/2}$$
$$\leq C_* \frac{\rho_\lambda(n)^2}{\sqrt{n}}\|\boldsymbol{u}\|_2\|\boldsymbol{v}\|_2\|\boldsymbol{B}\|_{op}.$$

Also, from (Misiakiewicz & Saeed, 2024, Proposition 6), we obtain that

$$\left|\frac{n\boldsymbol{v}^\top\boldsymbol{u}\mathbb{E}[\text{Tr}(\boldsymbol{M}_{-1}\boldsymbol{B}\boldsymbol{M}_{-1})]}{(1+\kappa)^2} - \frac{\mu_*^2}{n}\frac{\boldsymbol{v}^\top\boldsymbol{u}\text{Tr}(\boldsymbol{B}\overline{\boldsymbol{M}}^2)}{1 - \frac{\mu_*^2}{n}\text{Tr}(\overline{\boldsymbol{M}}^2)}\right| \leq C_* \frac{\rho_\lambda(n)^6}{\sqrt{n}}\frac{\mu_*^2}{n}\frac{\boldsymbol{v}^\top\boldsymbol{u}\text{Tr}(\boldsymbol{B}\overline{\boldsymbol{M}}^2)}{1 - \frac{\mu_*^2}{n}\text{Tr}(\overline{\boldsymbol{M}}^2)}.$$

Thus, we have

$$\left| \frac{n\mathbb{E}[(z_1^\top v u^\top z_1)(z_1 M_{-1} B M_{-1} z_1)]}{(1+\kappa)^2} - \frac{\mu_*^2}{n} \frac{v^\top u \mathrm{Tr}(B\overline{M}^2)}{1 - \frac{\mu_*^2}{n}\mathrm{Tr}(\overline{M}^2)} \right| \le C_* \frac{\rho_\lambda(n)^6}{\sqrt{n}} \left( \|u\|_2 \|v\|_2 \|B\|_{op} \right.$$

$$\left. + \frac{\mu_*^2}{n} \frac{v^\top u \mathrm{Tr}(B\overline{M}^2)}{1 - \frac{\mu_*^2}{n}\mathrm{Tr}(\overline{M}^2)} \right).$$

For the second term, we have

$$\left| \mathbb{E}\left[ \frac{(\kappa - S_1)(2 + S_1 + \kappa)}{(1+\kappa)^2(1+S_1)^2}(z_1^\top v u^\top z_1)(z_1 M_{-1} B M_{-1} z_1) \right] \right|$$

$$\le \mathbb{E}[(\kappa - S_1)^3]^{1/3}\mathbb{E}[(z_1^\top v u^\top z_1)^3]^{1/3}\mathbb{E}[(z_1 M_{-1} B M_{-1} z_1)^3]^{1/3}$$

$$\le C_* \frac{\rho_\lambda(n)}{\sqrt{n}} \|u\|_2 \|v\|_2 \mathbb{E}[\mathrm{Tr}(M_{-1} B M_{-1})]$$

$$\le C_* \frac{\rho_\lambda(n)}{\sqrt{n}} \|u\|_2 \|v\|_2 \|B\|_{op},$$

where in the last step we use Lemma 3.

**Deterministic equivalent of** $n(n-1)\mathbb{E}[\mathrm{Tr}(u^\top z_1 z_1^\top M B M z_2 z_2^\top)]$. We decompose the term as follows:

$$\mathbb{E}[\mathrm{Tr}(u^\top z_1 z_1^\top M B M z_2 z_2^\top)] = \mathbb{E}\left[ \frac{u^\top z_1 z_1^\top M_{12} B M_{12} z_2 z_2^\top v}{(1+S_1)(1+S_2)} \right] \tag{63}$$

$$- \mathbb{E}\left[ \frac{z_1^\top v u^\top z_2 z_2^\top M_{12} z_1 z_1^\top M_{12} B M_{12} z_1}{(1+S_1)(1+S_2)(1+\tilde{S}_1)} \right] \tag{64}$$

$$- \mathbb{E}\left[ \frac{z_1^\top v u^\top z_2 z_2^\top M_{12} B M_{12} z_2 z_2^\top M_{12} z_1}{(1+S_1)(1+S_2)(1+\tilde{S}_2)} \right] \tag{65}$$

$$+ \mathbb{E}\left[ \frac{z_1^\top v u^\top z_2 z_2^\top M_{12} z_1 z_1^\top M_{12} B M_{12} z_2 z_2^\top M_{12} z_1}{(1+S_1)(1+S_2)(1+\tilde{S}_1)(1+\tilde{S}_2)} \right]. \tag{66}$$

For (63), we have

$$(63) = \mathbb{E}\left[ \frac{u^\top z_1 z_1^\top M_{12} B M_{12} z_2 z_2^\top v}{(1+S_1)(1+S_2)} \right]$$

$$= \mathbb{E}\left[ \frac{z_1^\top B M_{12} A M_{12} z_1}{(1+\kappa)^2} \right] + \mathbb{E}\left[ \frac{(\kappa - S_1)(1+\kappa) + (\kappa - S_2)(1+S_1)}{(1+S_1)(1+S_2)(1+\kappa)^2} u^\top z_1 z_1^\top M_{12} B M_{12} z_2 z_2^\top v \right]$$

$$= \mathbb{E}\left[ \frac{\mathrm{Tr}(A M_{12} B M_{12})}{(1+\kappa)^2} \right] + \mathbb{E}\left[ \frac{(\kappa - S_1)(1+\kappa) + (\kappa - S_2)(1+S_1)}{(1+S_1)(1+S_2)(1+\kappa)^2} u^\top z_1 z_1^\top M_{12} B M_{12} z_2 z_2^\top v \right].$$

For the first term, we have

$$\left| n(n-1)\mathbb{E}\left[ \frac{\mathrm{Tr}(A M_{12} B M_{12})}{(1+\kappa)^2} \right] - \mu_*^2 \mathrm{Tr}(A\overline{M}B\overline{M}) - \frac{\overline{\mu}_*^4}{n} \frac{u^\top \overline{M}^2 v \mathrm{Tr}(B\overline{M}^2)}{1 - \frac{\overline{\mu}_*^2}{n}\mathrm{Tr}(\overline{M}^2)} \right|$$

$$\le C_* \frac{\rho_\lambda(n)^6}{\sqrt{n}} \left( \mu_*^2 \sqrt{u^\top \overline{M}u v^\top \overline{M}v} \frac{\|B\|_{op}}{n} + \frac{\mu_*^4}{n} \frac{|u^\top M^2 v| \mathrm{Tr}(B M^2)}{1 - \frac{\tilde{\mu}_*^2}{n}\mathrm{Tr}(\overline{M}^2)} \right).$$

For the second term, we have

$$
n(n-1)\left|\mathbb{E}\left[\frac{(\kappa-S_1)(1+\kappa)+(\kappa-S_2)(1+S_1)}{(1+S_1)(1+S_2)(1+\kappa)^2}\boldsymbol{u}^\top\boldsymbol{z}_1\boldsymbol{z}_1^\top\boldsymbol{M}_{12}\boldsymbol{B}\boldsymbol{M}_{12}\boldsymbol{z}_2\boldsymbol{z}_2^\top\boldsymbol{v}\right]\right|
$$

$$
\leq C_*\frac{\log(n)\rho_\lambda(n)}{\sqrt{n}}n^2\mathbb{E}[|\boldsymbol{z}_1^\top\boldsymbol{M}_{12}\boldsymbol{B}\boldsymbol{M}_{12}\boldsymbol{z}_2\boldsymbol{z}_2^\top\boldsymbol{v}\boldsymbol{u}^\top\boldsymbol{z}_1|^2]^{1/2}
$$

$$
\leq C_*\frac{\log(n)\rho_\lambda(n)}{\sqrt{n}}n^2\mathbb{E}[\|\boldsymbol{M}_{12}\boldsymbol{B}\boldsymbol{M}_{12}\boldsymbol{z}_2\boldsymbol{z}_2^\top\boldsymbol{v}\boldsymbol{u}^\top\|_F^2]^{1/2}
$$

$$
\leq C_*\frac{\log(n)\rho_\lambda(n)}{\sqrt{n}}n^2\mathbb{E}[(\boldsymbol{z}_2^\top\boldsymbol{v}\boldsymbol{v}\boldsymbol{z}_2)^2]^{1/4}\mathbb{E}[(\boldsymbol{z}_2^\top\boldsymbol{M}_{12}\boldsymbol{B}\boldsymbol{M}_{12}\boldsymbol{u}\boldsymbol{u}^\top\boldsymbol{M}_{12}\boldsymbol{B}\boldsymbol{M}_{12}\boldsymbol{z}_2)^2]^{1/4}
$$

$$
\leq C_*\frac{\log(n)\rho_\lambda(n)}{\sqrt{n}}n^2\|\boldsymbol{v}\|_2\mathbb{E}[\boldsymbol{u}^\top\boldsymbol{M}_{12}\boldsymbol{B}\boldsymbol{M}_{12}^2\boldsymbol{B}\boldsymbol{M}_{12}\boldsymbol{u}]^{1/2}
$$

$$
\leq C_*\frac{\log(n)\rho_\lambda(n)}{\sqrt{n}}n^2\|\boldsymbol{u}\|_2\|\boldsymbol{v}\|_2\|\boldsymbol{B}\|_{op}\mathbb{E}[\|\boldsymbol{M}_{12}^2\|_{op}^2]^{1/2}
$$

$$
\leq C_*\frac{\log(n)\rho_\lambda(n)}{\sqrt{n}}\|\boldsymbol{u}\|_2\|\boldsymbol{v}\|_2\|\boldsymbol{B}\|_{op},
$$

where in the last step we use (Misiakiewicz & Saeed, 2024, Lemma 4.(b)).

For (64) and (65), note that (64) = (65) by exchangability of $\boldsymbol{z}_1, \boldsymbol{z}_2$, and we have

$$
(64) = \mathbb{E}\left[\frac{\boldsymbol{z}_1^\top\boldsymbol{v}\boldsymbol{u}^\top\boldsymbol{z}_2\boldsymbol{z}_2^\top\boldsymbol{M}_{12}\boldsymbol{z}_1\boldsymbol{z}_1^\top\boldsymbol{M}_{12}\boldsymbol{B}\boldsymbol{M}_{12}\boldsymbol{z}_1}{(1+S_1)(1+S_2)(1+\tilde{S}_1)}\right]
$$

$$
= \mathbb{E}\left[\frac{\boldsymbol{z}_1^\top\boldsymbol{v}\boldsymbol{u}^\top\boldsymbol{z}_2\boldsymbol{z}_2^\top\boldsymbol{M}_{12}\boldsymbol{z}_1\boldsymbol{z}_1^\top\boldsymbol{M}_{12}\boldsymbol{B}\boldsymbol{M}_{12}\boldsymbol{z}_1}{(1+\kappa)^3}\right]
$$

$$
+ \mathbb{E}\left[\frac{(\kappa-S_1)(1+\kappa)^2+(\kappa-S_2)(1+S_1)(1+\kappa)+(\kappa-\tilde{S}_1)(1+S_1)(1+S_2)}{(1+\kappa)^3(1+S_1)(1+S_2)(1+\tilde{S}_1)}\boldsymbol{z}_1^\top\boldsymbol{v}\boldsymbol{u}^\top\boldsymbol{z}_2\boldsymbol{z}_2^\top\boldsymbol{M}_{12}\boldsymbol{z}_1\boldsymbol{z}_1^\top\boldsymbol{M}_{12}\boldsymbol{B}\boldsymbol{M}_{12}\boldsymbol{z}_1\right].
$$

For the first term, we have

$$
\left|n^2\mathbb{E}\left[\frac{\boldsymbol{z}_1^\top\boldsymbol{v}\boldsymbol{u}^\top\boldsymbol{z}_2\boldsymbol{z}_2^\top\boldsymbol{M}_{12}\boldsymbol{z}_1\boldsymbol{z}_1^\top\boldsymbol{M}_{12}\boldsymbol{B}\boldsymbol{M}_{12}\boldsymbol{z}_1}{(1+\kappa)^3}\right] - \frac{n^2}{(1+\kappa)^3}\mathbb{E}[\mathrm{Tr}(\boldsymbol{v}\boldsymbol{u}^\top\boldsymbol{M}_{12})]\mathbb{E}[\mathrm{Tr}(\boldsymbol{B}\boldsymbol{M}_{12}^2)]\right|
$$

$$
= \left|\mathbb{E}\left[\frac{\tilde{\mu}_-^3}{n}\boldsymbol{z}_1^\top\boldsymbol{v}\boldsymbol{u}^\top\boldsymbol{M}_{12}\boldsymbol{z}_1\boldsymbol{z}_1^\top\boldsymbol{M}_{12}\boldsymbol{B}\boldsymbol{M}_{12}\boldsymbol{z}_1\right] - \frac{\tilde{\mu}_-^3}{n}\mathbb{E}[\mathrm{Tr}(\boldsymbol{v}\boldsymbol{u}^\top\boldsymbol{M}_{12})]\mathbb{E}[\mathrm{Tr}(\boldsymbol{B}\boldsymbol{M}_{12}^2)]\right|
$$

$$
\leq n^2\mathbb{E}\left[(\boldsymbol{z}_1^\top\boldsymbol{v}\boldsymbol{u}^\top\boldsymbol{M}_{12}\boldsymbol{z}_1 - \mathrm{Tr}(\boldsymbol{v}\boldsymbol{u}^\top\boldsymbol{M}_{12}))^2\right]^{1/2}\mathbb{E}\left[(\boldsymbol{z}_1^\top\boldsymbol{M}_{12}\boldsymbol{B}\boldsymbol{M}_{12}\boldsymbol{z}_1 - \mathbb{E}[\mathrm{Tr}(\boldsymbol{B}\boldsymbol{M}_{12}^2)^2]\right]^{1/2}
$$

$$
\leq C_*n^2\mathbb{E}[\|\boldsymbol{v}\boldsymbol{u}^\top\boldsymbol{M}_{12}\|_F^2]^{1/2}\mathbb{E}[\|\boldsymbol{M}_{12}\boldsymbol{B}\boldsymbol{M}_{12}\|_F^2]^{1/2}
$$

$$
\leq C_*\frac{\mu_\lambda^3(n)}{\sqrt{n}}\|\boldsymbol{u}\|_2\|\boldsymbol{v}\|_2\|\boldsymbol{B}\|_{op}.
$$

Next, we derive

$$
\left|\frac{\tilde{\mu}_-^3}{n}\mathbb{E}[\mathrm{Tr}(\boldsymbol{v}\boldsymbol{u}^\top\boldsymbol{M}_{12})]\mathbb{E}[\mathrm{Tr}(\boldsymbol{B}\boldsymbol{M}_{12}^2)] - \frac{\mu_*^3}{n}\frac{\boldsymbol{u}^\top\overline{\boldsymbol{M}}\boldsymbol{v}\mathrm{Tr}(\boldsymbol{B}\overline{\boldsymbol{M}}^2)}{1-\frac{\mu_*^2}{n}\mathrm{Tr}(\overline{\boldsymbol{M}}^2)}\right|
$$

$$
\leq n^2\left|\mathbb{E}[\mathrm{Tr}(\boldsymbol{v}\boldsymbol{u}^\top\boldsymbol{M}_{12}) - \boldsymbol{u}^\top\overline{\boldsymbol{M}}\boldsymbol{v}]\right|\mathbb{E}[\mathrm{Tr}(\boldsymbol{B}\boldsymbol{M}_{12}^2)] + n^2|\mathbb{E}[\boldsymbol{u}^\top\boldsymbol{M}_{12}\boldsymbol{v}]|\left|\mathbb{E}[\mathrm{Tr}(\boldsymbol{B}\boldsymbol{M}_{12}^2)] - \frac{\mathrm{Tr}(\boldsymbol{B}\overline{\boldsymbol{M}}^2)}{1-\frac{\mu_*^2}{n}\mathrm{Tr}(\overline{\boldsymbol{M}}^2)}\right|
$$

$$
\leq C_*\frac{\rho_\lambda^6(n)}{\sqrt{n}}\|\boldsymbol{u}\|_2\|\boldsymbol{v}\|_2\|B\|_{op},
$$

where we use (Misiakiewicz & Saeed, 2024, Proposition 4).

For (66), we can easily see that it has lower order compared to (66), due to an extra $\boldsymbol{z}_2^\top \boldsymbol{M}_{12}\boldsymbol{z}_1$ factor.

Finally, by combining the upper bounds for (63)–(66) and noting that

$$\Psi_3(\lambda_*; \boldsymbol{A}, \boldsymbol{B}) = \mu_*^2 \mathrm{Tr}(\boldsymbol{A}\overline{\boldsymbol{M}}\boldsymbol{B}) + \frac{\mu_*^2}{n}\frac{\boldsymbol{v}^\top \boldsymbol{u}\,\mathrm{Tr}(\boldsymbol{B}\overline{\boldsymbol{M}}^2)}{1 - \frac{\mu_*^2}{n}\mathrm{Tr}(\overline{\boldsymbol{M}}^2)} - 2\frac{\mu_*^3}{n}\frac{\boldsymbol{u}^\top \overline{\boldsymbol{M}}\boldsymbol{v}\,\mathrm{Tr}(\boldsymbol{B}\overline{\boldsymbol{M}}^2)}{1 - \frac{\mu_*^2}{n}\mathrm{Tr}(\overline{\boldsymbol{M}}^2)} + \frac{\overline{\mu}_*^4}{n}\frac{\boldsymbol{u}^\top \overline{\boldsymbol{M}}^2\boldsymbol{v}\,\mathrm{Tr}(\boldsymbol{B}\overline{\boldsymbol{M}}^2)}{1 - \frac{\overline{\mu}_*^2}{n}\mathrm{Tr}(\overline{\boldsymbol{M}}^2)},$$

we conclude that

$$|\mathbb{E}[\mathrm{Tr}(\boldsymbol{A}\boldsymbol{Z}^\top \boldsymbol{Z}\boldsymbol{M}\boldsymbol{B}\boldsymbol{M}\boldsymbol{Z}^\top \boldsymbol{Z})] - \Psi_3(\lambda_*; \boldsymbol{A}, \boldsymbol{B})| \leq C_* \frac{\rho_\lambda^6(n)}{\sqrt{n}}\tilde{\Psi}_3(\lambda_*; \boldsymbol{A}, \boldsymbol{B}).$$

### C.5.2. BOUNDING THE MARTINGALE PART

For the martingale part, we first define

$$\Delta_i = (\mathbb{E}_i - \mathbb{E}_{i-1})\boldsymbol{u}^\top (\boldsymbol{Z}^\top \boldsymbol{Z})\boldsymbol{M}\boldsymbol{B}\boldsymbol{M}(\boldsymbol{Z}^\top \boldsymbol{Z})\boldsymbol{v}.$$

We recall that $\boldsymbol{M} = \boldsymbol{\Sigma}^{1/2}(\boldsymbol{X}^\top \boldsymbol{X} + \lambda)^{-1}\boldsymbol{\Sigma}^{1/2} = (\boldsymbol{Z}^\top \boldsymbol{Z} + \lambda\boldsymbol{\Sigma}^{-1})^{-1}$ and thus $\|\boldsymbol{Z}^\top \boldsymbol{Z}\boldsymbol{M}\|_{op} \leq 1$. It is then easy to see that

$$\mathbb{E}_i[\Delta_i^2]^{1/2} \leq C_* \|\boldsymbol{u}\|_2 \|\boldsymbol{v}\|_2 \|\boldsymbol{B}\|_{op}.$$

It remains to obtain a high probability bounds on $\Delta_i$.

Using Sherman-Morrison formula and the fact that $\boldsymbol{I} - \boldsymbol{Z}^\top \boldsymbol{Z}\boldsymbol{M} = \lambda\boldsymbol{\Sigma}^{-1}\boldsymbol{M}$, we first compute

$$\boldsymbol{u}^\top (\boldsymbol{Z}^\top \boldsymbol{Z})\boldsymbol{M}\boldsymbol{B}\boldsymbol{M}(\boldsymbol{Z}^\top \boldsymbol{Z})\boldsymbol{v} - \boldsymbol{u}^\top (\boldsymbol{Z}_i^\top \boldsymbol{Z}_i)\boldsymbol{M}_i\boldsymbol{B}\boldsymbol{M}_i(\boldsymbol{Z}_i^\top \boldsymbol{Z}_i)\boldsymbol{v}$$

$$= \frac{\lambda}{1 + S_i}[\boldsymbol{u}^\top \boldsymbol{Z}_i^\top \boldsymbol{Z}_i\boldsymbol{M}_i\boldsymbol{B}\boldsymbol{M}_i\boldsymbol{z}_i\boldsymbol{z}_i^\top \boldsymbol{M}_i\boldsymbol{\Sigma}^{-1}\boldsymbol{v} + \boldsymbol{v}^\top \boldsymbol{Z}_i^\top \boldsymbol{Z}_i\boldsymbol{M}_i\boldsymbol{B}\boldsymbol{M}_i\boldsymbol{z}_i\boldsymbol{z}_i^\top \boldsymbol{M}_i\boldsymbol{\Sigma}^{-1}\boldsymbol{u}]$$

$$+ \frac{\lambda^2}{(1 + S_i)^2}\boldsymbol{z}_i^\top \boldsymbol{M}_i\boldsymbol{B}\boldsymbol{M}_i\boldsymbol{z}_i\boldsymbol{z}_i^\top \boldsymbol{M}_i\boldsymbol{\Sigma}^{-1}\boldsymbol{v}\boldsymbol{u}^\top \boldsymbol{\Sigma}^{-1}\boldsymbol{M}_i\boldsymbol{z}_i.$$

The first term can be bounded as follows with probability at least $1 - n^{-D}$:

$$\lambda\mathbb{E}_i[|\boldsymbol{u}^\top \boldsymbol{Z}_i^\top \boldsymbol{Z}_i\boldsymbol{M}_i\boldsymbol{B}\boldsymbol{M}_i\boldsymbol{z}_i\boldsymbol{z}_i^\top \boldsymbol{M}_i\boldsymbol{\Sigma}^{-1}\boldsymbol{v}|]$$

$$\leq C_* \|\boldsymbol{u}\|_2 \|\boldsymbol{v}\|_2 \mathbb{E}_i[\|\boldsymbol{Z}_i^\top \boldsymbol{Z}_i\boldsymbol{M}_i\boldsymbol{B}\boldsymbol{M}_i\|_{op}]$$

$$\leq C_* \frac{\log(n)\rho_\lambda^3(n)}{n}\|\boldsymbol{u}\|_2 \|\boldsymbol{v}\|_2 \|\boldsymbol{B}\|_{op},$$

where we use the fact that $\boldsymbol{M}_i\boldsymbol{\Sigma}^{-1} \preceq \lambda^{-1}$, $\boldsymbol{Z}_i^\top \boldsymbol{Z}_i\boldsymbol{M}_i \preceq 1$ and where we use (Misiakiewicz & Saeed, 2024, Lemma 1). The bound for $\mathbb{E}_{i-1}$ holds in the same way without the factor $\log(n)$.

For the second term, we have that, with probability at least $1 - n^{-D}$,

$$\lambda^2\mathbb{E}_i[|\boldsymbol{z}_i^\top \boldsymbol{M}_i\boldsymbol{B}\boldsymbol{M}_i\boldsymbol{z}_i\boldsymbol{z}_i^\top \boldsymbol{M}_i\boldsymbol{\Sigma}^{-1}\boldsymbol{v}\boldsymbol{u}^\top \boldsymbol{\Sigma}^{-1}\boldsymbol{M}_i\boldsymbol{z}_i|]$$

$$\leq \lambda^2\mathbb{E}_i[|\boldsymbol{z}_i^\top \boldsymbol{M}_i\boldsymbol{B}\boldsymbol{M}_i\boldsymbol{z}_i|^2]^{1/2}\mathbb{E}_i[|\boldsymbol{z}_i^\top \boldsymbol{M}_i\boldsymbol{\Sigma}^{-1}\boldsymbol{v}\boldsymbol{u}^\top \boldsymbol{\Sigma}^{-1}\boldsymbol{M}_i\boldsymbol{z}_i|^2]^{1/2}$$

$$\leq C_* \log(n)\mathbb{E}_i[\mathrm{Tr}(\boldsymbol{M}_i\boldsymbol{B}\boldsymbol{M}_i)]\|\boldsymbol{u}\|_2 \|\boldsymbol{v}\|_2$$

$$\leq C_*\mathbb{E}_i[\|\boldsymbol{M}_i\|_{op}\mathrm{Tr}(\boldsymbol{M}_i)]\|\boldsymbol{u}\|_2 \|\boldsymbol{v}\|_2$$

$$\leq C_* \frac{\log(n)\rho_\lambda^2(n)}{n}\|\boldsymbol{u}\|_2 \|\boldsymbol{v}\|_2 \|\boldsymbol{B}\|_{op},$$

where in the last step we use (Misiakiewicz & Saeed, 2024, Lemma 1). The bound for $\mathbb{E}_{i-1}$ holds in the same way without the factor $\log(n)$. Following the same steps described in Appendix C.1, we obtain that, with probability $1 - n^{-D}$,

$$|\mathrm{Tr}(\boldsymbol{A}\boldsymbol{Z}^\top \boldsymbol{Z}\boldsymbol{M}\boldsymbol{B}\boldsymbol{M}\boldsymbol{Z}^\top \boldsymbol{Z}) - \mathbb{E}[\mathrm{Tr}(\boldsymbol{A}\boldsymbol{Z}^\top \boldsymbol{Z}\boldsymbol{M}\boldsymbol{B}\boldsymbol{M}\boldsymbol{Z}^\top \boldsymbol{Z})]| \leq C_* \frac{\log(n)\rho_\lambda^2(n)}{n}\tilde{\Psi}_3(\lambda_*; \boldsymbol{A}, \boldsymbol{B}).$$

# D. Proof of Lemma 1

*Proof of Lemma 1.* **Bound the term $\langle v, (X^\top X + \lambda)^{-1} X^\top u \rangle$**

We first upper bound:

$$
\begin{aligned}
|\langle v, (X^\top X + \lambda)^{-1} X^\top u \rangle| \leq &|\langle v, (X^\top X + \lambda)^{-1} X^\top u \rangle - \mathbb{E}[\langle v, (X^\top X + \lambda)^{-1} X^\top u \rangle]| \\
&+ |\mathbb{E}[\langle v, (X^\top X + \lambda)^{-1} X^\top u \rangle]|,
\end{aligned}
$$

and bound the martingale part and the deterministic part separately.

For the martingale part, we follow the same procedure described in Section C.1, and define:

$$
\Delta_i = (\mathbb{E}_i - \mathbb{E}_{i-1}) \langle v, (X^\top X + \lambda)^{-1} X^\top u \rangle,
$$

where $\mathbb{E}_i$ is the expectation w.r.t $\{x_{i+1}, \ldots, x_n\}, \{u_{i+1}, \ldots, u_n\}$.

We have with probability at least $1 - n^{-D}$ that :

$$
\begin{aligned}
\mathbb{E}_{i-1}[\Delta_i^2]^{1/2} &\leq 2\mathbb{E}_{i-1}[(\langle v, (X^\top X + \lambda)^{-1} X^\top u \rangle)^2]^{1/2} \\
&= 2\mathbb{E}_{i-1}[(\langle \tilde{v}, MZ^\top u \rangle)^2]^{1/2} \\
&\leq 2\mathbb{E}_{i-1}[(\langle \tilde{v}, MZ^\top ZM\tilde{v} \rangle)^2]^{1/4} \mathbb{E}_{i-1}[\|u_{-i}\|_2^4]^{1/4} \\
&\leq 2\mathbb{E}_{i-1}[\|MZ^\top ZM\|_{op}^2 \|\tilde{v}\|_2^4]^{1/4} \mathbb{E}_{i-1}[\|u_{-i}\|_2^4]^{1/4} \\
&\leq C_* \frac{\rho_\lambda(n) \log^{1/2}(n)}{\sqrt{n}} \sqrt{n} \|\tilde{v}\|_2 \\
&\leq C_* \log^{1/2}(n) \rho_\lambda(n) \|\tilde{v}\|_2,
\end{aligned}
$$

where we use $\mathbb{E}[\|u\|_2^q]^{1/q} \leq C_* \sqrt{n}$.

Next, we prove high-probability bound on $\Delta_i$. Recall that

$$
\Delta_i = (\mathbb{E}_i - \mathbb{E}_{i-1})(\langle v, (X^\top X + \lambda)^{-1} X^\top u \rangle - \langle v, (X_{-i}^\top X_{-i} + \lambda)^{-1} X_{-i}^\top u_{-i} \rangle)
$$

We compute the martingale difference below:

$$
\begin{aligned}
&\langle v, (X^\top X + \lambda)^{-1} X^\top u \rangle - \langle v, (X_{-i}^\top X_{-i} + \lambda)^{-1} X_{-i}^\top u_{-i} \rangle \\
=&\frac{v^\top (X_{-i}^\top X_{-i} + \lambda)^{-1} x_i}{1 + x_i^\top (X_{-i}^\top X_{-i} + \lambda)^{-1} x_i} (u_i - x_i^\top (X_{-i}^\top X_{-i} + \lambda)^{-1} X_{-i}^\top u_{-i}) \\
=&\frac{\tilde{v}^\top M_i z_i}{1 + z_i M_i z_i} (u_i - z_i M_i Z_{-i} u_{-i}),
\end{aligned}
$$

and upper bound with probability $1 - n^{-D}$:

$$
\begin{aligned}
\mathbb{E}_i[|\tilde{v} M_i z_i u_i|^2]^{1/2} &\leq \mathbb{E}_i[|\tilde{v} M_i z_i|^4]^{1/4} \mathbb{E}[|u_i|^4]^{1/4} \\
&\leq C_* \log(n) \mathbb{E}_i[\mathrm{Tr}(\tilde{v}\tilde{v}^\top M_i^2)^2]^{\frac{1}{4}} \\
&\leq C_* \frac{\log(n) \rho_\lambda(n)}{n} \|\tilde{v}\|_2
\end{aligned}
$$

Further, we note that:

$$
\begin{aligned}
&\mathbb{E}_i[|u_i|^2]^{1/2} \leq C_* \log(n) \\
&\mathbb{E}_i[|z_i M_i Z_{-i} u_{-i}|^2]^{1/2} \leq C_* \log(n) \mathbb{E}_i[\|Z_{-i}^\top M_i^2 Z_{-i}\|_{op} \|u_i\|_2^2]^{1/2} \leq C_* \log(n) \rho_\lambda(n),
\end{aligned}
$$

which implies with probability at least $1 - n^{-D}$,

$$
|\Delta_i| \leq \mathbb{E}_i[|z_i M_i Z_{-i} u_{-i}|^2]^{1/2} \leq C_* \frac{\log(n) \rho_\lambda(n)}{n^{3/2}} \|\tilde{v}\|_2,
$$

and thus implies the bounds for the martingale part:

$$|\langle \boldsymbol{v}, (\boldsymbol{X}^\top \boldsymbol{X} + \lambda)^{-1} \boldsymbol{X}^\top \boldsymbol{u}\rangle - \mathbb{E}[\langle \boldsymbol{v}, (\boldsymbol{X}^\top \boldsymbol{X} + \lambda)^{-1} \boldsymbol{X}^\top \boldsymbol{u}\rangle]| \leq C_* \frac{\log(n)\rho_\lambda(n)}{\sqrt{n}} \|\tilde{\boldsymbol{v}}\|_2$$

To bound the deterministic part, we note by symmetry that:

$$\mathbb{E}[\langle \boldsymbol{v}, (\boldsymbol{X}^\top \boldsymbol{X} + \lambda)^{-1} \boldsymbol{X}^\top \boldsymbol{u}\rangle] = n\mathbb{E}[\frac{\tilde{\boldsymbol{v}}^\top \boldsymbol{M}_i \boldsymbol{z}_i u_i}{1 + \boldsymbol{z}_i^\top \boldsymbol{M}_i \boldsymbol{z}_i}].$$

Next we note that:

$$\left| n\mathbb{E}[\frac{\tilde{\boldsymbol{v}}^\top \boldsymbol{M}_i \boldsymbol{z}_i u_i}{1 + \boldsymbol{z}_i^\top \boldsymbol{M}_i \boldsymbol{z}_i}] - n\mathbb{E}[\frac{\tilde{\boldsymbol{v}}^\top \boldsymbol{M}_i \boldsymbol{z}_i u_i}{1 + \mathrm{Tr}(\boldsymbol{M}_i)}] \right|$$

$$\leq C_* n\mathbb{E}[(\boldsymbol{z}_i^\top \boldsymbol{M}_i \boldsymbol{z}_i - \mathrm{Tr}(\boldsymbol{M}_i))^3]^{1/3}\mathbb{E}[u_i^3]^{1/3}\mathbb{E}[(\tilde{\boldsymbol{v}}^\top \boldsymbol{M}_i \boldsymbol{z}_i)^3]^{1/3}$$

$$\leq C_* n \cdot \frac{\rho_\lambda(n)}{\sqrt{n}} \cdot \frac{\rho_\lambda(n)}{n} \|\tilde{\boldsymbol{v}}\|_2$$

$$= C_* \cdot \frac{\rho_\lambda(n)}{\sqrt{n}} \|\tilde{\boldsymbol{v}}\|_2$$

Finally, note that $\mathbb{E}[\frac{\tilde{\boldsymbol{v}}^\top \boldsymbol{M}_i \boldsymbol{z}_i u_i}{1 + \mathrm{Tr}(\boldsymbol{M}_i)}] = 0$, we have:

$$|\mathbb{E}[\langle \boldsymbol{v}, (\boldsymbol{X}^\top \boldsymbol{X} + \lambda)^{-1} \boldsymbol{X}^\top \boldsymbol{u}\rangle]| \leq C_* \cdot \frac{\rho_\lambda(n)}{\sqrt{n}} \|\tilde{\boldsymbol{v}}\|_2$$

**Bound the term** $|\langle \boldsymbol{v}, \boldsymbol{X}^\top \boldsymbol{X}(\boldsymbol{X}^\top \boldsymbol{X} + \lambda)^{-1} \boldsymbol{B}(\boldsymbol{X}^\top \boldsymbol{X} + \lambda)^{-1} \boldsymbol{X}^\top \boldsymbol{u}\rangle$

We bound the martingale and the deterministic part similarly, and we only state the differences. First note that $\langle \boldsymbol{v}, \boldsymbol{X}^\top \boldsymbol{X}(\boldsymbol{X}^\top \boldsymbol{X} + \lambda)^{-1} \boldsymbol{B}(\boldsymbol{X}^\top \boldsymbol{X} + \lambda)^{-1} \boldsymbol{X}^\top \boldsymbol{u}\rangle = \hat{\boldsymbol{v}}^\top \boldsymbol{Z}^\top \boldsymbol{Z} \boldsymbol{M} \tilde{\boldsymbol{B}} \boldsymbol{M} \boldsymbol{Z}^\top \boldsymbol{u}$, and we compute the leave-one-out different as

$$\hat{\boldsymbol{v}}^\top \boldsymbol{Z}^\top \boldsymbol{Z} \boldsymbol{M} \tilde{\boldsymbol{B}} \boldsymbol{M} \boldsymbol{Z}^\top \boldsymbol{u} - \hat{\boldsymbol{v}}^\top \boldsymbol{Z}_i^\top \boldsymbol{Z}_i \boldsymbol{M}_i \tilde{\boldsymbol{B}} \boldsymbol{M}_i \boldsymbol{Z}_i^\top \boldsymbol{u}_i$$

$$= -\frac{1}{1 + \kappa_i} \hat{\boldsymbol{v}}^\top \boldsymbol{Z}_i^\top \boldsymbol{Z}_i \boldsymbol{M}_i \boldsymbol{z}_i \boldsymbol{z}_i^\top \boldsymbol{M}_i \tilde{\boldsymbol{B}} \boldsymbol{M}_i \boldsymbol{Z}_i^\top \boldsymbol{u}_i$$

$$\quad -\frac{1}{1 + \kappa_i} \hat{\boldsymbol{v}}^\top \boldsymbol{Z}_i^\top \boldsymbol{Z}_i \boldsymbol{M}_i \tilde{\boldsymbol{B}} \boldsymbol{M}_i \boldsymbol{z}_i \boldsymbol{z}_i^\top \boldsymbol{M}_i \boldsymbol{Z}_i^\top \boldsymbol{u}_i$$

$$\quad +\frac{1}{(1 + \kappa_i)^2} \hat{\boldsymbol{v}}^\top \boldsymbol{Z}_i^\top \boldsymbol{Z}_i \boldsymbol{M}_i \boldsymbol{z}_i \boldsymbol{z}_i^\top \boldsymbol{M}_i \tilde{\boldsymbol{B}} \boldsymbol{M}_i \boldsymbol{z}_i \boldsymbol{z}_i^\top \boldsymbol{M}_i \boldsymbol{Z}_i^\top \boldsymbol{u}_i$$

$$\quad +\frac{1}{1 + \kappa_i} \hat{\boldsymbol{v}}^\top \boldsymbol{z}_i \boldsymbol{z}_i^\top \boldsymbol{M}_i \tilde{\boldsymbol{B}} \boldsymbol{M}_i \boldsymbol{Z}_i^\top \boldsymbol{u}_i$$

$$\quad -\frac{1}{(1 + \kappa_i)^2} \hat{\boldsymbol{v}}^\top \boldsymbol{z}_i \boldsymbol{z}_i^\top \boldsymbol{M}_i \tilde{\boldsymbol{B}} \boldsymbol{M}_i \boldsymbol{z}_i \boldsymbol{z}_i^\top \boldsymbol{M}_i \boldsymbol{Z}_i^\top \boldsymbol{u}_i$$

$$\quad +\frac{1}{(1 + \kappa_i)^2} \hat{\boldsymbol{v}}^\top \boldsymbol{z}_i \boldsymbol{z}_i^\top \boldsymbol{M}_i \tilde{\boldsymbol{B}} \boldsymbol{M}_i \boldsymbol{z}_i u_i$$

where we define:

$$\hat{\boldsymbol{v}} = \boldsymbol{\Sigma}^{1/2} \boldsymbol{v}, \quad \tilde{\boldsymbol{B}} = \boldsymbol{\Sigma}^{-1/2} \boldsymbol{B} \boldsymbol{\Sigma}^{-1/2}, \quad \kappa_i = \boldsymbol{z}_i^\top \boldsymbol{M}_i \boldsymbol{z}_i$$

For the martingale part, we upper bound the martingale difference $\Delta_i := (\mathbb{E}_i - \mathbb{E}_{i-1})(\hat{\boldsymbol{v}}^\top \boldsymbol{Z}^\top \boldsymbol{Z} \boldsymbol{M} \tilde{\boldsymbol{B}} \boldsymbol{M} \boldsymbol{Z}^\top \boldsymbol{u} -$

$\hat{\boldsymbol{v}}^\top \boldsymbol{Z}_i^\top \boldsymbol{Z}_i \boldsymbol{M}_i \tilde{\boldsymbol{B}} \boldsymbol{M}_i \boldsymbol{Z}_i^\top \boldsymbol{u}_i)$. We have the following upper bound with probability at least $1 - n^{-D}$:

$$
\begin{aligned}
\mathbb{E}_i[|\hat{\boldsymbol{v}}^\top \boldsymbol{Z}_i^\top \boldsymbol{Z}_i \boldsymbol{M}_i \boldsymbol{z}_i|^2]^{1/2} &\leq C_* \log(n) \mathbb{E}_i[(\hat{\boldsymbol{v}}^\top \boldsymbol{Z}_i^\top \boldsymbol{Z}_i \boldsymbol{M}_i^2 \boldsymbol{Z}_i^\top \boldsymbol{Z}_i \hat{\boldsymbol{v}})]^{1/2} \\
&\leq C_* \log(n) \|\hat{\boldsymbol{v}}\|_2 \\
\mathbb{E}[|\boldsymbol{z}_i^\top \boldsymbol{M}_i \tilde{\boldsymbol{B}} \boldsymbol{M}_i \boldsymbol{Z}_i^\top \boldsymbol{u}_i|^2]^{1/2} &\leq C_* \log(n) \mathbb{E}[\boldsymbol{u}_i^\top \boldsymbol{Z}_i \boldsymbol{M}_i \tilde{\boldsymbol{B}} \boldsymbol{M}_i^2 \tilde{\boldsymbol{B}} \boldsymbol{M}_i \boldsymbol{Z}_i^\top \boldsymbol{u}_i]^{1/2} \\
&\leq C_* \log(n) \|\tilde{\boldsymbol{B}}\|_{op} \mathbb{E}_i[\|\boldsymbol{u}_i\|_2^2 \|\boldsymbol{M}_i\|_{op}^3]^{1/2}, \\
&\leq \frac{C_* \log(n) \rho_\lambda(n)^{3/2}}{n^{3/2}} \|\tilde{\boldsymbol{B}}\|_{op} \mathbb{E}_i[\|\boldsymbol{u}_i\|_2^4]^{1/4} \\
&\leq \frac{C_* \log(n) \rho_\lambda(n)^{3/2}}{n} \|\tilde{\boldsymbol{B}}\|_{op}, \\
\mathbb{E}_i[|\hat{\boldsymbol{v}}^\top \boldsymbol{Z}_i^\top \boldsymbol{Z}_i \boldsymbol{M}_i \tilde{\boldsymbol{B}} \boldsymbol{M}_i \boldsymbol{z}_i|^2]^{1/2} &\leq C_* \log(n) \mathbb{E}[\hat{\boldsymbol{v}}^\top \boldsymbol{Z}_i^\top \boldsymbol{Z}_i \boldsymbol{M}_i \tilde{\boldsymbol{B}} \boldsymbol{M}_i^2 \tilde{\boldsymbol{B}} \boldsymbol{M}_i \boldsymbol{Z}_i^\top \boldsymbol{Z}_i \hat{\boldsymbol{v}}]^{1/2} \\
&\leq \frac{C_* \log(n) \rho_\lambda^2(n)}{n} \|\tilde{\boldsymbol{B}}\|_{op} \|\boldsymbol{v}\|_2 \\
\mathbb{E}_i[|\boldsymbol{z}_i^\top \boldsymbol{M}_i \boldsymbol{Z}_i^\top \boldsymbol{u}_i|^2]^{1/2} &\leq C_* \log(n) \mathbb{E}_i[\boldsymbol{u}_i^\top \boldsymbol{Z}_i \boldsymbol{M}_i^2 \boldsymbol{Z}_i^\top \boldsymbol{u}_i]^{1/2} \\
&\leq C_* \log(n) \rho_\lambda(n) \\
\mathbb{E}_i[|\boldsymbol{z}_i^\top \boldsymbol{M}_i \tilde{\boldsymbol{B}} \boldsymbol{M}_i \boldsymbol{z}_i|^2]^{1/2} &\leq \frac{C_* \log(n) \rho_\lambda^2(n)}{n} \|\tilde{\boldsymbol{B}}\|_{op}
\end{aligned}
$$

where we use $\mathbb{E}[\|\boldsymbol{u}_i\|_2^q]^{1/q} \leq \sqrt{n}$, and (Misiakiewicz & Saeed, 2024, Lemma 4).

Combining the above bounds, we have with probability at least $1 - n^{-D}$

$$
|\Delta_i| \leq \frac{C_* \log(n) \rho_\lambda^2(n)}{n} \|\tilde{\boldsymbol{B}}\|_{op} \|\hat{\boldsymbol{v}}\|_2
$$

Further, using the same arguments, it is easy to see:

$$
\mathbb{E}_i[|\Delta_i|^2]^{1/2} \leq C_* \log_\beta(n) \rho_\lambda^2(n) \|\tilde{\boldsymbol{B}}\|_{op} \|\hat{\boldsymbol{v}}\|_2,
$$

which implies that:

$$
|\hat{\boldsymbol{v}}^\top \boldsymbol{Z}^\top \boldsymbol{Z} \boldsymbol{M} \tilde{\boldsymbol{B}} \boldsymbol{M} \boldsymbol{Z}^\top \boldsymbol{u} - \mathbb{E}[\hat{\boldsymbol{v}}^\top \boldsymbol{Z}^\top \boldsymbol{Z} \boldsymbol{M} \tilde{\boldsymbol{B}} \boldsymbol{M} \boldsymbol{Z}^\top \boldsymbol{u}]| \leq \frac{C_* \log(n) \rho_\lambda^2(n)}{\sqrt{n}} \|\tilde{\boldsymbol{B}}\|_{op} \|\hat{\boldsymbol{v}}\|_2
$$

For the deterministic part, we first write:

$$
\begin{aligned}
\mathbb{E}[\hat{\boldsymbol{v}}^\top \boldsymbol{Z}^\top \boldsymbol{Z} \boldsymbol{M} \tilde{\boldsymbol{B}} \boldsymbol{M} \boldsymbol{Z}^\top \boldsymbol{u}] =& n \mathbb{E}[\hat{\boldsymbol{v}}^\top \boldsymbol{Z}^\top \boldsymbol{Z} \boldsymbol{M} \tilde{\boldsymbol{B}} \boldsymbol{M} \boldsymbol{z}_i u_i] \\
=& n \mathbb{E}[\frac{1}{1 + \kappa_i} \hat{\boldsymbol{v}}^\top \boldsymbol{Z}_i^\top \boldsymbol{Z}_i \boldsymbol{M}_i \tilde{\boldsymbol{B}} \boldsymbol{M}_i \boldsymbol{z}_i u_i] \\
& - n \mathbb{E}[\frac{1}{(1 + \kappa_i)^2} \hat{\boldsymbol{v}}^\top \boldsymbol{Z}_i^\top \boldsymbol{Z}_i \boldsymbol{M}_i \boldsymbol{z}_i \boldsymbol{z}_i^\top \boldsymbol{M}_i \tilde{\boldsymbol{B}} \boldsymbol{M}_i \boldsymbol{z}_i u_i] \\
& + n \mathbb{E}[\frac{1}{(1 + \kappa_i)^2} \hat{\boldsymbol{v}}^\top \boldsymbol{z}_i \boldsymbol{z}_i^\top \boldsymbol{M}_i \tilde{\boldsymbol{B}} \boldsymbol{M}_i \boldsymbol{z}_i u_i],
\end{aligned}
$$

and further bound:

$$n\big|\mathbb{E}[\frac{1}{1+\kappa_i}\hat{\boldsymbol{v}}^\top \boldsymbol{Z}_i^\top \boldsymbol{Z}_i \boldsymbol{M}_i \tilde{\boldsymbol{B}} \boldsymbol{M}_i \boldsymbol{z}_i u_i] - \mathbb{E}[\frac{1}{1+\mathrm{Tr}(\boldsymbol{M}_i)}\hat{\boldsymbol{v}}^\top \boldsymbol{Z}_i^\top \boldsymbol{Z}_i \boldsymbol{M}_i \tilde{\boldsymbol{B}} \boldsymbol{M}_i \boldsymbol{z}_i u_i]\big|$$

$$\leq C_* n \mathbb{E}[(\boldsymbol{z}_i^\top \boldsymbol{M}_i \boldsymbol{z}_i - \mathrm{Tr}(\boldsymbol{M}_i))^3]^{1/3}\mathbb{E}[(\hat{\boldsymbol{v}}^\top \boldsymbol{Z}_i^\top \boldsymbol{Z}_i \boldsymbol{M}_i \tilde{\boldsymbol{B}} \boldsymbol{M}_i \boldsymbol{z}_i)^3]^{1/3}\mathbb{E}[|u_i|^3]^{1/3}$$

$$\leq C_* n \cdot \frac{\rho_\lambda(n)}{\sqrt{n}} \cdot \frac{\rho_\lambda^2(n)}{n}\|\tilde{\boldsymbol{B}}\|_{op}\|\boldsymbol{v}\|_2 = \frac{C_* \rho_\lambda^3(n)}{\sqrt{n}}\|\tilde{\boldsymbol{B}}\|_{op}\|\boldsymbol{v}\|_2$$

$$n\big|\mathbb{E}[\frac{1}{(1+\kappa_i)^2}\hat{\boldsymbol{v}}^\top \boldsymbol{Z}_i^\top \boldsymbol{Z}_i \boldsymbol{M}_i \boldsymbol{z}_i \boldsymbol{z}_i^\top \boldsymbol{M}_i \tilde{\boldsymbol{B}} \boldsymbol{M}_i \boldsymbol{z}_i u_i]\big|$$

$$\leq n\big|\mathbb{E}[\hat{\boldsymbol{v}}^\top \boldsymbol{Z}_i^\top \boldsymbol{Z}_i \boldsymbol{M}_i \boldsymbol{z}_i \boldsymbol{z}_i^\top \boldsymbol{M}_i \tilde{\boldsymbol{B}} \boldsymbol{M}_i \boldsymbol{z}_i u_i] - \mathbb{E}[\hat{\boldsymbol{v}}^\top \boldsymbol{Z}_i^\top \boldsymbol{Z}_i \boldsymbol{M}_i \boldsymbol{z}_i \mathrm{Tr}(\boldsymbol{M}_i \tilde{\boldsymbol{B}} \boldsymbol{M}_i) u_i]\big| + n\big|\mathbb{E}[\hat{\boldsymbol{v}}^\top \boldsymbol{Z}_i^\top \boldsymbol{Z}_i \boldsymbol{M}_i \boldsymbol{z}_i \mathrm{Tr}(\boldsymbol{M}_i \tilde{\boldsymbol{B}} \boldsymbol{M}_i) u_i]\big|$$

$$= n\big|\mathbb{E}[\hat{\boldsymbol{v}}^\top \boldsymbol{Z}_i^\top \boldsymbol{Z}_i \boldsymbol{M}_i \boldsymbol{z}_i \boldsymbol{z}_i^\top \boldsymbol{M}_i \tilde{\boldsymbol{B}} \boldsymbol{M}_i \boldsymbol{z}_i u_i] - \mathbb{E}[\hat{\boldsymbol{v}}^\top \boldsymbol{Z}_i^\top \boldsymbol{Z}_i \boldsymbol{M}_i \boldsymbol{z}_i \mathrm{Tr}(\boldsymbol{M}_i \tilde{\boldsymbol{B}} \boldsymbol{M}_i) u_i]\big|$$

$$\leq n\mathbb{E}[|\hat{\boldsymbol{v}}^\top \boldsymbol{Z}_i^\top \boldsymbol{Z}_i \boldsymbol{M}_i \boldsymbol{z}_i|^3]^{1/3}\mathbb{E}[(\boldsymbol{z}_i^\top \boldsymbol{M}_i \tilde{\boldsymbol{B}} \boldsymbol{M}_i \boldsymbol{z}_i - \mathrm{Tr}(\boldsymbol{M}_i \tilde{\boldsymbol{B}} \boldsymbol{M}_i))^3]^{1/3}\mathbb{E}[u_i^3]^{1/3}$$

$$\leq C_* n \cdot \|\hat{\boldsymbol{v}}\|_2 \frac{\rho_\lambda^2(n)}{n^{3/2}}\|\tilde{\boldsymbol{B}}\|_{op} = C_* \frac{\rho_\lambda^2(n)}{n^{1/2}}\|\tilde{\boldsymbol{B}}\|_{op}\|\hat{\boldsymbol{v}}\|_2$$

$$n\big|\mathbb{E}[\frac{1}{(1+\kappa_i)^2}\hat{\boldsymbol{v}}^\top \boldsymbol{z}_i \boldsymbol{z}_i^\top \boldsymbol{M}_i \tilde{\boldsymbol{B}} \boldsymbol{M}_i \boldsymbol{z}_i u_i]\big|$$

$$\leq n\big|\mathbb{E}[\hat{\boldsymbol{v}}^\top \boldsymbol{z}_i \boldsymbol{z}_i^\top \boldsymbol{M}_i \tilde{\boldsymbol{B}} \boldsymbol{M}_i \boldsymbol{z}_i u_i] - \mathbb{E}[\hat{\boldsymbol{v}}^\top \boldsymbol{z}_i \mathrm{Tr}(\boldsymbol{M}_i \tilde{\boldsymbol{B}} \boldsymbol{M}_i) u_i]\big| + n\big|\mathbb{E}[\hat{\boldsymbol{v}}^\top \boldsymbol{z}_i \mathrm{Tr}(\boldsymbol{M}_i \tilde{\boldsymbol{B}} \boldsymbol{M}_i) u_i]\big|$$

$$\leq n\big|\mathbb{E}[\hat{\boldsymbol{v}}^\top \boldsymbol{z}_i \boldsymbol{z}_i^\top \boldsymbol{M}_i \tilde{\boldsymbol{B}} \boldsymbol{M}_i \boldsymbol{z}_i u_i] - \mathbb{E}[\hat{\boldsymbol{v}}^\top \boldsymbol{z}_i \mathrm{Tr}(\boldsymbol{M}_i \tilde{\boldsymbol{B}} \boldsymbol{M}_i) u_i]\big|$$

$$\leq C_* \frac{\rho_\lambda^2(n)}{n^{1/2}}\|\tilde{\boldsymbol{B}}\|_{op}\|\hat{\boldsymbol{v}}\|_2,$$

where we use:

$$\mathbb{E}[\hat{\boldsymbol{v}}^\top \boldsymbol{z}_i \mathrm{Tr}(\boldsymbol{M}_i \tilde{\boldsymbol{B}} \boldsymbol{M}_i) u_i] = 0, \quad \mathbb{E}[\hat{\boldsymbol{v}}^\top \boldsymbol{z}_i \mathrm{Tr}(\boldsymbol{M}_i \tilde{\boldsymbol{B}} \boldsymbol{M}_i) u_i] = 0.$$

Finally, by using that $\mathbb{E}[\frac{1}{1+\mathrm{Tr}(\boldsymbol{M}_i)}\hat{\boldsymbol{v}}^\top \boldsymbol{Z}_i^\top \boldsymbol{Z}_i \boldsymbol{M}_i \tilde{\boldsymbol{B}} \boldsymbol{M}_i \boldsymbol{z}_i u_i] = 0$, we have:

$$|\mathbb{E}[\hat{\boldsymbol{v}}^\top \boldsymbol{Z}^\top \boldsymbol{Z} \boldsymbol{M} \tilde{\boldsymbol{B}} \boldsymbol{M} \boldsymbol{Z}^\top \boldsymbol{u}]| \leq C_* \frac{\rho_\lambda^2(n)}{n^{1/2}}\|\tilde{\boldsymbol{B}}\|_{op}\|\hat{\boldsymbol{v}}\|_2$$

**Bound the term $\langle \boldsymbol{u}, \boldsymbol{X}(\boldsymbol{X}^\top \boldsymbol{X} + \lambda)^{-1}\boldsymbol{B}(\boldsymbol{X}^\top \boldsymbol{X} + \lambda)^{-1}\boldsymbol{X}^\top \boldsymbol{u}\rangle$**

We first note that $\langle \boldsymbol{u}, \boldsymbol{X}(\boldsymbol{X}^\top \boldsymbol{X} + \lambda)^{-1}\boldsymbol{B}(\boldsymbol{X}^\top \boldsymbol{X} + \lambda)^{-1}\boldsymbol{X}^\top \boldsymbol{u}\rangle = \boldsymbol{u}^\top \boldsymbol{Z} \boldsymbol{M} \tilde{\boldsymbol{B}} \boldsymbol{M} \boldsymbol{Z}^\top \boldsymbol{u}$. Then we follow the same steps as in (Defilippis et al., 2024, Proof Lemma 9 eq.(170), Proof Lemma 10 eq.(177)), and we only state the difference here.

For the martingale part, we first compute the leave-one-out difference as:

$$\boldsymbol{u}^\top \boldsymbol{Z} \boldsymbol{M} \tilde{\boldsymbol{B}} \boldsymbol{M} \boldsymbol{Z}^\top \boldsymbol{u} - \boldsymbol{u}_i^\top \boldsymbol{Z}_i \boldsymbol{M}_i \tilde{\boldsymbol{B}} \boldsymbol{M}_i \boldsymbol{Z}_i^\top \boldsymbol{u}_i$$

$$= \frac{\boldsymbol{z}_i^\top \boldsymbol{M}_i \tilde{\boldsymbol{B}} \boldsymbol{M}_i \boldsymbol{z}_i}{(1+\kappa_i)^2}(u_i - \boldsymbol{z}_i \boldsymbol{M}_i \boldsymbol{Z}_i^\top \boldsymbol{u}_i)^2 - \frac{2}{1+\kappa_i}\boldsymbol{z}_i^\top \boldsymbol{M}_i \tilde{\boldsymbol{B}} \boldsymbol{M}_i \boldsymbol{Z}_i^\top \boldsymbol{u}_i$$

For terms involving $\tilde{\boldsymbol{B}}$, recall that we have:

$$\mathbb{E}[|\boldsymbol{z}_i^\top \boldsymbol{M}_i \tilde{\boldsymbol{B}} \boldsymbol{M}_i \boldsymbol{z}_i|^2]^{1/2} \leq \frac{C_* \log(n)\rho_\lambda^2(n)}{n}\|\tilde{\boldsymbol{B}}\|_{op}$$

$$\mathbb{E}[|\boldsymbol{z}_i^\top \boldsymbol{M}_i \tilde{\boldsymbol{B}} \boldsymbol{M}_i \boldsymbol{Z}_i^\top \boldsymbol{u}_i|^2]^{1/2} \leq \frac{C_* \log(n)\rho_\lambda^{3/2}(n)}{n}\|\tilde{\boldsymbol{B}}\|_{op},$$

and the rest follows exactly the same as (Defilippis et al., 2024, Proof Lemma 9 eq.(170)), and we obtained that:

$$\boldsymbol{u}^\top \boldsymbol{Z} \boldsymbol{M} \tilde{\boldsymbol{B}} \boldsymbol{M} \boldsymbol{Z}^\top \boldsymbol{u} - \mathbb{E}[\boldsymbol{u}^\top \boldsymbol{Z} \boldsymbol{M} \tilde{\boldsymbol{B}} \boldsymbol{M} \boldsymbol{Z}^\top \boldsymbol{u}] \leq C_* \frac{\rho_\lambda(n)^3 \log^{7/2}(n)}{\sqrt{n}}\|\tilde{\boldsymbol{B}}\|_{op}$$

For the deterministic part, we have:

$$\mathbb{E}[\boldsymbol{u}^\top \boldsymbol{Z} \boldsymbol{M} \tilde{\boldsymbol{B}} \boldsymbol{M} \boldsymbol{Z}^\top \boldsymbol{u}] = n\mathbb{E}\Big[\frac{\boldsymbol{z}_i^\top \boldsymbol{M}_i \tilde{\boldsymbol{B}} \boldsymbol{M}_i \boldsymbol{z}_i}{(1+\kappa_i)^2} u_i(u_i - \boldsymbol{z}_i \boldsymbol{M}_i \boldsymbol{Z}_i^\top \boldsymbol{u}_i) - \frac{1}{1+\kappa_i} u_i \boldsymbol{z}_i^\top \boldsymbol{M}_i \tilde{\boldsymbol{B}} \boldsymbol{M}_i \boldsymbol{Z}_i^\top \boldsymbol{u}_i\Big].$$

We use

$$\mathbb{E}[|\boldsymbol{z}_i^\top \boldsymbol{M}_i \boldsymbol{B} \boldsymbol{M}_i \boldsymbol{z}_i - \mathrm{Tr}(\boldsymbol{M}_i \boldsymbol{B} \boldsymbol{M}_i)|] \le C_* \frac{\rho_\lambda^3(n)}{n^{3/2}}\|\boldsymbol{B}\|_{op}$$

$$\mathbb{E}[|\boldsymbol{z}_i^\top \boldsymbol{M}_i \tilde{\boldsymbol{B}} \boldsymbol{M}_i \boldsymbol{Z}_i^\top \boldsymbol{u}_i|^2]^{1/2} \le \frac{C_* \log \rho_\lambda^{3/2}(n)}{n}\|\tilde{\boldsymbol{B}}\|_{op}$$

and following the same steps as in (Defilippis et al., 2024, Proof Lemma 9 eq.(177)), we obtain that:

$$n\left|\mathbb{E}\Big[\frac{\boldsymbol{z}_i^\top \boldsymbol{M}_i \tilde{\boldsymbol{B}} \boldsymbol{M}_i \boldsymbol{z}_i}{(1+\kappa_i)^2} u_i(u_i - \boldsymbol{z}_i \boldsymbol{M}_i \boldsymbol{Z}_i^\top \boldsymbol{u}_i) - \frac{1}{1+\kappa_i} u_i \boldsymbol{z}_i^\top \boldsymbol{M}_i \tilde{\boldsymbol{B}} \boldsymbol{M}_i \boldsymbol{Z}_i^\top \boldsymbol{u}_i\Big] - \mathbb{E}\Big[\frac{\mathrm{Tr}(\tilde{\boldsymbol{B}} \boldsymbol{M}_i^2)}{(1+\mathrm{Tr}(\boldsymbol{M}_i^2))^2}\Big]\right| \le \frac{C_* \log \rho_\lambda^5(n)}{\sqrt{n}}\|\tilde{\boldsymbol{B}}\|_{op}$$

Finally, by (Misiakiewicz & Saeed, 2024, Proof of Proposition 6), we have:

$$\Big|\mathbb{E}\Big[\frac{\mathrm{Tr}(\tilde{\boldsymbol{B}} \boldsymbol{M}_i^2)}{(1+\mathrm{Tr}(\boldsymbol{M}_i^2))^2}\Big] - \psi_2(\lambda_*; \boldsymbol{A})\Big| \le C_* \frac{\rho_\lambda^6(n)}{n^{3/2}}$$

$\square$

# E. Technical lemmas

We first prove an auxiliary lemma, which is a generalization of (Misiakiewicz & Saeed, 2024, Lemma 2) to non-PSD matrix.

**Lemma 2.** *Assume $\boldsymbol{x}_1, \boldsymbol{x}_2 \in \mathbb{R}^d$ are independent and satisfy the same assumptions in Theorem 5. Then, for any constant $D > 0$, there exist $C_{x,D}$ such that for all matrix $\boldsymbol{B} \in \mathbb{R}^{d\times d}$ independent of $\boldsymbol{x}_1, \boldsymbol{x}_2$ we have with probability at least $1 - n^{-D}$*

$$|\boldsymbol{x}_1^\top \boldsymbol{B} \boldsymbol{x}_1 - \mathrm{Tr}(\boldsymbol{\Sigma} \boldsymbol{B})| \le C_{x,D} \log(n) \left\|\boldsymbol{\Sigma}^{\frac{1}{2}} \boldsymbol{B} \boldsymbol{\Sigma}^{\frac{1}{2}}\right\|_F.$$

*Moreover, for all integer $q$, there exist $C_{x,q}$ such that for all $\boldsymbol{B}$ independent of $\boldsymbol{x}_1, \boldsymbol{x}_1$, we have*

$$\mathbb{E}[|\boldsymbol{x}_1^\top \boldsymbol{B} \boldsymbol{x}_1 - \mathrm{Tr}(\boldsymbol{\Sigma} \boldsymbol{B})|^q]^{1/q} \le C_{x,q} \left\|\boldsymbol{\Sigma}^{\frac{1}{2}} \boldsymbol{B} \boldsymbol{\Sigma}^{\frac{1}{2}}\right\|_F.$$

*Proof of Lemma 2.* For any $\boldsymbol{B}$, we denote $\boldsymbol{A} = \boldsymbol{\Sigma}^{\frac{1}{2}}\left(\frac{\boldsymbol{B}+\boldsymbol{B}^\top}{2}\right)\boldsymbol{\Sigma}^{\frac{1}{2}}$, and we have $\boldsymbol{x}_1^\top \boldsymbol{B} \boldsymbol{x}_1 - \mathrm{Tr}(\boldsymbol{\Sigma} \boldsymbol{B}) = \boldsymbol{z}_1^\top \boldsymbol{A} \boldsymbol{z}_1 - \mathrm{Tr}(\boldsymbol{A})$, where we define $\boldsymbol{z}_1 = \boldsymbol{\Sigma}^{-\frac{1}{2}} \boldsymbol{x}_1$ and use $\boldsymbol{x}_1^\top \boldsymbol{B} \boldsymbol{x}_1 = \boldsymbol{x}_1^\top \boldsymbol{B} \boldsymbol{x}_1$ and $\mathrm{Tr}(\boldsymbol{\Sigma} \boldsymbol{B}) = \mathrm{Tr}(\boldsymbol{\Sigma} \boldsymbol{B}^\top)$ as $\boldsymbol{\Sigma}$ is symmetric.

Thus, it is sufficient to show the above bound for $\boldsymbol{A}$, and we note that $\boldsymbol{A}$ is symmetric. Define the eigen-decomposition of $\boldsymbol{A}$ as $\boldsymbol{A} = \boldsymbol{U} \boldsymbol{\Lambda} \boldsymbol{U}^\top$, with $\boldsymbol{\Lambda} = \mathrm{Diag}(\lambda_i, \ldots, \lambda_d)$. We define the positive part and the negative part of $\boldsymbol{A}$ as

$$\begin{aligned}
\boldsymbol{\Lambda}_+ &= \mathrm{Diag}((\lambda_i \vee 0)_{i=1}^d), & \boldsymbol{A}_+ &= \boldsymbol{U} \boldsymbol{\Lambda}_+ \boldsymbol{U}^\top, \\
\boldsymbol{\Lambda}_- &= -\mathrm{Diag}((\lambda_i \wedge 0)_{i=1}^d), & \boldsymbol{A}_- &= \boldsymbol{U} \boldsymbol{\Lambda}_- \boldsymbol{U}^\top,
\end{aligned}$$

where we have

$$\boldsymbol{A} = \boldsymbol{A}_+ - \boldsymbol{A}_-, \qquad \boldsymbol{A}_+, \boldsymbol{A}_- \succeq 0.$$

We have that, with probability at least $1 - n^{-D}$,

$$\begin{aligned}
|\boldsymbol{z}_1^\top \boldsymbol{A} \boldsymbol{z}_1 - \mathrm{Tr}(\boldsymbol{A})| &\le |\boldsymbol{z}_1^\top \boldsymbol{A}_+ \boldsymbol{z}_1 - \mathrm{Tr}(\boldsymbol{A}_+)| + |\boldsymbol{z}_1^\top \boldsymbol{A}_- \boldsymbol{z}_1 - \mathrm{Tr}(\boldsymbol{A}_-)| \\
&\le C_{x,D} \log(n)(\|\boldsymbol{A}_+\|_F + \|\boldsymbol{A}_-\|_F) \\
&\le C_{x,D} \log(n)\|\boldsymbol{A}\|_F,
\end{aligned}$$

where we use $\|\boldsymbol{A}_+\|_F, \|\boldsymbol{A}_-\|_F \leq \|\boldsymbol{A}\|_F$.

For the bound in expectation, we integrate the tail bound in (Misiakiewicz & Saeed, 2024, Proof of Lemma 2) and we get

$$
\begin{aligned}
\mathbb{E}_{\boldsymbol{x}}[|\boldsymbol{z}^\top \boldsymbol{A}\boldsymbol{x} - \mathrm{Tr}(\boldsymbol{A})|^q] =& q\int_0^\infty t^{q-1}\mathbb{P}\left(|\boldsymbol{z}^\top \boldsymbol{A}\boldsymbol{z} - \mathrm{Tr}(\boldsymbol{B})| \geq t\right)\,\mathrm{d}t \\
\leq& q\int_0^\infty t^{q-1}\left(\mathbb{P}\left(|\boldsymbol{z}^\top \boldsymbol{A}_+\boldsymbol{z} - \mathrm{Tr}(\boldsymbol{A}_+)| \geq \frac{t}{2}\right) + \mathbb{P}\left(|\boldsymbol{z}^\top \boldsymbol{A}_-\boldsymbol{z} - \mathrm{Tr}(\boldsymbol{A}_-)| \geq \frac{t}{2}\right)\right)\,\mathrm{d}t \\
\leq& C_{x,q}(\|\boldsymbol{A}_+\|_F^q + \|\boldsymbol{A}_-\|_F^q) \\
\leq& C_{x,q}\|\boldsymbol{A}\|_F^q,
\end{aligned}
$$

where again in the last inequality we use $\|\boldsymbol{A}_+\|_F, \|\boldsymbol{A}_-\|_F \leq \|\boldsymbol{A}\|_F$.

Finally, we note that $\|\boldsymbol{\Sigma}^{\frac{1}{2}}\boldsymbol{B}\boldsymbol{\Sigma}^{\frac{1}{2}}\|_F = \|\boldsymbol{\Sigma}^{\frac{1}{2}}\boldsymbol{B}^\top \boldsymbol{\Sigma}^{\frac{1}{2}}\|_F$, which implies that $\|\boldsymbol{A}\|_F \leq \|\boldsymbol{\Sigma}^{\frac{1}{2}}\boldsymbol{B}\boldsymbol{\Sigma}^{\frac{1}{2}}\|_F$ and concludes the proof.

$\square$

**Lemma 3.** *Under the same assumptions in Theorem 6, recall*

$$
\mu_* = \frac{n}{1 + \mathrm{Tr}(\overline{\boldsymbol{M}})}, \quad \tilde{\mu}_- = \frac{n}{1 + \mathbb{E}[\mathrm{Tr}(\boldsymbol{M}_-)]}.
$$

*Then, we have the following properties*

$$
\frac{n}{2\rho_\lambda(n)} \leq \mu_* \leq n, \quad \tilde{\mu}_- \leq \left(1 + C_*\rho_\lambda(n)^{5/2}\varphi_1(d)/n\right)\mu_*.
$$

*Proof.* The first inequality directly follows from (Misiakiewicz & Saeed, 2024, Lemma 2) using also that $1 + \mathrm{Tr}(\overline{\boldsymbol{M}}) \leq \rho_\lambda^2(n)$, and the second one follows from (Misiakiewicz & Saeed, 2024, Proof of Claim 3). $\square$

**Lemma 4.** *Under the same assumptions in (Misiakiewicz & Saeed, 2024, Lemma 4), for any p.s.d. matrix $\boldsymbol{B}$, $\ell \geq 1$ and $q > 1$, we have that, with probability at least $1 - n^{-D}$,*

$$
\mathbb{E}_i[\|\boldsymbol{M}^{\ell/2}\boldsymbol{B}\boldsymbol{M}^{\ell/2}\|_{op}^q]^{1/q} \leq C_*\frac{\rho_\lambda(n)^\ell}{n^{\ell-1}}\|\overline{\boldsymbol{M}}^{1/2}\boldsymbol{B}\overline{\boldsymbol{M}}^{1/2}\|_{op}.
$$

*Proof.* By (Misiakiewicz & Saeed, 2024, Proof Lemma 4.(b)), denoting $\mathcal{A}$ to be the event on which $\boldsymbol{M} \preceq C_*\frac{\rho_\lambda(n)}{n}$, we have $\mathrm{Pr}[\mathcal{A}] \geq 1 - n^{-D'}$. Furthermore, we have

$$
\mathbb{E}_i[\|\boldsymbol{M}^{\ell/2}\boldsymbol{B}\boldsymbol{M}^{\ell/2}\|_{op}^q]^{1/q} \leq \mathbb{E}_i[\|\boldsymbol{M}^{\ell/2}\boldsymbol{B}\boldsymbol{M}^{\ell/2}\|_{op}^q 1_{\mathcal{A}}]^{1/q} + \mathbb{E}_i[\|\boldsymbol{M}^{\ell/2}\boldsymbol{B}\boldsymbol{M}^{\ell/2}\|_{op}^q 1_{\mathcal{A}^c}]^{1/q}.
$$

For the first part, we have

$$
\begin{aligned}
\mathbb{E}_i[\|\boldsymbol{M}^{\ell/2}\boldsymbol{B}\boldsymbol{M}^{\ell/2}\|_{op}^q 1_{\mathcal{A}}]^{1/q} &\leq \mathbb{E}_i[\|\boldsymbol{M}^{\ell-1}\|_{op}\|\boldsymbol{M}^{1/2}\boldsymbol{B}\boldsymbol{M}^{1/2}\|_{op}^q 1_{\mathcal{A}}]^{1/q} \\
&\leq C_*\frac{\rho_\lambda(n)^{\ell-1}}{n^{\ell-1}}\mathbb{E}_i[\|\boldsymbol{M}^{1/2}\boldsymbol{B}\boldsymbol{M}^{1/2}\|_{op}^q 1_{\mathcal{A}}]^{1/q} \\
&\leq C_*\frac{\rho_\lambda(n)^\ell}{n^{\ell-1}}\|\overline{\boldsymbol{M}}^{1/2}\boldsymbol{B}\overline{\boldsymbol{M}}^{1/2}\|_{op},
\end{aligned}
$$

where we use $\|\boldsymbol{M}\|_{op} \leq C_*\frac{\rho_\lambda(n)}{n}$, and for the last step we define $\boldsymbol{v}$ to be the eigenvalue of $\boldsymbol{B}^{1/2}\boldsymbol{M}\boldsymbol{B}^{1/2}$ of unit norm and obtain:

$$
\begin{aligned}
\|\boldsymbol{M}^{1/2}\boldsymbol{B}\boldsymbol{M}^{1/2}\|_{op} =& \|\boldsymbol{B}^{1/2}\boldsymbol{M}\boldsymbol{B}^{1/2}\|_{op} \\
=& \boldsymbol{v}^\top \boldsymbol{B}^{1/2}\boldsymbol{M}\boldsymbol{B}^{1/2}\boldsymbol{v} \\
\leq& C_*\rho_\lambda(n)\boldsymbol{v}^\top \boldsymbol{B}^{1/2}\overline{\boldsymbol{M}}\boldsymbol{B}^{1/2}\boldsymbol{v} \\
\leq& C_*\rho_\lambda(n)\|\overline{\boldsymbol{M}}^{1/2}\boldsymbol{B}\overline{\boldsymbol{M}}^{1/2}\|_{op}.
\end{aligned}
$$

For the second part, we use the same argument as in (Misiakiewicz & Saeed, 2024, Proof of Lemma 4), and we have

$$
\begin{aligned}
\mathbb{E}_i[\|\boldsymbol{M}^{\ell/2}\boldsymbol{B}\boldsymbol{M}^{\ell/2}\|_{op}^q 1_{\mathcal{A}^c}]^{1/q} &\leq \frac{\|\boldsymbol{\Sigma}^{\ell/2}\boldsymbol{B}\boldsymbol{\Sigma}^{\ell/2}\|_{op}}{\lambda^\ell}\mathbb{E}[1_{\mathcal{A}^c}]^{1/q} \\
&\leq \frac{\|\boldsymbol{\Sigma}^{1/2}\boldsymbol{B}\boldsymbol{\Sigma}^{1/2}\|_{op}}{\lambda^\ell}\mathbb{E}[1_{\mathcal{A}^c}]^{1/q} \\
&\leq (\mu_* + \lambda)\frac{\|\overline{\boldsymbol{M}}^{1/2}\boldsymbol{B}\overline{\boldsymbol{M}}^{1/2}\|_{op}}{\lambda^\ell} \\
&\leq \frac{C_*\rho_\lambda(n)^\ell}{n^{\ell-1}}\|\overline{\boldsymbol{M}}^{1/2}\boldsymbol{B}\overline{\boldsymbol{M}}^{1/2}\|_{op},
\end{aligned}
$$

where the last step follows form the same steps in (Misiakiewicz & Saeed, 2024, Proof of Lemma 4.(b)), by picking $r = \ell - 1$. $\qquad\square$

## F. Technical results for scaling laws

### F.1. Approximation of the trace

We give a generalization of the estimation in (Defilippis et al., 2024, Appendix D). We have the conditions:

$$
s_1 + s_2 \in \{0, 1\}, 0 \leq \delta_1 \leq \gamma_1, 0 \leq \delta_2 \leq \gamma_2.
$$

Define

$$
T(\mu, \nu; (s_1, \delta_1, \gamma_1), (s_2, \delta_2, \gamma_2)) = \sum_{k=1}^\infty \frac{k^{-s_1-\delta_1\alpha}}{(k^{-\alpha}+\mu)^{\gamma_1}}\frac{k^{-s_2-\delta_2\alpha}}{(k^{-\alpha}+\nu)^{\gamma_2}}.
$$

We denote this quantity by $T^s_{\delta,(\gamma_1,\gamma_2)}$, with $s = s_1 + s_2$ and $\delta = \delta_1 + \delta_2$ for short. We assume $\mu = o_{n_t}(1)$, $\nu = o_{n_t}(1)$, noting that in all cases considered later (see Lemma 5) this condition is satisfied.

We first prove the following estimation:

$$
\sum_{k=N_1}^{N_2} k^{-b} = \begin{cases} \Theta(N_2^{1-b} - N_1^{1-b}), & 0 < b < 1, \\ \Theta(\log\frac{N_2}{N_1}), & b = 1, \\ \Theta(N_1^{1-b} - N_2^{1-b}), & b > 1, \end{cases} \tag{67}
$$

for all $N_1, N_2 \in \mathbb{N}$ such that $1 \leq N_1 < N_2$. To see this, simply note the following upper and lower bound on the sum:

$$
\int_{N_1}^{N_2+1} x^{-b}\,\mathrm{d}x \leq \sum_{k=N_1}^{N_2} k^{-b} \leq \int_{N_1-1}^{N_2} x^{-b}\,\mathrm{d}x,
$$

and in the case $N_1 = 1$, the upper bound becomes $1 + \int_1^{N_2} x^{-b}\,\mathrm{d}x$. By computing the integral on both sides, we obtain (67).

Next, to estimate the order of $T^s_{\delta,(\gamma_1,\gamma_2)}(\mu, \nu)$, we first define:

$$
\rho = \gamma_1 + \gamma_2 - \delta + \frac{1-s}{\alpha}
$$

for simplicity. Note that if $\mu = \nu$, then by (Defilippis et al., 2024, Appendix D) we immediately have $T^s_{\delta,(\gamma_1,\gamma_2)}(\mu, \nu) = \Theta(\mu^{(-\rho)\wedge 0})$. Thus w.l.o.g. we assume $0 < \mu < \nu$, and define $N_\nu = \lfloor \nu^{-\frac{1}{\alpha}} \rfloor$, $N_\mu = \lfloor \mu^{-\frac{1}{\alpha}} \rfloor$. We split $k$ into three regimes $[1, N_\nu], (N_\nu, N_\mu], (N_\mu, \infty)$, and compute the sum in the three regimes separately.

(a) For $1 \leq k \leq N_\nu$, we have $k^{-\alpha} \geq \nu > \mu$, and thus:

$$
2^{-(\gamma_1+\gamma_2)}\sum_{k=1}^{N_\nu} k^{-(1-\alpha\rho)} \leq \sum_{k=1}^{N_\nu}\frac{k^{-s-\delta\alpha}}{(k^{-\alpha}+\mu)^{\gamma_1}(k^{-\alpha}+\nu)^{\gamma_2}} \leq \sum_{k=1}^{N_\nu} k^{-(1-\alpha\rho)},
$$

implying that

$$\sum_{k=1}^{N_\nu} \frac{k^{-s-\delta\alpha}}{(k^{-\alpha}+\mu)^{\gamma_1}(k^{-\alpha}+\nu)^{\gamma_2}} = \Theta\left(\sum_{k=1}^{N_\nu} k^{-(1-\alpha\rho)}\right).$$

Therefore, by (67), we have

$$\sum_{k=1}^{N_\nu} \frac{k^{-s-\delta\alpha}}{(k^{-\alpha}+\mu)^{\gamma_1}(k^{-\alpha}+\nu)^{\gamma_2}} = \begin{cases} \Theta\left(N_\nu^{\alpha\rho}\right), & \rho > 0, \\ \Theta\left(\log N_\nu\right), & \rho = 0, \\ \Theta(1), & \rho < 0. \end{cases} \tag{68}$$

Since $N_\nu = \lfloor \nu^{-1/\alpha} \rfloor$, this can be written as

$$\sum_{k=1}^{N_\nu} \frac{k^{-s-\delta\alpha}}{(k^{-\alpha}+\mu)^{\gamma_1}(k^{-\alpha}+\nu)^{\gamma_2}} = \begin{cases} \Theta\left(\nu^{-\rho}\right), & \rho > 0, \\ \Theta\left(\log \frac{1}{\nu}\right), & \rho = 0, \\ \Theta(1), & \rho < 0. \end{cases} \tag{69}$$

(b) For $N_\nu < k \le N_\mu$, we have $\mu \le k^{-\alpha} < \nu$, and thus:

$$2^{-(\gamma_1+\gamma_2)}\nu^{-\gamma_2} \sum_{k=N_\nu+1}^{N_\mu} k^{-s-\delta\alpha+\alpha\gamma_1} \le \sum_{k=N_\nu+1}^{N_\mu} \frac{k^{-s-\delta\alpha}}{(k^{-\alpha}+\mu)^{\gamma_1}(k^{-\alpha}+\nu)^{\gamma_2}}$$

$$\le \nu^{-\gamma_2} \sum_{k=N_\nu+1}^{N_\mu} k^{-s-\delta\alpha+\alpha\gamma_1}.$$

Define

$$\rho_1 = \gamma_1 - \delta + \frac{1-s}{\alpha} = \rho - \gamma_2.$$

Then

$$-s - \delta\alpha + \alpha\gamma_1 = -(1-\alpha\rho_1).$$

Therefore,

$$\sum_{k=N_\nu+1}^{N_\mu} \frac{k^{-s-\delta\alpha}}{(k^{-\alpha}+\mu)^{\gamma_1}(k^{-\alpha}+\nu)^{\gamma_2}} = \Theta\left(\nu^{-\gamma_2} \sum_{k=N_\nu+1}^{N_\mu} k^{-(1-\alpha\rho_1)}\right).$$

Applying (67), we obtain

$$\sum_{k=N_\nu+1}^{N_\mu} \frac{k^{-s-\delta\alpha}}{(k^{-\alpha}+\mu)^{\gamma_1}(k^{-\alpha}+\nu)^{\gamma_2}} = \begin{cases} \Theta\left(\nu^{-\gamma_2}\left(N_\mu^{\alpha\rho_1}-N_\nu^{\alpha\rho_1}\right)\right), & \rho_1 > 0, \\ \Theta\left(\nu^{-\gamma_2}\log \frac{N_\mu}{N_\nu}\right), & \rho_1 = 0, \\ \Theta\left(\nu^{-\gamma_2}\left(N_\nu^{\alpha\rho_1}-N_\mu^{\alpha\rho_1}\right)\right), & \rho_1 < 0. \end{cases} \tag{70}$$

Since $N_\nu = \lfloor \nu^{-1/\alpha} \rfloor$ and $N_\mu = \lfloor \mu^{-1/\alpha} \rfloor$, this is equivalent to

$$\sum_{k=N_\nu+1}^{N_\mu} \frac{k^{-s-\delta\alpha}}{(k^{-\alpha}+\mu)^{\gamma_1}(k^{-\alpha}+\nu)^{\gamma_2}} = \begin{cases} \Theta\left(\nu^{-\gamma_2}\left(\mu^{-\rho_1}-\nu^{-\rho_1}\right)\right), & \rho_1 > 0, \\ \Theta\left(\nu^{-\gamma_2}\log \frac{\nu}{\mu}\right), & \rho_1 = 0, \\ \Theta\left(\nu^{-\gamma_2}\left(\nu^{-\rho_1}-\mu^{-\rho_1}\right)\right), & \rho_1 < 0. \end{cases} \tag{71}$$

Equivalently, using $\rho_1 = \rho - \gamma_2$,

$$\sum_{k=N_\nu+1}^{N_\mu} \frac{k^{-s-\delta\alpha}}{(k^{-\alpha}+\mu)^{\gamma_1}(k^{-\alpha}+\nu)^{\gamma_2}} = \begin{cases} \Theta\left(\nu^{-\gamma_2}\left(\mu^{-(\rho-\gamma_2)}-\nu^{-(\rho-\gamma_2)}\right)\right), & \rho > \gamma_2, \\ \Theta\left(\nu^{-\gamma_2}\log \frac{\nu}{\mu}\right), & \rho = \gamma_2, \\ \Theta\left(\nu^{-\gamma_2}\left(\nu^{-(\rho-\gamma_2)}-\mu^{-(\rho-\gamma_2)}\right)\right), & \rho < \gamma_2. \end{cases} \tag{72}$$

(c) For $k > N_\mu$, we have $k^{-\alpha} < \mu < \nu$, and thus:

$$2^{-(\gamma_1+\gamma_2)} \mu^{-\gamma_1} \nu^{-\gamma_2} \sum_{k=N_\mu+1}^{\infty} k^{-s-\delta\alpha} \leq \sum_{k=N_\mu+1}^{\infty} \frac{k^{-s-\delta\alpha}}{(k^{-\alpha}+\mu)^{\gamma_1}(k^{-\alpha}+\nu)^{\gamma_2}}$$

$$\leq \mu^{-\gamma_1} \nu^{-\gamma_2} \sum_{k=N_\mu+1}^{\infty} k^{-s-\delta\alpha}.$$

Assume $s + \delta\alpha > 1$. Then

$$\sum_{k=N_\mu+1}^{\infty} k^{-s-\delta\alpha} = \Theta\left(N_\mu^{1-s-\delta\alpha}\right).$$

Hence

$$\sum_{k=N_\mu+1}^{\infty} \frac{k^{-s-\delta\alpha}}{(k^{-\alpha}+\mu)^{\gamma_1}(k^{-\alpha}+\nu)^{\gamma_2}} = \Theta\left(\mu^{-\gamma_1}\nu^{-\gamma_2}N_\mu^{1-s-\delta\alpha}\right)$$

$$= \Theta\left(\mu^{-\gamma_1}\nu^{-\gamma_2}\mu^{\delta+\frac{s-1}{\alpha}}\right) \tag{73}$$

$$= \Theta\left(\nu^{-\gamma_2}\mu^{-\rho_1}\right)$$

$$= \Theta\left(\nu^{-\gamma_2}\mu^{-(\rho-\gamma_2)}\right).$$

Combining the three regimes, we have

$$T^s_{\delta,(\gamma_1,\gamma_2)}(\mu,\nu) = \Theta\left(A_\nu + B_{\mu,\nu} + C_{\mu,\nu}\right), \tag{74}$$

where

$$A_\nu = \begin{cases} \nu^{-\rho}, & \rho > 0, \\ \log\frac{1}{\nu}, & \rho = 0, \\ 1, & \rho < 0, \end{cases}$$

$$B_{\mu,\nu} = \begin{cases} \nu^{-\gamma_2}\left(\mu^{-(\rho-\gamma_2)}-\nu^{-(\rho-\gamma_2)}\right), & \rho > \gamma_2, \\ \nu^{-\gamma_2}\log\frac{\nu}{\mu}, & \rho = \gamma_2, \\ \nu^{-\gamma_2}\left(\nu^{-(\rho-\gamma_2)}-\mu^{-(\rho-\gamma_2)}\right), & \rho < \gamma_2, \end{cases}$$

and

$$C_{\mu,\nu} = \nu^{-\gamma_2}\mu^{-(\rho-\gamma_2)}.$$

Since $0 < \mu < \nu$, the above expression simplifies to

$$T^s_{\delta,(\gamma_1,\gamma_2)}(\mu,\nu) = \begin{cases} \Theta(1), & \rho < 0, \\ \Theta\left(\log\frac{1}{\nu}\right), & \rho = 0, \\ \Theta\left(\nu^{-\rho}\right), & 0 < \rho < \gamma_2, \\ \Theta\left(\nu^{-\gamma_2}\left(1+\log\frac{\nu}{\mu}\right)\right), & \rho = \gamma_2, \\ \Theta\left(\nu^{-\gamma_2}\mu^{-(\rho-\gamma_2)}\right), & \rho > \gamma_2. \end{cases} \tag{75}$$

Equivalently, ignoring logarithmic factors at the critical cases $\rho = 0$ and $\rho = \gamma_2$, we can write

$$T^s_{\delta,(\gamma_1,\gamma_2)}(\mu,\nu) = \Theta\left(\nu^{(-\rho)\wedge 0}\left(\nu\mu^{-1}\right)^{(\rho-\gamma_2)\vee 0}\right), \qquad 0 < \mu < \nu. \tag{76}$$

By symmetry, for $0 < \nu < \mu$,

$$T^s_{\delta,(\gamma_1,\gamma_2)}(\mu,\nu) = \Theta\left(\mu^{(-\rho)\wedge 0}\left(\mu\nu^{-1}\right)^{(\rho-\gamma_1)\vee 0}\right), \qquad 0 < \nu < \mu. \tag{77}$$

Therefore, up to logarithmic factors at the critical cases, we obtain

$$T^s_{\delta,(\gamma_1,\gamma_2)}(\mu,\nu) = \begin{cases} \Theta\left(\nu^{(-\rho)\wedge 0}\left(\nu\mu^{-1}\right)^{(\rho-\gamma_2)\vee 0}\right), & \text{if } \mu \leq \nu, \\ \Theta\left(\mu^{(-\rho)\wedge 0}\left(\mu\nu^{-1}\right)^{(\rho-\gamma_1)\vee 0}\right), & \text{if } \nu < \mu. \end{cases} \tag{78}$$

## F.2. Technical lemmas

**Lemma 5.** *We have the following approximations regarding the traces and inner products appearing in the expression of the test error:*

$$\text{Tr}(\boldsymbol{\Sigma}^2(\boldsymbol{\Sigma}+\mu_{s,2})^{-2}\boldsymbol{\Sigma}(\boldsymbol{\Sigma}+\mu_{t,2})^{-1}) \approx (\mu_{t,2}\vee\mu_{s,2})^{-\frac{1}{\alpha}},$$

$$\text{Tr}((\boldsymbol{\Sigma}+\mu_{s,2})^{-2}\boldsymbol{\Sigma}(\boldsymbol{\Sigma}+\mu_{t,2})^{-1}) \approx \begin{cases} \mu_{t,2}^{-1}\mu_{s,2}^{-\left(1+\frac{1}{\alpha}\right)}, & \mu_{s,2}\leq\mu_{t,2}, \\ \mu_{s,2}^{-2}\mu_{t,2}^{-\frac{1}{\alpha}}, & \mu_{t,2}<\mu_{s,2}, \end{cases}$$

$$\text{Tr}(\boldsymbol{\Sigma}(\boldsymbol{\Sigma}+\mu_{s,2})^{-2}\boldsymbol{\Sigma}(\boldsymbol{\Sigma}+\mu_{t,2})^{-1}) \approx \begin{cases} \mu_{t,2}^{-1}\mu_{s,2}^{-\frac{1}{\alpha}}, & \mu_{s,2}\leq\mu_{t,2}, \\ \mu_{s,2}^{-\left(1+\frac{1}{\alpha}\right)}, & \mu_{t,2}<\mu_{s,2}, \end{cases}$$

$$\text{Tr}(\boldsymbol{\Sigma}^2(\boldsymbol{\Sigma}+\mu_{s,2})^{-2}\boldsymbol{\Sigma}(\boldsymbol{\Sigma}+\mu_{t,2})^{-2}) \approx (\mu_{t,2}\vee\mu_{s,2})^{-\left(1+\frac{1}{\alpha}\right)},$$

$$\text{Tr}((\boldsymbol{\Sigma}+\mu_{s,2})^{-2}\boldsymbol{\Sigma}(\boldsymbol{\Sigma}+\mu_{t,2})^{-2}) \approx (\mu_{t,2}\vee\mu_{s,2})^{-2}(\mu_{t,2}\wedge\mu_{s,2})^{-\left(1+\frac{1}{\alpha}\right)},$$

$$\text{Tr}(\boldsymbol{\Sigma}(\boldsymbol{\Sigma}+\mu_{s,2})^{-2}\boldsymbol{\Sigma}(\boldsymbol{\Sigma}+\mu_{t,2})^{-2}) \approx (\mu_{t,2}\vee\mu_{s,2})^{-2}(\mu_{t,2}\wedge\mu_{s,2})^{-\frac{1}{\alpha}},$$

$$\text{Tr}(\boldsymbol{\Sigma}^2(\boldsymbol{\Sigma}+\mu_{s,2})^{-2}\boldsymbol{\Sigma}^2(\boldsymbol{\Sigma}+\mu_{t,2})^{-2}) \approx (\mu_{t,2}\vee\mu_{s,2})^{-\frac{1}{\alpha}}$$

$$\langle\boldsymbol{\beta}_*,\boldsymbol{\Sigma}^2(\boldsymbol{\Sigma}+\mu_{t,2})^{-2}(\boldsymbol{\Sigma}+\mu_{s,2})^{-2}\boldsymbol{\beta}_*\rangle \approx (\mu_{t,2}\vee\mu_{s,2})^{(2r-2)\wedge 0},$$

$$\langle\boldsymbol{\beta}_*,(\boldsymbol{\Sigma}+\mu_{s,2})^{-1}\boldsymbol{\beta}_*\rangle = T^1_{2r,1}(\mu_{s,2}) \approx \mu_{s,2}^{(2r-1)\wedge 0},$$

$$\langle\boldsymbol{\beta}_*,\boldsymbol{\Sigma}(\boldsymbol{\Sigma}+\mu_{s,2})^{-1}(\boldsymbol{\Sigma}+\mu_{t,2})^{-1}\boldsymbol{\beta}_*\rangle \approx (\mu_{t,2}\vee\mu_{s,2})^{(2r-1)\wedge 0},$$

$$\langle\boldsymbol{\beta}_*,\boldsymbol{\Sigma}^3(\boldsymbol{\Sigma}+\mu_{t,2})^{-2}(\boldsymbol{\Sigma}+\mu_{s,2})^{-2}\boldsymbol{\beta}_*\rangle \approx (\mu_{t,2}\vee\mu_{s,2})^{(2r-1)\wedge 0},$$

$$\langle\boldsymbol{\beta}_*,(\boldsymbol{\Sigma}+\mu_{t,2})^{-2}\boldsymbol{\beta}_*\rangle \approx \mu_{t,2}^{(2r-2)\wedge 0},$$

$$\langle\boldsymbol{\beta}_*,\boldsymbol{\Sigma}(\boldsymbol{\Sigma}+\mu_{t,2})^{-2}\boldsymbol{\beta}_*\rangle \approx \mu_{t,2}^{(2r-1)\wedge 0}.$$

*Proof.* We apply the approximation in Appendix F.1. For the trace terms, we obtain

$$\text{Tr}(\boldsymbol{\Sigma}^2(\boldsymbol{\Sigma}+\mu_{s,2})^{-2}\boldsymbol{\Sigma}(\boldsymbol{\Sigma}+\mu_{t,2})^{-1}) = T^0_{3,(2,1)}(\mu_{s,2},\mu_{t,2}) \approx (\mu_{t,2}\vee\mu_{s,2})^{-\frac{1}{\alpha}},$$

$$\text{Tr}((\boldsymbol{\Sigma}+\mu_{s,2})^{-2}\boldsymbol{\Sigma}(\boldsymbol{\Sigma}+\mu_{t,2})^{-1}) = T^0_{1,(2,1)}(\mu_{s,2},\mu_{t,2}) \approx \begin{cases} \mu_{t,2}^{-1}\mu_{s,2}^{-\left(1+\frac{1}{\alpha}\right)}, & \mu_{s,2}\leq\mu_{t,2}, \\ \mu_{s,2}^{-2}\mu_{t,2}^{-\frac{1}{\alpha}}, & \mu_{t,2}<\mu_{s,2}, \end{cases}$$

$$\text{Tr}(\boldsymbol{\Sigma}(\boldsymbol{\Sigma}+\mu_{s,2})^{-2}\boldsymbol{\Sigma}(\boldsymbol{\Sigma}+\mu_{t,2})^{-1}) = T^0_{2,(2,1)}(\mu_{s,2},\mu_{t,2}) \approx \begin{cases} \mu_{t,2}^{-1}\mu_{s,2}^{-\frac{1}{\alpha}}, & \mu_{s,2}\leq\mu_{t,2}, \\ \mu_{s,2}^{-\left(1+\frac{1}{\alpha}\right)}, & \mu_{t,2}<\mu_{s,2}, \end{cases}$$

$$\text{Tr}(\boldsymbol{\Sigma}^2(\boldsymbol{\Sigma}+\mu_{s,2})^{-2}\boldsymbol{\Sigma}(\boldsymbol{\Sigma}+\mu_{t,2})^{-2}) = T^0_{3,(2,2)}(\mu_{s,2},\mu_{t,2}) \approx (\mu_{t,2}\vee\mu_{s,2})^{-\left(1+\frac{1}{\alpha}\right)},$$

$$\text{Tr}((\boldsymbol{\Sigma}+\mu_{s,2})^{-2}\boldsymbol{\Sigma}(\boldsymbol{\Sigma}+\mu_{t,2})^{-2}) = T^0_{1,(2,2)}(\mu_{s,2},\mu_{t,2}) \approx (\mu_{t,2}\vee\mu_{s,2})^{-2}(\mu_{t,2}\wedge\mu_{s,2})^{-\left(1+\frac{1}{\alpha}\right)},$$

$$\text{Tr}(\boldsymbol{\Sigma}(\boldsymbol{\Sigma}+\mu_{s,2})^{-2}\boldsymbol{\Sigma}(\boldsymbol{\Sigma}+\mu_{t,2})^{-2}) = T^0_{2,(2,2)}(\mu_{s,2},\mu_{t,2}) \approx (\mu_{t,2}\vee\mu_{s,2})^{-2}(\mu_{t,2}\wedge\mu_{s,2})^{-\frac{1}{\alpha}}$$

$$\text{Tr}(\boldsymbol{\Sigma}^2(\boldsymbol{\Sigma}+\mu_{s,2})^{-2}\boldsymbol{\Sigma}^2(\boldsymbol{\Sigma}+\mu_{t,2})^{-2}) = T^0_{4,(2,2)}(\mu_{s,2},\mu_{t,2}) \approx (\mu_{t,2}\vee\mu_{s,2})^{-\frac{1}{\alpha}}$$

For the inner product terms, similarly,

$$\langle \boldsymbol{\beta}_*, \boldsymbol{\Sigma}^2 (\boldsymbol{\Sigma} + \mu_{\mathsf{t},2})^{-2} (\boldsymbol{\Sigma} + \mu_{\mathsf{s},2})^{-2} \boldsymbol{\beta}_* \rangle = \sum_{k=1}^{\infty} \frac{k^{-1-2(r+1)\alpha}}{(k^{-\alpha} + \mu_{\mathsf{t},2})^2 (k^{-\alpha} + \mu_{\mathsf{s},2})^2}$$
$$= T^1_{2r+2,(2,2)}(\mu_{\mathsf{t},2}, \mu_{\mathsf{s},2}) \approx (\mu_{\mathsf{t},2} \vee \mu_{\mathsf{s},2})^{(2r-2)\wedge 0},$$

$$\langle \boldsymbol{\beta}_*, (\boldsymbol{\Sigma} + \mu_{\mathsf{s},2})^{-1} \boldsymbol{\beta}_* \rangle = \sum_{k=1}^{\infty} \frac{k^{-1-2r\alpha}}{k^{-\alpha} + \mu_{\mathsf{s},2}} = T^1_{2r,1}(\mu_{\mathsf{s},2}) \approx \mu_{\mathsf{s},2}^{(2r-1)\wedge 0},$$

$$\langle \boldsymbol{\beta}_*, \boldsymbol{\Sigma}(\boldsymbol{\Sigma} + \mu_{\mathsf{s},2})^{-1} (\boldsymbol{\Sigma} + \mu_{\mathsf{t},2})^{-1} \boldsymbol{\beta}_* \rangle = \sum_{k=1}^{\infty} \frac{k^{-1-(2r+1)\alpha}}{(k^{-\alpha} + \mu_{\mathsf{t},2})(k^{-\alpha} + \mu_{\mathsf{s},2})}$$
$$= T^1_{2r+1,(1,1)}(\mu_{\mathsf{t},2}, \mu_{\mathsf{s},2}) \approx (\mu_{\mathsf{t},2} \vee \mu_{\mathsf{s},2})^{(2r-1)\wedge 0},$$

$$\langle \boldsymbol{\beta}_*, \boldsymbol{\Sigma}^3 (\boldsymbol{\Sigma} + \mu_{\mathsf{t},2})^{-2} (\boldsymbol{\Sigma} + \mu_{\mathsf{s},2})^{-2} \boldsymbol{\beta}_* \rangle = \sum_{k=1}^{\infty} \frac{k^{-1-(2r+3)\alpha}}{(k^{-\alpha} + \mu_{\mathsf{t},2})^2 (k^{-\alpha} + \mu_{\mathsf{s},2})^2}$$
$$= T^1_{2r+3,(2,2)}(\mu_{\mathsf{t},2}, \mu_{\mathsf{s},2}) \approx (\mu_{\mathsf{t},2} \vee \mu_{\mathsf{s},2})^{(2r-1)\wedge 0},$$

$$\langle \boldsymbol{\beta}_*, (\boldsymbol{\Sigma} + \mu_{\mathsf{t},2})^{-2} \boldsymbol{\beta}_* \rangle = \sum_{k=1}^{\infty} \frac{k^{-1-2r\alpha}}{(k^{-\alpha} + \mu_{\mathsf{t},2})^2} = T^1_{2r,2}(\mu_{\mathsf{t},2}) \approx \mu_{\mathsf{t},2}^{(2r-2)\wedge 0},$$

$$\langle \boldsymbol{\beta}_*, \boldsymbol{\Sigma}(\boldsymbol{\Sigma} + \mu_{\mathsf{t},2})^{-2} \boldsymbol{\beta}_* \rangle = \sum_{k=1}^{\infty} \frac{k^{-1-(2r+1)\alpha}}{(k^{-\alpha} + \mu_{\mathsf{t},2})^2} = T^1_{2r+1,2}(\mu_{\mathsf{t},2}) \approx \mu_{\mathsf{t},2}^{(2r-1)\wedge 0}.$$

This completes the proof. $\qquad\square$

Next, we compute the scaling of $\mu_{\mathsf{t},1}, \mu_{\mathsf{t},2}, \Upsilon_{n_{\mathsf{t}},p_{\mathsf{t}}}(\mu_{\mathsf{t},1}, \mu_{\mathsf{t},2}), \chi_{n_{\mathsf{t}},p_{\mathsf{t}}}(\mu_{\mathsf{t},1}, \mu_{\mathsf{t},2}), \mu_{\mathsf{s},1}, \mu_{\mathsf{s},2}, \Upsilon_{n_{\mathsf{s}},p_{\mathsf{s}}}(\mu_{\mathsf{s},1}, \mu_{\mathsf{s},2}), \chi_{n_{\mathsf{s}},p_{\mathsf{s}}}(\mu_{\mathsf{s},1}, \mu_{\mathsf{s},2})$.

**Lemma 6.** *We have the following scaling of the quantities appearing in the expression of the test error:*

$$\begin{cases} \mu_{\mathsf{t},2} \approx n_{\mathsf{t}}^{-\alpha\left(\frac{1+\gamma_{\lambda_{\mathsf{t}}}}{\alpha} \wedge 1 \wedge \gamma_{p_{\mathsf{t}}}\right)} \\ \mu_{\mathsf{t},1} \approx \begin{cases} n_{\mathsf{t}}^{-\alpha} \ if \ 1 < \left(\frac{1+\gamma_{\lambda_{\mathsf{t}}}}{\alpha} \wedge \gamma_{p_{\mathsf{t}}}\right) \\ n_{\mathsf{t}}^{-(1+\gamma_{\lambda_{\mathsf{t}}})}, \quad else. \end{cases} \end{cases} \qquad \begin{cases} \mu_{\mathsf{s},2} \approx n_{\mathsf{s}}^{-\alpha\left(\frac{\gamma_{n_{\mathsf{s}}}+\gamma_{\lambda_{\mathsf{s}}}}{\alpha\gamma_{n_{\mathsf{s}}}} \wedge 1 \wedge \frac{\gamma_{p_{\mathsf{s}}}}{\gamma_{n_{\mathsf{s}}}}\right)} \\ \mu_{\mathsf{s},1} \approx \begin{cases} n_{\mathsf{s}}^{-\alpha} \ if \ 1 < \frac{\gamma_{n_{\mathsf{s}}}+\gamma_{\lambda_{\mathsf{s}}}}{\alpha\gamma_{n_{\mathsf{s}}}} \wedge \frac{\gamma_{p_{\mathsf{s}}}}{\gamma_{n_{\mathsf{s}}}} \\ n_{\mathsf{s}}^{-\frac{\gamma_{n_{\mathsf{s}}}+\gamma_{\lambda_{\mathsf{s}}}}{\gamma_{n_{\mathsf{s}}}}}, \quad else. \end{cases} \end{cases}$$

$$\Upsilon_{n_{\mathsf{t}},p_{\mathsf{t}}}(\mu_{\mathsf{t},1}, \mu_{\mathsf{t},2}) \approx n_{\mathsf{t}}^{-1+\left(\frac{1+\gamma_{\lambda_{\mathsf{t}}}}{\alpha} \wedge 1 \wedge \gamma_{p_{\mathsf{t}}}\right)}, \qquad \Upsilon_{n_{\mathsf{s}},p_{\mathsf{s}}}(\mu_{\mathsf{s},1}, \mu_{\mathsf{s},2}) \approx n_{\mathsf{s}}^{-1+\left(\frac{1+\frac{\gamma_{\lambda_{\mathsf{s}}}}{\gamma_{n_{\mathsf{s}}}}}{\alpha} \wedge 1 \wedge \frac{\gamma_{p_{\mathsf{s}}}}{\gamma_{n_{\mathsf{s}}}}\right)},$$

$$\chi_{n_{\mathsf{t}},p_{\mathsf{t}}}(\mu_{\mathsf{t},1}, \mu_{\mathsf{t},2}) \approx p_{\mathsf{t}}^{-1} \mu_{\mathsf{t},2}^{-1-\frac{1}{\alpha}}, \qquad \chi_{n_{\mathsf{s}},p_{\mathsf{s}}}(\mu_{\mathsf{s},1}, \mu_{\mathsf{s},2}) \approx p_{\mathsf{s}}^{-1} \mu_{\mathsf{s},2}^{-1-\frac{1}{\alpha}}.$$

*Proof.* From (Defilippis et al., 2024, Appendix D) and using the relation $p_{\mathsf{s}} = n_{\mathsf{s}}^{\frac{\gamma_{p_{\mathsf{s}}}}{\gamma_{n_{\mathsf{s}}}}}, \lambda_{\mathsf{s}} = n_{\mathsf{s}}^{-\frac{\gamma_{\lambda_{\mathsf{s}}}}{\gamma_{n_{\mathsf{s}}}}}$, we have

$$\begin{cases} \mu_{\mathsf{t},2} = \frac{\mu_{\mathsf{t},2}}{p_{\mathsf{t}}} T^0_{11}(\mu_{\mathsf{t},2}) + \mu_{\mathsf{t},1} \\ \mu_{\mathsf{t},1} = \frac{\mu_{\mathsf{t},1}}{n_{\mathsf{t}}} T^0_{11}(\mu_{\mathsf{t},2}) + \frac{\lambda_{\mathsf{t}}}{n_{\mathsf{t}}} \end{cases} \qquad \begin{cases} \mu_{\mathsf{s},2} = \frac{\mu_{\mathsf{s},2}}{p_{\mathsf{s}}} T^0_{11}(\mu_{\mathsf{s},2}) + \mu_{\mathsf{s},1} \\ \mu_{\mathsf{s},1} = \frac{\mu_{\mathsf{s},1}}{n_{\mathsf{s}}} T^0_{11}(\mu_{\mathsf{s},2}) + \frac{\lambda_{\mathsf{s}}}{n_{\mathsf{s}}} \end{cases}$$

$$\Upsilon_{n_{\mathsf{t}},p_{\mathsf{t}}}(\mu_{\mathsf{t},1}, \mu_{\mathsf{t},2}) \approx n_{\mathsf{t}}^{-1+\left(\frac{1+\gamma_{\lambda_{\mathsf{t}}}}{\alpha} \wedge 1 \wedge \gamma_{p_{\mathsf{t}}}\right)}, \qquad \Upsilon_{n_{\mathsf{s}},p_{\mathsf{s}}}(\mu_{\mathsf{s},1}, \mu_{\mathsf{s},2}) \approx n_{\mathsf{s}}^{-1+\left(\frac{1+\frac{\gamma_{\lambda_{\mathsf{s}}}}{\gamma_{n_{\mathsf{s}}}}}{\alpha} \wedge 1 \wedge \frac{\gamma_{p_{\mathsf{s}}}}{\gamma_{n_{\mathsf{s}}}}\right)},$$

$$\chi_{n_{\mathsf{t}},p_{\mathsf{t}}}(\mu_{\mathsf{t},1}, \mu_{\mathsf{t},2}) \approx p_{\mathsf{t}}^{-1} \mu_{\mathsf{t},2}^{-1-\frac{1}{\alpha}}, \qquad \chi_{n_{\mathsf{s}},p_{\mathsf{s}}}(\mu_{\mathsf{s},1}, \mu_{\mathsf{s},2}) \approx p_{\mathsf{s}}^{-1} \mu_{\mathsf{s},2}^{-1-\frac{1}{\alpha}}.$$

Then, directly from (Defilippis et al., 2024, Appendix D) we get

$$\mu_{\mathsf{t},1} \approx \begin{cases} n_{\mathsf{t}}^{-\alpha} \ \text{if} \ 1 < \left(\frac{1+\gamma_{\lambda_{\mathsf{t}}}}{\alpha} \wedge \gamma_{p_{\mathsf{t}}}\right) \\ n_{\mathsf{t}}^{-(1+\gamma_{\lambda_{\mathsf{t}}})}, \quad \text{else.} \end{cases} \quad , \qquad \mu_{\mathsf{t},2} \approx n_{\mathsf{t}}^{-\alpha\left(\frac{1+\gamma_{\lambda_{\mathsf{t}}}}{\alpha} \wedge 1 \wedge \gamma_{p_{\mathsf{t}}}\right)},$$

and by further applying the same analysis to $\mu_{\mathsf{s},1}, \mu_{\mathsf{s},2}$, the result readily follows. $\qquad\square$

Next, we compute $\Lambda_0, \Upsilon_{n_t,p_t}(\Lambda_0), \chi_{n_t,p_t}(\Lambda_0)$.

**Lemma 7.** *We have the following scaling of the quantities appearing in the expression of the test error:*

$$\Lambda_0 \approx \Sigma^2(\Sigma + \mu_{s,2})^{-2} + n_s^{-1}\mu_{s,2}^{-\frac{1}{\alpha}}\mu_{s,2}^2(\Sigma + \mu_{s,2})^{-2} + p_s^{-1}\mu_{s,2}^{-\frac{(1+\alpha)}{\alpha}}\mu_{s,2}^2\Sigma(\Sigma + \mu_{s,2})^{-2},$$

$$\Upsilon_{n_t,p_t}(\Lambda_0) \approx n_t^{-1}\begin{cases}(\mu_{t,2})^{-\frac{1}{\alpha}} & \mu_{s,2} \leq \mu_{t,2}, \\ (\mu_{s,2})^{-\frac{1}{\alpha}} + n_s^{-1}\mu_{s,2}^{-\frac{1}{\alpha}}\mu_{t,2}^{-\frac{1}{\alpha}} & \mu_{t,2} < \mu_{s,2}.\end{cases}$$

$$\chi_{n_t,p_t}(\Lambda_0) \approx p_t^{-1} \cdot \begin{cases}\mu_{t,2}^{-\left(1+\frac{1}{\alpha}\right)}, & \mu_{s,2} \leq \mu_{t,2}, \\ \mu_{s,2}^{-\left(1+\frac{1}{\alpha}\right)} + n_s^{-1}\mu_{s,2}^{-\frac{1}{\alpha}}\mu_{t,2}^{-\left(1+\frac{1}{\alpha}\right)} + p_s^{-1}\mu_{s,2}^{-\left(1+\frac{1}{\alpha}\right)}\mu_{t,2}^{-\frac{1}{\alpha}}, & \mu_{t,2} < \mu_{s,2}.\end{cases}$$

*Proof.* We first recall that

$$\Lambda_0 = \Sigma^2(\Sigma + \mu_{s,2})^{-2} + \frac{\Upsilon_{n_s,p_s}(\mu_{s,1}, \mu_{s,2})}{1 - \Upsilon_{n_s,p_s}(\mu_{s,1}, \mu_{s,2})}\mu_{s,2}^2(\Sigma + \mu_{s,2})^{-2} + \frac{\chi_{n_s,p_s}(\mu_{s,2})}{1 - \Upsilon_{n_s,p_s}(\mu_{s,1}, \mu_{s,2})}\mu_{s,2}^2\Sigma(\Sigma + \mu_{s,2})^{-2}$$

$$= \Sigma^2(\Sigma + \mu_{s,2})^{-2} + \bar{\Lambda}_0.$$

From Lemma 6, we know that $\Upsilon_{n_s,p_s} \approx n_t^{-\gamma_{n_s}+\left(\frac{\gamma_{n_s}+\gamma_{\lambda_s}}{\alpha} \wedge \gamma_{n_s} \wedge \gamma_{p_s}\right)}, \chi_{n_s,p_s} \approx p_s^{-1}\mu_{s,2}^{-\frac{(1+\alpha)}{\alpha}}$. Thus, we know that $1 - \Upsilon_{n_s,p_s} = \Theta(1)$.

Hence, we have

$$\Lambda_0 \approx \Sigma^2(\Sigma + \mu_{s,2})^{-2} + n_s^{-1}\mu_{s,2}^{-\frac{1}{\alpha}}\mu_{s,2}^2(\Sigma + \mu_{s,2})^{-2} + p_s^{-1}\mu_{s,2}^{-\frac{(1+\alpha)}{\alpha}}\mu_{s,2}^2\Sigma(\Sigma + \mu_{s,2})^{-2}.$$

To compute $\Upsilon_{n_t,p_t}(\Lambda_0)$, we first express it as a sum of positive terms. Using the self-consistency equation $p_t - \frac{p_t\mu_{t,1}}{\mu_{t,2}} = \text{Tr}(\Sigma(\Sigma + \mu_{t,2})^{-1})$, we have:

$$\Upsilon_{n_t,p_t}(\Lambda_0) = \frac{1}{n_t}\left[\text{Tr}(\Lambda_0\Sigma(\Sigma + \mu_{t,2})^{-1}) - \mu_{t,1}\frac{\text{Tr}(\Lambda_0\Sigma(\Sigma + \mu_{t,2})^{-2})}{1 - \frac{1}{p_t}\text{Tr}(\Sigma^2(\Sigma + \mu_{t,2})^{-2})}\right]$$

$$= \frac{1}{n_t}\left[\text{Tr}(\Lambda_0\Sigma(\Sigma + \mu_{t,2})^{-1}) - \frac{1 - \frac{1}{p_t}\text{Tr}(\Sigma(\Sigma + \mu_{t,2})^{-1})}{1 - \frac{1}{p_t}\text{Tr}(\Sigma^2(\Sigma + \mu_{t,2})^{-2})}\mu_{t,2}\text{Tr}(\Lambda_0\Sigma(\Sigma + \mu_{t,2})^{-2})\right]$$

$$= \frac{1}{n_t}\left[\text{Tr}(\Lambda_0\Sigma(\Sigma + \mu_{t,2})^{-1}) - \frac{1 - \frac{1}{p_t}\text{Tr}(\Sigma(\Sigma + \mu_{t,2})^{-1})}{1 - \frac{1}{p_t}\text{Tr}(\Sigma^2(\Sigma + \mu_{t,2})^{-2})}(\text{Tr}(\Lambda_0\Sigma(\Sigma + \mu_{t,2})^{-1}) - \text{Tr}(\Lambda_0\Sigma^2(\Sigma + \mu_{t,2})^{-2}))\right]$$

$$= \frac{1}{n_t}\left[\frac{1}{p_t}\frac{\mu_{t,2}\text{Tr}(\Sigma(\Sigma + \mu_{t,2})^{-2})}{1 - \frac{1}{p_t}\text{Tr}(\Sigma^2(\Sigma + \mu_{t,2})^{-2})}\text{Tr}(\Lambda_0\Sigma(\Sigma + \mu_{t,2})^{-1}) + \frac{\mu_{t,1}}{\mu_{t,2}}\frac{\text{Tr}(\Lambda_0\Sigma^2(\Sigma + \mu_{t,2})^{-2})}{1 - \frac{1}{p_t}\text{Tr}(\Sigma^2(\Sigma + \mu_{t,2})^{-2})}\right]$$

From (Defilippis et al., 2024, Appendix D) we know $1 - \frac{1}{p_t}\text{Tr}(\Sigma^2(\Sigma + \mu_{t,2})^{-2}) = \Theta(1)$. Using the approximation of $\Lambda_0$, we have

$$\text{Tr}(\Lambda_0\Sigma(\Sigma + \mu_{t,2})^{-1}) \approx \text{Tr}(\Sigma^2(\Sigma + \mu_{s,2})^{-2}\Sigma(\Sigma + \mu_{t,2})^{-1})$$

$$+ n_s^{-1}\mu_{s,2}^{2-\frac{1}{\alpha}}\text{Tr}((\Sigma + \mu_{s,2})^{-2}\Sigma(\Sigma + \mu_{t,2})^{-1})$$

$$+ p_s^{-1}\mu_{s,2}^{1-\frac{1}{\alpha}}\text{Tr}(\Sigma(\Sigma + \mu_{s,2})^{-2}\Sigma(\Sigma + \mu_{t,2})^{-1})$$

$$\approx \begin{cases}(\mu_{t,2})^{-\frac{1}{\alpha}} + n_s^{-1}\mu_{t,2}^{-1}\mu_{s,2}^{1-\frac{2}{\alpha}} + p_s^{-1}\mu_{t,2}^{-1}\mu_{s,2}^{1-\frac{2}{\alpha}}, & \mu_{s,2} \leq \mu_{t,2}, \\ (\mu_{s,2})^{-\frac{1}{\alpha}} + n_s^{-1}\mu_{s,2}^{-\frac{1}{\alpha}}\mu_{t,2}^{-\frac{1}{\alpha}} + p_s^{-1}\mu_{s,2}^{-\frac{2}{\alpha}}, & \mu_{t,2} < \mu_{s,2}.\end{cases}$$

In case where $\mu_{s,2} \leq \mu_{t,2}$, by Lemma 6 we have $n_s^{-1}\mu_{s,2}^{-\frac{1}{\alpha}} = O(1), p_s^{-1}\mu_{s,2}^{-\frac{1}{\alpha}} = O(1)$. Further, $\mu_{t,2}^{-1}\mu_{s,2}^{1-\frac{1}{\alpha}} \leq \mu_{t,2}^{-\frac{1}{\alpha}}$ as $1 - \frac{1}{\alpha} > 0$. Thus, $\text{Tr}(\Lambda_0\Sigma(\Sigma + \mu_{t,2})^{-1}) = \Theta(\mu_{t,2}^{-\frac{1}{\alpha}})$.

In the case where $\mu_{t,2} < \mu_{s,2}$, as $p_s^{-1}\mu_{s,2}^{-\frac{1}{\alpha}} = O(1)$, we have $\mathrm{Tr}(\mathbf{\Lambda}_0\mathbf{\Sigma}(\mathbf{\Sigma} + \mu_{t,2})^{-1}) = \Theta((\mu_{s,2})^{-\frac{1}{\alpha}} + n_s^{-1}\mu_{s,2}^{-\frac{1}{\alpha}}\mu_{t,2}^{-\frac{1}{\alpha}})$.

Thus,

$$\mathrm{Tr}(\mathbf{\Lambda}_0\mathbf{\Sigma}(\mathbf{\Sigma} + \mu_{t,2})^{-1}) \approx \begin{cases} (\mu_{t,2})^{-\frac{1}{\alpha}} & \mu_{s,2} \leq \mu_{t,2}, \\ (\mu_{s,2})^{-\frac{1}{\alpha}} + n_s^{-1}\mu_{s,2}^{-\frac{1}{\alpha}}\mu_{t,2}^{-\frac{1}{\alpha}} & \mu_{t,2} < \mu_{s,2}. \end{cases}$$

Similarly,

$$\begin{aligned} \mathrm{Tr}(\mathbf{\Lambda}_0\mathbf{\Sigma}^2(\mathbf{\Sigma} + \mu_{t,2})^{-2}) \approx & \mathrm{Tr}(\mathbf{\Sigma}^2(\mathbf{\Sigma} + \mu_{s,2})^{-2}\mathbf{\Sigma}^2(\mathbf{\Sigma} + \mu_{t,2})^{-2}) \\ & + n_s^{-1}\mu_{s,2}^{2-\frac{1}{\alpha}}\mathrm{Tr}((\mathbf{\Sigma} + \mu_{s,2})^{-2}\mathbf{\Sigma}^2(\mathbf{\Sigma} + \mu_{t,2})^{-2}) \\ & + p_s^{-1}\mu_{s,2}^{1-\frac{1}{\alpha}}\mathrm{Tr}(\mathbf{\Sigma}(\mathbf{\Sigma} + \mu_{s,2})^{-2}\mathbf{\Sigma}^2(\mathbf{\Sigma} + \mu_{t,2})^{-2}) \\ \approx & \begin{cases} (\mu_{t,2})^{-\frac{1}{\alpha}} + n_s^{-1}\mu_{t,2}^{-2}\mu_{s,2}^{2-\frac{2}{\alpha}} + p_s^{-1}\mu_{s,2}^{1-\frac{1}{\alpha}}\mu_{t,2}^{-(1+\frac{1}{\alpha})}, & \mu_{s,2} \leq \mu_{t,2}, \\ (\mu_{s,2})^{-\frac{1}{\alpha}} + n_s^{-1}\mu_{s,2}^{-\frac{1}{\alpha}}\mu_{t,2}^{-\frac{1}{\alpha}} + p_s^{-1}\mu_{s,2}^{-\frac{2}{\alpha}}, & \mu_{t,2} < \mu_{s,2}. \end{cases} \end{aligned}$$

Using the same argument as for $\mathrm{Tr}(\mathbf{\Lambda}_0\mathbf{\Sigma}(\mathbf{\Sigma} + \mu_{t,2})^{-1})$, we have:

$$\mathrm{Tr}(\mathbf{\Lambda}_0\mathbf{\Sigma}^2(\mathbf{\Sigma} + \mu_{t,2})^{-2}) \approx \begin{cases} (\mu_{t,2})^{-\frac{1}{\alpha}} & \mu_{s,2} \leq \mu_{t,2}, \\ (\mu_{s,2})^{-\frac{1}{\alpha}} + n_s^{-1}\mu_{s,2}^{-\frac{1}{\alpha}}\mu_{t,2}^{-\frac{1}{\alpha}} & \mu_{t,2} < \mu_{s,2}. \end{cases}$$

From (Defilippis et al., 2024, Appendix D) and Lemma 6 we further have:

$$\mathrm{Tr}(\mathbf{\Sigma}(\mathbf{\Sigma} + \mu_{t,2})^{-2}) \approx \mu_{t,2}^{-\frac{(1+\alpha)}{\alpha}}$$

$$\frac{\mu_{t,1}}{\mu_{t,2}} = \begin{cases} \Theta(1), & \text{if } (1 \wedge \frac{1+\gamma_{\lambda_t}}{\alpha}) \leq \gamma_{p_t} \\ o(1), & \text{otherwise} \end{cases}$$

Thus, combining all the terms, we have:

$$\Upsilon_{n_t,p_t}(\mathbf{\Lambda}_0) = n_t^{-1}(p_t^{-1}\mu_{t,2}^{-\frac{1}{\alpha}} + \frac{\mu_{t,1}}{\mu_{t,2}}) \cdot \begin{cases} (\mu_{t,2})^{-\frac{1}{\alpha}} & \mu_{s,2} \leq \mu_{t,2}, \\ (\mu_{s,2})^{-\frac{1}{\alpha}} + n_s^{-1}\mu_{s,2}^{-\frac{1}{\alpha}}\mu_{t,2}^{-\frac{1}{\alpha}} & \mu_{t,2} < \mu_{s,2}. \end{cases}$$

We note that $p_t^{-1}\mu_{t,2}^{-\frac{1}{\alpha}} + \frac{\mu_{t,1}}{\mu_{t,2}} = \Theta(1)$ since when $(1 \wedge \frac{1+\gamma_{\lambda_t}}{\alpha}) \leq \gamma_{p_t}$, $\frac{\mu_{t,1}}{\mu_{t,2}} = \Theta(1), p_t^{-1}\mu_{t,2}^{-\frac{1}{\alpha}} = O(1)$, otherwise when $\gamma_{p_t} < (1 \wedge \frac{1+\gamma_{\lambda_t}}{\alpha}), p_t^{-1}\mu_{t,2}^{-\frac{1}{\alpha}} = \Theta(1)$. Thus, we finally have:

$$\Upsilon_{n_t,p_t}(\mathbf{\Lambda}_0) = n_t^{-1} \begin{cases} (\mu_{t,2})^{-\frac{1}{\alpha}} & \mu_{s,2} \leq \mu_{t,2}, \\ (\mu_{s,2})^{-\frac{1}{\alpha}} + n_s^{-1}\mu_{s,2}^{-\frac{1}{\alpha}}\mu_{t,2}^{-\frac{1}{\alpha}} & \mu_{t,2} < \mu_{s,2}. \end{cases}$$

Next we compute $\chi_{n_t,p_t}(\mathbf{\Lambda}_0)$ as:

$$\begin{aligned} \chi_{n_t,p_t}(\mathbf{\Lambda}_0) &= \frac{\mathrm{Tr}(\mathbf{\Lambda}_0\mathbf{\Sigma}(\mathbf{\Sigma} + \mu_{t,2})^{-2})}{p_t - \mathrm{Tr}(\mathbf{\Sigma}^2(\mathbf{\Sigma} + \mu_{t,2})^{-2})} \\ &\approx \frac{1}{p_t}\mathrm{Tr}(\mathbf{\Lambda}_0\mathbf{\Sigma}(\mathbf{\Sigma} + \mu_{t,2})^{-2}) \\ &\approx \frac{1}{p_t} \begin{cases} \mu_{t,2}^{-(1+\frac{1}{\alpha})} + n_s^{-1}\mu_{t,2}^{-2}\mu_{s,2}^{1-\frac{2}{\alpha}} + p_s^{-1}\mu_{t,2}^{-2}\mu_{s,2}^{1-\frac{2}{\alpha}}, & \mu_{s,2} \leq \mu_{t,2}, \\ \mu_{s,2}^{-(1+\frac{1}{\alpha})} + n_s^{-1}\mu_{s,2}^{-\frac{1}{\alpha}}\mu_{t,2}^{-(1+\frac{1}{\alpha})} + p_s^{-1}\mu_{s,2}^{-(1+\frac{1}{\alpha})}\mu_{t,2}^{-\frac{1}{\alpha}}, & \mu_{t,2} < \mu_{s,2}. \end{cases} \end{aligned}$$

In the case $\mu_{s,2} \leq \mu_{t,2}$, by Lemma 6 we have $n_s^{-1}\mu_{s,2}^{-\frac{1}{\alpha}} = O(1)$, $p_s^{-1}\mu_{s,2}^{-\frac{1}{\alpha}} = O(1)$. Further, $\mu_{t,2}^{-2}\mu_{s,2}^{1-\frac{1}{\alpha}} \leq \mu_{t,2}^{-(1+\frac{1}{\alpha})}$ as $1 - \frac{1}{\alpha} > 0$. Thus, $\chi_{n_t,p_t}(\mathbf{\Lambda}_0) = \Theta(\mu_{t,2}^{-(1+\frac{1}{\alpha})})$.

Thus we have:

$$\chi_{n_t,p_t}(\mathbf{\Lambda}_0) = p_t^{-1} \cdot \begin{cases} \mu_{t,2}^{-(1+\frac{1}{\alpha})}, & \mu_{s,2} \leq \mu_{t,2}, \\ \mu_{s,2}^{-(1+\frac{1}{\alpha})} + n_s^{-1}\mu_{s,2}^{-\frac{1}{\alpha}}\mu_{t,2}^{-(1+\frac{1}{\alpha})} + p_s^{-1}\mu_{s,2}^{-(1+\frac{1}{\alpha})}\mu_{t,2}^{-\frac{1}{\alpha}}, & \mu_{t,2} < \mu_{s,2}. \end{cases}$$

$\square$

# G. Decay rates under source and capacity conditions

## Notation

Since in this section we only care about the decay rate of the deterministic equivalent rather than a precise computation, we introduce the following notation. For two functions $f_{n_t}, g_{n_t}$, we define

- $f_{n_t} \approx g_{n_t}$ if $f_{n_t} = \Theta(g_{n_t})$.

- $f_{n_t} \lesssim g_{n_t}$ if $f_{n_t} \leq \mathcal{O}(g_{n_t})$.

- $f_{n_t} \ll g_{n_t}$ if $f_{n_t} \leq o(g_{n_t})$.

Recall that the bias term, variance term and test error of the teacher model are

$$B_{n_t,p_t}(\boldsymbol{\beta}_*, \lambda_t) := \frac{\mu_{t,2}^2}{1 - \Upsilon_{n_t,p_t}(\mu_{t,1}, \mu_{t,2})} \left[ \langle \boldsymbol{\beta}_*, (\mathbf{\Sigma} + \mu_{t,2})^{-2}\boldsymbol{\beta}_* \rangle + \chi_{n_t,p_t}(\mu_{t,2})\langle \boldsymbol{\beta}_*, \mathbf{\Sigma}(\mathbf{\Sigma} + \mu_{t,2})^{-2}\boldsymbol{\beta}_* \rangle \right],$$

$$V_{n_t,p_t}(\lambda_t) := \tau_t^2 \frac{\Upsilon_{n_t,p_t}(\mu_{t,1}, \mu_{t,2})}{1 - \Upsilon_{n_t,p_t}(\mu_{t,1}, \mu_{t,2})},$$

$$R_{n_t,p_t}(\boldsymbol{\beta}_*, \lambda_t) := B_{n_t,p_t}(\boldsymbol{\beta}_*, \lambda_t) + V_{n_t,p_t}(\lambda_t),$$

and the bias term of the student model is

$$B_{n_s,p_s} = \underbrace{\langle \boldsymbol{\beta}_*, \mathbf{\Lambda}\boldsymbol{\beta}_* \rangle + \tau_t^2 \cdot \frac{\Upsilon_{n_t,p_t}(\mathbf{\Lambda}_0; \mu_{t,1}, \mu_{t,2})}{1 - \Upsilon_{n_t,p_t}(\mu_{t,1}, \mu_{t,2})}}_{\text{Student bias term: teacher bias + teacher variance}},$$

where

$$\mathbf{\Lambda} = \left[ \boldsymbol{I} - \mathbf{\Sigma}^2(\mathbf{\Sigma} + \mu_{s,2})^{-1}(\mathbf{\Sigma} + \mu_{t,2})^{-1} \right]^2 + \bar{\mathbf{\Lambda}}_0\mathbf{\Sigma}^2(\mathbf{\Sigma} + \mu_{t,2})^{-2}$$

$$+ \frac{\Upsilon_{n_t,p_t}(\mathbf{\Lambda}_0; \mu_{t,1}, \mu_{t,2})}{1 - \Upsilon_{n_t,p_t}(\mu_{t,1}, \mu_{t,2})}\mu_{t,2}^2(\mathbf{\Sigma} + \mu_{t,2})^{-2}$$

$$+ \left[ \chi_{n_t,p_t}(\mathbf{\Lambda}_0; \mu_{t,1}, \mu_{t,2}) + \frac{\Upsilon_{n_t,p_t}(\mathbf{\Lambda}_0; \mu_{t,1}, \mu_{t,2})}{1 - \Upsilon_{n_t,p_t}(\mu_{t,1}, \mu_{t,2})}\chi_{n_t,p_t}(\mu_{t,1}, \mu_{t,2}) \right]\mu_{t,2}^2\mathbf{\Sigma}(\mathbf{\Sigma} + \mu_{t,2})^{-2},$$

and

$$\bar{\mathbf{\Lambda}}_0 = \frac{\Upsilon_{n_s,p_s}(\mu_{s,1}, \mu_{s,2})}{1 - \Upsilon_{n_s,p_s}(\mu_{s,1}, \mu_{s,2})}\mu_{s,2}^2(\mathbf{\Sigma} + \mu_{s,2})^{-2} + \frac{\chi_{n_s,p_s}(\mu_{s,2})}{1 - \Upsilon_{n_s,p_s}(\mu_{s,1}, \mu_{s,2})}\mu_{s,2}^2\mathbf{\Sigma}(\mathbf{\Sigma} + \mu_{s,2})^{-2}.$$

We will assume that

$$\mathbf{\Sigma} = \text{diag}(\xi_k^2)_{k \geq 1} = \text{diag}(k^{-\alpha})_{k \geq 1}, \qquad \boldsymbol{\beta}_* = (\beta_{*,k})_{k \geq 1} = (k^{-\frac{(1+2\alpha r)}{2}})_{k \geq 1}.$$

We fix $n_t$, i.e. the number of training samples of the teacher's model, and define

$$p_t = n_t^{\gamma_{p_t}}, \qquad \lambda_t = n_t^{-\gamma_{\lambda_t}}, \qquad n_s = n_t^{\gamma_{n_s}}, \qquad p_s = n_t^{\gamma_{p_s}}, \qquad \lambda_s = n_t^{-\gamma_{\lambda_s}}.$$

## G.1. Scaling law of the teacher model

**Derivation of Optimal Scaling Law.** In order to derive the optimal scaling law, we fix a dataset of size $n_t$ and want to understand what is the optimal scaling $\gamma_*$ such that $R_t = n_t^{\gamma_*}$.

Define

$$\gamma_{B_1} = -2\alpha(r \wedge 1)z_t, \qquad \gamma_{B_2} = -\gamma_{p_t} + (1 - 2\alpha(r \wedge \frac{1}{2}))z_t, \qquad \gamma_V = -1 + z_t,$$

and we have

$$\gamma_* = \min_{z_t} \max\{\gamma_{B_1}(z_t), \gamma_{B_2}(z_t), \gamma_V(z_t)\}.$$

It is easy to see the following lower bounds on $\gamma_*$:

$$\gamma_* \geq \min_{z_t} \max\{\gamma_{B_1}(z_t), \gamma_V(z_t)\} = \frac{-2\alpha(r \wedge 1)}{1 + 2\alpha(r \wedge 1)},$$

achieved by $z_t^* = \frac{1}{1+2\alpha(r\wedge 1)}$.

Next, we just require $\gamma_{B_2} < \gamma_*$, which implies that:

$$\gamma_{p_t} > \frac{1 + 2\alpha(r \wedge 1) - 2\alpha(r \wedge \frac{1}{2})}{1 + 2\alpha(r \wedge 1)}$$

This lower bound is always achievable, by taking $\gamma_{\lambda_t} = \frac{\alpha}{1+2\alpha(r\wedge 1)} - 1, \gamma_{p_t} > \frac{1+2\alpha(r\wedge 1)-2\alpha(r\wedge\frac{1}{2})}{1+2\alpha(r\wedge 1)}$.

## G.2. Scaling law of the student model: Proof of Theorem 4

### G.2.1. SCALING LAW FOR $\mu_{t,2} \geq \mu_{s,2}$

First of all, by Lemma 6, $\mu_{t,2} \gg \mu_{s,2}$ implies that

$$\frac{\gamma_{n_s} + \gamma_{\lambda_s}}{\alpha} \wedge \gamma_{n_s} \wedge \gamma_{p_s} > \frac{1 + \gamma_{\lambda_t}}{\alpha} \wedge 1 \wedge \gamma_{p_t}.$$

Recall that

$$B_{bias,s} = \langle \boldsymbol{\beta}_*, \boldsymbol{\Lambda}\boldsymbol{\beta}_* \rangle, \qquad B_{var,s} = \tau_t^2 \cdot \frac{\Upsilon_{n_t,p_t}(\boldsymbol{\Lambda}_0; \mu_{t,1}, \mu_{t,2})}{1 - \Upsilon_{n_t,p_t}(\mu_{t,1}, \mu_{t,2})},$$

$$\boldsymbol{\Lambda} = \left[\boldsymbol{I} - \boldsymbol{\Sigma}^2(\boldsymbol{\Sigma} + \mu_{s,2})^{-1}(\boldsymbol{\Sigma} + \mu_{t,2})^{-1}\right]^2 + \bar{\boldsymbol{\Lambda}}_0 \boldsymbol{\Sigma}^2(\boldsymbol{\Sigma} + \mu_{t,2})^{-2}$$

$$+ \frac{\Upsilon_{n_t,p_t}(\boldsymbol{\Lambda}_0; \mu_{t,1}, \mu_{t,2})}{1 - \Upsilon_{n_t,p_t}(\mu_{t,1}, \mu_{t,2})}\mu_{t,2}^2(\boldsymbol{\Sigma} + \mu_{t,2})^{-2}$$

$$+ \left[\chi_{n_t,p_t}(\boldsymbol{\Lambda}_0; \mu_{t,1}, \mu_{t,2}) + \frac{\Upsilon_{n_t,p_t}(\boldsymbol{\Lambda}_0; \mu_{t,1}, \mu_{t,2})}{1 - \Upsilon_{n_t,p_t}(\mu_{t,1}, \mu_{t,2})}\chi_{n_t,p_t}(\mu_{t,1}, \mu_{t,2})\right]\mu_{t,2}^2\boldsymbol{\Sigma}(\boldsymbol{\Sigma} + \mu_{t,2})^{-2},$$

$$\bar{\boldsymbol{\Lambda}}_0 = \frac{\Upsilon_{n_s,p_s}(\mu_{s,1}, \mu_{s,2})}{1 - \Upsilon_{n_s,p_s}(\mu_{s,1}, \mu_{s,2})}\mu_{s,2}^2(\boldsymbol{\Sigma} + \mu_{s,2})^{-2} + \frac{\chi_{n_s,p_s}(\mu_{s,2})}{1 - \Upsilon_{n_s,p_s}(\mu_{s,1}, \mu_{s,2})}\mu_{s,2}^2\boldsymbol{\Sigma}(\boldsymbol{\Sigma} + \mu_{s,2})^{-2}.$$

Using Lemma 7, we have

$$\chi_{n_t,p_t}(\boldsymbol{\Lambda}_0) \approx p_t^{-1}\mu_{t,2}^{-\frac{(1+\alpha)}{\alpha}}, \quad \Upsilon_{n_t,p_t}(\boldsymbol{\Lambda}_0) \approx n_t^{-1}\mu_{t,2}^{-\frac{1}{\alpha}}.$$

Now, it remains to compute $B_{bias,s}$. Recall that

$$\langle \boldsymbol{\beta}_*, \left[\boldsymbol{I} - \boldsymbol{\Sigma}^2(\boldsymbol{\Sigma} + \mu_{s,2})^{-1}(\boldsymbol{\Sigma} + \mu_{t,2})^{-1}\right]^2 \boldsymbol{\beta}_* \rangle = \langle \boldsymbol{\beta}_*, \left(\mu_{s,2}^2(\boldsymbol{\Sigma} + \mu_{s,2})^{-2} + 2\mu_{t,2}\mu_{s,2}\boldsymbol{\Sigma}(\boldsymbol{\Sigma} + \mu_{s,2})^{-2}(\boldsymbol{\Sigma} + \mu_{t,2})^{-1}\right.$$

$$\left. + \mu_{t,2}^2\boldsymbol{\Sigma}^2(\boldsymbol{\Sigma} + \mu_{s,2})^{-2}(\boldsymbol{\Sigma} + \mu_{t,2})^{-2}\right)\boldsymbol{\beta}_* \rangle$$

$$= \mu_{s,2}^2\mu_{s,2}^{(2r-2)\wedge 0} + \mu_{t,2}\mu_{s,2}(\mu_{t,2} \vee \mu_{s,2})^{(2r-2)\wedge 0}$$

$$+ \mu_{t,2}^2(\mu_{t,2} \vee \mu_{s,2})^{(2r-2)\wedge 0}$$

$$\approx \mu_{t,2}^{2(r\wedge 1)}.$$

Hence, by plugging in the above, $\mu_{\mathsf{s},2} = n_{\mathsf{t}}^{-\alpha\left(\frac{\gamma_{n_{\mathsf{s}}} + \gamma_{\lambda_{\mathsf{s}}}}{\alpha} \wedge \gamma_{n_{\mathsf{s}}} \wedge \gamma_{p_{\mathsf{s}}}\right)}$ and $\mu_{\mathsf{t},2} = n_{\mathsf{t}}^{-\alpha\left(\frac{1+\gamma_{\lambda_{\mathsf{t}}}}{\alpha} \wedge 1 \wedge \gamma_{p_{\mathsf{t}}}\right)}$, we have

$$\frac{\Upsilon_{n_{\mathsf{s}},p_{\mathsf{s}}}(\mu_{\mathsf{s},1}, \mu_{\mathsf{s},2})}{1 - \Upsilon_{n_{\mathsf{s}},p_{\mathsf{s}}}(\mu_{\mathsf{s},1}, \mu_{\mathsf{s},2})} \mu_{\mathsf{s},2}^2 \langle \boldsymbol{\beta}_*, \boldsymbol{\Sigma}^2(\boldsymbol{\Sigma} + \mu_{\mathsf{t},2})^{-2}(\boldsymbol{\Sigma} + \mu_{\mathsf{s},2})^{-2}\boldsymbol{\beta}_* \rangle \approx n_{\mathsf{s}}^{-1}\mu_{\mathsf{s},2}^{-\frac{1}{\alpha}}\mu_{\mathsf{s},2}^2\mu_{\mathsf{t},2}^{(2r-2)\wedge 0}$$

$$\ll \mu_{\mathsf{t},2}^{2(r\wedge 1)} \qquad (\text{use } n_{\mathsf{s}}^{-1}\mu_{\mathsf{s},2}^{-\frac{1}{\alpha}} = O(1), \mu_{\mathsf{s},2}^2\mu_{\mathsf{t},2}^{-2} \ll 1),$$

$$\frac{\chi_{n_{\mathsf{s}},p_{\mathsf{s}}}(\mu_{\mathsf{s},1}, \mu_{\mathsf{s},2})}{1 - \Upsilon_{n_{\mathsf{s}},p_{\mathsf{s}}}(\mu_{\mathsf{s},1}, \mu_{\mathsf{s},2})} \mu_{\mathsf{s},2}^2 \langle \boldsymbol{\beta}_*, \boldsymbol{\Sigma}^3(\boldsymbol{\Sigma} + \mu_{\mathsf{t},2})^{-2}(\boldsymbol{\Sigma} + \mu_{\mathsf{s},2})^{-2}\boldsymbol{\beta}_* \rangle \approx p_{\mathsf{s}}^{-1}\mu_{\mathsf{s},2}^{-\frac{(1+\alpha)}{\alpha}}\mu_{\mathsf{s},2}^2\mu_{\mathsf{t},2}^{(2r-1)\wedge 0},$$

$$\frac{\Upsilon_{n_{\mathsf{t}},p_{\mathsf{t}}}(\boldsymbol{\Lambda}_0; \mu_{\mathsf{t},1}, \mu_{\mathsf{t},2})}{1 - \Upsilon_{n_{\mathsf{t}},p_{\mathsf{t}}}(\mu_{\mathsf{t},1}, \mu_{\mathsf{t},2})} \mu_{\mathsf{t},2}^2 \langle \boldsymbol{\beta}_*, (\boldsymbol{\Sigma} + \mu_{\mathsf{t},2})^{-2}\boldsymbol{\beta}_* \rangle \approx n_{\mathsf{t}}^{-1}\mu_{\mathsf{t},2}^{-\frac{1}{\alpha}}\mu_{\mathsf{t},2}^2\mu_{\mathsf{t},2}^{(2r-2)\wedge 0}$$

$$= O(\mu_{\mathsf{t},2}^{2(r\wedge 1)}) \qquad (\text{use } n_{\mathsf{t}}^{-1}\mu_{\mathsf{t},2}^{-\frac{1}{\alpha}} = O(1)),$$

$$\left[\chi_{n_{\mathsf{t}},p_{\mathsf{t}}}(\boldsymbol{\Lambda}_0) + \frac{\Upsilon_{n_{\mathsf{t}},p_{\mathsf{t}}}(\boldsymbol{\Lambda}_0)}{1 - \Upsilon_{n_{\mathsf{t}},p_{\mathsf{t}}}}\chi_{n_{\mathsf{t}},p_{\mathsf{t}}}\right]\mu_{\mathsf{t},2}^2 \langle \boldsymbol{\beta}_*, \boldsymbol{\Sigma}(\boldsymbol{\Sigma} + \mu_{\mathsf{t},2})^{-2}\boldsymbol{\beta}_* \rangle \approx (p_{\mathsf{t}}^{-1}\mu_{\mathsf{t},2}^{-\frac{(\alpha+1)}{\alpha}} + n_{\mathsf{t}}^{-1}\mu_{\mathsf{t},2}^{-\frac{1}{\alpha}}p_{\mathsf{t}}^{-1}\mu_{\mathsf{t},2}^{-\frac{(\alpha+1)}{\alpha}})\mu_{\mathsf{t},2}^2\mu_{\mathsf{t},2}^{(2r-1)\wedge 0}$$

$$= (p_{\mathsf{t}}^{-1}\mu_{\mathsf{t},2}^{-\frac{(\alpha+1)}{\alpha}} + n_{\mathsf{t}}^{-1}\mu_{\mathsf{t},2}^{-\frac{1}{\alpha}}p_{\mathsf{t}}^{-1}\mu_{\mathsf{t},2}^{-\frac{(\alpha+1)}{\alpha}})\mu_{\mathsf{t},2}\mu_{\mathsf{t},2}^{2(r\wedge\frac{1}{2})}$$

$$\approx p_{\mathsf{t}}^{-1}\mu_{\mathsf{t},2}^{-\frac{(\alpha+1)}{\alpha}}\mu_{\mathsf{t},2}\mu_{\mathsf{t},2}^{2(r\wedge\frac{1}{2})} \qquad (\text{use } n_{\mathsf{t}}^{-1}\mu_{\mathsf{t},2}^{-\frac{1}{\alpha}} = O(1))$$

$$\approx p_{\mathsf{t}}^{-1}\mu_{\mathsf{t},2}^{-\frac{1}{\alpha}}\mu_{\mathsf{t},2}^{2(r\wedge\frac{1}{2})}.$$

Thus, combining all terms, we have

$$\mathsf{B}_{var,\mathsf{s}} \approx n_{\mathsf{t}}^{-1+\left(\frac{1+\gamma_{\lambda_{\mathsf{t}}}}{\alpha} \wedge 1 \wedge \gamma_{p_{\mathsf{t}}}\right)},$$

$$\mathsf{B}_{bias,\mathsf{s}} \approx \mu_{\mathsf{t},2}^{2(r\wedge 1)} + p_{\mathsf{s}}^{-1}\frac{\mu_{\mathsf{s},2}}{\mu_{\mathsf{t},2}}\mu_{\mathsf{s},2}^{-\frac{1}{\alpha}}\mu_{\mathsf{t},2}^{2(r\wedge\frac{1}{2})} + p_{\mathsf{t}}^{-1}\mu_{\mathsf{t},2}^{-\frac{1}{\alpha}}\mu_{\mathsf{t},2}^{2(r\wedge\frac{1}{2})}$$

$$\approx n_{\mathsf{t}}^{-2\alpha(r\wedge 1)\left(\frac{1+\gamma_{\lambda_{\mathsf{t}}}}{\alpha} \wedge 1 \wedge \gamma_{p_{\mathsf{t}}}\right)} + (n_{\mathsf{t}}^{-\gamma_{p_{\mathsf{s}}}+\alpha(z_{\mathsf{t}}-z_{\mathsf{s}})+\left(\frac{\gamma_{n_{\mathsf{s}}}+\gamma_{\lambda_{\mathsf{s}}}}{\alpha} \wedge \gamma_{n_{\mathsf{s}}} \wedge \gamma_{p_{\mathsf{s}}}\right)} + n_{\mathsf{t}}^{-\gamma_{p_{\mathsf{t}}}+\left(\frac{1+\gamma_{\lambda_{\mathsf{t}}}}{\alpha} \wedge 1 \wedge \gamma_{p_{\mathsf{t}}}\right)}) \cdot n_{\mathsf{t}}^{-2\alpha(r\wedge\frac{1}{2})\left(\frac{1+\gamma_{\lambda_{\mathsf{t}}}}{\alpha} \wedge 1 \wedge \gamma_{p_{\mathsf{t}}}\right)}$$

$$= n_{\mathsf{t}}^{-2\alpha(r\wedge 1)z_{\mathsf{t}}} + (n_{\mathsf{t}}^{-\gamma_{p_{\mathsf{s}}}+z_{\mathsf{s}}+\alpha(z_{\mathsf{t}}-z_{\mathsf{s}})} + n_{\mathsf{t}}^{-\gamma_{p_{\mathsf{t}}}+z_{\mathsf{t}}}) \cdot n_{\mathsf{t}}^{-2\alpha(r\wedge\frac{1}{2})z_{\mathsf{t}}}.$$

### G.2.2. SCALING LAW FOR $\mu_{\mathsf{t},2} < \mu_{\mathsf{s},2}$

First of all, by Lemma 6, $\mu_{\mathsf{t},2} < \mu_{\mathsf{s},2}$ implies that

$$\frac{\gamma_{n_{\mathsf{s}}} + \gamma_{\lambda_{\mathsf{s}}}}{\alpha} \wedge \gamma_{n_{\mathsf{s}}} \wedge \gamma_{p_{\mathsf{s}}} \le \frac{1 + \gamma_{\lambda_{\mathsf{t}}}}{\alpha} \wedge 1 \wedge \gamma_{p_{\mathsf{t}}}.$$

Using Lemma 7, we have

$$\chi_{n_{\mathsf{t}},p_{\mathsf{t}}}(\boldsymbol{\Lambda}_0) \approx p_{\mathsf{t}}^{-1}(\mu_{\mathsf{s},2}^{-\left(1+\frac{1}{\alpha}\right)} + n_{\mathsf{s}}^{-1}\mu_{\mathsf{s},2}^{-\frac{1}{\alpha}}\mu_{\mathsf{t},2}^{-\left(1+\frac{1}{\alpha}\right)} + p_{\mathsf{s}}^{-1}\mu_{\mathsf{s},2}^{-\left(1+\frac{1}{\alpha}\right)}\mu_{\mathsf{t},2}^{-\frac{1}{\alpha}}),$$

$$\Upsilon_{n_{\mathsf{t}},p_{\mathsf{t}}}(\boldsymbol{\Lambda}_0) \approx n_{\mathsf{t}}^{-1}((\mu_{\mathsf{s},2})^{-\frac{1}{\alpha}} + n_{\mathsf{s}}^{-1}\mu_{\mathsf{s},2}^{-\frac{1}{\alpha}}\mu_{\mathsf{t},2}^{-\frac{1}{\alpha}}).$$

Now, it remains to compute $\mathsf{B}_{bias,\mathsf{s}}$. Recall that

$$\langle \boldsymbol{\beta}_*, \left[\boldsymbol{I} - \boldsymbol{\Sigma}^2(\boldsymbol{\Sigma} + \mu_{\mathsf{s},2})^{-1}(\boldsymbol{\Sigma} + \mu_{\mathsf{t},2})^{-1}\right]^2 \boldsymbol{\beta}_* \rangle$$

$$= \mu_{\mathsf{s},2}^2 \langle \boldsymbol{\beta}_*, (\boldsymbol{\Sigma} + \mu_{\mathsf{s},2})^{-2}\boldsymbol{\beta}_* \rangle + 2\mu_{\mathsf{t},2}\mu_{\mathsf{s},2}\langle \boldsymbol{\beta}_*, \boldsymbol{\Sigma}(\boldsymbol{\Sigma} + \mu_{\mathsf{s},2})^{-2}(\boldsymbol{\Sigma} + \mu_{\mathsf{t},2})^{-1}\boldsymbol{\beta}_* \rangle$$

$$+ \mu_{\mathsf{t},2}^2 \langle \boldsymbol{\beta}_*, \boldsymbol{\Sigma}^2(\boldsymbol{\Sigma} + \mu_{\mathsf{s},2})^{-2}(\boldsymbol{\Sigma} + \mu_{\mathsf{t},2})^{-2}\boldsymbol{\beta}_* \rangle$$

$$\approx \mu_{\mathsf{s},2}^{2(r\wedge 1)} + \mu_{\mathsf{t},2}\mu_{\mathsf{s},2}\mu_{\mathsf{s},2}^{(2r-2)\wedge 0} + \mu_{\mathsf{t},2}^2\mu_{\mathsf{s},2}^{(2r-2)\wedge 0}$$

$$\approx \mu_{\mathsf{s},2}^{2(r\wedge 1)},$$

where we used $\mu_{\mathsf{t},2} \le \mu_{\mathsf{s},2}$ in the last step.

Next, using Lemma 7, we have

$$\frac{\Upsilon_{n_{\mathsf{s}},p_{\mathsf{s}}}(\mu_{\mathsf{s},1},\mu_{\mathsf{s},2})}{1-\Upsilon_{n_{\mathsf{s}},p_{\mathsf{s}}}(\mu_{\mathsf{s},1},\mu_{\mathsf{s},2})}\mu_{\mathsf{s},2}^2\langle\boldsymbol{\beta}_*,\boldsymbol{\Sigma}^2(\boldsymbol{\Sigma}+\mu_{\mathsf{t},2})^{-2}(\boldsymbol{\Sigma}+\mu_{\mathsf{s},2})^{-2}\boldsymbol{\beta}_*\rangle$$
$$\approx n_{\mathsf{s}}^{-1}\mu_{\mathsf{s},2}^{-\frac{1}{\alpha}}\mu_{\mathsf{s},2}^2\mu_{\mathsf{s},2}^{(2r-2)\wedge 0}$$
$$= n_{\mathsf{s}}^{-1}\mu_{\mathsf{s},2}^{-\frac{1}{\alpha}}\mu_{\mathsf{s},2}^{2(r\wedge 1)} = O\left(\mu_{\mathsf{s},2}^{2(r\wedge 1)}\right),\qquad\left(\text{use }n_{\mathsf{s}}^{-1}\mu_{\mathsf{s},2}^{-\frac{1}{\alpha}}=O(1)\right),$$

$$\frac{\chi_{n_{\mathsf{s}},p_{\mathsf{s}}}(\mu_{\mathsf{s},1},\mu_{\mathsf{s},2})}{1-\Upsilon_{n_{\mathsf{s}},p_{\mathsf{s}}}(\mu_{\mathsf{s},1},\mu_{\mathsf{s},2})}\mu_{\mathsf{s},2}^2\langle\boldsymbol{\beta}_*,\boldsymbol{\Sigma}^3(\boldsymbol{\Sigma}+\mu_{\mathsf{t},2})^{-2}(\boldsymbol{\Sigma}+\mu_{\mathsf{s},2})^{-2}\boldsymbol{\beta}_*\rangle$$
$$\approx p_{\mathsf{s}}^{-1}\mu_{\mathsf{s},2}^{-\frac{1+\alpha}{\alpha}}\mu_{\mathsf{s},2}^2\mu_{\mathsf{s},2}^{(2r-1)\wedge 0}$$
$$= p_{\mathsf{s}}^{-1}\mu_{\mathsf{s},2}^{-\frac{1}{\alpha}}\mu_{\mathsf{s},2}^{2(r\wedge\frac{1}{2})},$$

$$\frac{\Upsilon_{n_{\mathsf{t}},p_{\mathsf{t}}}(\boldsymbol{\Lambda}_0;\mu_{\mathsf{t},1},\mu_{\mathsf{t},2})}{1-\Upsilon_{n_{\mathsf{t}},p_{\mathsf{t}}}(\mu_{\mathsf{t},1},\mu_{\mathsf{t},2})}\mu_{\mathsf{t},2}^2\langle\boldsymbol{\beta}_*,(\boldsymbol{\Sigma}+\mu_{\mathsf{t},2})^{-2}\boldsymbol{\beta}_*\rangle$$
$$\approx n_{\mathsf{t}}^{-1}\left(\mu_{\mathsf{s},2}^{-\frac{1}{\alpha}}+n_{\mathsf{s}}^{-1}\mu_{\mathsf{s},2}^{-\frac{1}{\alpha}}\mu_{\mathsf{t},2}^{-\frac{1}{\alpha}}\right)\mu_{\mathsf{t},2}^2\mu_{\mathsf{t},2}^{(2r-2)\wedge 0}$$
$$= n_{\mathsf{t}}^{-1}\left(\mu_{\mathsf{s},2}^{-\frac{1}{\alpha}}+n_{\mathsf{s}}^{-1}\mu_{\mathsf{s},2}^{-\frac{1}{\alpha}}\mu_{\mathsf{t},2}^{-\frac{1}{\alpha}}\right)\mu_{\mathsf{t},2}^{2(r\wedge 1)}$$
$$= O\left(\mu_{\mathsf{s},2}^{2(r\wedge 1)}\right),\qquad\left(\text{use }n_{\mathsf{t}}^{-1}\mu_{\mathsf{t},2}^{-\frac{1}{\alpha}}=O(1),\ n_{\mathsf{s}}^{-1}\mu_{\mathsf{s},2}^{-\frac{1}{\alpha}}=O(1),\ \mu_{\mathsf{t},2}\le\mu_{\mathsf{s},2}\right),$$

$$\left[\chi_{n_{\mathsf{t}},p_{\mathsf{t}}}(\boldsymbol{\Lambda}_0)+\frac{\Upsilon_{n_{\mathsf{t}},p_{\mathsf{t}}}(\boldsymbol{\Lambda}_0)}{1-\Upsilon_{n_{\mathsf{t}},p_{\mathsf{t}}}}\chi_{n_{\mathsf{t}},p_{\mathsf{t}}}\right]\mu_{\mathsf{t},2}^2\langle\boldsymbol{\beta}_*,\boldsymbol{\Sigma}(\boldsymbol{\Sigma}+\mu_{\mathsf{t},2})^{-2}\boldsymbol{\beta}_*\rangle$$
$$\approx p_{\mathsf{t}}^{-1}\left[\mu_{\mathsf{s},2}^{-\left(1+\frac{1}{\alpha}\right)}\mu_{\mathsf{t},2}+n_{\mathsf{s}}^{-1}\mu_{\mathsf{s},2}^{-\frac{1}{\alpha}}\mu_{\mathsf{t},2}^{-\frac{1}{\alpha}}+p_{\mathsf{s}}^{-1}\mu_{\mathsf{s},2}^{-\left(1+\frac{1}{\alpha}\right)}\mu_{\mathsf{t},2}^{1-\frac{1}{\alpha}}+n_{\mathsf{t}}^{-1}\mu_{\mathsf{s},2}^{-\frac{1}{\alpha}}\mu_{\mathsf{t},2}^{-\frac{1}{\alpha}}\right]\mu_{\mathsf{t},2}^{2(r\wedge\frac{1}{2})}.$$

Thus, combining all terms, we have

$$\mathsf{B}_{var,\mathsf{s}}\approx n_{\mathsf{t}}^{-1}\left(\mu_{\mathsf{s},2}^{-\frac{1}{\alpha}}+n_{\mathsf{s}}^{-1}\mu_{\mathsf{s},2}^{-\frac{1}{\alpha}}\mu_{\mathsf{t},2}^{-\frac{1}{\alpha}}\right),$$
$$\mathsf{B}_{bias,\mathsf{s}}\approx\mu_{\mathsf{s},2}^{2(r\wedge 1)}+p_{\mathsf{s}}^{-1}\mu_{\mathsf{s},2}^{-\frac{1}{\alpha}}\mu_{\mathsf{s},2}^{2(r\wedge\frac{1}{2})}$$
$$+p_{\mathsf{t}}^{-1}\left[\mu_{\mathsf{s},2}^{-\left(1+\frac{1}{\alpha}\right)}\mu_{\mathsf{t},2}+n_{\mathsf{s}}^{-1}\mu_{\mathsf{s},2}^{-\frac{1}{\alpha}}\mu_{\mathsf{t},2}^{-\frac{1}{\alpha}}+n_{\mathsf{t}}^{-1}\mu_{\mathsf{s},2}^{-\frac{1}{\alpha}}\mu_{\mathsf{t},2}^{-\frac{1}{\alpha}}\right.$$
$$\left.+p_{\mathsf{s}}^{-1}\mu_{\mathsf{s},2}^{-\left(1+\frac{1}{\alpha}\right)}\mu_{\mathsf{t},2}^{1-\frac{1}{\alpha}}\right]\mu_{\mathsf{t},2}^{2(r\wedge\frac{1}{2})}.$$

Note that

$$p_{\mathsf{t}}^{-1}p_{\mathsf{s}}^{-1}\mu_{\mathsf{s},2}^{-\left(1+\frac{1}{\alpha}\right)}\mu_{\mathsf{t},2}^{1-\frac{1}{\alpha}}\mu_{\mathsf{t},2}^{2(r\wedge\frac{1}{2})}\le p_{\mathsf{s}}^{-1}\mu_{\mathsf{s},2}^{-\frac{1}{\alpha}}\mu_{\mathsf{s},2}^{2(r\wedge\frac{1}{2})},$$

since $\mu_{\mathsf{t},2}\mu_{\mathsf{s},2}^{-1}\le 1$ and $p_{\mathsf{t}}^{-1}\mu_{\mathsf{t},2}^{-\frac{1}{\alpha}}=O(1)$. Finally, plugging in

$$\mu_{\mathsf{s},2}=n_{\mathsf{t}}^{-\alpha z_{\mathsf{s}}},\qquad\mu_{\mathsf{t},2}=n_{\mathsf{t}}^{-\alpha z_{\mathsf{t}}},$$

where

$$z_{\mathsf{s}}=\frac{\gamma_{n_{\mathsf{s}}}+\gamma_{\lambda_{\mathsf{s}}}}{\alpha}\wedge\gamma_{n_{\mathsf{s}}}\wedge\gamma_{p_{\mathsf{s}}},\qquad z_{\mathsf{t}}=\frac{1+\gamma_{\lambda_{\mathsf{t}}}}{\alpha}\wedge 1\wedge\gamma_{p_{\mathsf{t}}},$$

and using $z_{\mathsf{s}} \leq z_{\mathsf{t}}$, we obtain

$$\mathsf{B}_{var,\mathsf{s}} \approx n_{\mathsf{t}}^{-1+z_{\mathsf{s}}} + n_{\mathsf{t}}^{-1-\gamma_{n_{\mathsf{s}}}+z_{\mathsf{s}}+z_{\mathsf{t}}},$$

$$\mathsf{B}_{bias,\mathsf{s}} \approx n_{\mathsf{t}}^{-2\alpha(r\wedge 1)z_{\mathsf{s}}} + n_{\mathsf{t}}^{-\gamma_{p_{\mathsf{s}}}+z_{\mathsf{s}}-2\alpha(r\wedge\frac{1}{2})z_{\mathsf{s}}}$$

$$+ \left[ n_{\mathsf{t}}^{-\gamma_{p_{\mathsf{t}}}+(\alpha+1)z_{\mathsf{s}}-\alpha z_{\mathsf{t}}} \right.$$

$$+ n_{\mathsf{t}}^{-\gamma_{p_{\mathsf{t}}}-\gamma_{n_{\mathsf{s}}}+z_{\mathsf{s}}+z_{\mathsf{t}}} + n_{\mathsf{t}}^{-\gamma_{p_{\mathsf{t}}}-1+z_{\mathsf{s}}+z_{\mathsf{t}}}$$

$$\left. + n_{\mathsf{t}}^{-\gamma_{p_{\mathsf{t}}}-\gamma_{p_{\mathsf{s}}}+(\alpha+1)z_{\mathsf{s}}-(\alpha-1)z_{\mathsf{t}}} \right] n_{\mathsf{t}}^{-2\alpha(r\wedge\frac{1}{2})z_{\mathsf{t}}}$$

$$= n_{\mathsf{t}}^{-2\alpha(r\wedge 1)z_{\mathsf{s}}} + n_{\mathsf{t}}^{-\gamma_{p_{\mathsf{s}}}+z_{\mathsf{s}}-2\alpha(r\wedge\frac{1}{2})z_{\mathsf{s}}}$$

$$+ n_{\mathsf{t}}^{-\gamma_{p_{\mathsf{t}}}+(\alpha+1)z_{\mathsf{s}}-\alpha z_{\mathsf{t}}-2\alpha(r\wedge\frac{1}{2})z_{\mathsf{t}}}$$

$$+ n_{\mathsf{t}}^{-\gamma_{p_{\mathsf{t}}}-\gamma_{n_{\mathsf{s}}}+z_{\mathsf{s}}+z_{\mathsf{t}}-2\alpha(r\wedge\frac{1}{2})z_{\mathsf{t}}}$$

$$+ n_{\mathsf{t}}^{-\gamma_{p_{\mathsf{t}}}-1+z_{\mathsf{s}}+z_{\mathsf{t}}-2\alpha(r\wedge\frac{1}{2})z_{\mathsf{t}}}.$$

**Derivation of Optimal Scaling Law.** In order to derive the optimal scaling law, we separately consider the two cases $z_{\mathsf{t}} \leq z_{\mathsf{s}}$ and $z_{\mathsf{t}} > z_{\mathsf{s}}$. For simplicity, denote

$$R := r \wedge 1, \qquad q := r \wedge \frac{1}{2}.$$

The target optimal exponent is obtained by balancing the first term in the bias and the first term in the variance:

$$2\alpha R z = 1 - z.$$

Thus,

$$z_* = \frac{1}{1+2\alpha R}, \qquad \gamma_* = \frac{2\alpha R}{1+2\alpha R}.$$

When $z_{\mathsf{t}} \leq z_{\mathsf{s}}$, or equivalently $\mu_{\mathsf{t},2} \geq \mu_{\mathsf{s},2}$, we have

$$\mathsf{B}_{var,\mathsf{s}} \approx n_{\mathsf{t}}^{-1+z_{\mathsf{t}}},$$

$$\mathsf{B}_{bias,\mathsf{s}} \approx n_{\mathsf{t}}^{-2\alpha R z_{\mathsf{t}}} + \left( n_{\mathsf{t}}^{-\gamma_{p_{\mathsf{s}}}+z_{\mathsf{t}}+\alpha(z_{\mathsf{t}}-z_{\mathsf{s}})} + n_{\mathsf{t}}^{-\gamma_{p_{\mathsf{t}}}+z_{\mathsf{t}}} \right) n_{\mathsf{t}}^{-2\alpha q z_{\mathsf{t}}}.$$

Therefore,

$$\gamma_{\mathsf{s},bias} \wedge \gamma_{\mathsf{s},var} \leq [2\alpha R z_{\mathsf{t}}] \wedge [1 - z_{\mathsf{t}}] \leq \frac{2\alpha R}{1+2\alpha R}.$$

The upper bound is achieved by taking

$$z_{\mathsf{t}} = z_* = \frac{1}{1+2\alpha R},$$

provided that $z_{\mathsf{t}} \leq z_{\mathsf{s}}$ and the remaining bias terms are no larger than the target order. Equivalently, we require

$$\gamma_{p_{\mathsf{t}}} - z_{\mathsf{t}} + 2\alpha q z_{\mathsf{t}} \geq \gamma_*,$$

$$\gamma_{p_{\mathsf{s}}} - z_{\mathsf{s}} - \alpha(z_{\mathsf{t}} - z_{\mathsf{s}}) + 2\alpha q z_{\mathsf{t}} \geq \gamma_*.$$

Plugging in $z_{\mathsf{t}} = \frac{1}{1+2\alpha R}$, we obtain (18).

When $z_{\mathsf{t}} > z_{\mathsf{s}}$, or equivalently $\mu_{\mathsf{t},2} < \mu_{\mathsf{s},2}$, we have

$$\mathsf{B}_{var,\mathsf{s}} \approx n_{\mathsf{t}}^{-1+z_{\mathsf{s}}} + n_{\mathsf{t}}^{-1-\gamma_{n_{\mathsf{s}}}+z_{\mathsf{s}}+z_{\mathsf{t}}},$$

$$\mathsf{B}_{bias,\mathsf{s}} \approx n_{\mathsf{t}}^{-2\alpha R z_{\mathsf{s}}} + n_{\mathsf{t}}^{-\gamma_{p_{\mathsf{s}}}+z_{\mathsf{s}}-2\alpha q z_{\mathsf{s}}}$$

$$+ n_{\mathsf{t}}^{-\gamma_{p_{\mathsf{t}}}+(\alpha+1)z_{\mathsf{s}}-\alpha z_{\mathsf{t}}-2\alpha q z_{\mathsf{t}}}$$

$$+ n_{\mathsf{t}}^{-\gamma_{p_{\mathsf{t}}}-\gamma_{n_{\mathsf{s}}}+z_{\mathsf{s}}+z_{\mathsf{t}}-2\alpha q z_{\mathsf{t}}}$$

$$+ n_{\mathsf{t}}^{-\gamma_{p_{\mathsf{t}}}-1+z_{\mathsf{s}}+z_{\mathsf{t}}-2\alpha q z_{\mathsf{t}}}.$$

Therefore,

$$\gamma_{\mathsf{s},bias} \wedge \gamma_{\mathsf{s},var} \leq [2\alpha R z_{\mathsf{s}}] \wedge [1 - z_{\mathsf{s}}] \leq \frac{2\alpha R}{1 + 2\alpha R}.$$

The upper bound is achieved by taking

$$z_{\mathsf{s}} = z_{*} = \frac{1}{1 + 2\alpha R},$$

provided that $z_{\mathsf{t}} > z_{\mathsf{s}}$ and all remaining variance and bias terms are no larger than the target order. Equivalently, we require

$$1 + \gamma_{n_{\mathsf{s}}} - z_{\mathsf{s}} - z_{\mathsf{t}} \geq \gamma_{*},$$
$$\gamma_{p_{\mathsf{s}}} - z_{\mathsf{s}} + 2\alpha q z_{\mathsf{s}} \geq \gamma_{*},$$
$$\gamma_{p_{\mathsf{t}}} - (\alpha + 1)z_{\mathsf{s}} + \alpha z_{\mathsf{t}} + 2\alpha q z_{\mathsf{t}} \geq \gamma_{*},$$
$$\gamma_{p_{\mathsf{t}}} + \gamma_{n_{\mathsf{s}}} - z_{\mathsf{s}} - z_{\mathsf{t}} + 2\alpha q z_{\mathsf{t}} \geq \gamma_{*},$$
$$\gamma_{p_{\mathsf{t}}} + 1 - z_{\mathsf{s}} - z_{\mathsf{t}} + 2\alpha q z_{\mathsf{t}} \geq \gamma_{*}.$$

Plugging in $z_{\mathsf{s}} = \frac{1}{1+2\alpha R}$, we obtain

$$\gamma_{n_{\mathsf{s}}} \geq z_{\mathsf{t}},$$

$$\gamma_{p_{\mathsf{s}}} \geq 1 - \frac{2\alpha q}{1 + 2\alpha R},$$

$$\gamma_{p_{\mathsf{t}}} \geq \max\left\{\frac{2\alpha R + \alpha + 1}{1 + 2\alpha R} - (\alpha + 2\alpha q)z_{\mathsf{t}},\ 1 + (1 - 2\alpha q)z_{\mathsf{t}} - \gamma_{n_{\mathsf{s}}},\ (1 - 2\alpha q)z_{\mathsf{t}}\right\},$$

which corresponds to (19).

### G.3. Proof of Corollary 2

*Proof.* By Corollary 1, W2SG only occurs when $z_{\mathsf{s}} < z_{\mathsf{t}}$. In this case, $\mu_{\mathsf{t},2} \leq \mu_{\mathsf{s},2}$, and from the previous derivation,

$$B_{var,\mathsf{s}} \approx n_{\mathsf{t}}^{-1+z_{\mathsf{s}}} + n_{\mathsf{t}}^{-1-\gamma_{n_{\mathsf{s}}}+z_{\mathsf{s}}+z_{\mathsf{t}}},$$
$$B_{bias,\mathsf{s}} \approx n_{\mathsf{t}}^{-2\alpha(r\wedge 1)z_{\mathsf{s}}} + n_{\mathsf{t}}^{-\gamma_{p_{\mathsf{s}}}+z_{\mathsf{s}}-2\alpha(r\wedge\frac{1}{2})z_{\mathsf{s}}}$$
$$+ n_{\mathsf{t}}^{-\gamma_{p_{\mathsf{t}}}+(\alpha+1)z_{\mathsf{s}}-\alpha z_{\mathsf{t}}-2\alpha(r\wedge\frac{1}{2})z_{\mathsf{t}}}$$
$$+ n_{\mathsf{t}}^{-\gamma_{p_{\mathsf{t}}}-\gamma_{n_{\mathsf{s}}}+z_{\mathsf{s}}+z_{\mathsf{t}}-2\alpha(r\wedge\frac{1}{2})z_{\mathsf{t}}}$$
$$+ n_{\mathsf{t}}^{-\gamma_{p_{\mathsf{t}}}-1+z_{\mathsf{s}}+z_{\mathsf{t}}-2\alpha(r\wedge\frac{1}{2})z_{\mathsf{t}}}.$$

In order for $B_{var,\mathsf{s}} \ll V_{\mathsf{t}}$, we require

$$1 - z_{\mathsf{s}} > 1 - z_{\mathsf{t}}, \qquad 1 + \gamma_{n_{\mathsf{s}}} - z_{\mathsf{s}} - z_{\mathsf{t}} > 1 - z_{\mathsf{t}},$$

which holds as $z_{\mathsf{s}} < z_{\mathsf{t}}$ and $z_{\mathsf{s}} < \gamma_{n_{\mathsf{s}}}$.

Recall that

$$V_{\mathsf{t}} \approx n_{\mathsf{t}}^{-(1-z_{\mathsf{t}})},$$

and

$$B_{\mathsf{t}} \approx n_{\mathsf{t}}^{-2\alpha(r\wedge 1)z_{\mathsf{t}}} + n_{\mathsf{t}}^{-\gamma_{p_{\mathsf{t}}}-(2\alpha(r\wedge\frac{1}{2})-1)z_{\mathsf{t}}}.$$

Thus, $V_{\mathsf{t}} \gg B_{\mathsf{t}}$ is equivalent to

$$1 - z_{\mathsf{t}} < 2\alpha(r \wedge 1)z_{\mathsf{t}},$$

$$1 - z_{\mathsf{t}} < \gamma_{p_{\mathsf{t}}} + \left(2\alpha(r \wedge \frac{1}{2}) - 1\right)z_{\mathsf{t}}.$$

Equivalently,

$$z_t > \frac{1}{1 + 2\alpha(r \wedge 1)}, \qquad z_t > \frac{1 - \gamma_{p_t}}{2\alpha(r \wedge \frac{1}{2})}.$$

Moreover, by definition we have $z_t \leq \gamma_{p_t}$. Hence, for the above interval to be feasible, it is necessary that

$$\frac{1 - \gamma_{p_t}}{2\alpha(r \wedge \frac{1}{2})} < \gamma_{p_t},$$

which gives

$$\gamma_{p_t} > \frac{1}{1 + 2\alpha(r \wedge \frac{1}{2})}.$$

It remains to impose $V_t \gg B_{bias,s}$. The first two terms in the bias give the conditions

$$1 - z_t < 2\alpha(r \wedge 1)z_s,$$
$$1 - z_t < \gamma_{p_s} + \left(2\alpha(r \wedge \frac{1}{2}) - 1\right)z_s. \tag{79}$$

Equivalently,

$$z_s > \frac{1 - z_t}{2\alpha(r \wedge 1)}, \qquad \gamma_{p_s} + \left(2\alpha(r \wedge \frac{1}{2}) - 1\right)z_s > 1 - z_t.$$

We now check that the remaining terms in $B_{bias,s}$ are automatically controlled by the requirement $V_t \gg B_t$. Indeed, the corresponding exponents are

$$E_1 = \gamma_{p_t} - (\alpha + 1)z_s + \alpha z_t + 2\alpha(r \wedge \frac{1}{2})z_t,$$
$$E_2 = \gamma_{p_t} + \gamma_{n_s} - z_s - z_t + 2\alpha(r \wedge \frac{1}{2})z_t,$$
$$E_3 = \gamma_{p_t} + 1 - z_s - z_t + 2\alpha(r \wedge \frac{1}{2})z_t.$$

The teacher's bias exponent satisfies

$$\gamma_{t,V} \leq E_t = \gamma_{p_t} + \left(2\alpha(r \wedge \frac{1}{2}) - 1\right)z_t.$$

Since $z_t > z_s$, we have

$$E_1 - E_t = (\alpha + 1)(z_t - z_s) > 0,$$
$$E_2 - E_t = \gamma_{n_s} - z_s \geq 0,$$
$$E_3 - E_t = 1 - z_s > 0.$$

Therefore, once

$$1 - z_t < E_t$$

holds, namely once $V_t \gg B_t$, these three bias terms are also dominated by $V_t$.

Combining the above, $V_t$ dominates both $B_t$ and $B_{bias,s}$ provided that

$$z_t > \frac{1}{1 + 2\alpha(r \wedge 1)},$$
$$z_t > \frac{1 - \gamma_{p_t}}{2\alpha(r \wedge \frac{1}{2})},$$
$$z_s > \frac{1 - z_t}{2\alpha(r \wedge 1)},$$
$$\gamma_{p_s} + \left(2\alpha(r \wedge \frac{1}{2}) - 1\right)z_s > 1 - z_t,$$
$$z_s < z_t \leq \gamma_{p_t}.$$

In particular, the feasibility of $z_t \leq \gamma_{p_t}$ together with

$$z_t > \frac{1 - \gamma_{p_t}}{2\alpha(r \wedge \frac{1}{2})}$$

requires

$$\gamma_{p_t} > \frac{1}{1 + 2\alpha(r \wedge \frac{1}{2})}.$$

This completes the proof. $\square$

### G.4. Proof of Corollary 3

*Proof.* Let

$$R := r \wedge 1, \qquad q := r \wedge \frac{1}{2}.$$

We focus on the regime $z_t > z_s$. In this regime, we have

$$\mu_{t,2} < \mu_{s,2}.$$

Recall from the previous derivation that

$$B_{var,s} \approx n_t^{-1+z_s} + n_t^{-1-\gamma_{n_s}+z_s+z_t},$$
$$B_{bias,s} \approx n_t^{-2\alpha R z_s} + n_t^{-\gamma_{p_s}+z_s-2\alpha q z_s}$$
$$+ n_t^{-\gamma_{p_t}+(\alpha+1)z_s-\alpha z_t-2\alpha q z_t}$$
$$+ n_t^{-\gamma_{p_t}-\gamma_{n_s}+z_s+z_t-2\alpha q z_t}$$
$$+ n_t^{-\gamma_{p_t}-1+z_s+z_t-2\alpha q z_t}.$$

Furthermore, the teacher bias exponent is

$$\gamma_{t,B} = [2\alpha R z_t] \wedge [\gamma_{p_t} + (2\alpha q - 1)z_t].$$

We first observe that if

$$2\alpha R z_t \leq \gamma_{p_t} + (2\alpha q - 1)z_t,$$

or equivalently

$$\gamma_{p_t} \geq (1 + 2\alpha R - 2\alpha q) z_t,$$

then

$$\gamma_{t,B} = 2\alpha R z_t.$$

Since $z_t > z_s$, we have

$$2\alpha R z_s < 2\alpha R z_t.$$

Thus the term $n_t^{-2\alpha R z_s}$ in $B_{bias,s}$ is larger than $B_t$. Hence

$$B_{bias,s} \gg B_t,$$

and weak-to-strong generalization does not occur.

Therefore, we focus on the complementary regime

$$\gamma_{p_t} < (1 + 2\alpha R - 2\alpha q) z_t. \tag{80}$$

In this regime,

$$\gamma_{t,B} = \gamma_{p_t} + (2\alpha q - 1)z_t.$$

Moreover, since $z_t \leq \gamma_{p_t}$ by the definition of $z_t$, condition (80) cannot hold when $r \leq 1/2$. Hence we must have $r > 1/2$, and therefore

$$q = \frac{1}{2}.$$

Thus

$$\gamma_{t,B} = \gamma_{p_t} + (\alpha - 1)z_t.$$

For teacher's bias to dominate the student's test error, we require

$$\begin{aligned}
&\gamma_{p_t} + (\alpha - 1)z_t < 1 - z_s, \\
&\gamma_{p_t} + (\alpha - 1)z_t < 1 + \gamma_{n_s} - z_s - z_t, \\
&\gamma_{p_t} + (\alpha - 1)z_t < 2\alpha R z_s, \\
&\gamma_{p_t} + (\alpha - 1)z_t < \gamma_{p_s} + (\alpha - 1)z_s, \\
&\gamma_{p_t} + (\alpha - 1)z_t < \gamma_{p_t} + \gamma_{n_s} - z_s - z_t + \alpha z_t
\end{aligned} \tag{81}$$

The remaining two bias terms are automatically controlled as

$$z_s < z_t \leq 1.$$

Indeed,

$$\begin{aligned}
&\gamma_{p_t} - (\alpha + 1)z_s + 2\alpha z_t - (\gamma_{p_t} + (\alpha - 1)z_t) \\
&\quad = (\alpha + 1)(z_t - z_s) > 0,
\end{aligned}$$

and

$$\gamma_{p_t} + 1 - z_s - z_t + \alpha z_t - (\gamma_{p_t} + (\alpha - 1)z_t) = 1 - z_s.$$

The first and third inequalities in (81) imply

$$1 - (\alpha - 1)z_t - \gamma_{p_t} > z_s > \frac{(\alpha - 1)z_t + \gamma_{p_t}}{2\alpha R}.$$

The second inequality further requires

$$z_s < 1 + \gamma_{n_s} - \alpha z_t - \gamma_{p_t},$$

and the fourth inequality requires

$$z_s > \frac{(\alpha - 1)z_t + \gamma_{p_t} - \gamma_{p_s}}{\alpha - 1}.$$

The fifth inequality requires

$$z_s < \gamma_{n_s}.$$

Therefore, the sufficient and necessary condition for weak-to-strong improvement in this regime is that

$$\max\left\{\frac{(\alpha - 1)z_t + \gamma_{p_t}}{2\alpha R}, \frac{(\alpha - 1)z_t + \gamma_{p_t} - \gamma_{p_s}}{\alpha - 1}\right\} < z_s < \min\left\{z_t, 1 - (\alpha - 1)z_t - \gamma_{p_t}, 1 + \gamma_{n_s} - \alpha z_t - \gamma_{p_t}, \gamma_{n_s}\right\}. \tag{82}$$

In order for the interval in (82) to be non-empty, we require

$$\max\left\{\frac{(\alpha - 1)z_t + \gamma_{p_t}}{2\alpha R}, \frac{(\alpha - 1)z_t + \gamma_{p_t} - \gamma_{p_s}}{\alpha - 1}\right\} < \min\left\{z_t, 1 - (\alpha - 1)z_t - \gamma_{p_t}, 1 + \gamma_{n_s} - \alpha z_t - \gamma_{p_t}, \gamma_{n_s}\right\},$$

which is equivalent to each term on the LHS of (82) to be smaller than each term on the RHS.

We first study the terms that only depend on the teacher's scaling, and identify a necessary condition for the teacher. In particular, we require:

$$\frac{(\alpha - 1)z_\mathsf{t} + \gamma_{p_\mathsf{t}}}{2\alpha R} < 1 - (\alpha - 1)z_\mathsf{t} - \gamma_{p_\mathsf{t}}$$

$$\frac{(\alpha - 1)z_\mathsf{t} + \gamma_{p_\mathsf{t}}}{2\alpha R} < z_\mathsf{t}.$$

The above is equivalent to

$$\frac{\gamma_{p_\mathsf{t}}}{1 + 2\alpha R - \alpha} < z_\mathsf{t} < \frac{1}{\alpha - 1}\left(\frac{2\alpha R}{1 + 2\alpha R} - \gamma_{p_\mathsf{t}}\right),$$

where the lower bound is equivalent to (80). This interval for $z_\mathsf{t}$ is non-empty whenever

$$\gamma_{p_\mathsf{t}} < 1 - \frac{\alpha}{1 + 2\alpha R}.$$

Finally, we study the terms that depend both the teacher's and the student's scaling, which leads to the conditions

$$\gamma_{p_\mathsf{s}} > \max\left\{\gamma_{p_\mathsf{t}}, \, \alpha\big((\alpha - 1)z_\mathsf{t} + \gamma_{p_\mathsf{t}}\big) - (\alpha - 1)\right\},$$

$$\gamma_{n_\mathsf{s}} > \max\left\{\frac{(\alpha - 1)z_\mathsf{t} + \gamma_{p_\mathsf{t}}}{2\alpha R}, \, \alpha z_\mathsf{t} + \gamma_{p_\mathsf{t}} - 1 + \frac{(\alpha - 1)z_\mathsf{t} + \gamma_{p_\mathsf{t}}}{2\alpha R}\right\}, \tag{83}$$

$$\gamma_{p_\mathsf{s}} + (\alpha - 1)\gamma_{n_\mathsf{s}} > \max\left\{(\alpha - 1)z_\mathsf{t} + \gamma_{p_\mathsf{t}}, \, (\alpha^2 - 1)z_\mathsf{t} + \alpha\gamma_{p_\mathsf{t}} - (\alpha - 1)\right\},$$

thus concluding the proof. $\qquad\square$

# H. Validity of deterministic equivalent

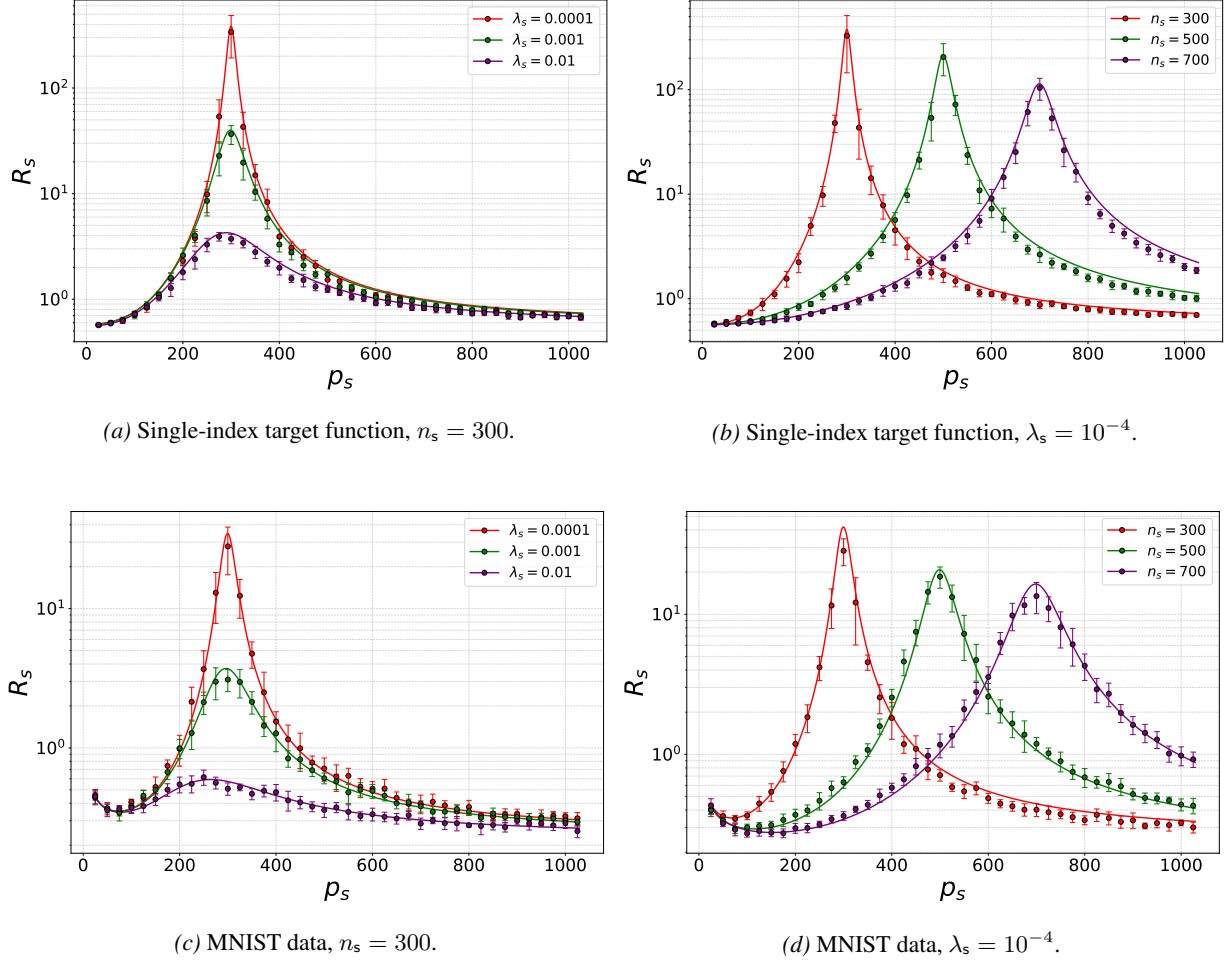

*(a)* Single-index target function, $n_{\mathsf{s}} = 300$.

*(b)* Single-index target function, $\lambda_{\mathsf{s}} = 10^{-4}$.

*(c)* MNIST data, $n_{\mathsf{s}} = 300$.

*(d)* MNIST data, $\lambda_{\mathsf{s}} = 10^{-4}$.

*Figure 2.* Excess test errors of various students trained on teacher labels, together with the corresponding deterministic equivalents (Theorem 2), as a function of the number of student features $p_{\mathsf{s}}$. The teacher and the student are random feature models with the same number of features ($p_{\mathsf{t}} = p_{\mathsf{s}}$), trained with the same sample size ($n_{\mathsf{t}} = n_{\mathsf{s}}$) and with the teacher regularization being half of that of the student ($\lambda_{\mathsf{t}} = \lambda_{\mathsf{s}}/2$). In the top two plots, the data is given by $\boldsymbol{x}_i \sim \mathcal{N}(0, I_d)$ and $y_i = \mathrm{erf}(\langle \boldsymbol{x}_i, \boldsymbol{w}_* \rangle) + \varepsilon_i$, with $\boldsymbol{w}_* \sim \mathrm{Unif}(\mathbb{S}^{d-1})$, and the random feature model is $\varphi(\boldsymbol{x}; \boldsymbol{w}) = \tanh(\langle \boldsymbol{x}, \boldsymbol{w} \rangle)$, $\boldsymbol{w} \sim \mathrm{Unif}(\mathbb{S}^{d-1})$; in the bottom two plots, the data is obtained from the MNIST dataset and the random feature model is $\varphi(\boldsymbol{x}; \boldsymbol{w}) = \mathrm{erf}(\langle \boldsymbol{x}, \boldsymbol{w} \rangle)$, $\boldsymbol{w} \sim \mathrm{Unif}(\mathbb{S}^{d-1})$. In the left figures, we fix the sample size $n_{\mathsf{s}}$ and consider three values of the regularization $\lambda_{\mathsf{s}}$ (in three different colors); in the right figures, we fix the regularization $\lambda_{\mathsf{s}}$ and consider three values of the sample size $n_{\mathsf{s}}$ (in three different colors). We run 10 independent experiments, reporting the average and the confidence interval at 1 standard deviation. The circles represent the test errors obtained experimentally and they match well the continuous curves which represent the deterministic equivalents of Definition 2.

