# OpenReview forum: "Improved Scaling Laws via Weak-to-Strong Generalization in Random Features Ridge Regression"
_ICML.cc/2026/Conference — ICML 2026 regular_

### Official Review · Reviewer_v6NU · 2026-03-03

**Soundness:** 4
**Presentation:** 1
**Significance:** 4
**Originality:** 2
**Overall Recommendation:** 4
**Confidence:** 2

**Summary:**

This paper studies the phenomenon of Weak-to-Strong Generalization (W2SG) in random features regression. The authors derive a deterministic approximation for the test error of a student trained on teacher-generated labels, but tested in the original target function. This is the fundamental contribution of this paper to the theoretical analysis of random-feature models. Thanks to this deterministic equivalent, the authors can derive joint teacher and student scaling laws under standard source and capacity conditions on the target function. In particular, the authors find that there are regimes where the student not only outperforms the teacher, but also enjoys a faster scaling law. I believe this is the fundamental contribution of this paper to the field of W2SG. Illustrative experiments corroborate the theoretical results.

**Compliance With Llm Reviewing Policy:**

Affirmed.

**Final Justification:**

I lack the technical expertise to judge such a technical submission and my requests for clarification were not satisfactorily addressed. I remain of the idea that this contribution can only be appreciated within the corresponding subfield, but I'll leave the final judgement to other reviewers

**Key Questions For Authors:**

No further questions.

**Limitations:**

I did not see an address of the limitations in the manuscript. For instance, what's the role of assumptions 1 and 2? In addition, the different regimes of W2SG behavior are identified in terms of the $z$ exponents, whose current definition is not particularly illuminating.  What are the implications in terms of the various exponents of Eq. (17)? For instance, what are the conditions on the scaling of the number of samples, number of random features and regularization of the teacher for the student to be able to outperform it? How do these conditions compare to practically relevant settings?

**Strengths And Weaknesses:**

## Significance

The result on the possibility of a student achieving a better scaling than the teacher is significant to the theory of W2SG. In addition, the deterministic equivalent for the student's test error could be of interest to the sub-community studying the theory of random features regression.

## Soundness

Due to the complexity of the technical arguments, I did not verify any of the proofs. However, all the results seem sound, as they closely mirror established techniques and results in the literature.

## Originality

For the same reason, is hard for me to judge the originality of the results. While this seems to be the first mathematically provable case where the student's scaling law is faster than the teacher's, I cannot judge whether there is any novelty in the analysis and techniques used. I will probably revise my score during rebuttal, considering also the other reviews.

## Presentation

While the exposition is clear and well-articulated up to page 3, the exposition of the results becomes quickly technical. As a result, the manuscript is obscure to readers not familiar with the literature on deterministic equivalents. This severely limits the impact that such a paper could have for the broad ICML audience, and would make the contribution more suitable for a specialised venue. I am adding a list of improvements that would help me understand the manuscript better:

* The quantities introduced in equations 8-to-10, then (12) and (13) could be named, and the author could illustrate their meaning/role in deriving deterministic equivalents in words (e.g. Upsilon connects to the variance of the bias + variance decomposition), or connect them to analogous quantities that the general deep learning theory public could be familiar with.

* The same reasoning applies to assumptions in and outside theorems. When are they satisfied? Why are they required? A positive example of this type of explanation is the comment after Theorem 1, where the authors explain that "the approximation rate is $o(1)$ under the source-capacity conditions". It would be great to have additional explanations of such statements, i.e. how to use source-capacity to infer that the approximation rate is small.

* The complexity of the definition of the exponents makes them hard to parse, especially $\gamma_{s,\text{bias}}$. Having a few examples could help, e.g. " in the limit of ..., where $z_s$ converges to ... $\gamma$ is controlled by ...". This would also help understanding the scope of the main results for W2SG, beyond the random features setting (see limitations).

---

> ### Author Rebuttal · Authors · 2026-03-30
>
> We thank the reviewer for the constructive comments. We respond below and we will revise accordingly.
>
> **Equation (8)** is a self-consistency system of the type that is standard in random matrix theory. The fixed point allows to characterize the deterministic quantities around which matrix functionals concentrate, eg spectral statistics. In our setting, $(\mu_1,\mu_2) $ is related to the fixed point of the matrix Dyson equation associated with a $3 \times 3$ block linearization (or linear pencil), whose off-diagonal blocks contain the empirical data matrix $G \in R^{n \times d}$ and the random feature matrix $F \in R^{p \times d}$.
>
> **Equation (9)** corresponds to the fixed points associated with teacher random matrices $(G_t,F_t)$ and student random matrices $(G_s,F_s)$.
>
> **Equation (10)**: When $A = I$, $\Upsilon$ is related to the deterministic equivalent of the variance term as $\Upsilon/(1 - \Upsilon)$. This provides a sharper characterization than the proxy typically used only to upper bound the variance based on effective dimension, see eg “Optimal Rates of Kernel Ridge Regression under Source Condition in Large Dimensions” (Zhang et al., 2024) and refs therein.
>
> **Equation (12)**: We do not impose structural assumptions on $\Sigma$. This is in sharp contrast with typical high-dimensional stats/RMT literature, where one often assumes eg bounded condition number or proportional asymptotics. Instead of constraining $\Sigma$, the quality of our approximation depends explicitly on $\Sigma$: the bound depends only on the intrinsic dimension of the tail $r_\Sigma (n)$ (see Definition 3 in Appendix B). This dependence is mild enough that it contributes only a ${\rm polylog}(n)$ factor even for power-law spectra, which are usually outside the scope of standard RMT results. Our approximation guarantee depends on $\rho$ (due to the random feature matrix $F$) and $\tilde \rho$ (due to the random data matrix $G$).
>
> **Equation (13)**: We get a contribution of $\rho$ / $\tilde{\rho}$ for each of the four random matrices $F_t,G_t,F_s,G_s$. We will further discuss when the quantities appearing in the theorem are negligible, see our response **Regimes covered by our theorem** to Reviewer XkRU.
>
> **Role of Assumption 1**: To derive deterministic equivalents, one needs some control on the concentration of quadratic forms of the features; this is already necessary in basic settings, eg to recover the Marchenko-Pastur law. Assumption 1 is chosen because it is relatively simple and yields sharp dimension-free deterministic equivalents (ie bounds that do not explicitly depend on the ambient feature dimension), allowing power-law spectral decay. We expect the deterministic equivalents to remain accurate well beyond this assumption and, to showcase this, Figure 1 at the anonymous link [1] considers a non-linear feature map and data coming from either a single-index target function (top plots) or a standard dataset (bottom plots).
>
> **Role of Assumption 2**: We stress that Assumption 2 is not required for our main result, Theorem 2. It is only used in our presentation of the prior result from [Defilippis et al., 2024] (Theorem 1). In that earlier work, the assumption was needed to obtain a sharp multiplicative bound, see also our response to **Q1** of Reviewer XkRU.
>
> **Assumptions in and outside theorems** We point to our response **Regimes covered by our theorem** to Reviewer XkRU, which will be incorporated in the revision.
>
> **Regimes of W2SG/conditions on scaling of number of samples, number of random features and regularization of the teacher for the student to outperform it** Corollaries 2-3 identify a set of conditions on the number of training samples, model size and regularization allowing the student to improve upon the scaling law of the teacher. To interpret such conditions, we can focus on two special cases: (i) the *kernel limit*, in which both teacher and student have a very large number of features, i.e., $\gamma_{p_t}=\gamma_{p_s}=\infty$, and (ii) the *approximation limit*, in which the student has a very large number of samples labeled by the teacher, i.e., $\gamma_{n_s}=\infty$. We discuss these two special cases in detail in our response to Reviewer QMTZ and we will include the discussion in the revision.
>
> **Limitations of the work** In general, our framework is better suited to handle ridge regression. As discussed in our response to **Q3** of Reviewer XkRU, our framework does not directly capture the effect of early stopping in W2SG, despite the high-level connection between early stopping and regularization. We consider this an interesting future direction. We will include a detailed discussion of limitations in the revision.
>
> [1]  https://github.com/conferenceanonymous46/ICML26.git

---

> > ### Author Rebuttal · Reviewer_v6NU · 2026-04-03
> >
> > The answers to my concern about the technical difficulty of this submission are even harder to parse than the submission itself. How is the sentence "$(\mu_1,\mu_2) $ is related to the fixed point of the matrix Dyson equation associated with a $3 \times 3$ block linearization (or linear pencil)" supposed to help a reader who is confused about equation 8?
> >
> > I will maintain my mark out of pure ignorance. I still believe that this contribution is way too technical to be appreciated by anybody outside of the random features / deterministic equivalents literature

---

> > > ### Author Response · Authors · 2026-04-04
> > >
> > > We thank the reviewer for this feedback.
> > >
> > > We would like to offer another perspective on the quantities $\mu_{1}$ and $\mu_{2}$ for readers outside the random features / deterministic equivalents literature. The quantities $\mu_{1}$ and $\mu_{2}$ can be viewed as effective regularization parameters that appear in the deterministic equivalent for the random feature model. Informally, we have that $(Z^\top Z +\lambda)^{-1} \approx (\mu_1/\lambda) (\frac{1}{p} F^\top F + \mu_1)^{-1} \approx (\mu_2/\lambda) (\Sigma + \mu_2)^{-1} $. Note that this intuition holds for linear functionals and there will be correction terms for functionals of higher order, see Eqs. (32)-(33) in Theorem 5 and (35)-(37) in Theorem 6. We will add this to the revision.

---

### Official Review · Reviewer_LZyi · 2026-03-04

**Soundness:** 3
**Presentation:** 3
**Significance:** 3
**Originality:** 3
**Overall Recommendation:** 5
**Confidence:** 3

**Summary:**

This paper provides a theoretical framework for Weak-to-Strong Generalization (W2SG) by deriving deterministic equivalents and scaling laws for student models trained on teacher-generated labels in the RFRR setting. It identifies the conditions under which a student can achieve a faster error decay rate than its teacher, providing intuition into the two underlying mechanisms: bias reduction and variance reduction.

**Compliance With Llm Reviewing Policy:**

Affirmed.

**Key Questions For Authors:**

Corollary 1 establishes that z_t > z_s is a necessary condition for an improved scaling law. How should this condition be interpreted? Intuitively, it seems to me that this suggests that the teacher must have been trained in a regime where it was "richer" in relative data or capacity than the student is. Does this imply that in the reverse case, where the teacher is limited compared to the student in both data and capacity (which seems highly likely in real-world scenario), improved scaling laws cannot occur? Furthermore, how does this result connect to the findings of Wu & Sahai 2024, which demonstrate W2SG in cases where the teacher is smaller than the student in both data size and model capacity?

**Limitations:**

Yes

**Strengths And Weaknesses:**

Strengths:
1. This paper presents novel insights into scaling laws under W2SG and provides multiple interpretable theoretical results regarding the conditions under which a student achieves better scaling laws than its teacher. It sheds light on how regularization and relative sample size interact with bias and variance, contributing to the current understanding of W2SG.
2. The presentation is largely clear, and the key results are well-explained.
3. Numerical experiments support the main theoretical findings.

Weaknesses:
1. The work would be strengthened by experiments beyond Gaussian linear models. More realistic setups would help determine if the characterized phenomenon is relevant in real-world scenarios

---

> ### Author Rebuttal · Authors · 2026-03-30
>
> We thank the reviewer for the positive evaluation of our work. We reply to comments below.
>
> **Weakness** This is a great point and, to address it, we have performed a numerical simulation as suggested by the reviewer. Figure 1 reported in the anonymous link [1] shows that the deterministic equivalent in Definition 2 closely tracks the behavior of the student test error for a non-linear feature map and data coming from either a single-index target function (top plots) or from a standard dataset (MNIST, bottom plots). In both cases, the deterministic equivalents are evaluated by estimating $(\beta_*,\Sigma)$ from a large dataset, see Appendix C.3 in [Defilippis et al., 2024]. We will add this in the revision.
>
> **Condition $z_t>z_s$ and connection to [Wu & Sahai, 2024]** At a high level, we agree with the reviewer that $z_t > z_s$ suggests that the teacher is “richer” than the student. However, this condition cannot be directly translated into the teacher being limited in data or capacity compared to the student, due to the effect of regularization. We discuss this below.
>
> On one hand, if the student is substantially richer in data and capacity compared to the teacher, i.e. $\gamma_{p_s}, \gamma_{n_s} \rightarrow \infty$, then $z_s \rightarrow \infty$ so that the condition $z_t > z_s$ cannot be satisfied by any choice $\gamma_{p_t}, \gamma_{\lambda_t}$ and W2SG cannot happen. This is actually expected. Since the student’s data is noiseless, the student will overfit the teacher given sufficiently large data and capacity.
>
> On the other hand, if $\gamma_{p_s}, \gamma_{n_s}$ are large but finite compared to the teacher, it is possible to exhibit concrete cases in which W2SG happens with teacher smaller than the student both in data size and model size ($\gamma_{p_t} < \gamma_{p_s}$, $1 < \gamma_{n_s}$) by finding a proper regularization $\gamma_{\lambda_s}.$ For a concrete example, we can pick $\alpha = 1.2, r = 0.45, \gamma_{p_t} = 0.5, \gamma_{\lambda_t} > -0.4$. This implies $z_t = 0.5$ and the teacher exponent is $\gamma_{\*, t} = 0.5$. For the student model, we pick $\gamma_{p_s} = 1.1, \gamma_{n_s} = 1.1, \gamma_{\lambda_s} = -0.524,$ which implies $z_s = 0.48$. This satisfies the conditions in Corollary 2 and leads to the student exponent $\gamma_{\*, s} = 0.52$. As a result, the scaling law of the student is better than that of the teacher.
>
> We discuss further regimes in which W2SG occurs in our response to Reviewer QTMZ, where we focus on two special cases (*kernel regime* and *approximation regime*).
>
> We finally note that Assumption (3) of Theorem 3.2 in [Wu & Sahai, 2024] requires an upper bound on the number of samples of the student, which complies with the above discussion that the student cannot be substantially richer than the teacher in both data and capacity for improvements in scaling law to happen. Besides, [Wu & Sahai, 2024] construct a case where the teacher’s test error is $\Theta(1)$ while the student’s is $o(1),$ which, at high-level, is similar to ours in the variance-reduction setting, despite the different setup considered (binary classification v.s. ridge regression).
>
> We will add the above discussions in the revision.
>
> [1] https://github.com/conferenceanonymous46/ICML26.git

---

> > ### Author Rebuttal · Reviewer_LZyi · 2026-04-03
> >
> > Thanks for the response. I am satisfied with the authors’ responses and will keep my positive score.

---

### Official Review · Reviewer_QMTZ · 2026-03-13

**Soundness:** 3
**Presentation:** 2
**Significance:** 4
**Originality:** 3
**Overall Recommendation:** 5
**Confidence:** 3

**Summary:**

The article studies the theoretical scaling laws of two models in the context of random feature ridge regression, where a strong model is trained using labels generated by a weak model. Under strong concentrations of the eigenfunctions of the feature map and other technical assumptions, it is proved that the scaling law of the strong model can be more favorable than that of the weak learner.

**Compliance With Llm Reviewing Policy:**

Affirmed.

**Key Questions For Authors:**

Could you explain the results of Theorem 4 and the following corollaries?  Currently, the results show that the student can achieve better scaling laws if certain constraints among different parameters are satisfied - but what do these constraints mean?

**Limitations:**

The results are limited to random feature ridge regression, but it is not a significant limitation.

**Strengths And Weaknesses:**

Strengths: The paper provides a rigorous analysis of weak-to-strong generalization and improves upon the existing error rates. In addition, the article provides an interpretable mechanism for reducing the error of the strong model depending on whether the weak model has a larger variance than bias or vice versa.

Weaknesses: The article is very theoretical, and navigating through the notations is challenging. Interpretations of some technical assumptions and theorem results are not clear.

---

> ### Author Rebuttal · Authors · 2026-03-30
>
> We thank the reviewer for appreciating our work.
>
> As concerns the interpretation of the various notations, see our answer to Reviewer v6NU.
>
>
> As concerns the interpretation of Theorem 4 and the following corollaries, let us focus on two special cases: (i) the *kernel limit*, in which both teacher and student have a very large number of features, i.e., $\gamma_{p_t}=\gamma_{p_s}=\infty$, and (ii) the *approximation limit*, in which the student has a very large number of samples labeled by the teacher, i.e., $\gamma_{n_s}=\infty$.
>
> In the *kernel limit*, W2SG improvements come only from variance reduction, since the upper bound on $\gamma_{p_t}$ required by Corollary 3 does not hold. Furthermore, the student improves the scaling law only of teachers that are not already optimal, due to a sub-optimal choice of the regularization. Formally, this corresponds to $z_t$ larger than its optimal value $\frac{1}{1+2\alpha(r \wedge 1)}$ (Theorem 3 shows that the optimal teacher exponent is reached when $z_t=\frac{1}{1+2\alpha(r\wedge 1)}$, and Corollary 2 requires $z_t>\frac{1}{1+2\alpha(r\wedge 1)}$). Now, for any sub-optimal teacher, W2SG holds upon choosing properly sample size and regularization of the student:
>
> * If the student has enough samples ($\gamma_{n_s}>z_t$, implied when the student has more samples than the teacher), the scaling law improvement occurs for a range of regularizations ($\gamma_{\lambda_{s}}\in (\frac{1-z_t}{2(r \wedge 1)}-\gamma_{n_s}, \alpha z_t-\gamma_{n_s})$).
>
> * For small regularizations ($\gamma_{\lambda_s}\ge (\alpha-1)\gamma_{n_s}$), the scaling law improvement occurs for a range of sample sizes ($\gamma_{n_s}\in (\frac{1-z_t}{2\alpha(r\wedge 1)}, z_t)$); the improvement also occurs for large regularizations ($\gamma_{\lambda_s}< (\alpha-1)\gamma_{n_s}$), albeit in a different range of sample sizes ($\gamma_{n_s}\in (\frac{1-z_t}{2(r\wedge 1)}-\gamma_{\lambda_s}, \alpha z_t-\gamma_{\lambda_s})$).
>
> In the *approximation limit*, W2SG improvements can come from either variance or bias reduction. Given that the number of student samples is large, its regularization becomes irrelevant: $\gamma_{n_s}=\infty$ implies $z_s=\gamma_{p_s}$ and, hence, $\gamma_{\lambda_s}$ does not appear in any of the conditions of Corollaries 2-3. Thus, the only relevant student quantity is its number of features, which has to be chosen carefully in order to improve the scaling law. As mentioned in the paper, when W2SG occurs due to a bias reduction, the student width has to be larger than the teacher width; in contrast, when W2SG occurs due to a variance reduction, the student width has to be smaller than the teacher width.
>
> We will add this discussion in the revision.

---

> > ### Author Rebuttal · Reviewer_QMTZ · 2026-04-05
> >
> > The authors have adequately responded to my comments.

---

### Official Review · Reviewer_XkRU · 2026-03-13

**Soundness:** 3
**Presentation:** 2
**Significance:** 2
**Originality:** 2
**Overall Recommendation:** 4
**Confidence:** 4

**Summary:**

This work investigates the excess risk of random feature ridge regression trained via teacher student training, extending the results from Defilippis et al. (2024). Based on these results and the power law assumption on the spectrum and target decay, the authors establish a scaling law for the student. They further connect this to weak to strong generalization by comparing the student scaling law with the teacher scaling law, identifying the conditions under which the student achieves a better scaling law than its teacher.

**Compliance With Llm Reviewing Policy:**

Affirmed.

**Final Justification:**

The authors successfully addressed my primary concern regarding technical novelty and significance during the rebuttal. I now acknowledge the significance of their technical contribution and that it is challenging to obtain multiplicative bounds.

**Key Questions For Authors:**

1. In Defilippis et al. (2024), Assumption 3.2 is required to establish their results. Furthermore, they seem to require an additional assumption, specifically Equation 16 in their work, which builds upon settings similar to Assumption 2 in this current paper. I am curious whether this additional condition is unnecessary for the present analysis due to specific differences in the setup, or if it was unintentionally omitted.
2. Could the authors clarify whether any new proof techniques were applied to prove Theorem 2? Specifically, I am interested in how the proof of Theorem 2 differs from the proof techniques used in Defilippis et al. (2024). If a novel analytical approach was introduced  I believe it would be beneficial to present these technical highlights in the main text.
3. In the weak to strong generalization literature, the importance of early stopping has been extensively addressed in both practice (Burns et al., 2024) and theory (Medvedev et al., 2025; Oh et al., 2025), particularly in regimes where population data or an abundance of data is available. Since early stopping often prevent the student from overfitting to the teacher's errors, I am curious whether the proposed framework can also capture the importance of early stopping.

---
Reference

[1] Defilippis et al. Dimension-free deterministic equivalents and scaling laws for random feature regression. NeurIPS 2024

[2] Burns et al. Weak-to-Strong Generalization: Eliciting Strong Capabilities with Weak Supervision. ICML 2024

[3] Medvedev et al. Weak-to-Strong Generalization Even in Random Feature Networks, Provably. ICML 2025

[4] Oh et al. From Linear to Nonlinear: Provable Weak-to-Strong Generalization through Feature Learning. NeurIPS 2025

**Limitations:**

Yes

**Strengths And Weaknesses:**

### **Strengths**

To the best of my knowledge, this is the first work to investigate weak to strong generalization from the perspective of scaling laws. The findings are interesting; in particular, it is surprising that the **s**tudent can achieve the optimal error rate even in regimes where the teacher error does not decay with the sample size.

### **Weaknesses**

- My first concern relates to technical soundness and novelty. I found that a significant portion of the content depends heavily on the settings and results from Defilippis et al. (2024). While I acknowledge that adapting existing frameworks is not necessarily a major weakness in itself, the weak theoretical results regarding the student deterministic equivalent amplify this concern. While the authors leave showing multiplicative bound for student deterministic equivalent as future work, I believe this is very important problem that cannot be overlooked. I also believe that discussions in line 286-line 299 (left) should be more specifically support why student deterministic equivalent bound is sufficiently small, thus justifying why focusing on $R_S$ is sufficient for the remainder of the analysis.
- In Section 5, the authors focus on the question of when a student can achieve a better scaling law than its teacher to address weak to strong generalization. However, I believe the current draft does not sufficiently address the other essential aspect: the definition of a weak teacher and a strong student. It is difficult to grasp why the specific regime where the student outperforms its teacher corresponds to the weak teacher and strong student scenario.

---

> ### Author Rebuttal · Authors · 2026-03-30
>
> We thank the reviewer for detailed comments to which we reply below. We will revise accordingly.
>
> **Technical soundness and novelty** While we build on [Misiakiewicz & Saeed, 2024; Defilippis et al., 2024], we respectfully disagree that our theoretical contributions are weak. Our setting is much more involved than [Defilippis et al., 2024]: our student test error depends on 4 sources of randomness (student data, teacher data, student random features, teacher random features), and the test function is different from the train target function (eliminating various simplifications). Thus, we have to control many extra terms, including new asymmetric functionals.
>
> More precisely, (i) we study the new functional ${\rm Tr}(AMBM)$, and (ii) [Misiakiewicz & Saeed, 2024; Defilippis et al., 2024] focus on functionals involving a PSD matrix, while ours involve non-PSD matrices. The non-PSD extension is hard, also because of the difficulty of getting multiplicative bounds (see below). Beyond this, (iii) deriving the student test error itself requires substantial work (including finding a decomposition with non-vacuous deterministic-equivalent bounds), and (iv) obtaining scaling-law consequences from deterministic equivalents is non-trivial (requiring derivations specific to W2S setting).
>
> **Multiplicative bounds** Multiplicative bounds are not possible for general functionals. Eg for $u^\top M v$, Eq. (34) implies
> $| u^\top Mv-u^\top\bar M,v|\leq O(n^{-1/2}) \sqrt{u^\top M u v^\top M v }$, which is tight in experiments. This makes sense as when $u^\top M v\approx 0$, one cannot hope to have a multiplicative bound (which would imply zero fluctuations).
>
> In general, multiplicative bounds are delicate and [Defilippis et al., 2024] introduce Assumption 2 to that aim. In our setting, because many more terms appear, analogous multiplicative estimates might require several additional assumptions and a much longer argument. While we believe such an extension to be possible, it is not essential to our main goal of understanding when W2S training improves scaling laws.
>
> **Regimes covered by our theorem** Theorem 2 suffices to capture the phenomenology of Section 5. In particular, when $r$ is close to $0$ and $\alpha$ is close to $1$, one can choose parameters in (17) s.t. $(p_t \mu_{t, 1}, p_t \lambda_t/n_t, p_s\lambda_s/n_s)$ decay arbitrarily slowly in $n_s$ (ie as $n_s^{-a}$ for some $a$ very close to $0$). Thus, $\mathcal E_{n_s, p_s, n_t, p_t}$ decays roughly as $\min(n_t, n_s, p_t, p_s)^{-1/2}$ and, even if the bound is not multiplicative, we still have $\mathcal E_{n_s, p_s, n_t, p_t}\tilde{\mathcal{R}}_s \ll \text{R}_s$ since the minimax rate for $\text{R}_s$ also decays slowly with $n_s$. This already covers various W2SG regimes.
>
> In addition, the simulations of Figure 1 match theory well, indicating small fluctuations compared to deterministic equivalents and the correctness of the scaling law based on deterministic equivalence. To those, we add a plot capturing W2SG more clearly in a bias-dominated regime, see Figure 2 at the anonymous link [1]. More broadly, we expect deterministic equivalents to be accurate well beyond the assumptions of Theorem 2. To showcase this, Figure 1 at [1] considers a non-linear feature map and data coming from a single-index target (top plots) or a standard dataset (bottom plots).
>
> **Definition of weak/strong models** We say the teacher is weaker than the student whenever it has worse test error on the target task than the student. Thus, Corollaries 2-3 establish under what conditions the teacher is weaker than the student. It is also common in the literature to define the weak teacher as a model having less capacity than the student. This corresponds to $\gamma_{p_t} < \gamma_{p_s}$, and Corollaries 2-3 allow us to deduce when the student scaling law improves under this extra constraint.
>
> **Q1.** [Defilippis et al., 2024] introduce Eq. (16) to reduce the dependence on $\lambda$ in multiplicative bounds, replacing it by $\lambda_{>m}$. As we do not pursue multiplicative bounds, we do not impose Eq. (16). Technically, incorporating $\lambda_{>m}$ requires splitting both student and teacher features into low- and high-frequency components, creating many more terms than in [Defilippis et al., 2024].
>
> **Q2.** See above (**Technical soundness and novelty**).
>
> **Q3.** Our framework is better suited to handle ridge regression than early stopping. However, both ridge and early stopping have the same high-level goal of preventing overfitting, and the connection between the two has been investigated, see eg “Early stopping and non-parametric regression: An optimal data-dependent stopping rule” (Raskutti et al., 2014). Early-stopped RFRR was considered in “Privacy for free in the over-parameterized regime” (Bombari et al., 2025) for differentially-private learning, and we regard incorporating early stopping as an interesting future direction.
>
> [1]  https://github.com/conferenceanonymous46/ICML26.git

---

> > ### Author Rebuttal · Reviewer_XkRU · 2026-04-03
> >
> > Thanks to the authors for their detailed rebuttal. The rebuttal successfully addressed my concerns. In particular, clarifying the technical novelty and the discussion on multiplicative bounds were helpful. I suggest that the authors include a discussion on why the framework in this work already captures the importance of early stopping (using ridge to prevent overfitting instead of early stopping), as mentioned in the reference I provided, for the benefit of the readers. I have increased my score as the authors resolved my concerns.

---

### Decision · Program_Chairs · 2026-04-30

**Decision:**

Accept (regular)

**Comment:**

This paper studies weak-to-strong generalization in a two-stage teacher–student setting within random features ridge regression. The authors derive a deterministic equivalent for the student’s test error and use it to characterize scaling laws, identifying regimes in which the student can achieve improved rates compared to the teacher.

The reviewers agree that the paper is technically sound, and the rebuttal clarified several of the initial concerns, in particular regarding the relation to prior work and the nontrivial nature of extending deterministic-equivalent techniques to the two-stage setting. The analysis is careful and builds in a meaningful way on recent advances in the theory of random features and scaling laws.

The main strength of the paper lies in providing a clean and tractable framework in which weak-to-strong generalization can be analyzed at the level of scaling exponents. While the setting remains stylized and some limitations persist (e.g., additive rather than multiplicative control of the error, and a focus on ridge-based training), the results offer a useful step toward a more systematic understanding of this phenomenon.

Overall, I find the contribution to be solid and technically well-executed, and suitable for acceptance at ICML.